# From Linear to Nonlinear: Provable Weak-to-Strong Generalization through Feature Learning

**Junsoo Oh**     **Jerry Song**     **Chulhee Yun**
KAIST
{junsoo.oh, sjerry0316, chulhee.yun}@kaist.ac.kr

## Abstract

*Weak-to-strong generalization* refers to the phenomenon where a stronger model trained under supervision from a weaker one can outperform its teacher. While prior studies aim to explain this effect, most theoretical insights are limited to abstract frameworks or linear/random feature models. In this paper, we provide a formal analysis of weak-to-strong generalization from a *linear CNN (weak)* to a *two-layer ReLU CNN (strong)*. We consider structured data composed of label-dependent signals of varying difficulty and label-independent noise, and analyze gradient descent dynamics when the strong model is trained on data labeled by the pretrained weak model. Our analysis identifies two regimes—data-scarce and data-abundant—based on the signal-to-noise characteristics of the dataset, and reveals distinct mechanisms of weak-to-strong generalization. In the *data-scarce* regime, generalization occurs via benign overfitting or fails via harmful overfitting, depending on the amount of data, and we characterize the transition boundary. In the *data-abundant* regime, generalization emerges in the early phase through label correction, but we observe that overtraining can subsequently degrade performance.

## 1   Introduction

As the capabilities of today's AI models grow, recent models such as state-of-the-art large language models (LLMs) increasingly demonstrate superhuman performance in various domains. The complex and often unpredictable behaviors of superhuman models make it crucial to align them with human intent, a challenge known as superalignment. In order to tackle this challenge, human-level supervision techniques, such as reinforcement learning from human feedback (RLHF), are commonly applied. This situation, where a less capable supervisor guides a more advanced model, reverses the traditional teaching paradigm and raises an important question: What happens when a model with stronger capabilities is trained under the supervision of a weaker one?

To address this question, Burns et al. (2024) performed extensive experiments training strong student models, like GPT-4 (OpenAI, 2023), with supervision from a weaker teacher model, such as a fine-tuned GPT-2 (Radford et al., 2019). They observe that the strong model consistently surpasses their supervisor's performance, and refer to this phenomenon as *weak-to-strong generalization*. This surprising phenomenon has attracted considerable attention, and several recent studies have investigated it from theoretical perspectives.

Lang et al. (2024) introduce a theoretical framework that establishes weak-to-strong generalization when the strong model is unable to fit the weak model's mistakes. Building on this framework, Shin et al. (2025) propose a mechanism for weak-to-strong generalization in data exhibiting both easy and hard patterns. Concurrently, another line of work has focused on quantifying the weak-to-strong gain. Charikar et al. (2024) investigate the relationship between weak-to-strong gains and the misfit between weak and strong models in regression with squared loss. Specifically, they show that the

gain in weak-to-strong generalization correlates with the degree of misfit between the weak and strong models. Mulgund and Pabbaraju (2025) and Yao et al. (2025) extend this analysis to a broader class of loss functions, including the reversed Kullback–Leibler divergence. However, both lines of work often rely on abstract theoretical frameworks and typically do not guarantee that weak-to-strong generalization can be achieved through practical training procedures such as gradient-based optimization.

Wu and Sahai (2025) explore weak-to-strong generalization in an overparameterized spiked covariance model and prove transitions between generalization and random guessing by considering both weak and strong models as minimum $\ell_2$ norm interpolating solutions on feature spaces of differing expressivity. Ildiz et al. (2024) investigate a more general form of knowledge distillation (Hinton et al., 2015) in a high-dimensional regression setting and show that distillation from a weak model can outperform distillation from a strong model, while it fails to improve the overall scaling law. Dong et al. (2025) also study a linear regression setting from a variance reduction perspective via the intrinsic dimension of feature spaces. However, these works are limited to linear models and rely on specific assumptions on structural differences between the feature spaces of weak and strong models. A more recent work by Medvedev et al. (2025) alleviates some of these limitations by using random feature networks of differing widths for the strong and weak models. However, in their approach, the trainable component is still linear. These limitations motivate the following question:

*When and how does weak-to-strong generalization emerge through nonlinear feature learning?*

## 1.1 Summary of Contributions

In this paper, we investigate a classification problem on structured data composed of patches, which consist of signals and noise. We employ linear CNNs as the weak model and two-layer ReLU CNNs as the strong model. We focus on the following training scenario: training the weak model under true supervision and then training the strong model under supervision from the pretrained weak model. We investigate how models trained through this scenario perform, particularly focusing on when and how weak-to-strong generalization emerges. We summarize our contributions as follows:

- We compare the capability of weak models and strong models in our data distribution, showing that any weak model makes non-negligible errors (Proposition 2.1) while there exists a strong model that exhibits zero errors (Proposition 2.2).

- We prove that training a weak model using a finite number of training samples and gradient descent can result in a test error that is close to the best possible error achievable by the weak model architecture (Theorem 3.3).

- We also demonstrate that when a strong model is trained on a finite set of samples using supervision from a weak model that makes non-negligible errors, it either achieves near-optimal generalization via benign overfitting or suffers from degraded performance due to harmful overfitting. We further characterize the conditions under which this transition occurs (Theorem 3.4).

- We further explore weak-to-strong training in the regime where more data is available than the previously considered scenario, and perhaps surprisingly, we find that it exhibits a notably different behavior. The strong model can achieve near-zero test error even while the training error on pseudo-labels remains non-negligible (Theorem 3.6). However, we also empirically observe that "overtraining" until convergence to zero training loss eliminates this benefit, resulting in test error levels close to those of the weak model.

## 2 Problem Setting

In this section, we introduce the data distribution and weak/strong model architectures that we focus on, and formally describe the training scenario considered in our work.

In our analysis, we adopt a patch-wise structured data distribution and patch-wise convolutional neural network architectures. This approach follows a recent line of work on feature learning theory starting from Allen-Zhu and Li (2020). This type of setting provides a simple but useful framework for studying training dynamics in deep learning. Similar problem settings have been widely used to understand several aspects of deep learning, such as benign overfitting (Cao et al., 2022; Kou et al., 2023; Meng et al., 2024), optimizer (Jelassi and Li, 2022; Zou et al., 2023b; Chen et al., 2023), data

augmentation (Shen et al., 2022; Chidambaram et al., 2023; Zou et al., 2023a; Oh and Yun, 2024; Li et al., 2025), and architecture (Huang et al., 2023; Jiang et al., 2024). The broad utility of such settings confirms their value in understanding fundamental aspects of deep learning.

## 2.1 Data Distribution

We investigate a binary classification problem on structured data consisting of multiple patches. These patches contain label-dependent vectors (called *signal*) and label-independent vectors (called *noise*).

**Definition 1.** We define a data distribution $\mathcal{D}$ on $\mathbb{R}^{d \times 3} \times \{\pm 1\}$ such that a sample $(\boldsymbol{X}, y) \sim \mathcal{D}$ with $\boldsymbol{X} = \left(\boldsymbol{x}^{(1)}, \boldsymbol{x}^{(2)}, \boldsymbol{x}^{(3)}\right)$ and $y \in \{\pm 1\}$ is constructed as follows.

1. Choose the *label* $y \in \{\pm 1\}$ uniformly at random.

2. Let $\{\boldsymbol{\mu}_1, \boldsymbol{\mu}_{-1}, \boldsymbol{\nu}_1, \boldsymbol{\nu}_{-1}\}$ be a set of mutually orthogonal *signal vectors*. We choose two signal vectors $\boldsymbol{v}^{(1)}, \boldsymbol{v}^{(2)} \in \mathbb{R}^d$ for data point $\boldsymbol{X}$ associated with the label $y$ as follows:

$$\left(\boldsymbol{v}^{(1)}, \boldsymbol{v}^{(2)}\right) \sim \begin{cases} (\boldsymbol{\mu}_y, \boldsymbol{\mu}_y) & \text{with probability } p_{\text{e}} \\ \text{Unif}\{(\boldsymbol{\nu}_y, \boldsymbol{\nu}_y), (\boldsymbol{\nu}_y, -\boldsymbol{\nu}_y), (-\boldsymbol{\nu}_y, \boldsymbol{\nu}_y), (-\boldsymbol{\nu}_y, -\boldsymbol{\nu}_y)\} & \text{with probability } p_{\text{h}} \\ \text{Unif}\{(\boldsymbol{\mu}_y, \boldsymbol{\nu}_y), (\boldsymbol{\mu}_y, -\boldsymbol{\nu}_y), (\boldsymbol{\nu}_y, \boldsymbol{\mu}_y), (-\boldsymbol{\nu}_y, \boldsymbol{\mu}_y)\} & \text{with probability } p_{\text{b}} \end{cases}$$

   For simplicity, we assume $\|\boldsymbol{\mu}_1\| = \|\boldsymbol{\mu}_{-1}\|$ and $\|\boldsymbol{\nu}_1\| = \|\boldsymbol{\nu}_{-1}\|$, and refer to their norms as $\|\boldsymbol{\mu}\|$ and $\|\boldsymbol{\nu}\|$, respectively, omitting the subscripts.

3. A *noise vector* $\boldsymbol{\xi}$ is drawn from a Gaussian distribution $\mathcal{N}\left(\boldsymbol{0}, \sigma_p^2 \boldsymbol{\Lambda}\right)$, where the covariance matrix is given by $\boldsymbol{\Lambda} = \boldsymbol{I}_d - \frac{\boldsymbol{\mu}_1 \boldsymbol{\mu}_1^\top}{\|\boldsymbol{\mu}\|^2} - \frac{\boldsymbol{\mu}_{-1} \boldsymbol{\mu}_{-1}^\top}{\|\boldsymbol{\mu}\|^2} - \frac{\boldsymbol{\nu}_1 \boldsymbol{\nu}_1^\top}{\|\boldsymbol{\nu}\|^2} - \frac{\boldsymbol{\nu}_{-1} \boldsymbol{\nu}_{-1}^\top}{\|\boldsymbol{\nu}\|^2}$.

4. The components $\boldsymbol{x}^{(1)}, \boldsymbol{x}^{(2)}, \boldsymbol{x}^{(3)}$ of the data point $\boldsymbol{X}$ are formed by assigning the generated vectors $\boldsymbol{v}^{(1)}, \boldsymbol{v}^{(2)}, \boldsymbol{\xi}$ in a randomly shuffled order.

Our data distribution is based on characteristics of image data, where inputs consist of multiple patches. Some patches contain information relevant to the label (such as a face or a tail for "dog"), while others contain irrelevant information, like grass in the background. Intuitively, a model can fit data by learning signals and/or memorizing noise. However, relying primarily on noise memorization instead of learning signals leads to poor generalization since noise is label-irrelevant. Therefore, effectively learning signals is crucial for achieving better generalization.

Real-world data often contains multiple types of label-relevant information, and these corresponding signals can exhibit varying levels of learning difficulty. For example, both a face and a tail are useful for recognizing "dog", but learning the tail could be harder since it occupies only a small region of the image or appears only in a small number of images. To reflect this difference, we consider two types of signals. We refer to $\boldsymbol{\mu}_1, \boldsymbol{\mu}_{-1}$ as *easy signals* and $\boldsymbol{\nu}_1, \boldsymbol{\nu}_{-1}$ as *hard signals*. These signal types are designed to have different levels of learning difficulty within the architectures we focus on, as detailed in the following subsection. We categorize a data point having only easy signals as *easy-only data*, only hard signals as *hard-only data*, and both types of signals as *both-signal data*. We denote by $\mathcal{S}_{\text{e}}, \mathcal{S}_{\text{h}},$ and $\mathcal{S}_{\text{b}}$ the supports of these data categories, respectively.

## 2.2 Neural Network Architecture

We now define the weak and strong model architectures in our analysis. First, weak models are linear convolutional neural networks where patch-wise convolution is applied.

**Definition 2** (Weak Model). We consider our weak model as linear CNN $f_{\text{wk}}(\boldsymbol{w}, \cdot) : \mathbb{R}^{d \times 3} \to \mathbb{R}$ parameterized by $\boldsymbol{w} \in \mathbb{R}^d$ defined as follows. For each input $\boldsymbol{X} = \left(\boldsymbol{x}^{(1)}, \boldsymbol{x}^{(2)}, \boldsymbol{x}^{(3)}\right) \in \mathbb{R}^{d \times 3}$, we define

$$f_{\text{wk}}(\boldsymbol{w}, \boldsymbol{X}) = \left\langle \boldsymbol{w}, \boldsymbol{x}^{(1)} \right\rangle + \left\langle \boldsymbol{w}, \boldsymbol{x}^{(2)} \right\rangle + \left\langle \boldsymbol{w}, \boldsymbol{x}^{(3)} \right\rangle$$

Our choice of weak model has limited capability for learning our data distribution $\mathcal{D}$. In particular, any weak model shows random-guess level performance on hard-only data, as formalized below.

**Proposition 2.1.** *Let $(\boldsymbol{X}, y) \sim \mathcal{D}$ be a test example. For any weak model $f_{\text{wk}}(\boldsymbol{w}, \cdot)$, it satisfies $\mathbb{P}[y f_{\text{wk}}(\boldsymbol{w}, \boldsymbol{X}) < 0 \mid (\boldsymbol{X}, y) \in \mathcal{S}_{\text{h}}] = \frac{1}{2}$.*

*Proof.* Consider a hard-only data $(\boldsymbol{X}, y) \in \mathcal{S}_{\mathrm{h}}$ with the noise vector $\boldsymbol{\xi}$. If the two underlying signals in a hard-only data point have opposite signs, the weak model's output $f_{\mathrm{wk}}(\boldsymbol{w}, \boldsymbol{X})$ simplifies to $\langle \boldsymbol{w}, \boldsymbol{\xi} \rangle$. This results in a $1/2$ conditional error rate due to symmetry of noise. For a hard-only data having two signal vectors of identical signs, we may assume two signal vectors of $(\boldsymbol{X}, y) \in \mathcal{S}_{\mathrm{h}}$ are both $\boldsymbol{\nu}_y$, without loss of generality. Define $(\tilde{\boldsymbol{X}}, y) \in \mathcal{S}_{\mathrm{h}}$ to be a data point where both signal vectors are $-\boldsymbol{\nu}_y$ and the noise vector is $-\boldsymbol{\xi}$. Then, $y f_{\mathrm{wk}}(\boldsymbol{w}, \boldsymbol{X}) = -y f_{\mathrm{wk}}(\boldsymbol{w}, \tilde{\boldsymbol{X}})$. From the symmetry of $\boldsymbol{\xi}$, it implies the model has $1/2$ error rate conditioned on the case where two signal vectors are identical. By combining two cases, we have the desired conclusion. □

The strong model is a 2-layer convolutional neural network with ReLU activation, also applying patch-wise convolution, where the second layer weights are fixed and only the first layer is trainable.

**Definition 3** (Strong Model)**.** We consider our strong model as 2-layer CNN $f_{\mathrm{st}}(\boldsymbol{W}, \cdot) : \mathbb{R}^{d \times 3} \to \mathbb{R}$ parameterized by $\boldsymbol{W} = \{\boldsymbol{W}_1, \boldsymbol{W}_{-1}\}$ where $\boldsymbol{W}_s = \{\boldsymbol{w}_{s,r}\}_{r \in [m]}$ for $s \in \{\pm 1\}$ represents the set of positive/negative filters, each containing $m$ filters $\boldsymbol{w}_{s,r} \in \mathbb{R}^d$. For each input $\boldsymbol{X} = \left( \boldsymbol{x}^{(1)}, \boldsymbol{x}^{(2)}, \boldsymbol{x}^{(3)} \right) \in \mathbb{R}^{d \times 3}$, we define

$$f_{\mathrm{st}}(\boldsymbol{W}, \boldsymbol{X}) = F_1(\boldsymbol{W}_1, \boldsymbol{X}) - F_{-1}(\boldsymbol{W}_{-1}, \boldsymbol{X}),$$

where for each $s \in \{\pm 1\}$,

$$F_s(\boldsymbol{W}_s, \boldsymbol{X}) = \frac{1}{m} \sum_{r \in [m]} \left[ \sigma \left( \left\langle \boldsymbol{w}_{s,r}, \boldsymbol{x}^{(1)} \right\rangle \right) + \sigma \left( \left\langle \boldsymbol{w}_{s,r}, \boldsymbol{x}^{(2)} \right\rangle \right) + \sigma \left( \left\langle \boldsymbol{w}_{s,r}, \boldsymbol{x}^{(3)} \right\rangle \right) \right]$$

and $\sigma(\cdot)$ denotes the ReLU activation function.

In contrast to the limitations of the weak model, the following proposition demonstrates the strong model architecture's capability for perfect generalization.

**Proposition 2.2.** *Let* $(\boldsymbol{X}, y) \sim \mathcal{D}$ *be a test example. If* $m \geq 2$, *then there exists a strong model with parameter* $\boldsymbol{W}^*$ *that achieves zero test error:* $\mathbb{P}\left[ y f_{\mathrm{st}}(\boldsymbol{W}^*, \boldsymbol{X}) < 0 \right] = 0$.

*Proof.* We construct $\boldsymbol{W}^*$ by defining, for each $s \in \{\pm 1\}$, the filters $\boldsymbol{w}^*_{s,1} = \boldsymbol{\mu}_s + \boldsymbol{\nu}_s$, $\boldsymbol{w}^*_{s,2} = \boldsymbol{\mu}_s - \boldsymbol{\nu}_s$, and setting $\boldsymbol{w}^*_{s,r} = \boldsymbol{0}$ for $r > 2$. Direct calculation shows that $y f_{\mathrm{st}}(\boldsymbol{W}^*, \boldsymbol{X}) > 0$ for all $(\boldsymbol{X}, y) \sim \mathcal{D}$, leading to zero test error. □

## 2.3 Training Scenario

Our goal is to train the weak and strong models, using a finite training set sampled from the distribution $\mathcal{D}$, to correctly classify unseen test examples from $\mathcal{D}$. We first outline the training procedure of the weak model and then describe the training of the strong model supervised by the weak model.

### 2.3.1 Weak Model Training

In weak model training, we use $n_{\mathrm{wk}}$ labeled data points $\{(\boldsymbol{X}_i, y_i)\}_{i=1}^{n_{\mathrm{wk}}} \stackrel{i.i.d.}{\sim} \mathcal{D}$ and training loss is defined as

$$L_{\mathrm{wk}}(\boldsymbol{w}) = \frac{1}{n_{\mathrm{wk}}} \sum_{i \in [n_{\mathrm{wk}}]} \ell \left( y_i f_{\mathrm{wk}}(\boldsymbol{w}, \boldsymbol{X}_i) \right),$$

where $\ell(z) = \log(1 + \exp(-z))$ is the logistic loss. We consider using gradient descent with learning rate $\eta$ to minimize training loss $L_{\mathrm{wk}}(\boldsymbol{w})$ and model parameters are initialized as $\boldsymbol{w}^{(0)} = \boldsymbol{0}$. The parameters are updated at each iteration $t$ as

$$\boldsymbol{w}^{(t+1)} = \boldsymbol{w}^{(t)} - \eta \nabla_{\boldsymbol{w}} L_{\mathrm{wk}} \left( \boldsymbol{w}^{(t)} \right)$$

$$= \boldsymbol{w}^{(t)} - \frac{\eta}{n_{\mathrm{wk}}} \sum_{i \in [n_{\mathrm{wk}}]} y_i \ell' \left( y_i f_{\mathrm{wk}} \left( \boldsymbol{w}^{(t)}, \boldsymbol{X}_i \right) \right) \left( \boldsymbol{x}_i^{(1)} + \boldsymbol{x}_i^{(2)} + \boldsymbol{x}_i^{(3)} \right)$$

$$= \boldsymbol{w}^{(t)} + \frac{\eta}{n_{\mathrm{wk}}} \sum_{i \in [n_{\mathrm{wk}}]} y_i g_i^{(t)} \left( \boldsymbol{x}_i^{(1)} + \boldsymbol{x}_i^{(2)} + \boldsymbol{x}_i^{(3)} \right) \tag{1}$$

where $\boldsymbol{X}_i = \left( \boldsymbol{x}_i^{(1)}, \boldsymbol{x}_i^{(2)}, \boldsymbol{x}_i^{(3)} \right)$ and we use $g_i^{(t)} = -\ell' \left( y_i f_{\mathrm{wk}} \left( \boldsymbol{w}^{(t)}, \boldsymbol{X}_i \right) \right)$.

### 2.3.2 Weak-to-Strong Training

Let $\{(\tilde{\boldsymbol{X}}_i, \tilde{y}_i)\}_{i=1}^{n_{\mathrm{st}}} \overset{i.i.d.}{\sim} \mathcal{D}$ denote a dataset drawn from the data distribution $\mathcal{D}$. Then the strong model is trained on the dataset $\{(\tilde{\boldsymbol{X}}_i, \hat{y}_i)\}_{i=1}^{n_{\mathrm{st}}}$, where the supervision $\hat{y}_i$ is provided by a pretrained weak model $f_{\mathrm{wk}}(\boldsymbol{w}^*, \cdot)$, i.e., $\hat{y}_i = \mathrm{sign}(f_{\mathrm{wk}}(\boldsymbol{w}^*, \tilde{\boldsymbol{X}}_i))$ instead of using true label $\tilde{y}_i$. The training objective is defined as

$$L_{\mathrm{st}}(\boldsymbol{W}) = \frac{1}{n_{\mathrm{st}}} \sum_{i \in [n_{\mathrm{st}}]} \ell\left(\hat{y}_i f_{\mathrm{st}}(\boldsymbol{W}, \tilde{\boldsymbol{X}}_i)\right)$$

and we use gradient descent with learning rate $\eta$ to minimize $L_{\mathrm{st}}(\boldsymbol{W})$, where the model parameters are initialized as $\boldsymbol{w}_{s,r}^{(0)} \sim \mathcal{N}(\boldsymbol{0}, \sigma_0^2 \boldsymbol{I}_d)$ for all $s \in \{\pm 1\}$ and $r \in [m]$. The parameters are updated at each iteration $t$ as

$$
\begin{aligned}
\boldsymbol{w}_{s,r}^{(t+1)} &= \boldsymbol{w}_{s,r}^{(t)} - \eta \nabla_{\boldsymbol{w}_{s,r}} L_{\mathrm{st}}(\boldsymbol{W}^{(t)}) \\
&= \boldsymbol{w}_{s,r}^{(t)} - \frac{s\eta}{mn_{\mathrm{st}}} \sum_{i \in [n_{\mathrm{st}}]} \hat{y}_i \, \ell'\left(\hat{y}_i f_{\mathrm{st}}(\boldsymbol{W}^{(t)}, \tilde{\boldsymbol{X}}_i)\right) \sum_{p \in [3]} \sigma'\left(\left\langle \boldsymbol{w}_{s,r}^{(t)}, \tilde{\boldsymbol{x}}_i^{(p)} \right\rangle\right) \tilde{\boldsymbol{x}}_i^{(p)} \\
&= \boldsymbol{w}_{s,r}^{(t)} + \frac{s\eta}{mn_{\mathrm{st}}} \sum_{i \in [n_{\mathrm{st}}]} \hat{y}_i \, \tilde{g}_i^{(t)} \sum_{p \in [3]} \sigma'\left(\left\langle \boldsymbol{w}_{s,r}^{(t)}, \tilde{\boldsymbol{x}}_i^{(p)} \right\rangle\right) \tilde{\boldsymbol{x}}_i^{(p)},
\end{aligned}
\tag{2}
$$

where $\tilde{\boldsymbol{X}}_i = \left(\tilde{\boldsymbol{x}}_i^{(1)}, \tilde{\boldsymbol{x}}_i^{(2)}, \tilde{\boldsymbol{x}}_i^{(3)}\right)$ and we use $\tilde{g}_i^{(t)} = -\ell'\left(\hat{y}_i f_{\mathrm{st}}(\boldsymbol{W}^{(t)}, \tilde{\boldsymbol{X}}_i)\right)$ for each $i \in [n_{\mathrm{st}}]$.

## 3 Provable Weak-to-Strong Generalization

In this section, we provide theoretical results on when and how weak-to-strong generalization occurs in our setting. For our analysis, we denote by $T^*$ the maximum admissible training iterates and we assume $T^* = \eta^{-1}\mathrm{poly}(\varepsilon^{-1}, d, n_{\mathrm{st}}, n_{\mathrm{wk}}, m)$, where $\varepsilon$ is a target training loss and $\mathrm{poly}(\cdot)$ is a sufficiently large polynomial. Furthermore, we focus on an asymptotic regime where parameters $\varepsilon^{-1}, d, n_{\mathrm{st}}, n_{\mathrm{wk}}, m$ are considered sufficiently large. Consequently, our theoretical guarantees will often be expressed using asymptotic notation such as $\mathcal{O}(\cdot), \Omega(\cdot), o(\cdot), \omega(\cdot)$, as well as $\widetilde{\mathcal{O}}(\cdot), \widetilde{\Omega}(\cdot)$, which hide logarithmic factors. Our main results depend on the conditions detailed below.

**Condition 3.1.** There exists a sufficiently large constant $C > 0$ such that the following hold:

(C1) $d \geq C \max\left\{n_{\mathrm{wk}}^2 \log\left(\frac{Cn_{\mathrm{wk}}^2}{\delta}\right), n_{\mathrm{st}} \log\left(\frac{Cn_{\mathrm{st}}}{\delta}\right)\right\}(\log T^*)^2$

(C2) $n_{\mathrm{wk}}, n_{\mathrm{st}} \geq C \max\left\{p_{\mathrm{e}}^{-2}, p_{\mathrm{b}}^{-2}, p_{\mathrm{h}}^{-2}\right\} \log\left(\frac{C}{\delta}\right), \quad m \geq C \log\left(\frac{Cn_{\mathrm{st}}}{\delta}\right)$

(C3) $\sigma_0 \leq C^{-1} \min\left\{\frac{1}{\|\boldsymbol{\mu}\|}, \frac{1}{\|\boldsymbol{\nu}\|}, \frac{1}{\sigma_p \sqrt{d}}\right\} \min\left\{\frac{n_{\mathrm{st}} p_{\mathrm{b}} \|\boldsymbol{\nu}\|^2}{\sigma_p^2 d}, \frac{\sigma_p^2 d}{(2p_{\mathrm{e}}+p_{\mathrm{b}})n_{\mathrm{st}}^1 \|\boldsymbol{\mu}\|^2}\right\} \left(\log\left(\frac{Cmn_{\mathrm{st}}}{\delta}\right)\right)^{-\frac{1}{2}}$

(C4) $\eta \leq C^{-1} \sigma_p^{-2} d^{-\frac{3}{2}}$

(C5) $(2p_{\mathrm{e}} + p_{\mathrm{b}})\|\boldsymbol{\mu}\|^2 \geq Cp_{\mathrm{b}}\|\boldsymbol{\nu}\|^2, \quad n_{\mathrm{wk}}, n_{\mathrm{st}} = \omega\left(\frac{\sigma_p^4 d}{(2p_{\mathrm{e}}+p_{\mathrm{b}})^2\|\boldsymbol{\mu}\|^4}\right)$

(C6) $p_{\mathrm{b}} \geq C \max\{p_{\mathrm{h}}, \sigma_p \|\boldsymbol{\mu}\| \|\boldsymbol{\nu}\|^{-2}(\log T^*)^{\frac{1}{2}}\}$

(C1) and (C2) allow us to apply concentration inequalities and ensure that our training data samples and initial model parameters satisfy certain desirable properties with high probability. (C3) and (C4) ensure that initialization is negligible compared to the update and that learning dynamics are stable. They facilitate our analysis of the learning dynamics. (C5) guarantees that easy signals are easier to learn than hard signals for both weak and strong models, as the difficulty of learning signals is determined by their frequency and strength. This also ensures that both models are guaranteed to learn these easy signals. Furthermore, a large enough portion of both-signal data stated in (C6) is essential to weak-to-strong generalization, in line with the insights discussed in Shin et al. (2025).

As we mentioned before, in our problem setting, there are two mechanisms for minimizing training loss: learning signals and memorizing noise. Since signals repeatedly appear in data while noise is

independent across data points, the amount of data affects which mechanism predominantly influences the learning dynamics. In our analysis, we consider two regimes based on this observation: the *data-scarce regime* and the *data-abundant regime*.

In the data-scarce regime, we demonstrate two key findings. First, we prove that weak model training can achieve performance close to the optimal limit of the weak model class even in this regime. Second, we demonstrate that even within the data-scarce regime, weak-to-strong training can achieve low test error through benign overfitting, provided that the given dataset size is not too small. We also characterize tight conditions under which the weak-to-strong training exhibits benign or harmful overfitting. In the data-abundant regime, we analyze weak-to-strong training and, perhaps surprisingly, observe that weak-to-strong generalization behaves differently compared to the data-scarce regime: Early stopping plays a crucial role.

### 3.1 Data-Scarce Regime

In this regime, the amount of available data is small. Consequently, noise memorization is more prevalent than signal learning, leading to model outputs on training data points mainly determined by activations from noise vectors. We formalize this regime as follows.

**Condition 3.2** (Data-Scarce Regime). All conditions in Condition 3.1 hold, using the same constant $C > 0$ as introduced therein, and the following additional condition holds: $n_{\mathrm{wk}}, n_{\mathrm{st}} \leq C^{-1}\sigma_p^2 d/((2p_{\mathrm{e}} + p_{\mathrm{b}})\|\boldsymbol{\mu}\|^2 \log T^*)$.

The following theorem provides convergence and test error guarantees for weak model training.

**Theorem 3.3** (Weak Model Training). *Let $\boldsymbol{w}^{(t)}$ be the iterates of weak model training. For any $\varepsilon > 0$ and $\delta \in (0, 1)$ satisfying Condition 3.2, with probability at least $1 - \delta$, there exists $T_{\mathrm{wk}} = \widetilde{\mathcal{O}}(\eta^{-1}\varepsilon^{-1}n_{\mathrm{wk}}d^{-1}\sigma_p^{-2})$ such that for all $t \in [T_{\mathrm{wk}}, T^*]$, the following statements hold:*

1. *The training loss converges below $\varepsilon$: $L_{\mathrm{wk}}\left(\boldsymbol{w}^{(t)}\right) < \varepsilon$.*

2. *Let $(\boldsymbol{X}, y) \sim \mathcal{D}$ be an unseen test example, independent of the training set $\{(\boldsymbol{X}_i, y_i)\}_{i=1}^{n_{\mathrm{wk}}}$. Then, we have*

$$\mathbb{P}\left[yf_{\mathrm{wk}}\left(\boldsymbol{w}^{(t)}, \boldsymbol{X}\right) < 0 \,\Big|\, (\boldsymbol{X}, y) \in \mathcal{S}_{\mathrm{e}} \cup \mathcal{S}_{\mathrm{b}}\right] \leq \exp\left(-\frac{n_{\mathrm{wk}}(2p_{\mathrm{e}} + p_{\mathrm{b}})^2\|\boldsymbol{\mu}\|^4}{C_1\sigma_p^4 d}\right) = o(1).$$

*Here, $C_1 > 0$ is a constant.*

The proof is provided in Appendix B. Combined with Proposition 2.1, Theorem 3.3 guarantees the convergence of training loss and shows that the trained weak model achieves low test error on easy-only data and both-signal data, while performing random guessing on unseen hard-only data. This corresponds to the near optimal error attainable by the weak model, but not perfect because the overall test error will be of order $\frac{p_{\mathrm{h}}}{2} + o(1)$.

The following theorem provides convergence and test error guarantees for weak-to-strong training.

**Theorem 3.4** (Weak-to-Strong Training, Data-Scarce Regime). *Let $\boldsymbol{W}^{(t)}$ be the iterates of weak-to-strong training, with the weak model $f_{\mathrm{wk}}(\boldsymbol{w}^*, \cdot)$ satisfying the conclusion of Theorem 3.3. For any $\varepsilon > 0$ and $\delta \in (0, 1)$ satisfying Condition 3.2, with probability at least $1 - \delta$, there exists $T_{\mathrm{w2s}} = \mathcal{O}(\eta^{-1}\varepsilon^{-1}mn_{\mathrm{st}}d^{-1}\sigma_p^{-2})$ such that for any $t \in [T_{\mathrm{w2s}}, T^*]$ the following statements hold:*

1. *The training loss converges below $\varepsilon$: $L_{\mathrm{st}}\left(\boldsymbol{W}^{(t)}\right) < \varepsilon$.*

2. *Let $(\boldsymbol{X}, y) \sim \mathcal{D}$ be an unseen test example, independent of the training set $\{(\tilde{\boldsymbol{X}}_i, \hat{y}_i)\}_{i=1}^{n_{\mathrm{st}}}$.*

   - *(Benign Overfitting) When $n_{\mathrm{st}}p_{\mathrm{b}}^2\|\boldsymbol{\nu}\|^4/(\sigma_p^4 d) \geq C_2,$[1] we have*

$$\mathbb{P}\left[yf_{\mathrm{st}}\left(\boldsymbol{W}^{(t)}, \boldsymbol{X}\right) < 0\right] \leq (p_{\mathrm{e}} + p_{\mathrm{b}})\exp\left(-\frac{n_{\mathrm{st}}(2p_{\mathrm{e}} + p_{\mathrm{b}})^2\|\boldsymbol{\mu}\|^4}{C_3\sigma_p^4 d}\right) + p_{\mathrm{h}}\exp\left(-\frac{n_{\mathrm{st}}p_{\mathrm{b}}^2\|\boldsymbol{\nu}\|^4}{C_3\sigma_p^4 d}\right).$$

   - *(Harmful Overfitting) When $n_{\mathrm{st}}p_{\mathrm{b}}^2\|\boldsymbol{\nu}\|^4/(\sigma_p^4 d) \leq C_4,$*

$$\mathbb{P}\left[yf_{\mathrm{st}}\left(\boldsymbol{W}^{(t)}, \boldsymbol{X}\right) < 0\right] \geq 0.12p_{\mathrm{h}}.$$

---

[1]We emphasize that this condition does not contradict Condition 3.2 due to (C6).

*Here, $C_2, C_3, C_4 > 0$ are constants.*

The proof is provided in Appendix C. Theorem 3.4 guarantees training loss convergence and further characterizes the overall test error in the weak-to-strong scenario. Specifically, it shows that the error is much smaller than the lower bound for the weak model's error (Proposition 2.1) when the number of data $n_{st}$ exceeds a certain threshold. Conversely, when $n_{st}$ falls below a similar threshold, the error remains lower-bounded by a constant multiple of $p_h$, as in the case of the supervising weak model's error. The fact that these two thresholds differ only by constant factors provides a tight characterization of these distinct regimes.

## 3.2 Data-Abundant Regime

In this regime, a sufficient amount of data is available, allowing signal learning to dominate the effects of noise memorization. We formalize this regime as follows.

**Condition 3.5** (Data-Abundant Regime). All conditions in Condition 3.1 hold, using the same constant $C > 0$ as introduced therein, and the following additional condition holds: $n_{st} \geq C\sigma_p^2 d \log T^* / (p_b \|\boldsymbol{\nu}\|^2)$.

Due to the limited availability of costly true-labeled data, the defining conditions for this data-abundant regime primarily focus on $n_{st}$. Thus, the characteristics of the supervising weak model, as established in Theorem 3.3, remain applicable in this regime. The following theorem demonstrates the emergence of weak-to-strong generalization in the early phase, where training loss remains large.

**Theorem 3.6** (Weak-to-Strong Training, Data-Abundant Regime). *Let $\boldsymbol{W}^{(t)}$ be the iterates of the weak-to-strong training, with the weak model $f_{wk}(\boldsymbol{w}^*, \cdot)$ satisfying the conclusion of Theorem 3.3. For any $\delta \in (0, 1)$ satisfying Condition 3.5, with probability at least $1 - \delta$, there exists early stopping time $T_{es} = \mathcal{O}(\eta^{-1} m(2p_e + p_b)^{-1} \|\boldsymbol{\mu}\|^{-2})$ such that the following statements hold:*

1. *The early stopped strong model $f_{st}\left(\boldsymbol{W}^{(T_{es})}, \cdot\right)$ perfectly fits all training data points having correct labels (i.e. $\hat{y}_i = \tilde{y}_i$) but fails on all training data points with flipped labels (i.e. $\hat{y}_i \neq \tilde{y}_i$). In other words, the model predicts the true label $\tilde{y}_i$ for any training data point $\tilde{\boldsymbol{X}}_i$.*

2. *Let $(\boldsymbol{X}, y) \sim \mathcal{D}$ be an unseen test example, independent of the training set $\{(\tilde{\boldsymbol{X}}_i, \hat{y}_i)\}_{i=1}^{n_{st}}$. We have*

$$\mathbb{P}\left[yf_{st}\left(\boldsymbol{W}^{(T_{es})}, \boldsymbol{X}\right) < 0\right] \leq (p_e + p_b) \exp\left(-\frac{n_{st}(2p_e + p_b)^2 \|\boldsymbol{\mu}\|^4}{C_5 \sigma_p^4 d}\right) + p_h \exp\left(-\frac{n_{st} p_b^2 \|\boldsymbol{\nu}\|^4}{C_5 \sigma_p^4 d}\right).$$

*Here, $C_5 > 0$ is a constant.*

The proof is provided in Appendix D. Theorem 3.6 shows that weak-to-strong generalization can arise via early stopping in this regime. It provides guarantees for an early-stopped model and thus does not provide guarantees on the model's performance at convergence. One might therefore be curious how training until convergence influences performance. We conducted experiments in our setting and observed that after this early phase, performance often degrades and then plateaus, exhibiting accuracy similar to or even lower than that of the supervising weak model. While we leave a rigorous proof for this late-phase behavior open, we provide an intuitive explanation in Section 4.

The role of early stopping for weak-to-strong generalization is also discussed in the literature. Burns et al. (2024) observe that in ChatGPT Reward Modeling tasks and a subset of NLP tasks, early stopping can improve weak-to-strong generalization, while overtraining can lead to degradation. Medvedev et al. (2025) also discuss early stopping in their theoretical setting, where it becomes essential due to their consideration of training on the population risk over the distribution of pseudo-labeled data. In contrast, in our finite-sample setting, early stopping is not strictly required to achieve weak-to-strong generalization. In fact, a strong model that perfectly fits the pseudo-labeled training data may lead to either low or high test error, as observed in the data-scarce regime. Thus, the fact that training dynamics can converge to a solution with poor generalization, even under abundant data and the existence of good solutions, is somewhat surprising.

# 4 Key Theoretical Insights

In this section, we provide key insights behind our theoretical analysis. We formally prove this intuition using several theoretical tools, such as the signal-noise decomposition (Cao et al., 2022).

For weak model training, its update rule (1) implies that the model weight vector $\boldsymbol{w}$ is updated in directions determined by the signal and noise vectors within the training samples. The evolution of $\boldsymbol{w}$ along each such vector's direction is influenced by that vector's strength and its frequency of appearance in the dataset. Due to the limited capability of the weak model, it cannot learn hard signals with opposite signs (e.g., $\boldsymbol{\nu}_1, -\boldsymbol{\nu}_1$). Furthermore, the cancellation of updates along hard signal directions and our condition (C5) ensure that the learning of easy signals predominates over that of hard signals. This dominance means that while easy signals are effectively learned, the learning of hard signals is largely suppressed. Consequently, in both-signal data, the contribution from the poorly learned hard signal component is not large enough to disrupt the classification guided by the well-learned easy signals. Therefore, the weak model can correctly predict not only easy-only data but also both-signal data.

We now explain how the supervision from the pretrained weak model affects the learning dynamics of weak-to-strong training. Let us first introduce some notation. For each $i \in [n_{\mathrm{st}}]$, we denote by $\tilde{\boldsymbol{v}}_i^{(1)}, \tilde{\boldsymbol{v}}_i^{(2)}$, and $\tilde{\boldsymbol{\xi}}_i$ the signal vectors and noise vector of the $i$-th input $\tilde{\boldsymbol{X}}_i$, respectively. For each $\boldsymbol{v} \in \{\boldsymbol{\mu}_1, \boldsymbol{\mu}_{-1}, \pm\boldsymbol{\nu}_1, \pm\boldsymbol{\nu}_{-1}\}$ and $l \in [2]$, we define $\mathcal{C}_{\boldsymbol{v}}^{(l)}$ and $\mathcal{F}_{\boldsymbol{v}}^{(l)}$ as the sets of indices $i \in [n_{\mathrm{st}}]$ such that $\tilde{\boldsymbol{v}}_i^{(l)} = \boldsymbol{v}$ and the supervision corresponds to the clean label (i.e., $\hat{y}_i = \tilde{y}_i$) or the flipped label (i.e., $\hat{y}_i = -\tilde{y}_i$), respectively. Lastly, recall that $\tilde{g}_i^{(t)} = -\ell'(\hat{y}_i f_{\mathrm{st}}(\boldsymbol{W}^{(t)}, \tilde{\boldsymbol{X}}_i))$ denotes the negative of the loss derivative for $i$-th sample.

Update rule for weak-to-strong training (2) implies that for any $s \in \{\pm 1\}$ and $r \in [m]$,

$$\left\langle \boldsymbol{w}_{s,r}^{(t+1)}, \boldsymbol{\mu}_s \right\rangle = \left\langle \boldsymbol{w}_{s,r}^{(t)}, \boldsymbol{\mu}_s \right\rangle + \frac{\eta}{mn_{\mathrm{st}}} \sum_{l \in [2]} \left( \sum_{i \in \mathcal{C}_{\boldsymbol{\mu}_s}^{(l)}} \tilde{g}_i^{(t)} - \sum_{i \in \mathcal{F}_{\boldsymbol{\mu}_s}^{(l)}} \tilde{g}_i^{(t)} \right) \|\boldsymbol{\mu}\|^2 \, \mathbb{1} \left[ \left\langle \boldsymbol{w}_{s,r}^{(t)}, \boldsymbol{\mu}_s \right\rangle > 0 \right].$$

Since the supervising weak model achieves low test error on easy-only and both-signal data, the pseudo-labels for training samples involving $\boldsymbol{\mu}_s$ have a low flipping probability, and this implies $|\mathcal{F}_{\boldsymbol{\mu}_s}^{(l)}|/n_{\mathrm{st}} \approx 0$. This ensures that, in both data-scarce and data-abundant regimes, $\langle \boldsymbol{w}_{s,r}^{(t)}, \boldsymbol{\mu}_s \rangle$ increases if it is positive.

Similarly, an update for learning hard signals can be written as follows:

$$\left\langle \boldsymbol{w}_{s,r}^{(t+1)}, \boldsymbol{\nu}_s \right\rangle = \left\langle \boldsymbol{w}_{s,r}^{(t)}, \boldsymbol{\nu}_s \right\rangle + \frac{\eta}{mn_{\mathrm{st}}} \sum_{l \in [2]} \left( \sum_{i \in \mathcal{C}_{\boldsymbol{\nu}_s}^{(l)}} \tilde{g}_i^{(t)} - \sum_{i \in \mathcal{F}_{\boldsymbol{\nu}_s}^{(l)}} \tilde{g}_i^{(t)} \right) \|\boldsymbol{\nu}\|^2 \, \mathbb{1} \left[ \left\langle \boldsymbol{w}_{s,r}^{(t)}, \boldsymbol{\nu}_s \right\rangle > 0 \right]$$
$$- \frac{\eta}{mn_{\mathrm{st}}} \sum_{l \in [2]} \left( \sum_{i \in \mathcal{C}_{-\boldsymbol{\nu}_s}^{(l)}} \tilde{g}_i^{(t)} - \sum_{i \in \mathcal{F}_{-\boldsymbol{\nu}_s}^{(l)}} \tilde{g}_i^{(t)} \right) \|\boldsymbol{\nu}\|^2 \, \mathbb{1} \left[ \left\langle \boldsymbol{w}_{s,r}^{(t)}, \boldsymbol{\nu}_s \right\rangle < 0 \right].$$

However, weak-to-strong generalization exhibits different behaviors across the two regimes, influenced by the presence of a non-negligible fraction of data containing hard signals with flipped pseudo-labels. In the data-scarce regime, noise memorization is a dominant component of the learning process. This can lead to the learning effort being more balanced across different data points. A sufficient fraction of both-signal data guarantees $|\mathcal{C}_{\boldsymbol{\nu}_s}^{(l)}|, |\mathcal{C}_{-\boldsymbol{\nu}_s}^{(l)}| \gg |\mathcal{F}_{\boldsymbol{\nu}_s}^{(l)}|, |\mathcal{F}_{-\boldsymbol{\nu}_s}^{(l)}|$ and this indicates that $\langle \boldsymbol{w}_{s,r}^{(t+1)}, \boldsymbol{\nu}_s \rangle > \langle \boldsymbol{w}_{s,r}^{(t)}, \boldsymbol{\nu}_s \rangle$ if $\langle \boldsymbol{w}_{s,r}^{(t)}, \boldsymbol{\nu}_s \rangle > 0$ and $\langle \boldsymbol{w}_{s,r}^{(t+1)}, \boldsymbol{\nu}_s \rangle < \langle \boldsymbol{w}_{s,r}^{(t)}, \boldsymbol{\nu}_s \rangle$ if $\langle \boldsymbol{w}_{s,r}^{(t)}, \boldsymbol{\nu}_s \rangle < 0$. Therefore, the strong model can learn hard signals with opposite signs $\boldsymbol{\nu}_s$ and $-\boldsymbol{\nu}_s$, simultaneously, by utilizing different sets of filters $\{r \in [m] : \langle \boldsymbol{w}_{s,r}^{(0)}, \boldsymbol{\nu}_s \rangle > 0\}$ and $\{r \in [m] : \langle \boldsymbol{w}_{s,r}^{(0)}, \boldsymbol{\nu}_s \rangle < 0\}$.

In contrast to the data-scarce regime, in the early phase of the data-abundant regime, the strong model can learn both easy and hard signals quickly (even faster than noise is memorized) due to the significant abundance of signal vectors from the clean-labeled training data. This leads to almost perfect generalization on unseen data. Let us describe our intuition for why overtraining can lead to performance degradation. Rapid learning of signals also creates a growing discrepancy in the negative loss derivatives $\tilde{g}_i^{(t)}$'s between clean-label data and flipped-label data. The non-negligible portion of flipped-label hard-only data combined with the imbalance in loss derivatives can lead to the

contributions from these flipped-label data points (e.g., $\sum_{i \in \mathcal{F}_{\nu_s}^{(l)}} \tilde{g}_i^{(t)}$) predominating over those from clean-labeled data points (e.g., $\sum_{i \in \mathcal{C}_{\nu_s}^{(l)}} \tilde{g}_i^{(t)}$). Consequently, the strong model may start "forgetting" learned signals as it continues to minimize the training loss defined by these pseudo-labels.

**Practical Insights.** Our analysis reveals the following mechanism for weak-to-strong generalization: the weak model first successfully labels data containing easy-to-learn information. This data includes a subset that contains both easy information and harder-to-learn information (which the weak model fails to capture). The strong model then utilizes this correctly labeled subset to successfully learn the harder-to-learn information. We believe that this insight can be applied to practical scenarios, potentially leading to the development of data selection techniques that preferentially select such beneficial data for better weak-to-strong generalization.

## 5 Experiments

We conduct experiments to support our findings, using NVIDIA RTX A6000 GPUs.

### 5.1 Experiments on Our Theoretical Setting

We perform experiments in our setting described in Section 2. We set the dimension $d = 2000$ and the signal vectors $\boldsymbol{\mu}_1, \boldsymbol{\mu}_{-1}, \boldsymbol{\nu}_1, \boldsymbol{\nu}_{-1}$ are constructed from randomly generated orthonormal vectors, which are subsequently scaled so that their respective norms are $\|\boldsymbol{\mu}\| = 0.4$ and $\|\boldsymbol{\nu}\| = 0.35$. The noise strength is $\sigma_p = 0.1$ and the data type probabilities are $p_\mathrm{e} = 0.4$ and $p_\mathrm{h} = p_\mathrm{b} = 0.3$.

We first train the weak model using $n_\mathrm{wk} = 5000$ true-labeled data points. The training is conducted for 1000 epochs using stochastic gradient descent with batch size 256 and learning rate $\eta = 0.1$, which results in a test accuracy of $0.851$. For weak-to-strong training, we use the strong model with $m = 50$ filters and an initialization scale $\sigma_0 = 0.01$. We train the strong model using stochastic gradient descent with batch size 256 and learning rate $\eta = 0.1$ on the dataset labeled by the pretrained weak model. We use three different values for the number of data points, $n_\mathrm{st} = 75, 2000, 20000$.

Figure 1 provides the training and test accuracy for weak-to-strong training with three different training dataset sizes. We train the strong model for 2000 training epochs when $n_\mathrm{st} = 75$ or $n_\mathrm{st} = 2000$, and for 10000 epochs when $n_\mathrm{st} = 20000$, as this requires more iterations for convergence compared to the other cases. We observe three different types of results revealed in our analysis.

The cases $n_\mathrm{st} = 75$ and $n_\mathrm{st} = 2000$ support our analysis in the data-scarce regime. In both cases, the training accuracy initially increases faster than the test accuracy. However, their final test accuracies differ. In the case of $n_\mathrm{st} = 75$, the strong model achieves perfect training accuracy, while its test accuracy remains close to that of the supervising weak model. This aligns with our findings on the failure of weak-to-strong generalization due to harmful overfitting. In contrast, for $n_\mathrm{st} = 2000$, the increased amount of data allows the test accuracy to sufficiently increase, eventually exceeding the weak model's test accuracy. This aligns with our findings on the emergence of weak-to-strong generalization via benign overfitting.

The case of $n_\mathrm{st} = 20000$ corresponds to the data-abundant regime in our analysis. Unlike the prior two cases, test accuracy grows faster than training accuracy and achieves near-perfect accuracy, while training accuracy remains comparable to that of the weak model; this aligns with Theorem 3.6. We also observe that continued training deteriorates test accuracy, while training accuracy increases.

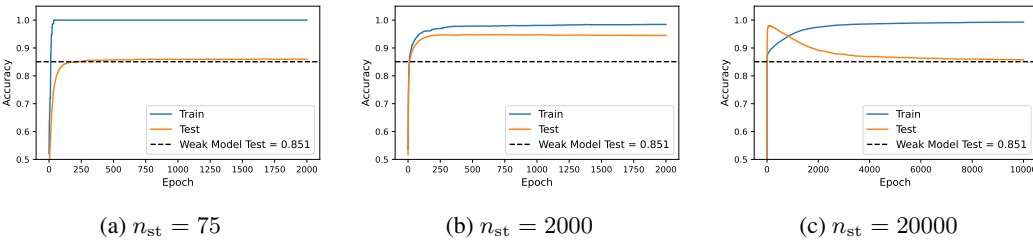

    (a) $n_\mathrm{st} = 75$                 (b) $n_\mathrm{st} = 2000$                 (c) $n_\mathrm{st} = 20000$

Figure 1: Weak-to-strong training with varying training dataset sizes ($n_\mathrm{st}$). These align with our theoretical findings: (a) harmful overfitting for $n_\mathrm{st} = 75$; (b) benign overfitting for $n_\mathrm{st} = 2000$; and (c) for $n_\mathrm{st} = 20000$, an early emergence of generalization and degradation with overtraining.

## 5.2 Experiments on MNIST

We also provide empirical results using a real-world dataset MNIST. Since it is hard to clearly delineate signal and noise in real data, we modify the MNIST dataset to emphasize their roles.

First, we multiply each pixel in the original images of digits 4, 5, 6, 7, 8, and 9 by 0.02, while keeping the images of other digits unchanged. This corresponds to the presence of hard signals, with digits 4–9 serving as the hard signals. To emphasize the role of noise, we replace the border region of each 28×28 image—a 5-pixel-wide frame along the edges—with standard Gaussian noise. This results in images where the central 18×18 region contains the digit, surrounded by Gaussian noise. Finally, we randomly concatenate two such modified images that share the same parity (i.e., both even or both odd), producing 28×56 images. We assign binary labels based on their parity.

Figure 2 provides examples of the modified data. The resulting data includes a variety of signal types: some pairs contain two bright digits, others contain one bright and one dark digit, and some consist of two dark digits. These types serve as easy-only data, both-signal data, hard-only data in our setting.

For the weak model, we use an MLP consisting of a single hidden layer with 128 units followed by a ReLU activation. For the strong model, we use a CNN with three convolutional layers of increasing channels (64, 128, 256), each followed by batch normalization, ReLU, and max pooling. The extracted features are then flattened and passed through a fully connected layer with 512 units. We first train the weak model using 500 samples. Then, we train the strong model using labels predicted by the trained weak model, with varying numbers of training samples $n_{st} = 500, 1000, 1500, 2000, 2500$. We train each model for 300 epochs using the full-batch Adam optimizer with default parameters.

In Table 1, we observe a trend in which the weak-to-strong gain increases with $n_{st}$ and then decreases. These observations are consistent with our theoretical findings, which describe a transition from harmful overfitting to benign overfitting, and eventually to the data-abundant regime.

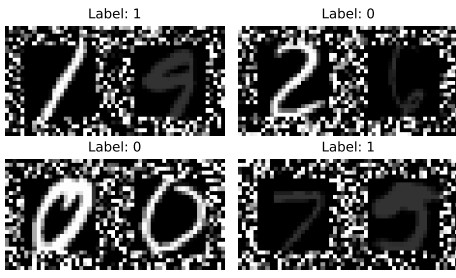

Figure 2: Examples of the modified MNIST.

Table 1: Test accuracy (%) for the weak model and the resulting weak-to-strong model. Results are calculated as the mean and standard deviation over five independent runs.

| $n_{st}$ | Weak Model | Weak-to-Strong |
|---|---|---|
| 500 | 86.06 (0.45) | 84.62 (3.95) |
| 1000 | 85.93 (0.13) | 88.34 (1.28) |
| 1500 | 86.42 (1.50) | 88.46 (2.85) |
| 2000 | 86.69 (0.52) | 87.44 (2.73) |
| 2500 | 85.86 (1.39) | 86.55 (1.22) |

## 6 Conclusion

We theoretically investigated weak-to-strong generalization by analyzing the training dynamics of a two-layer ReLU CNN under supervision from a pre-trained linear CNN on patch-wise data containing both signals and noise. Interestingly, our results reveal that weak-to-strong training exhibits distinct behaviors across different data regimes. In the data-scarce regime, we prove that weak-to-strong training converges and that generalization can emerge via benign overfitting when data availability is not extremely limited. Furthermore, we characterize the conditions leading to a transition from this benign overfitting to harmful overfitting. In the data-abundant regime, we show that weak-to-strong generalization arises in an early phase of training, and we observe that overtraining leads to performance degradation. We hope our theoretical approaches provide valuable insights.

**Limitation and Future Work.** Our work has some limitations regarding the simplified data distribution and model architectures used for theoretical analysis. Extending our analysis to more complex data or models could be a future direction. Also, it would be interesting to analyze methods for improving weak-to-strong generalization (e.g., auxiliary confidence loss (Burns et al., 2024)) in our theoretical framework. Lastly, developing techniques for better weak-to-strong generalization based on our theoretical insights is an important future direction.

## Acknowledgement

This work was supported by a National Research Foundation of Korea (NRF) grant funded by the Korean government (MSIT) (No. RS-2024-00421203) and the InnoCORE program of the Ministry of Science and ICT (No. N10250156).

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

# Contents

# A Proof Preliminaries

We use the following notation for the proof.

**Notation.** We define $\mathrm{SNR}_{\boldsymbol{\mu}} = \|\boldsymbol{\mu}\| / (\sigma_p \sqrt{d}), \mathrm{SNR}_{\boldsymbol{\nu}} = \|\boldsymbol{\nu}\| / (\sigma_p \sqrt{d})$. Let $S$ be the orthogonal complement of the span of the signal vectors $\{\boldsymbol{\mu}_1, \boldsymbol{\mu}_{-1}, \boldsymbol{\nu}_1, \boldsymbol{\nu}_{-1}\}$. We denote an orthonormal basis for $S$ by $\{\boldsymbol{b}_1, \dots, \boldsymbol{b}_{d-4}\}$. For any vector $\boldsymbol{v} \in \mathbb{R}^d$, $\Pi_S \boldsymbol{v}$ represents the projection of $\boldsymbol{v}$ onto $S$.

## A.1 Proof Preliminaries for Weak Model Training

In this subsection, we sequentially introduce signal-noise decomposition (Cao et al., 2022; Kou et al., 2023) in our setting, high-probability properties of data sampling, and quantitative bounds frequently used throughout the proof for weak model training.

We use the following notation for the analysis of weak model training.

**Notation.** For each $i \in [n_{\mathrm{wk}}]$, we denote by $\boldsymbol{v}_i^{(1)}$, $\boldsymbol{v}_i^{(2)}$, and $\boldsymbol{\xi}_i$ the signal vectors and noise vector of the $i$-th input $\boldsymbol{X}_i$, respectively. For each $\boldsymbol{v} \in \{\boldsymbol{\mu}_1, \boldsymbol{\mu}_{-1}, \pm\boldsymbol{\nu}_1, \pm\boldsymbol{\nu}_{-1}\}$, we define $\mathcal{S}_{\boldsymbol{v}}^{(1)}$ and $\mathcal{S}_{\boldsymbol{v}}^{(2)}$ as the sets of indices $i \in [n_{\mathrm{wk}}]$ such that $\boldsymbol{v}_i^{(1)} = \boldsymbol{v}$ and $\boldsymbol{v}_i^{(2)} = \boldsymbol{v}$, respectively.

### A.1.1 Signal-Noise Decomposition

**Lemma A.1.** *For any iteration $t \geq 0$, we can write $\boldsymbol{w}^{(t)}$ as*

$$\boldsymbol{w}^{(t)} = M_1^{(t)} \frac{\boldsymbol{\mu}_1}{\|\boldsymbol{\mu}\|^2} - M_{-1}^{(t)} \frac{\boldsymbol{\mu}_{-1}}{\|\boldsymbol{\mu}\|^2} + N_1^{(t)} \frac{\boldsymbol{\nu}_1}{\|\boldsymbol{\nu}\|^2} - N_{-1}^{(t)} \frac{\boldsymbol{\nu}_{-1}}{\|\boldsymbol{\nu}\|^2} + \sum_{i \in [n_{\mathrm{wk}}]} y_i \rho_i^{(t)} \frac{\boldsymbol{\xi}_i}{\|\boldsymbol{\xi}_i\|^2},$$

*where $M_s^{(t)}, N_s^{(t)}, \rho_i^{(t)}$ are recursively defined as*

$$M_s^{(t+1)} = M_s^{(t)} + \frac{\eta}{n_{\mathrm{wk}}} \left( \sum_{i \in \mathcal{S}_{\boldsymbol{\mu}_s}^{(1)}} g_i^{(t)} + \sum_{i \in \mathcal{S}_{\boldsymbol{\mu}_s}^{(2)}} g_i^{(t)} \right) \|\boldsymbol{\mu}\|^2,$$

$$N_s^{(t+1)} = N_s^{(t)} + \frac{\eta}{n_{\mathrm{wk}}} \left( \sum_{i \in \mathcal{S}_{\boldsymbol{\nu}_s}^{(1)}} g_i^{(t)} + \sum_{i \in \mathcal{S}_{\boldsymbol{\nu}_s}^{(2)}} g_i^{(t)} - \sum_{i \in \mathcal{S}_{-\boldsymbol{\nu}_s}^{(1)}} g_i^{(t)} - \sum_{i \in \mathcal{S}_{-\boldsymbol{\nu}_s}^{(2)}} g_i^{(t)} \right) \|\boldsymbol{\nu}\|^2,$$

$$\rho_i^{(t+1)} = \rho_i^{(t)} + \frac{\eta}{n_{\mathrm{wk}}} g_i^{(t)} \|\boldsymbol{\xi}_i\|^2,$$

*starting from $M_s^{(0)} = N_s^{(0)} = \rho_i^{(0)} = 0$. It follows that $M_s^{(t)}$ and $\rho_i^{(t)}$ are increasing in iteration $t$.*

*Proof of Lemma A.1.* It is trivial for the case $t = 0$. Suppose that it holds at iteration $\tau$. From the update rule, we have

$$\boldsymbol{w}^{(\tau+1)} = \boldsymbol{w}^{(\tau)} + \frac{\eta}{n_{\mathrm{wk}}} \sum_{i \in [n_{\mathrm{wk}}]} y_i g_i^{(\tau)} \sum_{p \in [3]} \boldsymbol{x}_i^{(p)}$$

$$= M_1^{(\tau)} \frac{\boldsymbol{\mu}_1}{\|\boldsymbol{\mu}\|^2} - M_{-1}^{(\tau)} \frac{\boldsymbol{\mu}_{-1}}{\|\boldsymbol{\mu}\|^2} + N_1^{(\tau)} \frac{\boldsymbol{\nu}_1}{\|\boldsymbol{\nu}\|^2} - N_{-1}^{(\tau)} \frac{\boldsymbol{\nu}_{-1}}{\|\boldsymbol{\nu}\|^2} + \sum_{i \in [n_{\mathrm{wk}}]} y_i \rho_i^{(\tau)} \frac{\boldsymbol{\xi}_i}{\|\boldsymbol{\xi}_i\|^2}$$

$$+ \frac{\eta}{n_{\mathrm{wk}}} \sum_{i \in [n_{\mathrm{wk}}]} y_i g_i^{(\tau)} \sum_{p \in [3]} \boldsymbol{x}_i^{(p)}.$$

Here $\boldsymbol{x}_i^{(p)}$'s are one of $\boldsymbol{\mu}_1, \boldsymbol{\mu}_{-1}, \boldsymbol{\nu}_1, \boldsymbol{\nu}_{-1}$, and $\boldsymbol{\xi}_i$. By grouping the terms accordingly, we obtain

$$\boldsymbol{w}^{(\tau+1)} = M_1^{(\tau+1)} \frac{\boldsymbol{\mu}_1}{\|\boldsymbol{\mu}\|^2} - M_{-1}^{(\tau+1)} \frac{\boldsymbol{\mu}_{-1}}{\|\boldsymbol{\mu}\|^2} + N_1^{(\tau+1)} \frac{\boldsymbol{\nu}_1}{\|\boldsymbol{\nu}\|^2} - N_{-1}^{(\tau+1)} \frac{\boldsymbol{\nu}_{-1}}{\|\boldsymbol{\nu}\|^2} + \sum_{i \in [n_{\mathrm{wk}}]} y_i \rho_i^{(\tau+1)} \frac{\boldsymbol{\xi}_i}{\|\boldsymbol{\xi}_i\|^2},$$

with

$$M_s^{(\tau+1)} = M_s^{(\tau)} + \frac{\eta}{n_{\mathrm{wk}}} \left( \sum_{i \in \mathcal{S}_{\boldsymbol{\mu}_s}^{(1)}} g_i^{(\tau)} + \sum_{i \in \mathcal{S}_{\boldsymbol{\mu}_s}^{(2)}} g_i^{(\tau)} \right) \|\boldsymbol{\mu}\|^2,$$

$$N_s^{(\tau+1)} = N_s^{(\tau)} + \frac{\eta}{n_{\mathrm{wk}}} \left( \sum_{i \in \mathcal{S}_{\boldsymbol{\nu}_s}^{(1)}} g_i^{(\tau)} + \sum_{i \in \mathcal{S}_{\boldsymbol{\nu}_s}^{(2)}} g_i^{(\tau)} - \sum_{i \in \mathcal{S}_{-\boldsymbol{\nu}_s}^{(1)}} g_i^{(\tau)} - \sum_{i \in \mathcal{S}_{-\boldsymbol{\nu}_s}^{(2)}} g_i^{(\tau)} \right) \|\boldsymbol{\nu}\|^2,$$

$$\rho_i^{(\tau+1)} = \rho_i^{(\tau)} + \frac{\eta}{n_{\mathrm{wk}}} g_i^{(\tau)} \|\boldsymbol{\xi}_i\|^2.$$

Hence, we have desired conclusion. $\qquad\square$

### A.1.2 Properties of Data Sampling

We establish concentration results for the data sampling.

**Lemma A.2.** *Let $E_{\mathrm{wk}}$ denote the event in which all the following hold for some large enough universal constant $C_{\mathrm{wk}} > 0$:*

*1. For each $s \in \{\pm1\}$ and $l \in [2]$, we have*

$$\left| \left| \mathcal{S}_{\boldsymbol{\mu}_s}^{(l)} \right| - \left( \frac{p_{\mathrm{e}}}{2} + \frac{p_{\mathrm{b}}}{4} \right) n_{\mathrm{wk}} \right|, \left| \left| \mathcal{S}_{\pm\boldsymbol{\nu}_s}^{(l)} \right| - \left( \frac{p_{\mathrm{h}}}{4} + \frac{p_{\mathrm{b}}}{8} \right) n_{\mathrm{wk}} \right| \le \sqrt{\frac{n_{\mathrm{wk}}}{2} \log\left( \frac{C_{\mathrm{wk}}}{\delta} \right)}.$$

*2. For any $i \in [n_{\mathrm{wk}}]$,*

$$\left| \|\boldsymbol{\xi}_i\|^2 - \sigma_p^2(d-4) \right| \le C_{\mathrm{wk}} \sigma_p^2 d^{\frac{1}{2}} \sqrt{\log\left( \frac{C_{\mathrm{wk}} n_{\mathrm{wk}}}{\delta} \right)}.$$

*3. For any $i, j \in [n_{\mathrm{wk}}]$ with $i \ne j$,*

$$|\langle \boldsymbol{\xi}_i, \boldsymbol{\xi}_j \rangle| \le C_{\mathrm{wk}} \sigma_p^2 d^{\frac{1}{2}} \sqrt{\log\left( \frac{C_{\mathrm{wk}} n_{\mathrm{wk}}^2}{\delta} \right)}.$$

*Then, the event $E_{\mathrm{wk}}$ occurs with probability at least $1 - \delta$.*

*Proof of Lemma A.2.* For each $s \in \{\pm1\}, l \in [2]$, and $i \in [n_{\mathrm{wk}}]$,

$$\mathbb{P}\left[ \boldsymbol{v}_i^{(l)} = \boldsymbol{\mu}_s \right] = \frac{p_{\mathrm{e}}}{2} + \frac{p_{\mathrm{b}}}{4}, \quad \mathbb{P}\left[ \boldsymbol{v}_i^{(l)} = \boldsymbol{\nu}_s \right] = \mathbb{P}\left[ \boldsymbol{v}_i^{(l)} = -\boldsymbol{\nu}_s \right] = \frac{p_{\mathrm{h}}}{4} + \frac{p_{\mathrm{b}}}{8}.$$

Hence, by Höeffding's inequality, we have

$$\mathbb{P}\left[ \left| \left| \mathcal{S}_{\boldsymbol{\mu}_s}^{(l)} \right| - \left( \frac{p_{\mathrm{e}}}{2} + \frac{p_{\mathrm{b}}}{4} \right) n_{\mathrm{wk}} \right| \ge \sqrt{\frac{n_{\mathrm{wk}}}{2} \log\left( \frac{C_{\mathrm{wk}}}{\delta} \right)} \right] \le \frac{2\delta}{C_{\mathrm{wk}}}$$

and

$$\mathbb{P}\left[ \left| \left| \mathcal{S}_{\pm\boldsymbol{\nu}_s}^{(l)} \right| - \left( \frac{p_{\mathrm{h}}}{4} + \frac{p_{\mathrm{b}}}{8} \right) n_{\mathrm{wk}} \right| \ge \sqrt{\frac{n_{\mathrm{wk}}}{2} \log\left( \frac{C_{\mathrm{wk}}}{\delta} \right)} \right] \le \frac{2\delta}{C_{\mathrm{wk}}}.$$

Note that for each $i \in [n_{\mathrm{wk}}]$, we can write $\boldsymbol{\xi}_i$ as

$$\boldsymbol{\xi}_i = \sigma_p \sum_{h \in [d-4]} \mathbf{z}_{i,h} \boldsymbol{b}_h,$$

where $\mathbf{z}_{i,h} \overset{i.i.d.}{\sim} \mathcal{N}(0,1)$. The sub-gaussian norm of standard normal distribution $\mathcal{N}(0,1)$ is $\sqrt{\frac{8}{3}}$ and then $(\mathbf{z}_{i,h})^2 - 1$'s are mean zero sub-exponential random variables with sub-exponential norm $\frac{8}{3}$

(Lemma 2.7.6 in Vershynin (2018)). In addition, $\mathbf{z}_{i,h}\mathbf{z}_{j,h}$'s with $i \neq j$ are mean zero sub-exponential random variables with sub-exponential norm less than or equal to $\frac{8}{3}$ (Lemma 2.7.7 in Vershynin (2018)). We use Bernstein's inequality (Theorem 2.8.1 in Vershynin (2018)), with $c$ being the absolute constant stated therein. We then have the following:

$$
\mathbb{P}\left[\left|\|\boldsymbol{\xi}_i\|^2 - \sigma_p^2(d-4)\right| \geq C_{\mathrm{wk}}\sigma_p^2 d^{\frac{1}{2}}\sqrt{\log\left(\frac{C_{\mathrm{wk}}n_{\mathrm{wk}}}{\delta}\right)}\right]
$$

$$
= \mathbb{P}\left[\left|\sum_{h\in[d-4]}\left((\mathbf{z}_{i,h})^2 - 1\right)\right| \geq C_{\mathrm{wk}}d^{\frac{1}{2}}\sqrt{\log\left(\frac{C_{\mathrm{wk}}n_{\mathrm{wk}}}{\delta}\right)}\right]
$$

$$
\leq 2\exp\left(-\frac{9cC_{\mathrm{wk}}^2 d}{64(d-4)}\log\left(\frac{C_{\mathrm{wk}}n_{\mathrm{wk}}}{\delta}\right)\right)
$$

$$
\leq 2\exp\left(-\log\left(\frac{C_{\mathrm{wk}}n_{\mathrm{wk}}}{\delta}\right)\right) \leq \frac{2\delta}{C_{\mathrm{wk}}n_{\mathrm{wk}}}.
$$

In addition, for $i,j \in [n_{\mathrm{wk}}]$ with $i \neq j$, we have

$$
\mathbb{P}\left[|\langle\boldsymbol{\xi}_i,\boldsymbol{\xi}_j\rangle| \geq C_{\mathrm{wk}}\sigma_p^2 d^{\frac{1}{2}}\sqrt{\log\left(\frac{C_{\mathrm{wk}}n_{\mathrm{wk}}^2}{\delta}\right)}\right]
$$

$$
= \mathbb{P}\left[\left|\sum_{h\in[d-4]}\mathbf{z}_{i,h}\mathbf{z}_{j,h}\right| \geq C_{\mathrm{wk}}d^{\frac{1}{2}}\sqrt{\log\left(\frac{C_{\mathrm{wk}}n_{\mathrm{wk}}^2}{\delta}\right)}\right]
$$

$$
\leq 2\exp\left(-\frac{9cC_{\mathrm{wk}}^2 d}{64(d-4)}\log\left(\frac{C_{\mathrm{wk}}n_{\mathrm{wk}}^2}{\delta}\right)\right) \leq \frac{2\delta}{C_{\mathrm{wk}}n_{\mathrm{wk}}^2}.
$$

From union bound and a large choice of universal constant $C_{\mathrm{wk}} > 0$, we conclude that the event $E_{\mathrm{wk}}$ occurs with probability at least $1 - \delta$. $\qquad\square$

### A.1.3 Properties Used Throughout the Proof

We introduce some notation and properties that are frequently used throughout the proof.

Let us define

$$
\beta_{\mathrm{wk}} := 4C_{\mathrm{wk}}n_{\mathrm{wk}}\sqrt{\frac{1}{d}\log\left(\frac{C_{\mathrm{wk}}n_{\mathrm{wk}}}{\delta}\right)}, \quad \gamma_{\mathrm{wk}} = \sqrt{\frac{1}{2n_{\mathrm{wk}}}\log\left(\frac{C_{\mathrm{wk}}}{\delta}\right)}, \quad \kappa_{\mathrm{wk}} = \frac{1}{2}.
$$

Under Condition 3.2 and the event $E_{\mathrm{wk}}$, the following hold:

- By combining (C1) and (C5), applying (C2), and from Condition 3.2, $\beta_{\mathrm{wk}}$ and $\gamma_{\mathrm{wk}}$ satisfy the following:

$$
\beta_{\mathrm{wk}} \leq \frac{\kappa_{\mathrm{wk}}}{256\log T^*}, \quad \gamma_{\mathrm{wk}} \leq \frac{\min\{p_{\mathrm{e}},p_{\mathrm{h}},p_{\mathrm{b}}\}}{8}. \tag{3}
$$

- From (C1), the following holds for any $i,j \in [n_{\mathrm{wk}}]$ with $i \neq j$:

$$
\frac{\sigma_p^2 d}{2} \leq \|\boldsymbol{\xi}_i\|^2 \leq \frac{3\sigma_p^2 d}{2}, \quad \frac{|\langle\boldsymbol{\xi}_i,\boldsymbol{\xi}_j\rangle|}{\|\boldsymbol{\xi}_i\|^2} \leq \frac{\beta_{\mathrm{wk}}}{n_{\mathrm{wk}}}, \quad \left|1 - \frac{\|\boldsymbol{\xi}_j\|^2}{\|\boldsymbol{\xi}_i\|^2}\right| \leq \frac{\beta_{\mathrm{wk}}}{n_{\mathrm{wk}}}. \tag{4}
$$

- For any $s,l \in \{\pm 1\}$, we have

$$
\left|\left|\mathcal{S}_{\boldsymbol{\mu}_s}^{(l)}\right| - \left(\frac{p_{\mathrm{e}}}{2} + \frac{p_{\mathrm{b}}}{4}\right)n_{\mathrm{wk}}\right|, \left|\left|\mathcal{S}_{\pm\boldsymbol{\nu}_s}^{(l)}\right| - \left(\frac{p_{\mathrm{h}}}{4} + \frac{p_{\mathrm{b}}}{8}\right)n_{\mathrm{wk}}\right| \leq n_{\mathrm{wk}}\gamma_{\mathrm{wk}}. \tag{5}
$$

## A.2 Proof Preliminaries for Weak-to-Strong Training

In this subsection, we sequentially introduce signal-noise decomposition (Cao et al., 2022; Kou et al., 2023) in our setting, high-probability properties of data sampling, quantitative bounds frequently used throughout the proof, and a technical lemma (Meng et al., 2024) for the analysis of weak-to-strong training.

We use the following notation for the analysis of weak-to-strong training.

**Notation.** For each $i \in [n_{\mathrm{st}}]$, we denote by $\tilde{\boldsymbol{v}}_i^{(1)}$, $\tilde{\boldsymbol{v}}_i^{(2)}$, and $\tilde{\boldsymbol{\xi}}_i$ the signal vectors and noise vector of the $i$-th input $\tilde{\boldsymbol{X}}_i$, respectively. For each $\boldsymbol{v} \in \{\boldsymbol{\mu}_1, \boldsymbol{\mu}_{-1}, \pm\boldsymbol{\nu}_1, \pm\boldsymbol{\nu}_{-1}\}$ and $l \in [2]$, we define $\mathcal{C}_{\boldsymbol{v}}^{(l)}$ and $\mathcal{F}_{\boldsymbol{v}}^{(l)}$ as the sets of indices $i \in [n_{\mathrm{st}}]$ such that $\tilde{\boldsymbol{v}}_i^{(l)} = \boldsymbol{v}$ and the supervision corresponds to the clean label (i.e., $\hat{y}_i = \tilde{y}_i$) or the flipped label (i.e., $\hat{y}_i = -\tilde{y}_i$), respectively.

### A.2.1 Signal-Noise Decomposition

**Lemma A.3.** *For any iteration $t \geq 0$, we can write each weights $\boldsymbol{w}_{s,r}^{(t)}$ with $s \in \{\pm 1\}, r \in [m]$ as*

$$
\boldsymbol{w}_{s,r}^{(t)} = \boldsymbol{w}_{s,r}^{(0)} + \overline{M}_{s,r}^{(t)} \frac{\boldsymbol{\mu}_s}{\|\boldsymbol{\mu}\|^2} + \underline{M}_{s,r}^{(t)} \frac{\boldsymbol{\mu}_{-s}}{\|\boldsymbol{\mu}\|^2} + \overline{N}_{s,r}^{(t)} \frac{\boldsymbol{\nu}_s}{\|\boldsymbol{\nu}\|^2} + \underline{N}_{s,r}^{(t)} \frac{\boldsymbol{\nu}_{-s}}{\|\boldsymbol{\nu}\|^2} + \sum_{i \in [n_{\mathrm{st}}]} \rho_{s,r,i}^{(t)} \frac{\tilde{\boldsymbol{\xi}}_i}{\|\tilde{\boldsymbol{\xi}}_i\|^2},
$$

*where $\overline{M}_{s,r}^{(t)}, \underline{M}_{s,r}^{(t)}, \overline{N}_{s,r}^{(t)}, \underline{N}_{s,r}^{(t)}, \rho_{s,r,i}^{(t)}$ are recursively defined as*

$$
\overline{M}_{s,r}^{(t+1)} = \overline{M}_{s,r}^{(t)} + \frac{\eta}{mn_{\mathrm{st}}} \sum_{l \in [2]} \left( \sum_{i \in \mathcal{C}_{\boldsymbol{\mu}_s}^{(l)}} \tilde{g}_i^{(t)} - \sum_{i \in \mathcal{F}_{\boldsymbol{\mu}_s}^{(l)}} \tilde{g}_i^{(t)} \right) \|\boldsymbol{\mu}\|^2 \cdot \mathbb{1}\left[ \left\langle \boldsymbol{w}_{s,r}^{(t)}, \boldsymbol{\mu}_s \right\rangle > 0 \right],
$$

$$
\underline{M}_{s,r}^{(t+1)} = \underline{M}_{s,r}^{(t)} - \frac{\eta}{mn_{\mathrm{st}}} \sum_{l \in [2]} \left( \sum_{i \in \mathcal{C}_{\boldsymbol{\mu}_{-s}}^{(l)}} \tilde{g}_i^{(t)} - \sum_{i \in \mathcal{F}_{\boldsymbol{\mu}_{-s}}^{(l)}} \tilde{g}_i^{(t)} \right) \|\boldsymbol{\mu}\|^2 \cdot \mathbb{1}\left[ \left\langle \boldsymbol{w}_{s,r}^{(t)}, \boldsymbol{\mu}_{-s} \right\rangle > 0 \right],
$$

$$
\overline{N}_{s,r}^{(t+1)} = \overline{N}_{s,r}^{(t)} + \frac{\eta}{mn_{\mathrm{st}}} \sum_{l \in [2]} \left( \sum_{i \in \mathcal{C}_{\boldsymbol{\nu}_s}^{(l)}} \tilde{g}_i^{(t)} - \sum_{i \in \mathcal{F}_{\boldsymbol{\nu}_s}^{(l)}} \tilde{g}_i^{(t)} \right) \|\boldsymbol{\nu}\|^2 \cdot \mathbb{1}\left[ \left\langle \boldsymbol{w}_{s,r}^{(t)}, \boldsymbol{\nu}_s \right\rangle > 0 \right],
$$

$$
- \frac{\eta}{mn_{\mathrm{st}}} \sum_{l \in [2]} \left( \sum_{i \in \mathcal{C}_{-\boldsymbol{\nu}_s}^{(l)}} \tilde{g}_i^{(t)} - \sum_{i \in \mathcal{F}_{-\boldsymbol{\nu}_s}^{(l)}} \tilde{g}_i^{(t)} \right) \|\boldsymbol{\nu}\|^2 \cdot \mathbb{1}\left[ \left\langle \boldsymbol{w}_{s,r}^{(t)}, \boldsymbol{\nu}_s \right\rangle < 0 \right],
$$

$$
\underline{N}_{s,r}^{(t+1)} = \underline{N}_{s,r}^{(t)} - \frac{\eta}{mn_{\mathrm{st}}} \sum_{l \in [2]} \left( \sum_{i \in \mathcal{C}_{\boldsymbol{\nu}_{-s}}^{(l)}} \tilde{g}_i^{(t)} - \sum_{i \in \mathcal{F}_{\boldsymbol{\nu}_{-s}}^{(l)}} \tilde{g}_i^{(t)} \right) \|\boldsymbol{\nu}\|^2 \cdot \mathbb{1}\left[ \left\langle \boldsymbol{w}_{s,r}^{(t)}, \boldsymbol{\nu}_{-s} \right\rangle > 0 \right]
$$

$$
+ \frac{\eta}{mn_{\mathrm{st}}} \sum_{l \in [2]} \left( \sum_{i \in \mathcal{C}_{-\boldsymbol{\nu}_{-s}}^{(l)}} \tilde{g}_i^{(t)} - \sum_{i \in \mathcal{F}_{-\boldsymbol{\nu}_{-s}}^{(l)}} \tilde{g}_i^{(t)} \right) \|\boldsymbol{\nu}\|^2 \cdot \mathbb{1}\left[ \left\langle \boldsymbol{w}_{s,r}^{(t)}, \boldsymbol{\nu}_{-s} \right\rangle < 0 \right],
$$

$$
\rho_{s,r,i}^{(t+1)} = \rho_{s,r,i}^{(t)} + \frac{s\hat{y}_i \eta}{mn_{\mathrm{st}}} \tilde{g}_i^{(t)} \|\tilde{\boldsymbol{\xi}}_i\|^2 \cdot \mathbb{1}\left[ \left\langle \boldsymbol{w}_{s,r}^{(t)}, \tilde{\boldsymbol{\xi}}_i \right\rangle > 0 \right],
$$

*starting from $\overline{M}_{s,r}^{(t)} = \underline{M}_{s,r}^{(t)} = \overline{N}_{s,r}^{(t)} = \underline{N}_{s,r}^{(t)} = \rho_{s,r,i}^{(t)} = 0$. For simplicity, for any iteration $t \in [0, T^*]$, $r \in [m]$ and $i \in [n_{\mathrm{st}}]$, we define $\overline{\rho}_{r,i}^{(t)} := \rho_{\hat{y}_i, r, i}^{(t)}$ and $\underline{\rho}_{r,i}^{(t)} := \rho_{-\hat{y}_i, r, i}^{(t)}$. It follows that $\overline{\rho}_{r,i}^{(t)}$ is increasing and $\underline{\rho}_{r,i}^{(t)}$ is decreasing in iteration $t$.*

*Proof of Lemma A.3.* It is trivial for the case $t = 0$. Suppose it holds at iteration $\tau$. From the update rule, we have

$$
\boldsymbol{w}_{s,r}^{(\tau+1)} = \boldsymbol{w}_{s,r}^{(\tau)} + \frac{s\eta}{mn_{\text{st}}} \sum_{p\in[3]} \sum_{i\in[n_{\text{st}}]} \hat{y}_i\, \tilde{g}_i^{(\tau)} \mathbb{1}\left[\left\langle \boldsymbol{w}_{s,r}^{(\tau)}, \tilde{\boldsymbol{x}}_i^{(p)} \right\rangle > 0\right] \tilde{\boldsymbol{x}}_i^{(p)}
$$

$$
= \boldsymbol{w}_{s,r}^{(0)} + \overline{M}_{s,r}^{(\tau)} \frac{\boldsymbol{\mu}_s}{\|\boldsymbol{\mu}\|^2} + \underline{M}_{s,r}^{(\tau)} \frac{\boldsymbol{\mu}_{-s}}{\|\boldsymbol{\mu}\|^2} + \overline{N}_{s,r}^{(\tau)} \frac{\boldsymbol{\nu}_s}{\|\boldsymbol{\nu}\|^2} + \underline{N}_{s,r}^{(\tau)} \frac{\boldsymbol{\nu}_{-s}}{\|\boldsymbol{\nu}\|^2} + \sum_{i\in[n_{\text{st}}]} \rho_{s,r,i}^{(\tau)} \frac{\tilde{\boldsymbol{\xi}}_i}{\|\tilde{\boldsymbol{\xi}}_i\|^2}
$$

$$
+ \frac{s\eta}{mn_{\text{st}}} \sum_{p\in[3]} \sum_{i\in[n_{\text{st}}]} \hat{y}_i\, \tilde{g}_i^{(\tau)} \mathbb{1}\left[\left\langle \boldsymbol{w}_{s,r}^{(\tau)}, \tilde{\boldsymbol{x}}_i^{(p)} \right\rangle > 0\right] \tilde{\boldsymbol{x}}_i^{(p)}.
$$

Here, $\tilde{x}_i^{(p)}$'s are one of $\boldsymbol{\mu}_1, \boldsymbol{\mu}_{-1}, \boldsymbol{\nu}_1, \boldsymbol{\nu}_{-1}$, and $\tilde{\boldsymbol{\xi}}_i$. By grouping the terms accordingly, we obtain

$$
\boldsymbol{w}_{s,r}^{(\tau+1)} = \boldsymbol{w}_{s,r}^{(0)} + \overline{M}_{s,r}^{(\tau+1)} \frac{\boldsymbol{\mu}_s}{\|\boldsymbol{\mu}\|^2} + \underline{M}_{s,r}^{(\tau+1)} \frac{\boldsymbol{\mu}_{-s}}{\|\boldsymbol{\mu}\|^2} + \overline{N}_{s,r}^{(\tau+1)} \frac{\boldsymbol{\nu}_s}{\|\boldsymbol{\nu}\|^2} + \underline{N}_{s,r}^{(\tau+1)} \frac{\boldsymbol{\nu}_{-s}}{\|\boldsymbol{\nu}\|^2} + \sum_{i\in[n_{\text{st}}]} \rho_{s,r,i}^{(\tau+1)} \frac{\tilde{\boldsymbol{\xi}}_i}{\|\tilde{\boldsymbol{\xi}}_i\|^2},
$$

with

$$
\overline{M}_{s,r}^{(\tau+1)} = \overline{M}_{s,r}^{(\tau)} + \frac{\eta}{mn_{\text{st}}} \sum_{l\in[2]} \left( \sum_{i\in\mathcal{C}_{\boldsymbol{\mu}_s}^{(l)}} \tilde{g}_i^{(\tau)} - \sum_{i\in\mathcal{F}_{\boldsymbol{\mu}_s}^{(l)}} \tilde{g}_i^{(\tau)} \right) \|\boldsymbol{\mu}\|^2 \cdot \mathbb{1}\left[\left\langle \boldsymbol{w}_{s,r}^{(\tau)}, \boldsymbol{\mu}_s \right\rangle > 0\right],
$$

$$
\underline{M}_{s,r}^{(\tau+1)} = \underline{M}_{s,r}^{(\tau)} - \frac{\eta}{mn_{\text{st}}} \sum_{l\in[2]} \left( \sum_{i\in\mathcal{C}_{\boldsymbol{\mu}_{-s}}^{(l)}} \tilde{g}_i^{(\tau)} - \sum_{i\in\mathcal{F}_{\boldsymbol{\mu}_{-s}}^{(l)}} \tilde{g}_i^{(\tau)} \right) \|\boldsymbol{\mu}\|^2 \cdot \mathbb{1}\left[\left\langle \boldsymbol{w}_{s,r}^{(\tau)}, \boldsymbol{\mu}_{-s} \right\rangle > 0\right],
$$

$$
\overline{N}_{s,r}^{(\tau+1)} = \overline{N}_{s,r}^{(\tau)} + \frac{\eta}{mn_{\text{st}}} \sum_{l\in[2]} \left( \sum_{i\in\mathcal{C}_{\boldsymbol{\nu}_s}^{(l)}} \tilde{g}_i^{(\tau)} - \sum_{i\in\mathcal{F}_{\boldsymbol{\nu}_s}^{(l)}} \tilde{g}_i^{(\tau)} \right) \|\boldsymbol{\nu}\|^2 \cdot \mathbb{1}\left[\left\langle \boldsymbol{w}_{s,r}^{(\tau)}, \boldsymbol{\nu}_s \right\rangle > 0\right],
$$

$$
- \frac{\eta}{mn_{\text{st}}} \sum_{l\in[2]} \left( \sum_{i\in\mathcal{C}_{-\boldsymbol{\nu}_s}^{(l)}} \tilde{g}_i^{(\tau)} - \sum_{i\in\mathcal{F}_{-\boldsymbol{\nu}_s}^{(l)}} \tilde{g}_i^{(\tau)} \right) \|\boldsymbol{\nu}\|^2 \cdot \mathbb{1}\left[\left\langle \boldsymbol{w}_{s,r}^{(\tau)}, \boldsymbol{\nu}_s \right\rangle < 0\right],
$$

$$
\underline{N}_{s,r}^{(\tau+1)} = \underline{N}_{s,r}^{(\tau)} - \frac{\eta}{mn_{\text{st}}} \sum_{l\in[2]} \left( \sum_{i\in\mathcal{C}_{\boldsymbol{\nu}_{-s}}^{(l)}} \tilde{g}_i^{(\tau)} - \sum_{i\in\mathcal{F}_{\boldsymbol{\nu}_{-s}}^{(l)}} \tilde{g}_i^{(\tau)} \right) \|\boldsymbol{\nu}\|^2 \cdot \mathbb{1}\left[\left\langle \boldsymbol{w}_{s,r}^{(\tau)}, \boldsymbol{\nu}_{-s} \right\rangle > 0\right]
$$

$$
+ \frac{\eta}{mn_{\text{st}}} \sum_{l\in[2]} \left( \sum_{i\in\mathcal{C}_{-\boldsymbol{\nu}_{-s}}^{(l)}} \tilde{g}_i^{(\tau)} - \sum_{i\in\mathcal{F}_{-\boldsymbol{\nu}_{-s}}^{(l)}} \tilde{g}_i^{(\tau)} \right) \|\boldsymbol{\nu}\|^2 \cdot \mathbb{1}\left[\left\langle \boldsymbol{w}_{s,r}^{(\tau)}, \boldsymbol{\nu}_{-s} \right\rangle < 0\right],
$$

$$
\rho_{s,r,i}^{(\tau+1)} = \rho_{s,r,i}^{(\tau)} + \frac{s\hat{y}_i\eta}{mn_{\text{st}}} \tilde{g}_i^{(\tau)} \|\tilde{\boldsymbol{\xi}}_i\|^2 \cdot \mathbb{1}\left[\left\langle \boldsymbol{w}_{s,r}^{(\tau)}, \tilde{\boldsymbol{\xi}}_i \right\rangle > 0\right].
$$

Hence, we have desired conclusion. $\qquad\square$

### A.2.2 Properties of Data Sampling and Model Initialization

We establish concentration results for data sampling and model initialization.

Throughout the proof, we frequently use the following quantities. For each $s \in \{\pm 1\}$ and $i \in [n_{\text{st}}]$, we define:

- $n_{\boldsymbol{\mu}} := \frac{(2p_{\text{e}}+p_{\text{b}})n_{\text{st}}}{4}, n_{\boldsymbol{\nu}} = \frac{p_{\text{b}}n_{\text{st}}}{8}$.
- $\mathcal{M}_s := \left\{ r \in [m] : \left\langle \boldsymbol{w}_{s,r}^{(0)}, \boldsymbol{\mu}_s \right\rangle > 0 \right\}$.
- $\mathcal{A}_s := \left\{ r \in [m] : \left\langle \boldsymbol{w}_{s,r}^{(0)}, \boldsymbol{\nu}_s \right\rangle > 0 \right\}, \mathcal{B}_s := \left\{ r \in [m] : \left\langle \boldsymbol{w}_{s,r}^{(0)}, \boldsymbol{\nu}_s \right\rangle < 0 \right\}$.

- $\mathcal{X}_i := \left\{ r \in [m] : \left\langle \boldsymbol{w}_{\hat{y}_i, r}^{(0)}, \tilde{\boldsymbol{\xi}}_i \right\rangle > 0 \right\}$.

**Lemma A.4.** *Let $E_{\mathrm{st}}$ denote the event in which all the following hold for some large enough universal constant $C_{\mathrm{st}} > 0$:*

1. *For each $s \in \{\pm 1\}$, $l \in [2]$, we have*

$$\left(1 - C_{\mathrm{st}}^{-1}\right) \cdot n_{\boldsymbol{\mu}} \leq \left| \mathcal{C}_{\boldsymbol{\mu}_s}^{(l)} \right| \leq \left(1 + C_{\mathrm{st}}^{-1}\right) \cdot n_{\boldsymbol{\mu}}, \quad \left| \mathcal{F}_{\boldsymbol{\mu}_s}^{(l)} \right| \leq C_{\mathrm{st}}^{-1} \cdot n_{\boldsymbol{\mu}}$$

*and*

$$\left(1 - C_{\mathrm{st}}^{-1}\right) \cdot n_{\boldsymbol{\nu}} \leq \left| \mathcal{C}_{\boldsymbol{\nu}_s}^{(l)} \right|, \left| \mathcal{C}_{-\boldsymbol{\nu}_s}^{(l)} \right| \leq \left(1 + C_{\mathrm{st}}^{-1}\right) \cdot n_{\boldsymbol{\nu}}, \quad \left| \mathcal{F}_{\boldsymbol{\nu}_s}^{(l)} \right|, \left| \mathcal{F}_{-\boldsymbol{\nu}_s}^{(l)} \right| \leq C_{\mathrm{st}}^{-1} \cdot n_{\boldsymbol{\nu}}$$

2. *For each $s \in \{\pm 1\}$, $r \in [m]$, and $i \in [n_{\mathrm{st}}]$,*

$$\left| |\mathcal{M}_s| - \frac{m}{2} \right|, \left| |\mathcal{A}_s| - \frac{m}{2} \right|, \left| |\mathcal{B}_s| - \frac{m}{2} \right| \leq \sqrt{\frac{m}{2} \log \left( \frac{C_{\mathrm{st}}}{\delta} \right)}$$

*and*

$$\left| |\mathcal{X}_i| - \frac{m}{2} \right| \leq \sqrt{\frac{m}{2} \log \left( \frac{C_{\mathrm{st}} n_{\mathrm{st}}}{\delta} \right)}.$$

3. *For each $s, s' \in \{\pm 1\}$ and $r \in [m]$,*

$$\left| \left\langle \boldsymbol{w}_{s,r}^{(0)}, \frac{\boldsymbol{\mu}_{s'}}{\|\boldsymbol{\mu}\|} \right\rangle \right|, \quad \left| \left\langle \boldsymbol{w}_{s,r}^{(0)}, \frac{\boldsymbol{\nu}_{s'}}{\|\boldsymbol{\nu}\|} \right\rangle \right| \leq \sigma_0 \sqrt{2 \log \left( \frac{C_{\mathrm{st}} m}{\delta} \right)}.$$

4. *For any $i \in [n_{\mathrm{st}}]$,*

$$\left| \|\tilde{\boldsymbol{\xi}}_i\|^2 - \sigma_p^2(d - 4) \right| \leq C_{\mathrm{st}} \sigma_p^2 d^{\frac{1}{2}} \sqrt{\log \left( \frac{C_{\mathrm{st}} n_{\mathrm{st}}}{\delta} \right)}.$$

5. *For any $i, j \in [n_{\mathrm{st}}]$ with $i \neq j$,*

$$\left| \left\langle \tilde{\boldsymbol{\xi}}_i, \tilde{\boldsymbol{\xi}}_j \right\rangle \right| \leq C_{\mathrm{st}} \sigma_p^2 d^{\frac{1}{2}} \sqrt{\log \left( \frac{C_{\mathrm{st}} n_{\mathrm{st}}^2}{\delta} \right)}.$$

6. *For any $s \in \{\pm 1\}$, $r \in [m]$, and $i \in [n_{\mathrm{st}}]$,*

$$\left| \left\langle \boldsymbol{w}_{s,r}^{(0)}, \tilde{\boldsymbol{\xi}}_i \right\rangle \right| \leq C_{\mathrm{st}} \sigma_0 \sigma_p d^{\frac{1}{2}} \sqrt{\log \left( \frac{C_{\mathrm{st}} m n_{\mathrm{st}}}{\delta} \right)}.$$

7. *For any $s \in \{\pm 1\}$ and $r \in [m]$,*

$$\left\| \Pi_S \boldsymbol{w}_{s,r}^{(0)} \right\|^2 \leq 2 \sigma_0^2 d.$$

*Then, the event $E_{\mathrm{st}}$ occurs with probability at least $1 - \delta$.*

*Proof of Lemma A.4.* We begin by showing that each statement holds with high probability, and conclude the proof by applying a union bound. We prove the statements one by one, marking each with ■ once established.

We fix an arbitrary $s \in \{\pm 1\}$, $l \in \{\pm 1\}$ and $i \in [n_{\mathrm{st}}]$. We have

$$\mathbb{P}\left[ i \in \mathcal{C}_{\boldsymbol{\mu}_s}^{(l)} \right]$$
$$= \mathbb{P}\left[ \tilde{y}_i f_{\mathrm{wk}}\left( \boldsymbol{w}^*, \tilde{\boldsymbol{X}}_i \right) > 0 \,\middle|\, \left( \tilde{\boldsymbol{v}}_i^{(l)}, \tilde{\boldsymbol{v}}_i^{(3-l)} \right) = (\boldsymbol{\mu}_s, \boldsymbol{\mu}_s) \right] \mathbb{P}\left[ \left( \tilde{\boldsymbol{v}}_i^{(l)}, \tilde{\boldsymbol{v}}_i^{(3-l)} \right) = (\boldsymbol{\mu}_s, \boldsymbol{\mu}_s) \right]$$

$$+ \mathbb{P}\left[\tilde{y}_i f_{\mathrm{wk}}\left(\boldsymbol{w}^*, \tilde{\boldsymbol{X}}_i\right) > 0 \,\bigg|\, \left(\tilde{\boldsymbol{v}}_i^{(l)}, \tilde{\boldsymbol{v}}_i^{(3-l)}\right) = (\boldsymbol{\mu}_s, \boldsymbol{\nu}_s)\right] \mathbb{P}\left[\left(\tilde{\boldsymbol{v}}_i^{(l)}, \tilde{\boldsymbol{v}}_i^{(3-l)}\right) = (\boldsymbol{\mu}_s, \boldsymbol{\nu}_s)\right]$$

$$+ \mathbb{P}\left[\tilde{y}_i f_{\mathrm{wk}}\left(\boldsymbol{w}^*, \tilde{\boldsymbol{X}}_i\right) > 0 \,\bigg|\, \left(\tilde{\boldsymbol{v}}_i^{(l)}, \tilde{\boldsymbol{v}}_i^{(3-l)}\right) = (\boldsymbol{\mu}_s, -\boldsymbol{\nu}_s)\right] \mathbb{P}\left[\left(\tilde{\boldsymbol{v}}_i^{(l)}, \tilde{\boldsymbol{v}}_i^{(3-l)}\right) = (\boldsymbol{\mu}_s, -\boldsymbol{\nu}_s)\right]$$

$$= \mathbb{P}\left[\tilde{y}_i f_{\mathrm{wk}}\left(\boldsymbol{w}^*, \tilde{\boldsymbol{X}}_i\right) > 0 \,\bigg|\, \left(\tilde{\boldsymbol{v}}_i^{(l)}, \tilde{\boldsymbol{v}}_i^{(3-l)}\right) = (\boldsymbol{\mu}_s, \boldsymbol{\mu}_s)\right] \cdot \frac{p_{\mathrm{e}}}{2}$$

$$+ \mathbb{P}\left[\tilde{y}_i f_{\mathrm{wk}}\left(\boldsymbol{w}^*, \tilde{\boldsymbol{X}}_i\right) > 0 \,\bigg|\, \left(\tilde{\boldsymbol{v}}_i^{(l)}, \tilde{\boldsymbol{v}}_i^{(3-l)}\right) = (\boldsymbol{\mu}_s, \boldsymbol{\nu}_s)\right] \cdot \frac{p_{\mathrm{b}}}{8}$$

$$+ \mathbb{P}\left[\tilde{y}_i f_{\mathrm{wk}}\left(\boldsymbol{w}^*, \tilde{\boldsymbol{X}}_i\right) > 0 \,\bigg|\, \left(\tilde{\boldsymbol{v}}_i^{(l)}, \tilde{\boldsymbol{v}}_i^{(3-l)}\right) = (\boldsymbol{\mu}_s, -\boldsymbol{\nu}_s)\right] \cdot \frac{p_{\mathrm{b}}}{8}.$$

From the conclusion of Theorem 3.3, we have

$$\mathbb{P}\left[\tilde{y}_i f_{\mathrm{wk}}\left(\boldsymbol{w}^*, \tilde{\boldsymbol{X}}_i\right) > 0 \,\bigg|\, \left(\tilde{\boldsymbol{v}}_i^{(l)}, \tilde{\boldsymbol{v}}_i^{(3-l)}\right) = (\boldsymbol{\mu}_s, \boldsymbol{\mu}_s)\right] \geq 1 - \frac{1}{2C_{\mathrm{st}}},$$

$$\mathbb{P}\left[\tilde{y}_i f_{\mathrm{wk}}\left(\boldsymbol{w}^*, \tilde{\boldsymbol{X}}_i\right) > 0 \,\bigg|\, \left(\tilde{\boldsymbol{v}}_i^{(l)}, \tilde{\boldsymbol{v}}_i^{(3-l)}\right) = (\boldsymbol{\mu}_s, \boldsymbol{\nu}_s)\right] \geq 1 - \frac{1}{2C_{\mathrm{st}}},$$

and

$$\mathbb{P}\left[\tilde{y}_i f_{\mathrm{wk}}\left(\boldsymbol{w}^*, \tilde{\boldsymbol{X}}_i\right) > 0 \,\bigg|\, \left(\tilde{\boldsymbol{v}}_i^{(l)}, \tilde{\boldsymbol{v}}_i^{(3-l)}\right) = (\boldsymbol{\mu}_s, -\boldsymbol{\nu}_s)\right] \geq 1 - \frac{1}{2C_{\mathrm{st}}}.$$

Therefore,

$$\left(1 - \frac{1}{2C_{\mathrm{st}}}\right) \cdot n_{\boldsymbol{\mu}} \leq \mathbb{E}\left[\left|\mathcal{C}_{\boldsymbol{\mu}_s}^{(l)}\right|\right] \leq n_{\boldsymbol{\mu}}$$

and

$$\mathbb{E}\left[\left|\mathcal{F}_{\boldsymbol{\mu}_s}^{(l)}\right|\right] = n_{\boldsymbol{\mu}} - \mathbb{E}\left[\left|\mathcal{C}_{\boldsymbol{\mu}_s}^{(l)}\right|\right] \leq \frac{n_{\boldsymbol{\mu}}}{2C_{\mathrm{st}}}$$

By Höeffding's inequality, we have

$$\mathbb{P}\left[\left|\left|\mathcal{C}_{\boldsymbol{\mu}_s}^{(l)}\right| - \mathbb{E}\left[\left|\mathcal{C}_{\boldsymbol{\mu}_s}^{(l)}\right|\right]\right| \geq \sqrt{\frac{n_{\mathrm{st}}}{2} \log\left(\frac{C_{\mathrm{st}}}{\delta}\right)}\right] \leq \frac{2\delta}{C_{\mathrm{st}}}$$

and

$$\mathbb{P}\left[\left|\left|\mathcal{F}_{\boldsymbol{\mu}_s}^{(l)}\right| - \mathbb{E}\left[\left|\mathcal{F}_{\boldsymbol{\mu}_s}^{(l)}\right|\right]\right| \geq \sqrt{\frac{n_{\mathrm{st}}}{2} \log\left(\frac{C_{\mathrm{st}}}{\delta}\right)}\right] \leq \frac{2\delta}{C_{\mathrm{st}}}.$$

Hence, combining with (C2),

$$\left(1 - C_{\mathrm{st}}^{-1}\right) \cdot n_{\boldsymbol{\mu}} \leq \left|\mathcal{C}_{\boldsymbol{\mu}_s}^{(l)}\right| \leq \left(1 + C_{\mathrm{st}}^{-1}\right) \cdot n_{\boldsymbol{\mu}}, \quad \left|\mathcal{F}_{\boldsymbol{\mu}_s}^{(l)}\right| \leq C_{\mathrm{st}}^{-1} \cdot n_{\boldsymbol{\mu}},$$

with probability at least $1 - \frac{4\delta}{C_{\mathrm{st}}}$.

Now we address the case $\boldsymbol{\nu}_s$. We have

$$\mathbb{P}\left[i \in \mathcal{C}_{\boldsymbol{\nu}_s}^{(l)}\right]$$

$$= \mathbb{P}\left[\tilde{y}_i f_{\mathrm{wk}}\left(\boldsymbol{w}^*, \tilde{\boldsymbol{X}}_i\right) > 0 \,\bigg|\, \left(\tilde{\boldsymbol{v}}_i^{(l)}, \tilde{\boldsymbol{v}}_i^{(3-l)}\right) = (\boldsymbol{\nu}_s, \boldsymbol{\mu}_s)\right] \mathbb{P}\left[\left(\tilde{\boldsymbol{v}}_i^{(l)}, \tilde{\boldsymbol{v}}_i^{(3-l)}\right) = (\boldsymbol{\nu}_s, \boldsymbol{\mu}_s)\right]$$

$$+ \mathbb{P}\left[\tilde{y}_i f_{\mathrm{wk}}\left(\boldsymbol{w}^*, \tilde{\boldsymbol{X}}_i\right) > 0 \,\bigg|\, \left(\tilde{\boldsymbol{v}}_i^{(l)}, \tilde{\boldsymbol{v}}_i^{(3-l)}\right) = (\boldsymbol{\nu}_s, \boldsymbol{\nu}_s)\right] \mathbb{P}\left[\left(\tilde{\boldsymbol{v}}_i^{(l)}, \tilde{\boldsymbol{v}}_i^{(3-l)}\right) = (\boldsymbol{\nu}_s, \boldsymbol{\nu}_s)\right]$$

$$+ \mathbb{P}\left[\tilde{y}_i f_{\mathrm{wk}}\left(\boldsymbol{w}^*, \tilde{\boldsymbol{X}}_i\right) > 0 \,\bigg|\, \left(\tilde{\boldsymbol{v}}_i^{(l)}, \tilde{\boldsymbol{v}}_i^{(3-l)}\right) = (\boldsymbol{\nu}_s, -\boldsymbol{\nu}_s)\right] \mathbb{P}\left[\left(\tilde{\boldsymbol{v}}_i^{(l)}, \tilde{\boldsymbol{v}}_i^{(3-l)}\right) = (\boldsymbol{\nu}_s, -\boldsymbol{\nu}_s)\right]$$

$$= \mathbb{P}\left[\tilde{y}_i f_{\mathrm{wk}}\left(\boldsymbol{w}^*, \tilde{\boldsymbol{X}}_i\right) > 0 \,\bigg|\, \left(\tilde{\boldsymbol{v}}_i^{(l)}, \tilde{\boldsymbol{v}}_i^{(3-l)}\right) = (\boldsymbol{\nu}_s, \boldsymbol{\mu}_s)\right] \cdot \frac{p_{\mathrm{b}}}{8}$$

$$+ \mathbb{P}\left[\tilde{y}_i f_{\mathrm{wk}}\left(\boldsymbol{w}^*, \tilde{\boldsymbol{X}}_i\right) > 0 \,\bigg|\, \left(\tilde{\boldsymbol{v}}_i^{(l)}, \tilde{\boldsymbol{v}}_i^{(3-l)}\right) = (\boldsymbol{\nu}_s, \boldsymbol{\nu}_s)\right] \cdot \frac{p_{\mathrm{h}}}{8}$$

$$+ \mathbb{P}\left[\tilde{y}_i f_{\mathrm{wk}}\left(\boldsymbol{w}^*, \tilde{\boldsymbol{X}}_i\right) > 0 \,\bigg|\, \left(\tilde{\boldsymbol{v}}_i^{(l)}, \tilde{\boldsymbol{v}}_i^{(3-l)}\right) = (\boldsymbol{\nu}_s, -\boldsymbol{\nu}_s)\right] \cdot \frac{p_{\mathrm{h}}}{8}.$$

From the conclusion of Theorem 3.3, we have

$$\mathbb{P}\left[\tilde{y}_i f_{\mathrm{wk}}\left(\boldsymbol{w}^*, \tilde{\boldsymbol{X}}_i\right) > 0 \,\bigg|\, \left(\tilde{\boldsymbol{v}}_i^{(l)}, \tilde{\boldsymbol{v}}_i^{(3-l)}\right) = (\boldsymbol{\nu}_s, \boldsymbol{\mu}_s)\right] \geq 1 - \frac{1}{2C_{\mathrm{st}}},$$

From (C6), we have

$$\mathbb{E}\left[\left|\mathcal{C}_{\boldsymbol{\nu}_s}^{(l)}\right|\right] \leq \left(\frac{p_{\mathrm{b}}}{8} + \frac{p_{\mathrm{h}}}{4}\right) n_{\mathrm{st}} \leq \left(1 + \frac{1}{2C_{\mathrm{st}}}\right) \cdot n_{\boldsymbol{\nu}}$$

and

$$\mathbb{E}\left[\left|\mathcal{C}_{\boldsymbol{\nu}_s}^{(l)}\right|\right] \geq \left(1 - \frac{1}{2C_{\mathrm{st}}}\right) \cdot \frac{p_{\mathrm{b}} n_{\mathrm{st}}}{8} = \left(1 - \frac{1}{2C_{\mathrm{st}}}\right) \cdot n_{\boldsymbol{\nu}}.$$

In addition, we have

$$\left|\mathbb{E}\left[\left|\mathcal{F}_{\boldsymbol{\nu}_s}^{(l)}\right|\right]\right| = \left|\left(\frac{p_{\mathrm{b}}}{8} + \frac{p_{\mathrm{h}}}{4}\right) n_{\mathrm{st}} - \mathbb{E}\left[\left|\mathcal{C}_{\boldsymbol{\nu}_s}^{(l)}\right|\right]\right| \leq \frac{1}{2C_{\mathrm{st}}} \cdot \frac{p_{\mathrm{b}}}{8} + \frac{p_{\mathrm{h}}}{4} \leq \frac{2n_{\boldsymbol{\nu}}}{3C_{\mathrm{st}}}.$$

By Höeffding's inequality, we have

$$\mathbb{P}\left[\left|\left|\mathcal{C}_{\boldsymbol{\nu}_s}^{(l)}\right| - \mathbb{E}\left[\left|\mathcal{C}_{\boldsymbol{\nu}_s}^{(l)}\right|\right]\right| \geq \sqrt{\frac{n_{\mathrm{st}}}{2}\log\left(\frac{C_{\mathrm{st}}}{\delta}\right)}\right] \leq \frac{2\delta}{C_{\mathrm{st}}}$$

and

$$\mathbb{P}\left[\left|\left|\mathcal{F}_{\boldsymbol{\nu}_s}^{(l)}\right| - \mathbb{E}\left[\left|\mathcal{F}_{\boldsymbol{\nu}_s}^{(l)}\right|\right]\right| \geq \sqrt{\frac{n_{\mathrm{st}}}{2}\log\left(\frac{C_{\mathrm{st}}}{\delta}\right)}\right] \leq \frac{2\delta}{C_{\mathrm{st}}}.$$

From (C2), we have

$$\left(1 - C_{\mathrm{st}}^{-1}\right) \cdot n_{\boldsymbol{\nu}} \leq \left|\mathcal{C}_{\boldsymbol{\nu}_s}^{(l)}\right| \leq \left(1 + C_{\mathrm{st}}^{-1}\right) \cdot n_{\boldsymbol{\nu}}, \quad \left|\mathcal{F}_{\boldsymbol{\nu}_s}^{(l)}\right| \leq C_{\mathrm{st}}^{-1} \cdot n_{\boldsymbol{\nu}}$$

with probability at least $1 - \frac{4\delta}{C_{\mathrm{st}}}$, where the last inequality follows from Condition 3.1.

Using a similar argument, we also have the desired conclusion for the case $-\boldsymbol{\nu}_s$. ∎

Let us prove that the second statement holds with high probability. we fix arbitrary $s \in \{\pm 1\}$ and $i \in [n_{\mathrm{st}}]$. For each $r \in [m]$, $\mathbb{P}[r \in \mathcal{M}_s] = \mathbb{P}[r \in \mathcal{A}_s] = \mathbb{P}[r \in \mathcal{B}_s] = \mathbb{P}[r \in \mathcal{X}_i] = \frac{1}{2}$. By Höeffding's inequality, we have

$$\mathbb{P}\left[\left||\mathcal{M}_s| - \frac{m}{2}\right| \geq \sqrt{\frac{m}{2}\log\left(\frac{C_{\mathrm{st}}}{\delta}\right)}\right] \leq \frac{2\delta}{C_{\mathrm{st}}},$$

$$\mathbb{P}\left[\left||\mathcal{A}_s| - \frac{m}{2}\right| \geq \sqrt{\frac{m}{2}\log\left(\frac{C_{\mathrm{st}}}{\delta}\right)}\right] \leq \frac{2\delta}{mC_{\mathrm{st}}},$$

$$\mathbb{P}\left[\left||\mathcal{B}_s| - \frac{m}{2}\right| \geq \sqrt{\frac{m}{2}\log\left(\frac{C_{\mathrm{st}}}{\delta}\right)}\right] \leq \frac{2\delta}{C_{\mathrm{st}}},$$

and

$$\mathbb{P}\left[\left||\mathcal{X}_i| - \frac{m}{2}\right| \geq \sqrt{\frac{m}{2}\log\left(\frac{C_{\mathrm{st}} n_{\mathrm{st}}}{\delta}\right)}\right] \leq \frac{2\delta}{C_{\mathrm{st}} n_{\mathrm{st}}}.$$

∎

For the third statement, we fix arbitrary $s, s' \in \{\pm 1\}$ and $r \in [m]$. We have

$$\left\langle \boldsymbol{w}_{s,r}^{(0)}, \frac{\boldsymbol{\mu}_{s'}}{\|\boldsymbol{\mu}\|} \right\rangle, \left\langle \boldsymbol{w}_{s,r}^{(0)}, \frac{\boldsymbol{\nu}_{s'}}{\|\boldsymbol{\nu}\|} \right\rangle \overset{i.i.d.}{\sim} \mathcal{N}(0, \sigma_0^2).$$

Hence, by Höeffding's inequality, we have

$$\mathbb{P}\left[\left|\left\langle \boldsymbol{w}_{s,r}^{(0)}, \frac{\boldsymbol{\mu}_{s'}}{\|\boldsymbol{\mu}\|} \right\rangle\right| > \sigma_0\sqrt{2\log\left(\frac{C_{\mathrm{st}} m}{\delta}\right)}\right] \leq \frac{2\delta}{C_{\mathrm{st}} m}.$$

Similarly, we also have

$$\mathbb{P}\left[\left|\left\langle \boldsymbol{w}_{s,r}^{(0)}, \frac{\boldsymbol{\nu}_{s'}}{\|\boldsymbol{\nu}\|} \right\rangle\right| > \sigma_0\sqrt{2\log\left(\frac{C_{\mathrm{st}} m}{\delta}\right)}\right] \leq \frac{2\delta}{C_{\mathrm{st}} m}.$$

■

Before moving on to the remaining part, note that for each $i \in [n_{\text{st}}], s \in \{\pm 1\}$, and $r \in [m]$, we can write $\tilde{\boldsymbol{\xi}}_i$ and $\Pi_S \boldsymbol{w}_{s,r}^{(0)}$ as

$$\tilde{\boldsymbol{\xi}}_i = \sigma_p \sum_{h \in [d-4]} \mathbf{z}_{i,h} \boldsymbol{b}_h, \quad \Pi_S \boldsymbol{w}_{s,r}^{(0)} = \sigma_0 \sum_{h \in [d-4]} \mathbf{z}_{s,r,h} \boldsymbol{b}_h$$

where $\mathbf{z}_{i,h}, \mathbf{z}_{s,r,h} \overset{i.i.d.}{\sim} \mathcal{N}(0,1)$. The sub-gaussian norm of standard normal distribution $\mathcal{N}(0,1)$ is $\sqrt{\frac{8}{3}}$ and then $(\mathbf{z}_{i,h})^2 - 1, (\mathbf{z}_{s,r,h})^2 - 1$'s are mean zero sub-exponential random variables with sub-exponential norm $\frac{8}{3}$ (Lemma 2.7.6 in Vershynin (2018)). In addition, $\mathbf{z}_{s,r,h}\mathbf{z}_{i,h}$'s and $\mathbf{z}_{i,h}\mathbf{z}_{j,h}$'s with $i \neq j$ are mean zero sub-exponential random variables with sub-exponential norm less than or equal to $\frac{8}{3}$ (Lemma 2.7.7 in Vershynin (2018)).

We use Bernstein's inequality (Theorem 2.8.1 in Vershynin (2018)), with $c$ being the absolute constant stated therein. We then have the following for any $i \in [n_{\text{st}}]$:

$$\mathbb{P}\left[\left|\|\tilde{\boldsymbol{\xi}}_i\|^2 - \sigma_p^2(d-4)\right| \geq C_{\text{st}}\sigma_p^2 d^{\frac{1}{2}}\sqrt{\log\left(\frac{C_{\text{st}}n_{\text{st}}}{\delta}\right)}\right]$$

$$= \mathbb{P}\left[\left|\sum_{h \in [d-4]}\left((\mathbf{z}_{i,h})^2 - 1\right)\right| \geq C_{\text{st}}d^{\frac{1}{2}}\sqrt{\log\left(\frac{C_{\text{st}}n_{\text{st}}}{\delta}\right)}\right]$$

$$\leq 2\exp\left(-\frac{9cC_{\text{st}}^2 d}{64(d-4)}\log\left(\frac{C_{\text{st}}n_{\text{st}}}{\delta}\right)\right)$$

$$\leq 2\exp\left(-\log\left(\frac{C_{\text{st}}n_{\text{st}}}{\delta}\right)\right) \leq \frac{2\delta}{C_{\text{st}}n_{\text{st}}}.$$

■

For $i, j \in [n_{\text{st}}]$ with $i \neq j$, we have

$$\mathbb{P}\left[\left|\langle\tilde{\boldsymbol{\xi}}_i, \tilde{\boldsymbol{\xi}}_j\rangle\right| \geq C_{\text{st}}\sigma_p^2 d^{\frac{1}{2}}\sqrt{\log\left(\frac{C_{\text{st}}n_{\text{st}}^2}{\delta}\right)}\right]$$

$$= \mathbb{P}\left[\left|\sum_{h \in [d-4]}\mathbf{z}_{i,h}\mathbf{z}_{j,h}\right| \geq C_{\text{st}}d^{\frac{1}{2}}\sqrt{\log\left(\frac{C_{\text{st}}n_{\text{st}}^2}{\delta}\right)}\right]$$

$$\leq 2\exp\left(-\frac{9cC_{\text{st}}^2 d}{64(d-4)}\log\left(\frac{C_{\text{st}}n_{\text{st}}^2}{\delta}\right)\right)$$

$$\leq \frac{2\delta}{C_{\text{st}}n_{\text{st}}^2}.$$

■

For any $s \in \{\pm 1\}, r \in [m]$ and $i \in [n_{\text{st}}]$, by applying Bernstein's inequality, we have

$$\mathbb{P}\left[\left|\left\langle\boldsymbol{w}_{s,r}^{(0)}, \tilde{\boldsymbol{\xi}}_i\right\rangle\right| \geq C_{\text{st}}\sigma_0\sigma_p d^{\frac{1}{2}}\sqrt{\log\left(\frac{C_{\text{st}}mn_{\text{st}}}{\delta}\right)}\right]$$

$$= \mathbb{P}\left[\left|\sum_{h \in [d-4]}\mathbf{z}_{i,h}\mathbf{z}_{s,r,h}\right| \geq C_{\text{st}}d^{\frac{1}{2}}\sqrt{\log\left(\frac{C_{\text{st}}mn_{\text{st}}}{\delta}\right)}\right]$$

$$\leq 2\exp\left(-\frac{9cC_{\text{st}}^2 d}{64(d-4)}\log\left(\frac{C_{\text{st}}mn_{\text{st}}}{\delta}\right)\right)$$

$$\leq \frac{2\delta}{16mn_{\text{st}}}.$$

By applying Bernstein's inequality, for any $s \in \{\pm 1\}$ and $r \in [m]$, we have

$$\mathbb{P}\left[\left\|\Pi_S \boldsymbol{w}_{s,r}^{(0)}\right\|^2 \geq 2\sigma_0^2 d\right]$$

$$\leq \mathbb{P}\left[\left|\left\|\Pi_S \boldsymbol{w}_{s,r}^{(0)}\right\|^2 - \sigma_0^2(d-4)\right| \geq C_{\mathrm{st}}\sigma_0^2 d^{\frac{1}{2}}\sqrt{\log\left(\frac{C_{\mathrm{st}}m}{\delta}\right)}\right]$$

$$= \mathbb{P}\left[\left|\sum_{h \in [d-4]}\left((\mathbf{z}_{s,r,h})^2 - 1\right)\right| \geq C_{\mathrm{st}}d^{\frac{1}{2}}\sqrt{\log\left(\frac{C_{\mathrm{st}}m}{\delta}\right)}\right]$$

$$\leq 2\exp\left(-\frac{9cC_{\mathrm{st}}^2 d}{64(d-4)}\log\left(\frac{C_{\mathrm{st}}m}{\delta}\right)\right)$$

$$\leq 2\exp\left(-\log\left(\frac{C_{\mathrm{st}}n_{\mathrm{st}}}{\delta}\right)\right) \leq \frac{2\delta}{C_{\mathrm{st}}m},$$

where the first inequality follows from (C1). ∎

From union bound and a large choice of universal constant $C_{\mathrm{st}} > 0$, we conclude that the event $E_{\mathrm{st}}$ occurs with probability at least $1 - \delta$. □

### A.2.3 Properties Used Throughout the Proof

We introduce some notation and properties that are frequently used throughout the proof.

Let us define

$$\alpha_{\mathrm{st}} := 2C_{\mathrm{st}}\sigma_0 \max\left\{\|\boldsymbol{\mu}\|, \|\boldsymbol{\nu}\|, \sigma_p d^{\frac{1}{2}}\right\}\sqrt{2\log\left(\frac{C_{\mathrm{st}}mn_{\mathrm{st}}}{\delta}\right)},$$

$$\beta_{\mathrm{st}} := 4C_{\mathrm{st}}n_{\mathrm{st}}\sqrt{\frac{1}{d}\log\left(\frac{C_{\mathrm{st}}n_{\mathrm{st}}}{\delta}\right)},$$

and

$$\kappa_{\mathrm{st}} := 8\log(12), \quad \lambda_{\mathrm{st}} := \exp(2\kappa_{\mathrm{st}}).$$

Under Condition 3.1 and the event $E_{\mathrm{st}}$, the following hold:

- $\alpha_{\mathrm{st}}$ and $\beta_{\mathrm{st}}$ are small enough to satisfy

$$\alpha_{\mathrm{st}} \leq \min\left\{\frac{1}{100}, \frac{p_{\mathrm{b}}n_{\mathrm{st}}\|\boldsymbol{\nu}\|^2}{\sigma_p^2 d}, \frac{\sigma_p^2 d}{(2p_{\mathrm{e}}+p_{\mathrm{b}})n_{\mathrm{st}}\|\boldsymbol{\mu}\|^2}\right\}, \quad \beta_{\mathrm{st}}\log T^* \leq \frac{1}{100}. \tag{6}$$

- For any $s, s' \in \{\pm 1\}$, $r \in [m]$, and $i \in [n_{\mathrm{st}}]$,

$$\left|\left\langle \boldsymbol{w}_{s,r}^{(0)}, \boldsymbol{\mu}_{s'}\right\rangle\right|, \left|\left\langle \boldsymbol{w}_{s,r}^{(0)}, \boldsymbol{\nu}_{s'}\right\rangle\right|, \left|\left\langle \boldsymbol{w}_{s,r}^{(0)}, \tilde{\boldsymbol{\xi}}_i\right\rangle\right| \leq \alpha_{\mathrm{st}}. \tag{7}$$

- From (C3), for any $i, j \in [n_{\mathrm{st}}]$ with $i \neq j$, we have

$$\frac{\sigma_p^2 d}{2} \leq \|\tilde{\boldsymbol{\xi}}_i\|^2 \leq \frac{3\sigma_p^2 d}{2}, \frac{\left|\langle\tilde{\boldsymbol{\xi}}_i, \tilde{\boldsymbol{\xi}}_j\rangle\right|}{\|\tilde{\boldsymbol{\xi}}_i\|^2} \leq \frac{\beta_{\mathrm{st}}}{n_{\mathrm{st}}}, \left|1 - \frac{\|\tilde{\boldsymbol{\xi}}_j\|^2}{\|\tilde{\boldsymbol{\xi}}_i\|^2}\right| \leq \frac{\beta_{\mathrm{st}}}{n_{\mathrm{st}}}, \left|\|\tilde{\boldsymbol{\xi}}_i\|^2 - \sigma_p^2(d-4)\right| \leq \frac{\beta_{\mathrm{st}}\sigma_p^2 d}{n_{\mathrm{st}}}. \tag{8}$$

- For any $s \in \{\pm 1\}$, $r \in [m]$, and $i \in [n_{\mathrm{st}}]$, we have

$$\left|\frac{|\mathcal{M}_s|}{m} - \frac{1}{2}\right|, \left|\frac{|\mathcal{A}_s|}{m} - \frac{1}{2}\right|, \left|\frac{|\mathcal{B}_s|}{m} - \frac{1}{2}\right|, \left|\frac{|\mathcal{X}_i|}{m} - \frac{1}{2}\right| \leq \frac{1}{10}. \tag{9}$$

- The learning rate $\eta$ is small enough to satisfy

$$\eta \leq \min\left\{\frac{\beta_{\mathrm{st}}mn_{\mathrm{st}}}{2\sigma_p^2 d}, \frac{\beta_{\mathrm{st}}m}{2\lambda_{\mathrm{st}}\|\boldsymbol{\mu}\|^2}, \frac{\beta_{\mathrm{st}}m}{2\lambda_{\mathrm{st}}\|\boldsymbol{\nu}\|^2}\right\}. \tag{10}$$

### A.2.4 Technical Lemma

We also introduce a technical lemma that enables a tight characterization of the learning dynamics.

**Lemma A.5** (Lemma D.1 in Meng et al. (2024)). *Suppose that a sequence $a_t, t \geq 0$ follows the iterative formula*

$$a_{t+1} = a_t + \frac{c}{1 + be^{a_t}},$$

*for some $c \in [0, 1]$ and $b \geq 0$. Then it holds that*

$$x_t \leq a_t \leq \frac{c}{1 + be^{a_0}} + x_t$$

*for all $t \geq 0$. Here, $x_t$ is the unique solution of*

$$x_t + be^{x_t} = ct + a_0 + be^{a_0}.$$

# B Proof of Theorem 3.3

For the proof, we first introduce properties preserved during training (Appendix B.1), then prove the convergence of the training loss (Appendix B.2), and finally establish a bound on the test error (Appendix B.3).

## B.1 Preserved Properties during Training

In this subsection, we present several properties that remain preserved throughout training.

**Lemma B.1.** *Under Condition 3.2 and the event $E_{\mathrm{wk}}$, we have the following for any iteration $t \in [0, T^*]$:*

*(W1)* $0 \leq \rho_i^{(t)} \leq 4 \log T^*$ *for any $i \in [n_{\mathrm{wk}}]$.*

*(W2)* $\frac{n_{\mathrm{wk}}(2p_{\mathrm{e}} + p_{\mathrm{b}})}{12} \mathrm{SNR}_{\boldsymbol{\mu}}^2 \cdot \rho_i^{(t)} \leq M_s^{(t)} \leq 3 n_{\mathrm{wk}}(2p_{\mathrm{e}} + p_{\mathrm{b}}) \mathrm{SNR}_{\boldsymbol{\mu}}^2 \cdot \rho_i^{(t)}$ *for any $i \in [n_{\mathrm{wk}}], s \in \{\pm 1\}$.*

*(W3)* $\left| \rho_i^{(t)} - \rho_j^{(t)} \right| \leq \frac{\kappa_{\mathrm{wk}}}{4}$ *for any $i, j \in [n_{\mathrm{wk}}]$.*

*(W4)* $\left| y_i f_{\mathrm{wk}} \left( \boldsymbol{w}^{(t)}, \boldsymbol{X}_i \right) - y_j f_{\mathrm{wk}} \left( \boldsymbol{w}^{(t)}, \boldsymbol{X}_j \right) \right| \leq \frac{\kappa_{\mathrm{wk}}}{2}$ *for any $i, j \in [n_{\mathrm{wk}}]$.*

*(W5)* $1 - \kappa_{\mathrm{wk}} \leq \frac{g_j^{(t)}}{g_i^{(t)}} \leq 1 + \kappa_{\mathrm{wk}}$ *for any $i, j \in [n_{\mathrm{wk}}]$.*

*(W6)* $\left| N_s^{(t)} \right| \leq (2p_{\mathrm{h}} + p_{\mathrm{b}}) n_{\mathrm{wk}} \mathrm{SNR}_{\boldsymbol{\nu}}^2 \cdot \rho_i^{(t)}$ *for any $s \in \{\pm 1\}, i \in [n]$.*

*Proof of Lemma B.1.* It is trivial for the case $t = 0$. Assume the conclusions hold at iteration $t = \tau$ and we will prove for the case $t = \tau + 1$. Note that (W2) and (W6) at iteration $t = \tau$, along with (C5) and (C6) imply that

$$\left| N_s^{(\tau)} \right| \leq (2p_{\mathrm{h}} + p_{\mathrm{b}}) n_{\mathrm{wk}} \mathrm{SNR}_{\boldsymbol{\nu}}^2 \cdot \rho_i^{(\tau)} \leq \frac{1}{24} n_{\mathrm{wk}}(2p_{\mathrm{e}} + p_{\mathrm{b}}) \mathrm{SNR}_{\boldsymbol{\mu}}^2 \cdot \rho_i^{(\tau)} \leq \frac{1}{2} M_{s'}^{(\tau)}, \quad (11)$$

for any $s, s' \in \{\pm 1\}$ and $i \in [n]$.

(W1): We fix an arbitrary $i \in [n_{\mathrm{wk}}]$ and we want to show $\rho_i^{(\tau+1)} \leq 4 \log T^*$. If $\rho_i^{(\tau)} \leq 2 \log T^*$, then we have

$$\rho_i^{(\tau+1)} = \rho_i^{(\tau)} + \frac{\eta}{n_{\mathrm{wk}}} g_i^{(\tau)} \|\boldsymbol{\xi}_i\|^2 \leq 2 \log T^* + \frac{\eta}{n_{\mathrm{wk}}} \cdot \frac{3\sigma_p^2 d}{2} \leq 4 \log T^*,$$

where the first inequality follows from $g_i^{(\tau)} \leq 1$ and (4), and the last inequality follows from (C4).

Otherwise, there exists $\hat{t} < \tau$ such that $\rho_i^{(\hat{t})} \leq 2 \log T^* < \rho_i^{(\hat{t}+1)}$ since $\rho_i^{(t)}$ is increasing in iteration $t$. From (4) and (C4), we have

$$\rho_i^{(\tau+1)} = \rho_i^{(\hat{t})} + \left( \rho_i^{(\hat{t}+1)} - \rho_i^{(\hat{t})} \right) + \sum_{t=\hat{t}+1}^{\tau} \left( \rho_i^{(t+1)} - \rho_i^{(t)} \right)$$

$$= \rho_i^{(\hat{t})} + \frac{\eta}{n_{\mathrm{wk}}} g_i^{(\hat{t})} \|\boldsymbol{\xi}_i\|^2 + \frac{\eta}{n_{\mathrm{wk}}} \sum_{t=\hat{t}+1}^{\tau} g_i^{(t)} \|\boldsymbol{\xi}_i\|^2$$

$$\leq 2 \log T^* + \frac{\eta}{n_{\mathrm{wk}}} \cdot \frac{3}{2} \sigma_p^2 d + \frac{\eta}{n_{\mathrm{wk}}} \cdot \frac{3}{2} \sigma_p^2 d \sum_{t=\hat{t}+1}^{\tau} g_i^{(t)}$$

$$\leq 3 \log T^* + \frac{3\eta\sigma_p^2 d}{2 n_{\mathrm{wk}}} \sum_{t=\hat{t}+1}^{\tau} \exp \left( -y_i f_{\mathrm{wk}} \left( \boldsymbol{w}^{(t)}, \boldsymbol{X}_i \right) \right).$$

For any iteration $t \in [\hat{t} + 1, \tau]$, we have

$$
\begin{aligned}
y_i f_{\mathrm{wk}}\left(\boldsymbol{w}^{(t)}, \boldsymbol{X}_i\right) &= \left\langle \boldsymbol{w}^{(t)}, y_i \boldsymbol{v}_i^{(1)} \right\rangle + \left\langle \boldsymbol{w}^{(t)}, y_i \boldsymbol{v}_i^{(2)} \right\rangle + \left\langle \boldsymbol{w}^{(t)}, y_i \boldsymbol{\xi}_i \right\rangle \\
&\geq -2 \max\left\{ \left| N_1^{(t)} \right|, \left| N_{-1}^{(t)} \right| \right\} + \rho_i^{(t)} + \sum_{j \in [n_{\mathrm{wk}}] \setminus \{i\}} y_i y_j \rho_j^{(t)} \frac{\langle \boldsymbol{\xi}_i, \boldsymbol{\xi}_j \rangle}{\|\boldsymbol{\xi}_j\|^2} \\
&\geq -2 \max\left\{ \left| N_1^{(t)} \right|, \left| N_{-1}^{(t)} \right| \right\} + \rho_i^{(t)} - \sum_{j \in [n_{\mathrm{wk}}] \setminus \{i\}} \rho_j^{(t)} \frac{|\langle \boldsymbol{\xi}_i, \boldsymbol{\xi}_j \rangle|}{\|\boldsymbol{\xi}_j\|^2} \\
&\geq -4 n_{\mathrm{wk}} (2 p_{\mathrm{h}} + p_{\mathrm{b}}) \mathrm{SNR}_{\boldsymbol{\nu}}^2 \cdot \rho_i^{(t)} + \rho_i^{(t)} - \sum_{j \in [n_{\mathrm{wk}}] \setminus \{i\}} \rho_j^{(t)} \frac{|\langle \boldsymbol{\xi}_i, \boldsymbol{\xi}_j \rangle|}{\|\boldsymbol{\xi}_j\|^2} \\
&\geq -4 n_{\mathrm{wk}} (2 p_{\mathrm{h}} + p_{\mathrm{b}}) \mathrm{SNR}_{\boldsymbol{\nu}}^2 \cdot 4 \log T^* + 2 \log T^* - 4 \log T^* \cdot \beta_{\mathrm{wk}} \\
&= \left( 1 - 8 n_{\mathrm{wk}} (2 p_{\mathrm{h}} + p_{\mathrm{b}}) \mathrm{SNR}_{\boldsymbol{\nu}}^2 - 2 \beta_{\mathrm{wk}} \right) \cdot 2 \log T^* \\
&\geq \log T^*,
\end{aligned}
$$

where the first inequality follows from the fact that $M_1^{(t)}, M_{-1}^{(t)} \geq 0$, the third from applying (W2) at iteration $t$, the fourth from (W1) at iteration $t$ and (4), and the last from (3) and Condition 3.2.

Now, we have our conclusion

$$
\begin{aligned}
\rho_i^{(\tau+1)} &\leq 3 \log T^* + \frac{3 \eta \sigma_p^2 d}{2 n_{\mathrm{wk}}} \sum_{t=\hat{t}+1}^{\tau} \exp\left( -y_i f_{\mathrm{wk}}\left( \boldsymbol{w}^{(t)}, \boldsymbol{X}_i \right) \right) \\
&\leq 3 \log T^* + \frac{3 \eta \sigma_p^2 d}{2 n_{\mathrm{wk}}} \sum_{t=\hat{t}+1}^{\tau} \exp\left( -\log T^* \right) \\
&\leq 3 \log T^* + \frac{3 \eta \sigma_p^2 d}{2 n_{\mathrm{wk}}} T^* \exp\left( -\log T^* \right) \\
&\leq 4 \log T^*,
\end{aligned}
$$

where we applied (C4) for the last inequality.

(W2): We fix arbitrary $s \in \{\pm 1\}$ and $i \in [n_{\mathrm{wk}}]$. We have

$$
\begin{aligned}
M_s^{(\tau+1)} - M_s^{(\tau)} &= \frac{\eta}{n_{\mathrm{wk}}} \left( \sum_{j \in \mathcal{S}_{\boldsymbol{\mu}_s}^{(1)}} g_j^{(\tau)} + \sum_{j \in \mathcal{S}_{\boldsymbol{\mu}_s}^{(2)}} g_j^{(\tau)} \right) \cdot \|\boldsymbol{\mu}\|^2 \\
&\leq \frac{\eta}{n_{\mathrm{wk}}} \cdot 2 \cdot \left( \frac{p_{\mathrm{e}}}{2} + \frac{p_{\mathrm{b}}}{4} + \gamma_{\mathrm{wk}} \right) n_{\mathrm{wk}} \cdot \left( g_i^{(\tau)} (1 + \kappa_{\mathrm{wk}}) \right) \cdot \|\boldsymbol{\mu}\|^2 \\
&\leq \frac{\eta}{n_{\mathrm{wk}}} \cdot 2 \cdot \frac{3}{2} \left( \frac{p_{\mathrm{e}}}{2} + \frac{p_{\mathrm{b}}}{4} \right) n_{\mathrm{wk}} \cdot 2 g_i^{(\tau)} \cdot \|\boldsymbol{\mu}\|^2 \\
&= \frac{3}{2} \eta \left( 2 p_{\mathrm{e}} + p_{\mathrm{b}} \right) g_i^{(\tau)} \|\boldsymbol{\mu}\|^2,
\end{aligned}
$$

where the first inequality follows from (W3) at iteration $\tau$ and (5), the second follows from (3). From (4), we have

$$
\rho_i^{(\tau+1)} - \rho_i^{(\tau)} = \frac{\eta}{n_{\mathrm{wk}}} g_i^{(\tau)} \|\boldsymbol{\xi}_i\|^2 \geq \frac{\eta \sigma_p^2 d}{2 n_{\mathrm{wk}}} g_i^{(\tau)},
$$

and thus,

$$
M_s^{(\tau+1)} - M_s^{(\tau)} \leq 3 n_{\mathrm{wk}} \left( 2 p_{\mathrm{e}} + p_{\mathrm{b}} \right) \mathrm{SNR}_{\boldsymbol{\mu}}^2 \left( \rho_i^{(\tau+1)} - \rho_i^{(\tau)} \right).
$$

Combining with (W2) at iteration $\tau$, we have

$$
\begin{aligned}
M_s^{(\tau+1)} &= M_s^{(\tau)} + \left( M_s^{(\tau+1)} - M_s^{(\tau)} \right) \\
&\leq 3 n_{\mathrm{wk}} (2 p_{\mathrm{e}} + p_{\mathrm{b}}) \mathrm{SNR}_{\boldsymbol{\mu}}^2 \cdot \rho_i^{(\tau)} + 3 n_{\mathrm{wk}} \left( 2 p_{\mathrm{e}} + p_{\mathrm{b}} \right) \mathrm{SNR}_{\boldsymbol{\mu}}^2 \left( \rho_i^{(\tau+1)} - \rho_i^{(\tau)} \right)
\end{aligned}
$$

$$= 3n_{\text{wk}}(2p_{\text{e}} + p_{\text{b}})\text{SNR}_{\boldsymbol{\mu}}^2 \cdot \rho_i^{(\tau+1)}.$$

Similarly, we have

$$
\begin{aligned}
M_s^{(\tau+1)} - M_s^{(\tau)} &= \frac{\eta}{n_{\text{wk}}} \left( \sum_{j \in \mathcal{S}_{\boldsymbol{\mu}_s}^{(1)}} g_j^{(\tau)} + \sum_{j \in \mathcal{S}_{\boldsymbol{\mu}_s}^{(2)}} g_j^{(\tau)} \right) \|\boldsymbol{\mu}\|^2 \\
&\geq \frac{\eta}{n_{\text{wk}}} \cdot 2 \cdot \left( \frac{p_{\text{e}}}{2} + \frac{p_{\text{b}}}{4} - \gamma_{\text{wk}} \right) n_{\text{wk}} \cdot \left( g_i^{(\tau)} \left(1 - \kappa_{\text{wk}}\right) \right) \|\boldsymbol{\mu}\|^2 \\
&\geq \frac{\eta}{n_{\text{wk}}} \cdot 2 \cdot \frac{1}{2} \left( \frac{p_{\text{e}}}{2} + \frac{p_{\text{b}}}{4} \right) n_{\text{wk}} \cdot \frac{1}{2} g_i^{(\tau)} \cdot \|\boldsymbol{\mu}\|^2 \\
&= \frac{1}{8} \eta \left( 2p_{\text{e}} + p_{\text{b}} \right) g_i^{(\tau)} \cdot \|\boldsymbol{\mu}\|^2,
\end{aligned}
$$

where the first inequality follows from (W5) at iteration $\tau$ and (5), and the second follows from (3). From (4), we have

$$\rho_i^{(\tau+1)} - \rho_i^{(\tau)} = \frac{\eta}{n_{\text{wk}}} g_i^{(\tau)} \|\boldsymbol{\xi}_i\|^2 \leq \frac{3\eta \sigma_p^2 d}{2n_{\text{wk}}} g_i^{(\tau)},$$

and thus, we have

$$M_s^{(\tau+1)} - M_s^{(\tau)} \geq \frac{1}{12} n_{\text{wk}} \left( 2p_{\text{e}} + p_{\text{b}} \right) \text{SNR}_{\boldsymbol{\mu}}^2 \left( \rho_i^{(\tau+1)} - \rho_i^{(\tau)} \right).$$

Combining with (W2) at iteration $\tau$, we have

$$
\begin{aligned}
M_s^{(\tau+1)} &= M_s^{(\tau)} + \left( M_s^{(\tau+1)} - M_s^{(\tau)} \right) \\
&\geq \frac{1}{12} n_{\text{wk}} (2p_{\text{e}} + p_{\text{b}}) \text{SNR}_{\boldsymbol{\mu}}^2 \cdot \rho_i^{(\tau)} + \frac{1}{12} n_{\text{wk}} \left( 2p_{\text{e}} + p_{\text{b}} \right) \text{SNR}_{\boldsymbol{\mu}}^2 \left( \rho_i^{(\tau+1)} - \rho_i^{(\tau)} \right) \\
&= \frac{1}{12} n_{\text{wk}} (2p_{\text{e}} + p_{\text{b}}) \text{SNR}_{\boldsymbol{\mu}}^2 \cdot \rho_i^{(\tau+1)}.
\end{aligned}
$$

(W3): We fix arbitrary $i, j \in [n_{\text{wk}}]$ with $i \neq j$. Without loss of generality, we assume that $\rho_i^{(\tau)} \geq \rho_j^{(\tau)}$. From (4) and (C4), we have

$$\rho_i^{(\tau+1)} - \rho_j^{(\tau+1)} = \rho_i^{(\tau)} - \rho_j^{(\tau)} + \frac{\eta}{n_{\text{wk}}} \left( g_i^{(\tau)} \|\boldsymbol{\xi}_i\|^2 - g_j^{(\tau)} \|\boldsymbol{\xi}_j\|^2 \right) \geq -\frac{\eta}{n_{\text{wk}}} \cdot \frac{3\sigma_p^2 d}{2} \geq -\frac{\kappa_{\text{wk}}}{4}.$$

Thus, we want to show that $\rho_i^{(\tau+1)} - \rho_j^{(\tau+1)} \leq \frac{\kappa_{\text{wk}}}{4}$.

If $\rho_i^{(\tau)} - \rho_j^{(\tau)} < \frac{\kappa_{\text{wk}}}{8}$, from triangular inequality, (4), and (C4), we have

$$\rho_i^{(\tau+1)} - \rho_j^{(\tau+1)} = \rho_i^{(\tau)} - \rho_j^{(\tau)} + \frac{\eta}{n_{\text{wk}}} \left( g_i^{(\tau)} \|\boldsymbol{\xi}_i\|^2 - g_j^{(\tau)} \|\boldsymbol{\xi}_j\|^2 \right) \leq \frac{\kappa_{\text{wk}}}{8} + \frac{\eta}{n_{\text{wk}}} \cdot \frac{3\sigma_p^2 d}{2} \leq \frac{\kappa_{\text{wk}}}{4}.$$

Otherwise, we have

$$
\begin{aligned}
& y_i f_{\text{wk}} \left( \boldsymbol{w}^{(\tau)}, \boldsymbol{X}_i \right) - y_j f_{\text{wk}} \left( \boldsymbol{w}^{(\tau)}, \boldsymbol{X}_j \right) \\
&= \left\langle \boldsymbol{w}^{(\tau)}, y_i \left( \boldsymbol{v}_i^{(1)} + \boldsymbol{v}_i^{(2)} + \boldsymbol{\xi}_i \right) \right\rangle - \left\langle \boldsymbol{w}^{(\tau)}, y_j \left( \boldsymbol{v}_j^{(1)} + \boldsymbol{v}_j^{(2)} + \boldsymbol{\xi}_j \right) \right\rangle \\
&\geq \left( \rho_i^{(\tau)} - \rho_j^{(\tau)} \right) - 3M_{y_j}^{(\tau)} + \sum_{i' \in [n_{\text{wk}}] \setminus \{i\}} y_i y_{i'} \rho_{i'}^{(\tau)} \frac{|\langle \boldsymbol{\xi}_i, \boldsymbol{\xi}_{i'} \rangle|}{\|\boldsymbol{\xi}_{i'}\|^2} - \sum_{j' \in [n_{\text{wk}}] \setminus \{j\}} y_j y_{j'} \rho_{j'}^{(\tau)} \frac{|\langle \boldsymbol{\xi}_j, \boldsymbol{\xi}_{j'} \rangle|}{\|\boldsymbol{\xi}_{j'}\|^2} \\
&\geq \left( \rho_i^{(\tau)} - \rho_j^{(\tau)} \right) - 3M_{y_j}^{(\tau)} - \sum_{i' \in [n_{\text{wk}}] \setminus \{i\}} \rho_{i'}^{(\tau)} \frac{|\langle \boldsymbol{\xi}_i, \boldsymbol{\xi}_{i'} \rangle|}{\|\boldsymbol{\xi}_{i'}\|^2} - \sum_{j' \in [n_{\text{wk}}] \setminus \{j\}} \rho_{j'}^{(\tau)} \frac{|\langle \boldsymbol{\xi}_j, \boldsymbol{\xi}_{j'} \rangle|}{\|\boldsymbol{\xi}_{j'}\|^2} \\
&\geq \frac{\kappa_{\text{wk}}}{8} - 3 \cdot 3n_{\text{wk}}(2p_{\text{e}} + p_{\text{b}})\text{SNR}_{\boldsymbol{\mu}}^2 \cdot 4 \log T^* - 2 \cdot 4 \log T^* \cdot \beta_{\text{wk}} \\
&\geq \frac{\kappa_{\text{wk}}}{16} > 0,
\end{aligned}
$$

where the first inequality follows from (11), and the fourth inequality follows from (3) and Condition 3.2. Then, we have

$$\frac{g_i^{(\tau)} \|\boldsymbol{\xi}_i\|^2}{g_j^{(\tau)} \|\boldsymbol{\xi}_j\|^2} = \frac{1 + \exp\left(y_j f_{\mathrm{wk}}\left(\boldsymbol{w}^{(\tau)}, \boldsymbol{X}_j\right)\right)}{1 + \exp\left(y_i f_{\mathrm{wk}}\left(\boldsymbol{w}^{(\tau)}, \boldsymbol{X}_i\right)\right)} \cdot \frac{\|\boldsymbol{\xi}_i\|^2}{\|\boldsymbol{\xi}_j\|^2}$$

$$\leq \exp\left[y_j f_{\mathrm{wk}}\left(\boldsymbol{w}^{(\tau)}, \boldsymbol{X}_j\right) - y_i f_{\mathrm{wk}}\left(\boldsymbol{w}^{(\tau)}, \boldsymbol{X}_i\right)\right] \cdot \left(1 + \frac{\beta_{\mathrm{wk}}}{n_{\mathrm{wk}}}\right)$$

$$\leq \exp\left[-\frac{\kappa_{\mathrm{wk}}}{16} + \frac{\beta_{\mathrm{wk}}}{n_{\mathrm{wk}}}\right]$$

$$\leq 1.$$

Therefore, we have

$$\rho_i^{(\tau+1)} - \rho_j^{(\tau+1)} = \rho_i^{(\tau)} - \rho_j^{(\tau)} + \frac{\eta}{n_{\mathrm{wk}}} \left(g_i^{(\tau)} \|\boldsymbol{\xi}_i\|^2 - g_j^{(\tau)} \|\boldsymbol{\xi}_j\|^2\right) \leq \rho_i^{(\tau)} - \rho_j^{(\tau)} \leq \frac{\kappa_{\mathrm{wk}}}{4}.$$

(W4): For any $i, j \in [n_{\mathrm{wk}}]$, we have

$$y_i f_{\mathrm{wk}}\left(\boldsymbol{w}^{(\tau+1)}, \boldsymbol{X}_i\right) - y_j f_{\mathrm{wk}}\left(\boldsymbol{w}^{(\tau+1)}, \boldsymbol{X}_j\right)$$

$$= \left\langle \boldsymbol{w}^{(\tau+1)}, y_i \left(\boldsymbol{v}_i^{(1)} + \boldsymbol{v}_i^{(2)} + \boldsymbol{\xi}_i\right)\right\rangle - \left\langle \boldsymbol{w}^{(\tau+1)}, y_j \left(\boldsymbol{v}_j^{(1)} + \boldsymbol{v}_j^{(2)} + \boldsymbol{\xi}_j\right)\right\rangle$$

$$\leq \left(\rho_i^{(\tau+1)} - \rho_j^{(\tau+1)}\right) + 3M_{y_j}^{(\tau+1)}$$

$$+ \sum_{i' \in [n_{\mathrm{wk}}] \setminus \{i\}} y_i y_{i'} \rho_{i'}^{(\tau+1)} \frac{\langle \boldsymbol{\xi}_i, \boldsymbol{\xi}_{i'}\rangle}{\|\boldsymbol{\xi}_{i'}\|^2} - \sum_{j' \in [n_{\mathrm{wk}}] \setminus \{j\}} y_j y_{j'} \rho_{j'}^{(\tau+1)} \frac{\langle \boldsymbol{\xi}_j, \boldsymbol{\xi}_{j'}\rangle}{\|\boldsymbol{\xi}_{j'}\|^2}$$

$$\leq \left(\rho_i^{(\tau+1)} - \rho_j^{(\tau+1)}\right) + 3M_{y_j}^{(\tau+1)} + \sum_{i' \in [n_{\mathrm{wk}}] \setminus \{i\}} \rho_{i'}^{(\tau+1)} \frac{|\langle \boldsymbol{\xi}_i, \boldsymbol{\xi}_{i'}\rangle|}{\|\boldsymbol{\xi}_{i'}\|^2} + \sum_{j' \in [n_{\mathrm{wk}}] \setminus \{j\}} \rho_{j'}^{(\tau+1)} \frac{|\langle \boldsymbol{\xi}_j, \boldsymbol{\xi}_{j'}\rangle|}{\|\boldsymbol{\xi}_{j'}\|^2}$$

$$\leq \frac{\kappa_{\mathrm{wk}}}{8} + 3 \cdot 3n_{\mathrm{wk}}(2p_{\mathrm{e}} + p_{\mathrm{b}})\mathrm{SNR}_{\boldsymbol{\mu}}^2 \cdot 4\log T^* + 2 \cdot 4\log T^* \cdot \beta_{\mathrm{wk}}$$

$$\leq \frac{\kappa_{\mathrm{wk}}}{2},$$

where the first inequality follows from (11), the third inequality follows from (W1) and (W2) at iteration $\tau + 1$, which we have shown earlier, and the last inequality is due to (3) and Condition 3.2.

(W5): Let us fix arbitrary $i, j \in [n_{\mathrm{wk}}]$ and assume $y_i f_{\mathrm{wk}}\left(\boldsymbol{w}^{(\tau+1)}, \boldsymbol{X}_i\right) \geq y_j f_{\mathrm{wk}}\left(\boldsymbol{w}^{(\tau+1)}, \boldsymbol{X}_j\right)$, without loss of generality. Then, we have

$$1 \leq \frac{g_j^{(\tau+1)}}{g_i^{(\tau+1)}} = \frac{1 + \exp\left(y_i f_{\mathrm{wk}}\left(\boldsymbol{w}^{(\tau+1)}, \boldsymbol{X}_i\right)\right)}{1 + \exp\left(y_j f_{\mathrm{wk}}\left(\boldsymbol{w}^{(\tau+1)}, \boldsymbol{X}_j\right)\right)}$$

$$\leq \exp\left[y_i f_{\mathrm{wk}}\left(\boldsymbol{w}^{(\tau+1)}, \boldsymbol{X}_i\right) - y_j f_{\mathrm{wk}}\left(\boldsymbol{w}^{(\tau+1)}, \boldsymbol{X}_j\right)\right]$$

$$\leq 1 + 2\left[y_i f_{\mathrm{wk}}\left(\boldsymbol{w}^{(\tau+1)}, \boldsymbol{X}_i\right) - y_j f_{\mathrm{wk}}\left(\boldsymbol{w}^{(\tau+1)}, \boldsymbol{X}_j\right)\right]$$

$$\leq 1 + \kappa_{\mathrm{wk}},$$

where we use the inequality $e^z \leq 1 + 2z$ for any $z \in (0, 1)$, which is applicable due to (W4) at iteration $\tau + 1$. In addition, we have

$$1 \geq \frac{g_i^{(\tau+1)}}{g_j^{(\tau+1)}} = \frac{1 + \exp\left(y_j f_{\mathrm{wk}}\left(\boldsymbol{w}^{(\tau+1)}, \boldsymbol{X}_j\right)\right)}{1 + \exp\left(y_i f_{\mathrm{wk}}\left(\boldsymbol{w}^{(\tau+1)}, \boldsymbol{X}_i\right)\right)}$$

$$\geq \exp\left[y_j f_{\mathrm{wk}}\left(\boldsymbol{w}^{(\tau+1)}, \boldsymbol{X}_j\right) - y_i f_{\mathrm{wk}}\left(\boldsymbol{w}^{(\tau+1)}, \boldsymbol{X}_i\right)\right]$$

$$\geq 1 + \left[y_j f_{\mathrm{wk}}\left(\boldsymbol{w}^{(\tau+1)}, \boldsymbol{X}_j\right) - y_i f_{\mathrm{wk}}\left(\boldsymbol{w}^{(\tau+1)}, \boldsymbol{X}_i\right)\right]$$

$$\geq 1 - \kappa_{\mathrm{wk}},$$

where we use the inequality $e^z \geq 1 + z$ for any $z \in \mathbb{R}$.

(W6): We fix arbitrary $s \in \{\pm 1\}$ and $i \in [n_{\mathrm{wk}}]$. We have

$$N_s^{(\tau+1)} - N_s^{(\tau)}$$

$$= \frac{\eta}{n_{\mathrm{wk}}} \left( \sum_{j \in \mathcal{S}_{\boldsymbol{\nu}_s}^{(1)}} g_j^{(\tau)} + \sum_{j \in \mathcal{S}_{\boldsymbol{\nu}_s}^{(2)}} g_j^{(\tau)} - \sum_{j \in \mathcal{S}_{-\boldsymbol{\nu}_s}^{(1)}} g_j^{(\tau)} - \sum_{j \in \mathcal{S}_{-\boldsymbol{\nu}_s}^{(2)}} g_j^{(\tau)} \right) \|\boldsymbol{\nu}\|^2$$

$$\leq \frac{\eta}{n_{\mathrm{wk}}} \left[ \left( \left| \mathcal{S}_{\boldsymbol{\nu}_s}^{(1)} \right| + \left| \mathcal{S}_{\boldsymbol{\nu}_s}^{(2)} \right| \right) (1 + \kappa_{\mathrm{wk}}) - \left( \left| \mathcal{S}_{-\boldsymbol{\nu}_s}^{(1)} \right| + \left| \mathcal{S}_{-\boldsymbol{\nu}_s}^{(2)} \right| \right) (1 - \kappa_{\mathrm{wk}}) \right] g_i^{(\tau)} \|\boldsymbol{\nu}\|^2$$

$$\leq \eta \left[ 2 \left( \frac{p_{\mathrm{h}}}{4} + \frac{p_{\mathrm{b}}}{8} + \gamma_{\mathrm{wk}} \right) (1 + \kappa_{\mathrm{wk}}) - 2 \left( \frac{p_{\mathrm{h}}}{4} + \frac{p_{\mathrm{b}}}{8} - \gamma_{\mathrm{wk}} \right) (1 - \kappa_{\mathrm{wk}}) \right] g_i^{(\tau)} \|\boldsymbol{\nu}\|^2$$

$$= \eta g_i^{(\tau)} \left( \frac{2p_{\mathrm{h}} + p_{\mathrm{b}}}{2} \cdot \kappa_{\mathrm{wk}} + 4\gamma_{\mathrm{wk}} \right) \|\boldsymbol{\nu}\|^2$$

$$\leq \frac{\eta(2p_{\mathrm{h}} + p_{\mathrm{b}})g_i^{(\tau)} \|\boldsymbol{\nu}\|^2}{2}$$

$$= \frac{(2p_{\mathrm{h}} + p_{\mathrm{b}})n_{\mathrm{wk}}\|\boldsymbol{\nu}\|^2}{2\|\boldsymbol{\xi}_i\|^2} \left( \rho_i^{(\tau+1)} - \rho_i^{(\tau)} \right),$$

where the inequalities follow from (W5) at iteration $\tau$, (3), and (4), respectively. Hence, we obtain

$$N_s^{(\tau+1)} \leq N_s^{(\tau)} + \frac{(2p_{\mathrm{h}} + p_{\mathrm{b}})n_{\mathrm{wk}}\|\boldsymbol{\nu}\|^2}{2\|\boldsymbol{\xi}_i\|^2} \left( \rho_i^{(\tau+1)} - \rho_i^{(\tau)} \right)$$

$$\leq N_s^{(\tau)} + (2p_{\mathrm{h}} + p_{\mathrm{b}})n_{\mathrm{wk}}\mathrm{SNR}_{\boldsymbol{\nu}}^2 \cdot \left( \rho_i^{(\tau+1)} - \rho_i^{(\tau)} \right)$$

$$\leq (2p_{\mathrm{h}} + p_{\mathrm{b}})n_{\mathrm{wk}}\mathrm{SNR}_{\boldsymbol{\nu}}^2 \cdot \rho_i^{(\tau)} + (2p_{\mathrm{h}} + p_{\mathrm{b}})n_{\mathrm{wk}}\mathrm{SNR}_{\boldsymbol{\nu}}^2 \cdot \left( \rho_i^{(\tau+1)} - \rho_i^{(\tau)} \right)$$

$$= (2p_{\mathrm{h}} + p_{\mathrm{b}})n_{\mathrm{wk}}\mathrm{SNR}_{\boldsymbol{\nu}}^2 \cdot \rho_i^{(\tau+1)},$$

where the second and last inequalities follow from (5) and (W6) at iteration $\tau$, respectively. Similarly, we have

$$N_s^{(\tau+1)} - N_s^{(\tau)}$$

$$= \frac{\eta}{n_{\mathrm{wk}}} \left( \sum_{j \in \mathcal{S}_{\boldsymbol{\nu}_s}^{(1)}} g_j^{(\tau)} + \sum_{j \in \mathcal{S}_{\boldsymbol{\nu}_s}^{(2)}} g_j^{(\tau)} - \sum_{j \in \mathcal{S}_{-\boldsymbol{\nu}_s}^{(1)}} g_j^{(\tau)} - \sum_{j \in \mathcal{S}_{-\boldsymbol{\nu}_s}^{(2)}} g_j^{(\tau)} \right) \|\boldsymbol{\nu}\|^2$$

$$\geq \frac{\eta}{n_{\mathrm{wk}}} \left[ \left( \left| \mathcal{S}_{\boldsymbol{\nu}_s}^{(1)} \right| + \left| \mathcal{S}_{\boldsymbol{\nu}_s}^{(2)} \right| \right) (1 - \kappa_{\mathrm{wk}}) - \left( \left| \mathcal{S}_{-\boldsymbol{\nu}_s}^{(1)} \right| + \left| \mathcal{S}_{-\boldsymbol{\nu}_s}^{(2)} \right| \right) (1 + \kappa_{\mathrm{wk}}) \right] g_i^{(\tau)} \|\boldsymbol{\nu}\|^2$$

$$\geq \eta \left[ 2 \left( \frac{p_{\mathrm{h}}}{4} + \frac{p_{\mathrm{b}}}{8} + \gamma_{\mathrm{wk}} \right) (1 - \kappa_{\mathrm{wk}}) - 2 \left( \frac{p_{\mathrm{h}}}{4} + \frac{p_{\mathrm{b}}}{8} - \gamma_{\mathrm{wk}} \right) (1 + \kappa_{\mathrm{wk}}) \right] g_i^{(\tau)} \|\boldsymbol{\nu}\|^2$$

$$= -\eta g_i^{(\tau)} \left( \frac{2p_{\mathrm{h}} + p_{\mathrm{b}}}{2} \cdot \kappa_{\mathrm{wk}} + 4\gamma_{\mathrm{wk}} \right) \|\boldsymbol{\nu}\|^2$$

$$\geq -\frac{\eta(2p_{\mathrm{h}} + p_{\mathrm{b}})g_i^{(\tau)} \|\boldsymbol{\nu}\|^2}{2}$$

$$= -\frac{(2p_{\mathrm{h}} + p_{\mathrm{b}})n_{\mathrm{wk}}\|\boldsymbol{\nu}\|^2}{2\|\boldsymbol{\xi}_i\|^2} \left( \rho_i^{(\tau+1)} - \rho_i^{(\tau)} \right),$$

where the inequalities follow from (W5) at iteration $\tau$, (5), and (3), respectively. Hence, we obtain

$$N_s^{(\tau+1)} \geq N_s^{(\tau)} - \frac{(2p_{\mathrm{h}} + p_{\mathrm{b}})n_{\mathrm{wk}}\|\boldsymbol{\nu}\|^2}{2\|\boldsymbol{\xi}_i\|^2} \left( \rho_i^{(\tau+1)} - \rho_i^{(\tau)} \right)$$

$$\geq N_s^{(\tau)} - (2p_{\mathrm{h}} + p_{\mathrm{b}})n_{\mathrm{wk}}\mathrm{SNR}_{\boldsymbol{\nu}}^2 \cdot \left( \rho_i^{(\tau+1)} - \rho_i^{(\tau)} \right)$$

$$\geq -(2p_{\mathrm{h}} + p_{\mathrm{b}})n_{\mathrm{wk}}\mathrm{SNR}_{\boldsymbol{\nu}}^2 \cdot \rho_i^{(\tau)} - (2p_{\mathrm{h}} + p_{\mathrm{b}})n_{\mathrm{wk}}\mathrm{SNR}_{\boldsymbol{\nu}}^2 \cdot \left( \rho_i^{(\tau+1)} - \rho_i^{(\tau)} \right)$$

$$= -(2p_{\mathrm{h}} + p_{\mathrm{b}})n_{\mathrm{wk}}\mathrm{SNR}_{\boldsymbol{\nu}}^2 \cdot \rho_i^{(\tau+1)},$$

where the second and last inequalities follow from (4) and (W6) at iteration $\tau$, respectively.

Therefore, the conclusions hold at any iteration $t \in [0, T^*]$. $\qquad\square$

## B.2 Convergence of Training Loss

In this subsection, we prove that the training loss converges below $\varepsilon$ within $\widetilde{\mathcal{O}}\left(\eta^{-1}\varepsilon^{-1}n_{\mathrm{wk}}d^{-1}\sigma_p^{-2}\right)$. All the arguments in this subsection are under Condition 3.2 and the event $E_{\mathrm{wk}}$.

Let us define
$$\hat{\boldsymbol{w}} := 2\log(4/\varepsilon)\sum_{i\in[n_{\mathrm{wk}}]} y_i\boldsymbol{\xi}_i \left\|\boldsymbol{\xi}_i\right\|^{-2},$$

which plays a crucial role in proving convergence.

**Lemma B.2.** *Under Condition 3.2 and the event $E_{\mathrm{wk}}$, we have the following:*

- $\|\hat{\boldsymbol{w}}\| \leq 3\log(4/\varepsilon)n_{\mathrm{wk}}^{\frac{1}{2}}d^{-\frac{1}{2}}\sigma_p^{-1}$.

- $y_i\left\langle\nabla_{\boldsymbol{w}}f_{\mathrm{wk}}\left(\boldsymbol{w}^{(t)}, \boldsymbol{X}_i\right), \hat{\boldsymbol{w}}\right\rangle \geq \log(4/\varepsilon)$ *for any* $t \in [T, T^*]$.

- $\left\|\nabla_{\boldsymbol{w}}L_{\mathrm{wk}}\left(\boldsymbol{w}^{(t)}\right)\right\|^2 \leq 2\sigma_p^2 d \cdot L_{\mathrm{wk}}\left(\boldsymbol{w}^{(t)}\right)$ *for any* $t \in [0, T^*]$.

*Proof of Lemma B.2.* The first statement follows from

$$\|\hat{\boldsymbol{w}}\|^2 = (2\log(4/\varepsilon))^2\left(\sum_{i\in[n_{\mathrm{wk}}]} y_i\boldsymbol{\xi}_i \left\|\boldsymbol{\xi}_i\right\|^{-2}\right)^2$$

$$= 4\log^2(4/\varepsilon)\left(\sum_{i\in[n_{\mathrm{wk}}]}\left\|\boldsymbol{\xi}_i\right\|^{-2} + \sum_{\substack{i,j\in[n_{\mathrm{wk}}]\\i\neq j}} y_iy_j\frac{\langle\boldsymbol{\xi}_i, \boldsymbol{\xi}_j\rangle}{\left\|\boldsymbol{\xi}_i\right\|^2\left\|\boldsymbol{\xi}_j\right\|^2}\right)$$

$$\leq 4\log^2(4/\varepsilon)\left(\sum_{i\in[n_{\mathrm{wk}}]}\left\|\boldsymbol{\xi}_i\right\|^{-2} + \sum_{\substack{i,j\in[n_{\mathrm{wk}}]\\i\neq j}}\frac{|\langle\boldsymbol{\xi}_i, \boldsymbol{\xi}_j\rangle|}{\left\|\boldsymbol{\xi}_i\right\|^2\left\|\boldsymbol{\xi}_j\right\|^2}\right)$$

$$\leq 4\log^2(4/\varepsilon)\left(n_{\mathrm{wk}}\cdot\frac{2}{\sigma_p^2 d} + n_{\mathrm{wk}}^2\cdot\frac{\beta_{\mathrm{wk}}}{n_{\mathrm{wk}}}\cdot\frac{2}{\sigma_p^2 d}\right)$$

$$= 4\log^2(4/\varepsilon)\frac{2n_{\mathrm{wk}}(1 + \beta_{\mathrm{wk}})}{\sigma_p^2 d}$$

$$\leq 9\log^2(4/\varepsilon)\frac{n_{\mathrm{wk}}}{\sigma_p^2 d},$$

where the second inequality follows from (4) and the last inequality follows from (3).

Next, let us prove the second statement. For any $t \in [0, T^*]$, we have

$$y_i\left\langle\nabla_{\boldsymbol{w}}f_{\mathrm{wk}}\left(\boldsymbol{w}^{(t)}, \boldsymbol{X}_i\right), \hat{\boldsymbol{w}}\right\rangle$$

$$= y_i\left\langle\boldsymbol{v}_i^{(1)} + \boldsymbol{v}_i^{(2)} + \boldsymbol{\xi}_i, 2\log(4/\varepsilon)\sum_{j\in[n_{\mathrm{wk}}]} y_j\boldsymbol{\xi}_j\left\|\boldsymbol{\xi}_j\right\|^{-2}\right\rangle$$

$$= 2\log(4/\varepsilon)\sum_{j\in[n_{\mathrm{wk}}]} y_iy_j\frac{\langle\boldsymbol{\xi}_i, \boldsymbol{\xi}_j\rangle}{\left\|\boldsymbol{\xi}_j\right\|^2}$$

$$\geq 2\log(4/\varepsilon) - \sum_{j\in[n_{\mathrm{wk}}]\backslash\{i\}} 2\log(4/\varepsilon)\frac{|\langle\boldsymbol{\xi}_i, \boldsymbol{\xi}_j\rangle|}{\left\|\boldsymbol{\xi}_j\right\|^2}$$

$$\geq 2(1 - \beta_{\text{wk}}) \log(4/\varepsilon)$$
$$\geq \log(4/\varepsilon)$$

where the second inequality follows from (4) and the last inequality follows from (3).

Let us prove the last statement. For any $t \in [0, T^*]$, we have

$$
\left\| \nabla_{\boldsymbol{w}} L_{\text{wk}} \left( \boldsymbol{w}^{(t)} \right) \right\|^2 = \left\| \frac{1}{n_{\text{wk}}} \sum_{i \in [n_{\text{wk}}]} g_i^{(t)} y_i \left( \boldsymbol{v}_i^{(1)} + \boldsymbol{v}_i^{(2)} + \boldsymbol{\xi}_i \right) \right\|^2
$$

$$
\leq \left[ \frac{1}{n_{\text{wk}}} \sum_{i \in [n_{\text{wk}}]} g_i^{(t)} \left\| \boldsymbol{v}_i^{(1)} + \boldsymbol{v}_i^{(2)} + \boldsymbol{\xi}_i \right\| \right]^2
$$

$$
\leq \left[ \frac{1}{n_{\text{wk}}} \sum_{i \in [n_{\text{wk}}]} g_i^{(t)} \right]^2 2\sigma_p^2 d
$$

$$
\leq 2\sigma_p^2 d \cdot \left[ \frac{1}{n_{\text{wk}}} \sum_{i \in [n_{\text{wk}}]} g_i^{(t)} \right]
$$

$$
\leq 2\sigma_p^2 d \cdot \left[ \frac{1}{n_{\text{wk}}} \sum_{i \in [n_{\text{wk}}]} \ell \left( y_i f_{\text{wk}} \left( \boldsymbol{w}, \boldsymbol{X}_i \right) \right) \right]
$$

$$
= 2\sigma_p^2 d \cdot L_{\text{wk}} \left( \boldsymbol{w}^{(t)} \right),
$$

where the first inequality follows from the triangle inequality, the second follows from (4) and the bound $\| \boldsymbol{\mu} \|^2, \| \boldsymbol{\nu} \|^2 \leq \frac{\sigma_p^2 d}{4}$ implied by Condition 3.2, the third follows from $\frac{1}{n_{\text{wk}}} \sum_{i \in [n_{\text{wk}}]} g_i^{(t)} \leq 1$, and the last follows from $-\ell'(z) \leq \ell(z)$ for all $z \in \mathbb{R}$. $\qquad \square$

**Lemma B.3.** *Under Condition 3.1 and the event $E_{\text{wk}}$, for any iteration $T \in [0, T^*]$, we have*

$$
\frac{1}{T} \sum_{t=0}^{T} L_{\text{wk}} \left( \boldsymbol{w}^{(t)} \right) \leq \frac{\| \hat{\boldsymbol{w}} \|^2}{\eta T} + \frac{\varepsilon}{2}.
$$

*Proof of Lemma B.3.* For any $t \in [0, T^*]$, we have

$$
\left\| \boldsymbol{w}^{(t)} - \hat{\boldsymbol{w}} \right\|^2 - \left\| \boldsymbol{w}^{(t+1)} - \hat{\boldsymbol{w}} \right\|^2
$$

$$
= \left\| \boldsymbol{w}^{(t)} - \hat{\boldsymbol{w}} \right\|^2 - \left\| \boldsymbol{w}^{(t)} - \hat{\boldsymbol{w}} - \eta \nabla L_{\text{wk}} \left( \boldsymbol{w}^{(t)} \right) \right\|^2
$$

$$
= 2\eta \left\langle \nabla L_{\text{wk}} \left( \boldsymbol{w}^{(t)} \right), \boldsymbol{w}^{(t)} - \hat{\boldsymbol{w}} \right\rangle - \eta^2 \left\| \nabla L_{\text{wk}} \left( \boldsymbol{w}^{(t)} \right) \right\|^2
$$

$$
= \frac{2\eta}{n_{\text{wk}}} \sum_{i \in [n_{\text{wk}}]} g_i^{(t)} \left( \left\langle y_i \nabla f_{\text{wk}} \left( \boldsymbol{w}^{(t)}, \boldsymbol{X}_i \right), \hat{\boldsymbol{w}} \right\rangle - y_i f_{\text{wk}} \left( \boldsymbol{w}^{(t)}, \boldsymbol{X}_i \right) \right) - \eta^2 \left\| \nabla L_{\text{wk}} \left( \boldsymbol{w}^{(t)} \right) \right\|^2
$$

$$
\geq \frac{2\eta}{n_{\text{wk}}} \sum_{i \in [n_{\text{wk}}]} g_i^{(t)} \left( \log(4/\varepsilon) - y_i f_{\text{wk}} \left( \boldsymbol{w}^{(t)}, \boldsymbol{X}_i \right) \right) - \eta^2 \left\| \nabla L_{\text{wk}} \left( \boldsymbol{w}^{(t)} \right) \right\|^2
$$

$$
\geq \frac{2\eta}{n_{\text{wk}}} \sum_{i \in [n_{\text{wk}}]} \left[ \ell \left( y_i f \left( \boldsymbol{w}^{(t)}, \boldsymbol{X}_i \right) \right) - \frac{\varepsilon}{4} \right] - \eta^2 \left\| \nabla L_{\text{wk}} \left( \boldsymbol{w}^{(t)} \right) \right\|^2
$$

$$
\geq \eta L_{\text{wk}} \left( \boldsymbol{w}^{(t)} \right) - \frac{\eta \varepsilon}{2},
$$

where the first inequality follows from Lemma B.2, the second follows from the convexity of $\ell$ and the bound $\ell(\log(4/\varepsilon)) \geq \varepsilon/4$, and the last follows from Lemma B.2 and (C4).

By applying a telescoping sum and using the fact that $\boldsymbol{w}^{(0)} = 0$, we obtain the desired conclusion. $\quad \square$

Using lemmas above, we can prove that the training loss converges to below $\varepsilon$. By applying Lemma B.3 with iteration $\tilde{T} = \lceil 18\eta^{-1}\varepsilon^{-1}\log(4/\varepsilon)n_{\mathrm{wk}}d^{-1}\sigma_p^{-2}\rceil = \widetilde{\mathcal{O}}(\eta^{-1}\varepsilon^{-1}n_{\mathrm{wk}}d^{-1}\sigma_p^{-2})$ and using Lemma B.2, we obtain

$$\frac{1}{\tilde{T}}\sum_{t=0}^{\tilde{T}}L_{\mathrm{wk}}\left(\boldsymbol{w}^{(t)}\right) \leq \frac{\|\hat{\boldsymbol{w}}\|^2}{\eta\tilde{T}} + \frac{\varepsilon}{2} \leq \frac{9\log^2(4/\varepsilon)n_{\mathrm{wk}}d^{-1}\sigma_p^{-2}}{\eta\tilde{T}} + \frac{\varepsilon}{2} \leq \varepsilon.$$

Therefore, there exists $T_{\mathrm{wk}} \in [0, \tilde{T}]$ such that $L_{\mathrm{wk}}(\boldsymbol{w}^{(T_{\mathrm{wk}})}) \leq \varepsilon$. In addition, for any $\boldsymbol{w}_1, \boldsymbol{w}_2 \in \mathbb{R}^d$, we have

$$\|\nabla_{\boldsymbol{w}}L_{\mathrm{wk}}(\boldsymbol{w}_1) - \nabla_{\boldsymbol{w}}L_{\mathrm{wk}}(\boldsymbol{w}_2)\|$$

$$= \frac{1}{n_{\mathrm{wk}}}\left\|\sum_{i\in[n_{\mathrm{wk}}]}\left[y_i(\ell'\big(y_if_{\mathrm{wk}}(\boldsymbol{w}_1, \boldsymbol{X}_i)\big) - \ell'\big(y_if_{\mathrm{wk}}(\boldsymbol{w}_2, \boldsymbol{X}_i)\big)\big)\big(\boldsymbol{v}_i^{(1)} + \boldsymbol{v}_i^{(2)} + \boldsymbol{\xi}_i\big)\right]\right\|$$

$$\leq \frac{1}{n_{\mathrm{wk}}}\sum_{i\in[n_{\mathrm{wk}}]}\left[\big|\ell'\big(y_if_{\mathrm{wk}}(\boldsymbol{w}_1, \boldsymbol{X}_i)\big) - \ell'\big(y_if_{\mathrm{wk}}(\boldsymbol{w}_2, \boldsymbol{X}_i)\big)\big| \cdot \left\|\boldsymbol{v}_i^{(1)} + \boldsymbol{v}_i^{(2)} + \boldsymbol{\xi}_i\right\|\right]$$

$$\leq \frac{\sqrt{2}\sigma_p d^{\frac{1}{2}}}{2n_{\mathrm{wk}}}\sum_{i\in[n_{\mathrm{wk}}]}\big|\big(f_{\mathrm{wk}}(\boldsymbol{w}_1, \boldsymbol{X}_i) - f_{\mathrm{wk}}(\boldsymbol{w}_2, \boldsymbol{X}_i)\big)\big|$$

$$\leq \frac{\sqrt{2}\sigma_p d^{\frac{1}{2}}}{2n_{\mathrm{wk}}}\sum_{i\in[n_{\mathrm{wk}}]}\left\|\boldsymbol{v}_i^{(1)} + \boldsymbol{v}_i^{(2)} + \boldsymbol{\xi}_i\right\| \cdot \|\boldsymbol{w}_1 - \boldsymbol{w}_2\|$$

$$\leq \sigma_p^2 d\|\boldsymbol{w}_1 - \boldsymbol{w}_2\|,$$

where the first and third inequalities follow from the Cauchy-Schwarz inequality, the second and last inequalities follow from (4) and the bound $\|\boldsymbol{\mu}\|^2, \|\boldsymbol{\nu}\|^2 \leq \frac{\sigma_p^2 d}{4}$ implied by Condition 3.2, and for the second inequality, we also use the fact that $0 \leq \ell' \leq \frac{1}{4}$.

Since $L_{\mathrm{wk}}(\boldsymbol{w})$ is $\sigma_p^2 d$-smooth and the learning rate satisfies (10), we can apply the descent lemma (Lemma 3.4 in Bubeck (2015)). This proves the first part of our conclusion.

$\square$

## B.3 Test Error

In this subsection, we prove the second part of our conclusion. All the arguments in this subsection are under Condition 3.2 and the event $E_{\mathrm{wk}}$.

Define $\boldsymbol{v}^{(1)}$, $\boldsymbol{v}^{(2)}$, and $\boldsymbol{\xi}$ as the signal vectors and the noise vector in the test data $(\boldsymbol{X}, y)$, respectively.

For any iteration $t \in [T_{\mathrm{wk}}, T^*]$ and for the case given $(\boldsymbol{X}, y) \in \mathcal{S}_{\mathrm{e}} \cup \mathcal{S}_{\mathrm{b}}$, we can express the test accuracy as

$$\mathbb{P}\left[yf_{\mathrm{wk}}\left(\boldsymbol{w}^{(t)}, \boldsymbol{X}\right) < 0 \,\middle|\, (\boldsymbol{X}, y) \in \mathcal{S}_{\mathrm{e}} \cup \mathcal{S}_{\mathrm{b}}\right]$$

$$= \mathbb{P}\left[\left\langle y\boldsymbol{w}^{(t)}, \boldsymbol{\xi}\right\rangle < -\left\langle y\boldsymbol{w}^{(t)}, \boldsymbol{v}^{(1)}\right\rangle - \left\langle y\boldsymbol{w}^{(t)}, \boldsymbol{v}^{(2)}\right\rangle \,\middle|\, (\boldsymbol{X}, y) \in \mathcal{S}_{\mathrm{e}} \cup \mathcal{S}_{\mathrm{b}}\right]$$

$$\leq \mathbb{P}\left[\left\langle y\boldsymbol{w}^{(t)}, \boldsymbol{\xi}\right\rangle < -\frac{M_y^{(t)}}{2}\right]$$

$$= \mathbb{P}\left[\mathbf{z} < -\frac{M_y^{(t)}}{2}\right],$$

where $\mathbf{z} \sim \mathcal{N}\left(0, \sigma_p^2\left\|\Pi_S\boldsymbol{w}^{(t)}\right\|^2\right)$, and the inequality follows from (11). By Höeffding's inequality, we have

$$\mathbb{P}\left[yf_{\mathrm{wk}}\left(\boldsymbol{w}^{(t)}, \boldsymbol{X}\right) < 0 \,\middle|\, (\boldsymbol{X}, y) \in \mathcal{S}_{\mathrm{e}} \cup \mathcal{S}_{\mathrm{b}}\right] \leq \exp\left(-\frac{\left(M_y^{(t)}\right)^2}{8\sigma_p^2\left\|\Pi_S\boldsymbol{w}^{(t)}\right\|^2}\right).$$

Let us characterize $\left\| \Pi_S \boldsymbol{w}^{(t)} \right\|^2$. We have

$$
\begin{aligned}
\left\| \Pi_S \boldsymbol{w}^{(t)} \right\|^2 &= \left\| \sum_{i \in [n_{\mathrm{wk}}]} y_i \rho_i^{(t)} \boldsymbol{\xi}_i \left\| \boldsymbol{\xi}_i \right\|^{-2} \right\|^2 \\
&\leq \sum_{i \in [n_{\mathrm{wk}}]} \left( \rho_i^{(t)} \right)^2 \left\| \boldsymbol{\xi}_i \right\|^{-2} + \sum_{i \in [n_{\mathrm{wk}}]} \sum_{j \in [n_{\mathrm{wk}}] \setminus \{i\}} \rho_i^{(t)} \rho_j^{(t)} \frac{|\langle \boldsymbol{\xi}_i, \boldsymbol{\xi}_j \rangle|}{\left\| \boldsymbol{\xi}_i \right\|^2 \left\| \boldsymbol{\xi}_j \right\|^2} \\
&\leq \frac{2}{\sigma_p^2 d} \sum_{i \in [n_{\mathrm{wk}}]} \left( \rho_i^{(t)} \right)^2 + \sum_{i \in [n_{\mathrm{wk}}]} \sum_{j \in [n_{\mathrm{wk}}]} \frac{\left( \rho_i^{(t)} \right)^2 + \left( \rho_j^{(t)} \right)^2}{2} \cdot \frac{\beta_{\mathrm{wk}}}{n_{\mathrm{wk}}} \cdot \frac{2}{\sigma_p^2 d} \\
&\leq \frac{4}{\sigma_p^2 d} \sum_{i \in [n_{\mathrm{wk}}]} \left( \rho_i^{(t)} \right)^2 \\
&\leq \frac{4}{\sigma_p^2 d} n_{\mathrm{wk}} \left( \frac{12}{n_{\mathrm{wk}} (2 p_{\mathrm{e}} + p_{\mathrm{b}}) \cdot \mathrm{SNR}_{\boldsymbol{\mu}}^2} \right)^2 \left( M_y^{(t)} \right)^2 \\
&= \frac{576 \sigma_p^2 d}{n_{\mathrm{wk}} (2 p_{\mathrm{e}} + p_{\mathrm{b}})^2 \left\| \boldsymbol{\mu} \right\|^4} \left( M_y^{(t)} \right)^2,
\end{aligned}
$$

where the first and second inequality follows from AM-GM inequality and (4), the third inequality follows from (3), and the last inequality follows from (W2). Hence, we have

$$
\mathbb{P} \left[ y f_{\mathrm{wk}} \left( \boldsymbol{w}^{(t)}, \boldsymbol{X} \right) < 0 \,\Big|\, (\boldsymbol{X}, y) \in \mathcal{S}_{\mathrm{e}} \cup \mathcal{S}_{\mathrm{b}} \right] \leq \exp \left( -\frac{n_{\mathrm{wk}} (2 p_{\mathrm{e}} + p_{\mathrm{b}})^2 \left\| \boldsymbol{\mu} \right\|^4}{4608 \sigma_p^4 d} \right).
$$

# C  Proof of Theorem 3.4

It suffices to prove the following restatement of Theorem 3.4.

**Theorem C.1** (Weak-to-Strong Training, Data-Scarce Regime). *Let $\boldsymbol{W}^{(t)}$ be the iterates of weak-to-strong training, with the weak model $f_{\mathrm{wk}}(\boldsymbol{w}^*, \cdot)$ satisfying the conclusion of Theorem 3.3. For any $\varepsilon > 0$ and $\delta \in (0,1)$ satisfying Condition 3.2, with probability at least $1 - \delta$, there exists $T_{\mathrm{w2s}} = \mathcal{O}(\eta^{-1}\varepsilon^{-1}mn_{\mathrm{st}}d^{-1}\sigma_p^{-2})$ such that for any $t \in [T_{\mathrm{w2s}}, T^*]$ the following statements hold:*

1. *The training loss converges below $\varepsilon$: $L_{\mathrm{st}}\left(\boldsymbol{W}^{(t)}\right) < \varepsilon$.*

2. *Let $(\boldsymbol{X}, y) \sim \mathcal{D}$ be an unseen test example, independent of the training set $\{(\tilde{\boldsymbol{X}}_i, \hat{y}_i)\}_{i=1}^{n_{\mathrm{st}}}$.*

   - *(Benign Overfitting) When $n_{\mathrm{st}}p_{\mathrm{b}}^2 \|\boldsymbol{\nu}\|^4/(\sigma_p^4 d) \geq C_2$, we have*

$$
\mathbb{P}\left[yf_{\mathrm{st}}\left(\boldsymbol{W}^{(t)}, \boldsymbol{X}\right) < 0 \,\middle|\, (\boldsymbol{X}, y) \in \mathcal{S}_{\mathrm{e}} \cup \mathcal{S}_{\mathrm{b}}\right] \leq \exp\left(-\frac{n_{\mathrm{st}}(2p_{\mathrm{e}} + p_{\mathrm{b}})^2\|\boldsymbol{\mu}\|^4}{C_3 \sigma_p^4 d}\right),
$$

   *and*

$$
\mathbb{P}\left[yf_{\mathrm{st}}\left(\boldsymbol{W}^{(t)}, \boldsymbol{X}\right) < 0 \,\middle|\, (\boldsymbol{X}, y) \in \mathcal{S}_{\mathrm{h}}\right] \leq \exp\left(-\frac{n_{\mathrm{st}}p_{\mathrm{b}}^2\|\boldsymbol{\nu}\|^4}{C_3 \sigma_p^4 d}\right).
$$

   - *(Harmful Overfitting) When $n_{\mathrm{st}}p_{\mathrm{b}}^2\|\boldsymbol{\nu}\|^4/(\sigma_p^4 d) \leq C_4$,*

$$
\mathbb{P}\left[yf_{\mathrm{st}}\left(\boldsymbol{W}^{(t)}, \boldsymbol{X}\right) < 0\right] \geq 0.12p_{\mathrm{h}}.
$$

   *Here, $C_2, C_3, C_4 > 0$ are constants.*

For the proof, we first introduce properties preserved during training (Appendix C.1), then prove the convergence of the training loss (Appendix C.2), and finally establish a bound on the test error (Appendix C.3).

## C.1  Preserved Properties during Training

In this subsection, we present several properties that remain preserved throughout training.

**Lemma C.2.** *Suppose for some iteration $t \in [0, T^*]$, it satisfies $\left|M_{s,r}^{(t)}\right|, \left|N_{s,r}^{(t)}\right| \leq \alpha_{\mathrm{st}} + \beta_{\mathrm{st}}$, $0 \leq \overline{\rho}_{r,i}^{(t)} \leq 4\log T^*$, and $-\alpha_{\mathrm{st}} - 5\beta_{\mathrm{st}}\log T^* \leq \underline{\rho}_{r,i}^{(t)} \leq 0$ for any $s \in \{\pm 1\}, r \in [m]$, and $i \in [n_{\mathrm{st}}]$. Then, for any $i \in [n_{\mathrm{st}}]$ it holds that*

$$
F_{-\hat{y}_i}\left(\boldsymbol{W}_{-\hat{y}_i}^{(t)}, \tilde{\boldsymbol{X}}_i\right) \leq \frac{\kappa_{\mathrm{st}}}{16}, \quad \left|\sigma\left(\left\langle \boldsymbol{w}_{\hat{y}_i,r}^{(t)}, \tilde{\boldsymbol{\xi}}_i \right\rangle\right) - \overline{\rho}_{r,i}^{(t)}\right| \leq \frac{\kappa_{\mathrm{st}}}{16}.
$$

*Proof of Lemma C.2.* For any $i \in [n_{\mathrm{st}}]$, we have

$$
\begin{aligned}
&F_{-\hat{y}_i}\left(\boldsymbol{W}_{-\hat{y}_i}^{(t)}, \tilde{\boldsymbol{X}}_i\right) \\
&= \frac{1}{m}\sum_{r \in [m]}\left[\sigma\left(\left\langle \boldsymbol{w}_{-\hat{y}_i,r}^{(t)}, \tilde{\boldsymbol{v}}_i^{(1)} \right\rangle\right) + \sigma\left(\left\langle \boldsymbol{w}_{-\hat{y}_i,r}^{(t)}, \tilde{\boldsymbol{v}}_i^{(2)} \right\rangle\right) + \sigma\left(\left\langle \boldsymbol{w}_{-\hat{y}_i,r}^{(t)}, \tilde{\boldsymbol{\xi}}_i \right\rangle\right)\right] \\
&\leq \frac{1}{m}\sum_{r \in [m]}\left[\left|\left\langle \boldsymbol{w}_{-\hat{y}_i,r}^{(t)}, \tilde{\boldsymbol{v}}_i^{(1)} \right\rangle\right| + \left|\left\langle \boldsymbol{w}_{-\hat{y}_i,r}^{(t)}, \tilde{\boldsymbol{v}}_i^{(2)} \right\rangle\right| + \left|\left\langle \boldsymbol{w}_{-\hat{y}_i,r}^{(t)}, \tilde{\boldsymbol{\xi}}_i \right\rangle\right|\right] \\
&\leq \frac{1}{m}\sum_{r \in [m]}\left[\left|\left\langle \boldsymbol{w}_{-\hat{y}_i,r}^{(0)}, \tilde{\boldsymbol{v}}_i^{(1)} \right\rangle\right| + \left|\left\langle \boldsymbol{w}_{-\hat{y}_i,r}^{(0)}, \tilde{\boldsymbol{v}}_i^{(2)} \right\rangle\right| + 2 \cdot (\alpha_{\mathrm{st}} + \beta_{\mathrm{st}}) + \left|\left\langle \boldsymbol{w}_{-\hat{y}_i,r}^{(t)}, \tilde{\boldsymbol{\xi}}_i \right\rangle\right|\right] \\
&\leq (4\alpha_{\mathrm{st}} + 2\beta_{\mathrm{st}}) + \frac{1}{m}\sum_{r \in [m]}\left|\left\langle \boldsymbol{w}_{-\hat{y}_i,r}^{(t)}, \tilde{\boldsymbol{\xi}}_i \right\rangle\right|,
\end{aligned}
$$

where the last two inequalities follow from the given bounds on $\left|M_{s,r}^{(t)}\right|, \left|N_{s,r}^{(t)}\right|$ and (7). In addition, for any $r \in [m]$, we have

$$
\begin{aligned}
\left\langle \boldsymbol{w}_{-\hat{y}_i,r}^{(t)}, \tilde{\boldsymbol{\xi}}_i \right\rangle &= \left\langle \boldsymbol{w}_{-\hat{y}_i,r}^{(0)}, \tilde{\boldsymbol{\xi}}_i \right\rangle + \underline{\rho}_{r,i}^{(t)} + \sum_{j \in [n_{\mathrm{st}}] \setminus \{i\}} \rho_{-\hat{y}_i,r,j}^{(t)} \frac{\langle \tilde{\boldsymbol{\xi}}_i, \tilde{\boldsymbol{\xi}}_j \rangle}{\|\tilde{\boldsymbol{\xi}}_j\|^2} \\
&\geq \left\langle \boldsymbol{w}_{-\hat{y}_i,r}^{(0)}, \tilde{\boldsymbol{\xi}}_i \right\rangle + \underline{\rho}_{r,i}^{(t)} - \sum_{j \in [n_{\mathrm{st}}] \setminus \{i\}} \left|\rho_{-\hat{y}_i,r,j}^{(t)}\right| \frac{|\langle \tilde{\boldsymbol{\xi}}_i, \tilde{\boldsymbol{\xi}}_j \rangle|}{\|\tilde{\boldsymbol{\xi}}_j\|^2} \\
&\geq -2\alpha_{\mathrm{st}} - 9\beta_{\mathrm{st}} \log T^*,
\end{aligned}
$$

where the last inequality follows from the given bound on $\overline{\rho}_{r,i}^{(t)}, \underline{\rho}_{r,i}^{(t)}$, (7), and (8). Similarly, for any $r \in [m]$, we have

$$
\begin{aligned}
\left\langle \boldsymbol{w}_{-\hat{y}_i,r}^{(t)}, \tilde{\boldsymbol{\xi}}_i \right\rangle &= \left\langle \boldsymbol{w}_{-\hat{y}_i,r}^{(0)}, \tilde{\boldsymbol{\xi}}_i \right\rangle + \underline{\rho}_{r,i}^{(t)} + \sum_{j \in [n_{\mathrm{st}}] \setminus \{i\}} \rho_{-\hat{y}_i,r,j}^{(t)} \frac{\langle \tilde{\boldsymbol{\xi}}_i, \tilde{\boldsymbol{\xi}}_j \rangle}{\|\tilde{\boldsymbol{\xi}}_j\|^2} \\
&\leq \left\langle \boldsymbol{w}_{-\hat{y}_i,r}^{(0)}, \tilde{\boldsymbol{\xi}}_i \right\rangle + \underline{\rho}_{r,i}^{(t)} + \sum_{j \in [n_{\mathrm{st}}] \setminus \{i\}} \left|\rho_{-\hat{y}_i,r,j}^{(t)}\right| \frac{|\langle \tilde{\boldsymbol{\xi}}_i, \tilde{\boldsymbol{\xi}}_j \rangle|}{\|\tilde{\boldsymbol{\xi}}_j\|^2} \\
&\leq \alpha_{\mathrm{st}} + 4\beta_{\mathrm{st}} \log T^*,
\end{aligned}
$$

where the last inequality follows from the given bound on $\overline{\rho}_{r,i}^{(t)}, \underline{\rho}_{r,i}^{(t)}$, (7), and (8). Hence, we have

$$
F_{-\hat{y}_i}\left(\boldsymbol{W}_{-\hat{y}_i}^{(t)}, \tilde{\boldsymbol{X}}_i\right) \leq 6\alpha_{\mathrm{st}} + 2\beta_{\mathrm{st}} + 9\beta_{\mathrm{st}} \log T^* \leq \frac{\kappa_{\mathrm{st}}}{16},
$$

where the last inequality follows from (6).

Next, we prove the second part. For any $i \in [n_{\mathrm{st}}]$ and $r \in [m]$, we have

$$
\begin{aligned}
\left|\sigma\left(\left\langle \boldsymbol{w}_{\hat{y}_i,r}^{(t)}, \tilde{\boldsymbol{\xi}}_i \right\rangle\right) - \overline{\rho}_{r,i}^{(t)}\right| &= \left|\sigma\left(\left\langle \boldsymbol{w}_{\hat{y}_i,r}^{(t)}, \tilde{\boldsymbol{\xi}}_i \right\rangle\right) - \sigma\left(\overline{\rho}_{r,i}^{(t)}\right)\right| \\
&\leq \left|\left\langle \boldsymbol{w}_{\hat{y}_i,r}^{(t)}, \tilde{\boldsymbol{\xi}}_i \right\rangle - \overline{\rho}_{r,i}^{(t)}\right| \\
&\leq \left\langle \boldsymbol{w}_{\hat{y}_i,r}^{(0)}, \tilde{\boldsymbol{\xi}}_i \right\rangle + \sum_{j \in [n_{\mathrm{st}}] \setminus \{i\}} \left|\rho_{\hat{y}_i,r,j}^{(t)}\right| \frac{|\langle \tilde{\boldsymbol{\xi}}_i, \tilde{\boldsymbol{\xi}}_j \rangle|}{\|\tilde{\boldsymbol{\xi}}_i\|^2} \\
&\leq \alpha_{\mathrm{st}} + 4\beta_{\mathrm{st}} \log T^* \\
&\leq \frac{\kappa_{\mathrm{st}}}{16},
\end{aligned}
$$

where the third inequality follows from the given bound on $\overline{\rho}_{r,i}^{(t)}, \underline{\rho}_{r,i}^{(t)}$, (7), and (8). □

**Lemma C.3.** *Under Condition 3.2 and the event $E_{\mathrm{st}}$, we have the following for any iteration $t \in [0, T^*]$:*

*(S1)* $-\alpha_{\mathrm{st}} - 5\beta_{\mathrm{st}} \log T^* \leq \underline{\rho}_{r,i}^{(t)} \leq 0$ *and* $0 \leq \overline{\rho}_{r,i}^{(t)} \leq 4\log T^*$ *for any* $i \in [n_{\mathrm{st}}]$ *and* $r \in [m]$.

*(S2) If $t \geq 1$, then for any $s \in \{\pm 1\}$, we have $\overline{M}_{s,r}^{(t)} \geq \overline{M}_{s,r}^{(t-1)}$ for all $r \in [m]$, $\overline{N}_{s,r}^{(t)} \geq \overline{N}_{s,r}^{(t-1)}$ for all $r \in \mathcal{A}_s$, and $\overline{N}_{s,r}^{(t)} \leq \overline{N}_{s,r}^{(t-1)}$ for all $r \in \mathcal{B}_s$. In addition, $\left|M_{s,r}^{(t)}\right|, \left|N_{s,r}^{(t)}\right| \leq \alpha_{\mathrm{st}} + \beta_{\mathrm{st}}$ for all $r \in [m]$.*

*(S3) For any $s \in \{\pm 1\}$ and $i \in [n_{\mathrm{st}}]$, we have*

$$
\frac{n_{\boldsymbol{\mu}} \mathrm{SNR}_{\boldsymbol{\mu}}^2}{12\lambda_{\mathrm{st}}} \cdot \sum_{r \in [m]} \overline{\rho}_{r,i}^{(t)} \leq \sum_{r \in [m]} \overline{M}_{s,r}^{(t)} \leq 6\lambda_{\mathrm{st}} n_{\boldsymbol{\mu}} \mathrm{SNR}_{\boldsymbol{\mu}}^2 \cdot \sum_{r \in [m]} \overline{\rho}_{r,i}^{(t)}
$$

$$
\frac{n_{\boldsymbol{\nu}} \mathrm{SNR}_{\boldsymbol{\nu}}^2}{12\lambda_{\mathrm{st}}} \cdot \sum_{r \in [m]} \overline{\rho}_{r,i}^{(t)} \leq \sum_{r \in \mathcal{A}_s} \overline{N}_{s,r}^{(t)} \leq 6\lambda_{\mathrm{st}} n_{\boldsymbol{\nu}} \mathrm{SNR}_{\boldsymbol{\nu}}^2 \cdot \sum_{r \in [m]} \overline{\rho}_{r,i}^{(t)}
$$

$$\frac{n_{\boldsymbol{\nu}}\mathrm{SNR}_{\boldsymbol{\nu}}^2}{12\lambda_{\mathrm{st}}} \cdot \sum_{r\in[m]} \overline{\rho}_{r,i}^{(t)} \leq - \sum_{r\in\mathcal{B}_s} \overline{N}_{s,r}^{(t)} \leq 6\lambda_{\mathrm{st}}n_{\boldsymbol{\nu}}\mathrm{SNR}_{\boldsymbol{\nu}}^2 \cdot \sum_{r\in[m]} \overline{\rho}_{r,i}^{(t)}.$$

(S4) $\left| \hat{y}_i f_{\mathrm{st}}\left(\boldsymbol{W}^{(t)}, \tilde{\boldsymbol{X}}_i\right) - \frac{1}{m}\sum_{r\in[m]}\overline{\rho}_{r,i}^{(t)} \right| \leq \frac{\kappa_{\mathrm{st}}}{4}$ *for any* $i \in [n_{\mathrm{st}}]$.

(S5) $\frac{1}{m}\left| \sum_{r\in[m]}\overline{\rho}_{r,i}^{(t)} - \sum_{r\in[m]}\overline{\rho}_{r,j}^{(t)} \right| \leq \kappa_{\mathrm{st}}$ *for any* $i, j \in [n_{\mathrm{st}}]$.

(S6) $\frac{\tilde{g}_j^{(t)}}{\tilde{g}_i^{(t)}} \leq \lambda_{\mathrm{st}}$ *for any* $i, j \in [n_{\mathrm{st}}]$.

(S7) *For any* $i \in [n_{\mathrm{st}}]$ *and* $r \in [m]$, $\left\langle \boldsymbol{w}_{\hat{y}_i,r}^{(t)}, \tilde{\boldsymbol{\xi}}_i \right\rangle > 0$ *if* $\left\langle \boldsymbol{w}_{\hat{y}_i,r}^{(0)}, \tilde{\boldsymbol{\xi}}_i \right\rangle > 0$. *Furthermore, for any* $i \in [n_{\mathrm{st}}]$ *and* $r \in \mathcal{X}_i$, $\overline{\rho}_{r,i}^{(t)} = \max_{r'\in[m]}\overline{\rho}_{r',i}^{(t)}$.

(S8) *Let* $x_t$ *be the unique solution of*

$$x_t + \exp(x_t + \kappa_{\mathrm{st}}/16) = \frac{\eta\sigma_p^2 d}{8mn_{\mathrm{st}}}t + \exp(\kappa_{\mathrm{st}}/4).$$

*It holds that for any* $i \in [n_{\mathrm{st}}]$,

$$x_t \leq \frac{1}{m}\sum_{r\in[m]}\overline{\rho}_{r,i}^{(t)}.$$

*Proof of Lemma C.3.* It is trivial for the case $t = 0$. Assume the conclusions hold at iteration $t \leq \tau$ and we will prove for the case $t = \tau + 1$.

(S1): We fix arbitrary $i \in [n_{\mathrm{st}}]$ and $r \in [m]$.

Let us prove the first statement. If $\underline{\rho}_{r,i}^{(\tau)} \geq -\alpha_{\mathrm{st}} - 4\beta_{\mathrm{st}}\log T^*$, then we have

$$\underline{\rho}_{r,i}^{(\tau+1)} = \underline{\rho}_{r,i}^{(\tau)} - \frac{\eta}{mn_{\mathrm{st}}}\tilde{g}_i^{(\tau)}\|\tilde{\boldsymbol{\xi}}_i\|^2 \geq -\alpha_{\mathrm{st}} - 4\beta_{\mathrm{st}}\log T^* - \frac{3\eta\sigma_p^2 d}{2mn_{\mathrm{st}}} \geq -\alpha_{\mathrm{st}} - 5\beta_{\mathrm{st}}\log T^*,$$

where the first inequality follows from (8) and the second inequality follows from (10). Otherwise, we have

$$\left\langle \boldsymbol{w}_{-\hat{y}_i,r}^{(\tau)}, \tilde{\boldsymbol{\xi}}_i \right\rangle = \left\langle \boldsymbol{w}_{-\hat{y}_i,r}^{(0)}, \tilde{\boldsymbol{\xi}}_i \right\rangle + \underline{\rho}_{r,i}^{(\tau)} + \sum_{j\in[n_{\mathrm{st}}]\setminus\{i\}} \rho_{-\hat{y}_i,r,j}^{(\tau)}\frac{\langle \tilde{\boldsymbol{\xi}}_i, \tilde{\boldsymbol{\xi}}_j \rangle}{\|\tilde{\boldsymbol{\xi}}_j\|^2}$$

$$\leq \alpha_{\mathrm{st}} + (-\alpha_{\mathrm{st}} - 4\beta_{\mathrm{st}}\log T^*) + \sum_{j\in[n_{\mathrm{st}}]\setminus\{i\}} \left| \rho_{-\hat{y}_i,r,j}^{(\tau)} \right| \frac{\left| \langle \tilde{\boldsymbol{\xi}}_i, \tilde{\boldsymbol{\xi}}_j \rangle \right|}{\|\tilde{\boldsymbol{\xi}}_j\|^2}$$

$$\leq -4\beta_{\mathrm{st}}\log T^* + n_{\mathrm{st}} \cdot 4\log T^* \cdot \frac{\beta_{\mathrm{st}}}{n_{\mathrm{st}}}$$

$$= 0.$$

It implies $\underline{\rho}_{r,i}^{(\tau+1)} = \underline{\rho}_{r,i}^{(\tau)} \geq -\alpha_{\mathrm{st}} - 5\beta_{\mathrm{st}}\log T^*$ and we have desired conclusion.

Next, we prove the second statement. If $\overline{\rho}_{r,i}^{(\tau)} < 3\log T^*$, then we have

$$\overline{\rho}_{r,i}^{(\tau+1)} \leq \overline{\rho}_{r,i}^{(\tau)} + \frac{\eta}{mn_{\mathrm{st}}}\tilde{g}_i^{(\tau)}\|\tilde{\boldsymbol{\xi}}_i\|^2 \leq 3\log T^* + \frac{3\eta\sigma_p^2 d}{2mn_{\mathrm{st}}} \leq 4\log T^*,$$

where the second inequality follows from (8) and the third inequality follows from (10). Otherwise, there exists $\hat{t} < \tau$ such that $\overline{\rho}_{r,i}^{(\hat{t})} \leq 3\log T^* < \overline{\rho}_{r,i}^{(\hat{t}+1)}$. Then, we have

$$\overline{\rho}_{r,i}^{(\tau+1)} = \overline{\rho}_{r,i}^{(\hat{t})} + \left( \overline{\rho}_{r,i}^{(\hat{t}+1)} - \overline{\rho}_{r,i}^{(\hat{t})} \right) + \sum_{t=\hat{t}+1}^{\tau} \left( \overline{\rho}_{r,i}^{(t+1)} - \overline{\rho}_{r,i}^{(t)} \right)$$

$$\leq 3 \log T^* + \frac{\eta}{mn_{\mathrm{st}}} \tilde{g}_i^{(\hat{t})} \|\tilde{\boldsymbol{\xi}}_i\|^2 + \frac{\eta \|\tilde{\boldsymbol{\xi}}_i\|^2}{mn_{\mathrm{st}}} \sum_{t=\hat{t}+1}^{\tau} \tilde{g}_i^{(t)}$$

$$\leq 3 \log T^* + \frac{\log T^*}{2} + \frac{3\eta \sigma_p^2 d}{2mn_{\mathrm{st}}} \sum_{t=\hat{t}+1}^{\tau} \frac{1}{1 + \exp\left(F_{\hat{y}_i}\left(\boldsymbol{W}_{\hat{y}_i}^{(t)}, \tilde{\boldsymbol{X}}_i\right) - F_{-\hat{y}_i}\left(\boldsymbol{W}_{-\hat{y}_i}^{(t)}, \tilde{\boldsymbol{X}}_i\right)\right)}$$

$$\leq \frac{7}{2} \log T^* + \frac{3\eta \sigma_p^2 d}{2mn_{\mathrm{st}}} \sum_{t=\hat{t}+1}^{\tau} \exp\left(-F_{\hat{y}_i}\left(\boldsymbol{W}_{\hat{y}_i}^{(t)}, \tilde{\boldsymbol{X}}_i\right) + F_{-\hat{y}_i}\left(\boldsymbol{W}_{-\hat{y}_i}^{(t)}, \tilde{\boldsymbol{X}}_i\right)\right)$$

$$\leq \frac{7}{2} \log T^* + \frac{3\eta \sigma_p^2 d}{2mn_{\mathrm{st}}} \sum_{t=\hat{t}+1}^{\tau} \exp\left(-F_{\hat{y}_i}\left(\boldsymbol{W}_{\hat{y}_i}^{(t)}, \tilde{\boldsymbol{X}}_i\right) + \frac{\kappa_{\mathrm{st}}}{16}\right),$$

where the second inequality follows from (10) and (8) and the last inequality follows from Lemma C.2. For any $t = \hat{t} + 1, \cdots, \tau$ and $r' \in \mathcal{X}_i$, by applying (S7) with iteration $t$, we have

$$\left\langle \boldsymbol{w}_{\hat{y}_i, r'}^{(t)}, \tilde{\boldsymbol{\xi}}_i \right\rangle = \left\langle \boldsymbol{w}_{\hat{y}_i, r'}^{(0)}, \tilde{\boldsymbol{\xi}}_i \right\rangle + \overline{\rho}_{r', i}^{(t)} + \sum_{j \in [n_{\mathrm{st}}] \setminus \{i\}} \rho_{\hat{y}_i, r, j}^{(t)} \cdot \frac{\langle \tilde{\boldsymbol{\xi}}_i, \tilde{\boldsymbol{\xi}}_j \rangle}{\|\tilde{\boldsymbol{\xi}}_j\|^2}$$

$$\geq \overline{\rho}_{r, i}^{(t)} - \alpha_{\mathrm{st}} - 4\beta_{\mathrm{st}} \log T^*$$

$$\geq 3 \log T^* - \alpha_{\mathrm{st}} - 4\beta_{\mathrm{st}} \log T^*.$$

Therefore, we have

$$\sum_{t=\hat{t}+1}^{\tau} \exp\left(-F_{\hat{y}_i}\left(\boldsymbol{W}_{\hat{y}_i}^{(t)}, \tilde{\boldsymbol{X}}_i\right)\right) \leq \sum_{t=\hat{t}+1}^{\tau} \exp\left(-\frac{1}{m} \sum_{r' \in \mathcal{X}_i} \left\langle \boldsymbol{w}_{\hat{y}_i, r'}^{(t)}, \tilde{\boldsymbol{\xi}}_i \right\rangle\right)$$

$$\leq \sum_{t=\hat{t}+1}^{\tau} \exp\left(-\frac{(3 \log T^* - \alpha_{\mathrm{st}} - 4\beta_{\mathrm{st}} \log T^*)|\mathcal{X}_i|}{m}\right)$$

$$\leq T^* \exp\left(-\frac{(3 \log T^* - \alpha_{\mathrm{st}} - 4\beta_{\mathrm{st}} \log T^*)|\mathcal{X}_i|}{m}\right)$$

$$\leq T^* \exp(-\log T^*) = 1,$$

where the last inequality follows from (6) and (9). Finally, we conclude

$$\overline{\rho}_{r, i}^{(\tau+1)} \leq \frac{7}{2} \log T^* + \frac{3\eta \sigma_p^2 d}{2mn_{\mathrm{st}}} \exp(\kappa_{\mathrm{st}}/16) \leq 4 \log T^*,$$

where the last inequality follows from (10).

(S2): We fix an arbitrary $s \in \{\pm 1\}$ and $i \in [n_{\mathrm{st}}]$.

For any $r \in [m]$, we have

$$\frac{mn_{\mathrm{st}}}{\eta \|\boldsymbol{\mu}\|^2} \left(\overline{M}_{s, r}^{(\tau+1)} - \overline{M}_{s, r}^{(\tau)}\right)$$

$$= \sum_{l \in [2]} \left(\sum_{j \in \mathcal{C}_{\boldsymbol{\mu}_s}^{(l)}} \tilde{g}_j^{(\tau)} - \sum_{j \in \mathcal{F}_{\boldsymbol{\mu}_s}^{(l)}} \tilde{g}_j^{(\tau)}\right) \cdot \mathbb{1}\left[\left\langle \boldsymbol{w}_{s, r}^{(\tau)}, \boldsymbol{\mu}_s \right\rangle > 0\right]$$

$$\geq \sum_{l \in [2]} \left(\left|\mathcal{C}_{\boldsymbol{\mu}_s}^{(l)}\right| / \lambda_{\mathrm{st}} - \left|\mathcal{F}_{\boldsymbol{\mu}_s}^{(l)}\right| \lambda_{\mathrm{st}}\right) \tilde{g}_i^{(\tau)} \cdot \mathbb{1}\left[\left\langle \boldsymbol{w}_{s, r}^{(\tau)}, \boldsymbol{\mu}_s \right\rangle > 0\right]$$

$$\geq 2\left(\left(1 - C_{\mathrm{st}}^{-1}\right) n_{\boldsymbol{\mu}} / \lambda_{\mathrm{st}} - C_{\mathrm{st}}^{-1} n_{\boldsymbol{\mu}} \lambda_{\mathrm{st}}\right) \tilde{g}_i^{(\tau)} \cdot \mathbb{1}\left[\left\langle \boldsymbol{w}_{s, r}^{(\tau)}, \boldsymbol{\mu}_s \right\rangle > 0\right]$$

$$\geq \frac{n_{\boldsymbol{\mu}} \tilde{g}_i^{(\tau)}}{\lambda_{\mathrm{st}}} \cdot \mathbb{1}\left[\left\langle \boldsymbol{w}_{s, r}^{(\tau)}, \boldsymbol{\mu}_s \right\rangle > 0\right] \tag{12}$$

$$\geq 0,$$

where the first inequality follows from (S6) with iteration $\tau$ and the third inequality follows from large choice of $C_{\mathrm{st}}$.

For any $r \in \mathcal{A}_s$, from (S2) at iteration $0, \ldots, \tau$, we have $\left\langle \boldsymbol{w}_{s,r}^{(\tau)}, \boldsymbol{\nu}_s \right\rangle > 0$. Hence, we have

$$
\begin{aligned}
\frac{mn_{\mathrm{st}}}{\eta \|\boldsymbol{\nu}\|^2} \left( \overline{N}_{s,r}^{(\tau+1)} - \overline{N}_{s,r}^{(\tau)} \right) &= \sum_{l \in [2]} \left( \sum_{j \in \mathcal{C}_{\boldsymbol{\nu}_s}^{(l)}} \tilde{g}_j^{(\tau)} - \sum_{j \in \mathcal{F}_{\boldsymbol{\nu}_s}^{(l)}} \tilde{g}_j^{(\tau)} \right) \\
&\geq \sum_{l \in [2]} \left( \left| \mathcal{C}_{\boldsymbol{\nu}_s}^{(l)} \right| / \lambda_{\mathrm{st}} - \left| \mathcal{F}_{\boldsymbol{\nu}_s}^{(l)} \right| \lambda_{\mathrm{st}} \right) \tilde{g}_i^{(\tau)} \\
&\geq 2\left( \left( 1 - C_{\mathrm{st}}^{-1} \right) n_{\boldsymbol{\nu}} / \lambda_{\mathrm{st}} - C_{\mathrm{st}}^{-1} n_{\boldsymbol{\nu}} \lambda_{\mathrm{st}} \right) \tilde{g}_i^{(\tau)} \\
&\geq \frac{n_{\boldsymbol{\nu}} \tilde{g}_i^{(\tau)}}{\lambda_{\mathrm{st}}} \\
&\geq 0,
\end{aligned}
\tag{13}
$$

where the first inequality follows from (S6) with iteration $\tau$ and the third inequality follows from the large choice of $C_{\mathrm{st}}$.

Similarly, for any $r \in \mathcal{B}_s$, from (S2) with iteration $0, \ldots, \tau$, we have $\left\langle \boldsymbol{w}_{s,r}^{(\tau)}, \boldsymbol{\nu}_s \right\rangle < 0$. Hence, we have

$$
\begin{aligned}
\frac{mn_{\mathrm{st}}}{\eta \|\boldsymbol{\nu}\|^2} \left( \overline{N}_{s,r}^{(\tau)} - \overline{N}_{s,r}^{(\tau+1)} \right) &= \sum_{l \in [2]} \left( \sum_{j \in \mathcal{C}_{-\boldsymbol{\nu}_s}^{(l)}} \tilde{g}_j^{(\tau)} - \sum_{j \in \mathcal{F}_{-\boldsymbol{\nu}_s}^{(l)}} \tilde{g}_j^{(\tau)} \right) \\
&\geq \sum_{l \in [2]} \left( \left| \mathcal{C}_{-\boldsymbol{\nu}_s}^{(l)} \right| / \lambda_{\mathrm{st}} - \left| \mathcal{F}_{-\boldsymbol{\nu}_s}^{(l)} \right| \lambda_{\mathrm{st}} \right) \tilde{g}_i^{(\tau)} \\
&\geq 2\left( \left( 1 - C_{\mathrm{st}}^{-1} \right) n_{\boldsymbol{\nu}_s} / \lambda_{\mathrm{st}} - C_{\mathrm{st}}^{-1} n_{\boldsymbol{\nu}_s} \lambda_{\mathrm{st}} \right) \tilde{g}_i^{(\tau)} \\
&\geq \frac{n_{\boldsymbol{\nu}_s} \tilde{g}_i^{(\tau)}}{\lambda_{\mathrm{st}}} \\
&\geq 0,
\end{aligned}
$$

where the first inequality follows from (S6) with iteration $\tau$ and the third inequality follows from large choice of $C_{\mathrm{st}} > 0$.

Let us prove the last part. For any $r \in [m]$, if $\underline{M}_{s,r}^{(\tau)} \leq -\alpha_{\mathrm{st}}$, then we have $\left\langle \boldsymbol{w}_{s,r}^{(\tau)}, \boldsymbol{\mu}_{-s} \right\rangle < 0$. Hence, $\left| \underline{M}_{s,r}^{(\tau+1)} \right| = \left| \underline{M}_{s,r}^{(\tau)} \right| \leq \alpha_{\mathrm{st}} + \beta_{\mathrm{st}}$ by Lemma A.3. Otherwise, $\underline{M}_{s,r}^{(\tau)} > -\alpha_{\mathrm{st}}$ implies

$$
\begin{aligned}
&\frac{mn_{\mathrm{st}}}{\eta \|\boldsymbol{\mu}\|^2} \left( \underline{M}_{s,r}^{(\tau+1)} - \underline{M}_{s,r}^{(\tau)} \right) \\
&= -\sum_{l \in [2]} \left( \sum_{j \in \mathcal{C}_{\boldsymbol{\mu}_{-s}}^{(l)}} \tilde{g}_j^{(\tau)} - \sum_{j \in \mathcal{F}_{\boldsymbol{\mu}_{-s}}^{(l)}} \tilde{g}_j^{(\tau)} \right) \cdot \mathbb{1}\left[ \left\langle \boldsymbol{w}_{s,r}^{(\tau)}, \boldsymbol{\mu}_{-s} \right\rangle > 0 \right] \\
&\leq -\sum_{l \in [2]} \left( \left| \mathcal{C}_{\boldsymbol{\mu}_{-s}}^{(l)} \right| / \lambda_{\mathrm{st}} - \left| \mathcal{F}_{\boldsymbol{\mu}_{-s}}^{(l)} \right| \lambda_{\mathrm{st}} \right) \tilde{g}_i^{(\tau)} \cdot \mathbb{1}\left[ \left\langle \boldsymbol{w}_{s,r}^{(\tau)}, \boldsymbol{\mu}_{-s} \right\rangle > 0 \right] \\
&\leq -2\left( \left( 1 - C_{\mathrm{st}}^{-1} \right) \cdot n_{\boldsymbol{\mu}} / \lambda_{\mathrm{st}} - C_{\mathrm{st}}^{-1} n_{\boldsymbol{\mu}} \lambda_{\mathrm{st}} \right) \tilde{g}_i^{(\tau)} \mathbb{1}\left[ \left\langle \boldsymbol{w}_{s,r}^{(\tau)}, \boldsymbol{\mu}_{-s} \right\rangle > 0 \right] \\
&\leq 0,
\end{aligned}
$$

where the first inequality follows from (S6) with iteration $\tau$ and the last inequality follows from the large choice of $C_{\mathrm{st}}$. Thus, $\underline{M}_{s,r}^{(\tau+1)} \leq \underline{M}_{s,r}^{(\tau)} \leq \alpha_{\mathrm{st}} + \beta_{\mathrm{st}}$. In addition,

$$
\frac{mn_{\mathrm{st}}}{\eta \|\boldsymbol{\mu}\|^2} \left( \underline{M}_{s,r}^{(\tau+1)} - \underline{M}_{s,r}^{(\tau)} \right)
$$

$$= -\sum_{l\in[2]}\left(\sum_{j\in\mathcal{C}_{\boldsymbol{\mu}_{-s}}^{(l)}}\tilde{g}_j^{(\tau)}-\sum_{j\in\mathcal{F}_{\boldsymbol{\mu}_{-s}}^{(l)}}\tilde{g}_j^{(\tau)}\right)\cdot\mathbb{1}\left[\left\langle\boldsymbol{w}_{s,r}^{(\tau)},\boldsymbol{\mu}_{-s}\right\rangle>0\right]$$

$$\geq -\sum_{l\in[2]}\left(\left|\mathcal{C}_{\boldsymbol{\mu}_{-s}}^{(l)}\right|\lambda_{\mathrm{st}}-\left|\mathcal{F}_{\boldsymbol{\mu}_{-s}}^{(l)}\right|/\lambda_{\mathrm{st}}\right)\tilde{g}_i^{(\tau)}\cdot\mathbb{1}\left[\left\langle\boldsymbol{w}_{s,r}^{(\tau)},\boldsymbol{\mu}_{-s}\right\rangle>0\right]$$

$$\geq -2\Big(\left(1-C_{\mathrm{st}}^{-1}\right)n_{\boldsymbol{\mu}}\lambda_{\mathrm{st}}-C_{\mathrm{st}}^{-1}n_{\boldsymbol{\mu}}/\lambda_{\mathrm{st}}\Big)\tilde{g}_i^{(\tau)}\cdot\mathbb{1}\left[\left\langle\boldsymbol{w}_{s,r}^{(\tau)},\boldsymbol{\mu}_{-s}\right\rangle>0\right]$$

$$\geq -2\lambda_{\mathrm{st}}n_{\boldsymbol{\mu}}$$

$$\geq -2\lambda_{\mathrm{st}}n_{\mathrm{st}},$$

where the first inequality follows from (S6) with iteration $\tau$. Therefore, we have

$$\underline{M}_{s,r}^{(\tau+1)}\geq\underline{M}_{s,r}^{(\tau)}-\frac{2\lambda_{\mathrm{st}}\eta\|\boldsymbol{\mu}\|^2}{m}\geq-\alpha_{\mathrm{st}}-\frac{2\lambda_{\mathrm{st}}\eta\|\boldsymbol{\mu}\|^2}{m}\geq-\alpha_{\mathrm{st}}-\beta_{\mathrm{st}},$$

where the last inequality follows from (10).

From Lemma A.3, for any $r\in[m]$,

$$\left|\underline{N}_{s,r}^{(\tau+1)}-\underline{N}_{s,r}^{(\tau)}\right|\leq\frac{2\eta\|\boldsymbol{\nu}\|^2}{m}\leq\alpha_{\mathrm{st}}.$$

Therefore, it suffices to show that $\underline{N}_{s,r}^{(\tau+1)}\leq\underline{N}_{s,r}^{(\tau)}$ when $\underline{N}_{s,r}^{(\tau)}>\alpha_{\mathrm{st}}$ and $\underline{N}_{s,r}^{(\tau+1)}\geq\underline{N}_{s,r}^{(\tau)}$ when $\underline{N}_{s,r}^{(\tau)}<-\alpha_{\mathrm{st}}$. If $\underline{N}_{s,r}^{(\tau)}>\alpha_{\mathrm{st}}$, then we have

$$\left\langle\boldsymbol{w}_{s,r}^{(\tau)},\boldsymbol{\nu}_{-s}\right\rangle=\left\langle\boldsymbol{w}_{s,r}^{(0)},\boldsymbol{\nu}_{-s}\right\rangle+\underline{N}_{s,r}^{(\tau)}>0.$$

Hence, we have

$$\frac{mn_{\mathrm{st}}}{\eta\|\boldsymbol{\nu}\|^2}\left(\underline{N}_{s,r}^{(\tau+1)}-\underline{N}_{s,r}^{(\tau)}\right)$$

$$=-\sum_{l\in[2]}\left(\sum_{j\in\mathcal{C}_{\boldsymbol{\nu}_{-s}}^{(l)}}\tilde{g}_j^{(\tau)}-\sum_{j\in\mathcal{F}_{\boldsymbol{\nu}_{-s}}^{(l)}}\tilde{g}_j^{(\tau)}\right)$$

$$\leq-\sum_{l\in[2]}\left(\left|\mathcal{C}_{\boldsymbol{\nu}_{-s}}^{(l)}\right|/\lambda_{\mathrm{st}}-\left|\mathcal{F}_{\boldsymbol{\nu}_{-s}}^{(l)}\right|\lambda_{\mathrm{st}}\right)\tilde{g}_i^{(\tau)}$$

$$\leq-2\Big(\left(1-C_{\mathrm{st}}^{-1}\right)n_{\boldsymbol{\nu}}/\lambda_{\mathrm{st}}-C_{\mathrm{st}}^{-1}\cdot n_{\boldsymbol{\nu}}\lambda_{\mathrm{st}}\Big)\tilde{g}_i^{(\tau)}$$

$$\leq 0,$$

where the first inequality follows from (S6) with iteration $\tau$ and the last inequality follows from the large choice of $C_{\mathrm{st}}$. Using the similar argument, we can also show that $\underline{N}_{s,r}^{(\tau+1)}\geq\underline{N}_{s,r}^{(\tau)}$ when $\underline{N}_{s,r}^{(\tau)}<-\alpha_{\mathrm{st}}$ and we have desired conclusion.

(S3): We fix arbitrary $s\in\{\pm1\}$ and $i\in[n_{\mathrm{st}}]$.

From (12) and (S2) at iteration $0,\ldots,\tau$, we have

$$\sum_{r\in[m]}\overline{M}_{s,r}^{(\tau+1)}-\sum_{r\in[m]}\overline{M}_{s,r}^{(\tau)}\geq\frac{\eta\|\boldsymbol{\mu}\|^2}{mn_{\mathrm{st}}}\cdot\frac{n_{\boldsymbol{\mu}}\tilde{g}_i^{(\tau)}}{\lambda_{\mathrm{st}}}\cdot|\mathcal{M}_s|$$

$$\geq\frac{n_{\boldsymbol{\mu}}\mathrm{SNR}_{\boldsymbol{\mu}}^2}{12\lambda_{\mathrm{st}}n_{\mathrm{st}}}\eta\tilde{g}_i^{(\tau)}\|\tilde{\boldsymbol{\xi}}_i\|^2$$

$$\geq\frac{n_{\boldsymbol{\mu}}\mathrm{SNR}_{\boldsymbol{\mu}}^2}{12\lambda_{\mathrm{st}}}\left(\sum_{r\in[m]}\overline{\rho}_{r,i}^{(\tau+1)}-\sum_{r\in[m]}\overline{\rho}_{r,i}^{(\tau)}\right),$$

where the second inequality follows from (8) and (9). Combining with (S3) at iteration $\tau$, we have

$$\frac{n_{\boldsymbol{\mu}}\mathrm{SNR}_{\boldsymbol{\mu}}^2}{12\lambda_{\mathrm{st}}} \cdot \sum_{r\in[m]} \overline{\rho}_{r,i}^{(\tau+1)} \leq \sum_{r\in[m]} \overline{M}_{s,r}^{(\tau+1)}.$$

For any $r \in [m]$, we have

$$\frac{mn_{\mathrm{st}}}{\eta\|\boldsymbol{\mu}\|^2}\left(\overline{M}_{s,r}^{(\tau+1)} - \overline{M}_{s,r}^{(\tau)}\right)$$

$$= \sum_{l\in[2]}\left(\sum_{j\in\mathcal{C}_{\boldsymbol{\mu}_s}^{(l)}}\tilde{g}_j^{(\tau)} - \sum_{j\in\mathcal{F}_{\boldsymbol{\mu}_s}^{(l)}}\tilde{g}_j^{(\tau)}\right)\cdot\mathbb{1}\left[\left\langle\boldsymbol{w}_{s,r}^{(\tau)},\boldsymbol{\mu}_s\right\rangle > 0\right]$$

$$\leq \sum_{l\in[2]}\left(\left|\mathcal{C}_{\boldsymbol{\mu}_s}^{(l)}\right|\lambda_{\mathrm{st}} - \left|\mathcal{F}_{\boldsymbol{\mu}_s}^{(l)}\right|/\lambda_{\mathrm{st}}\right)\tilde{g}_i^{(\tau)}\cdot\mathbb{1}\left[\left\langle\boldsymbol{w}_{s,r}^{(\tau)},\boldsymbol{\mu}_s\right\rangle > 0\right]$$

$$\leq \lambda_{\mathrm{st}}\sum_{l\in[2]}\left|\mathcal{C}_{\boldsymbol{\mu}_s}^{(l)}\right|\tilde{g}_i^{(\tau)}\cdot\mathbb{1}\left[\left\langle\boldsymbol{w}_{s,r}^{(\tau)},\boldsymbol{\mu}_s\right\rangle > 0\right]$$

$$\leq 2\lambda_{\mathrm{st}}\left(1 + C_{\mathrm{st}}^{-1}\right)n_{\boldsymbol{\mu}}\tilde{g}_i^{(\tau)}\cdot\mathbb{1}\left[\left\langle\boldsymbol{w}_{s,r}^{(\tau)},\boldsymbol{\mu}_s\right\rangle > 0\right]$$

$$\leq 3\lambda_{\mathrm{st}}n_{\boldsymbol{\mu}}\tilde{g}_i^{(\tau)}\cdot\mathbb{1}\left[\left\langle\boldsymbol{w}_{s,r}^{(\tau)},\boldsymbol{\mu}_s\right\rangle > 0\right]$$

where the first inequality follows from (S6) with iteration $\tau$. Hence, we have

$$\sum_{r\in[m]}\overline{M}_{s,r}^{(\tau+1)} - \sum_{r\in[m]}\overline{M}_{s,r}^{(\tau)} \leq \frac{\lambda_{\mathrm{st}}\eta\|\boldsymbol{\mu}\|^2}{mn_{\mathrm{st}}}n_{\boldsymbol{\mu}}\tilde{g}_i^{(\tau)}|\mathcal{M}_s|$$

$$\leq \frac{6\lambda_{\mathrm{st}}\eta}{n_{\mathrm{st}}}n_{\boldsymbol{\mu}}\mathrm{SNR}_{\boldsymbol{\mu}}^2\tilde{g}_i^{(\tau)}\|\tilde{\boldsymbol{\xi}}_i\|^2$$

$$\leq 6\lambda_{\mathrm{st}}n_{\boldsymbol{\mu}}\mathrm{SNR}_{\boldsymbol{\mu}}^2\left(\sum_{r\in[m]}\overline{\rho}_{r,i}^{(\tau+1)} - \sum_{r\in[m]}\overline{\rho}_{r,i}^{(\tau)}\right),$$

where the second and third inequalities follow from (8) and (9), and (S7) with iteration $\tau$. Combining with (S3) at iteration $\tau$, we have

$$\sum_{r\in[m]}\overline{M}_{s,r}^{(\tau+1)} \leq 6\lambda_{\mathrm{st}}n_{\boldsymbol{\mu}}\mathrm{SNR}_{\boldsymbol{\mu}}^2\cdot\sum_{r\in[m]}\overline{\rho}_{r,i}^{(\tau+1)}.$$

From (13) and (S2) at iteration $0,\ldots,\tau$, we have

$$\sum_{r\in\mathcal{A}_s}\overline{N}_{s,r}^{(\tau+1)} - \sum_{r\in\mathcal{A}_s}\overline{N}_{s,r}^{(\tau)} \geq \frac{\eta\|\boldsymbol{\nu}\|^2}{mn_{\mathrm{st}}}\cdot\frac{n_{\boldsymbol{\nu}}\tilde{g}_i^{(\tau)}}{2\lambda_{\mathrm{st}}}\cdot|\mathcal{A}_s|$$

$$\geq \frac{n_{\boldsymbol{\nu}}\mathrm{SNR}_{\boldsymbol{\nu}}^2}{12\lambda_{\mathrm{st}}n_{\mathrm{st}}}\eta\tilde{g}_i^{(\tau)}\|\tilde{\boldsymbol{\xi}}_i\|^2$$

$$\geq \frac{n_{\boldsymbol{\nu}}\mathrm{SNR}_{\boldsymbol{\nu}}^2}{12\lambda_{\mathrm{st}}}\left(\sum_{r\in[m]}\overline{\rho}_{r,i}^{(\tau+1)} - \sum_{r\in[m]}\overline{\rho}_{r,i}^{(\tau)}\right),$$

where the second inequality follows from (8) and (9). Combining with (S3) at iteration $\tau$, we have

$$\frac{n_{\boldsymbol{\nu}}\mathrm{SNR}_{\boldsymbol{\nu}}^2}{12\lambda_{\mathrm{st}}} \cdot \sum_{r\in[m]} \overline{\rho}_{r,i}^{(\tau+1)} \leq \sum_{r\in\mathcal{A}_s} \overline{N}_{s,r}^{(\tau+1)}.$$

For any $r \in \mathcal{A}_s$, we have

$$\frac{mn_{\mathrm{st}}}{\eta\|\boldsymbol{\nu}\|^2}\left(\overline{N}_{s,r}^{(\tau+1)} - \overline{N}_{s,r}^{(\tau)}\right) = \sum_{l\in[2]}\left(\sum_{j\in\mathcal{C}_{\boldsymbol{\nu}_s}^{(l)}}\tilde{g}_j^{(\tau)} - \sum_{j\in\mathcal{F}_{\boldsymbol{\nu}_s}^{(l)}}\tilde{g}_j^{(\tau)}\right)$$

$$\leq \lambda_{\text{st}} \sum_{l \in [2]} \left| \mathcal{C}_{\boldsymbol{\nu}_s}^{(l)} \right| \tilde{g}_i^{(\tau)}$$

$$\leq 2\lambda_{\text{st}} \left(1 + C_{\text{st}}^{-1}\right) n_{\boldsymbol{\nu}} \tilde{g}_i^{(\tau)}$$

$$\leq 3\lambda_{\text{st}} n_{\boldsymbol{\nu}} \tilde{g}_i^{(\tau)},$$

where the first inequality follows from (S6) and the third inequality follows from the large choice of $C_{\text{st}}$. Hence, we have

$$\sum_{r \in \mathcal{A}_s} \overline{N}_{s,r}^{(\tau+1)} - \sum_{r \in \mathcal{A}_s} \overline{N}_{s,r}^{(\tau)} \leq \frac{\eta \|\boldsymbol{\nu}\|^2}{mn_{\text{st}}} \cdot 3\lambda_{\text{st}} n_{\boldsymbol{\nu}} \tilde{g}_i^{(\tau)} |\mathcal{A}_s|$$

$$\leq \frac{6\lambda_{\text{st}} n_{\boldsymbol{\nu}} \text{SNR}_{\boldsymbol{\nu}}^2}{n_{\text{st}}} \eta \tilde{g}_i^{(\tau)} \|\tilde{\boldsymbol{\xi}}_i\|^2$$

$$\leq 6\lambda_{\text{st}} n_{\boldsymbol{\nu}} \text{SNR}_{\boldsymbol{\nu}}^2 \left( \sum_{r \in [m]} \overline{\rho}_{r,i}^{(\tau+1)} - \sum_{r \in [m]} \overline{\rho}_{r,i}^{(\tau)} \right),$$

where the second and third inequalities follow from (8) and (9). Combining with (S3) at iteration $\tau$, we have

$$\sum_{r \in \mathcal{A}_s} \overline{N}_{s,r}^{(\tau+1)} \leq 6\lambda_{\text{st}} n_{\boldsymbol{\nu}} \text{SNR}_{\boldsymbol{\nu}}^2 \cdot \sum_{r \in [m]} \overline{\rho}_{r,i}^{(\tau+1)}.$$

Using a similar argument, we can also show that

$$\frac{n_{\boldsymbol{\nu}} \text{SNR}_{\boldsymbol{\nu}}^2}{12\lambda_{\text{st}}} \cdot \sum_{r \in [m]} \overline{\rho}_{r,i}^{(\tau+1)} \leq - \sum_{r \in \mathcal{B}_s} \overline{N}_{s,r}^{(\tau+1)} \leq 6\lambda_{\text{st}} n_{\boldsymbol{\nu}} \text{SNR}_{\boldsymbol{\nu}}^2 \cdot \sum_{r \in [m]} \overline{\rho}_{r,i}^{(\tau+1)}.$$

(S4): We fix arbitrary $i \in [n_{\text{st}}]$. From (S3) at iteration $\tau + 1$ which we have already shown, we have

$$\frac{1}{m} \sum_{r \in \mathcal{M}_s} \overline{M}_{s,r}^{(\tau+1)} \leq \frac{1}{m} \sum_{r \in [m]} \overline{M}_{s,r}^{(\tau+1)}$$

$$\leq \frac{6\lambda_{\text{st}} n_{\boldsymbol{\mu}} \text{SNR}_{\boldsymbol{\mu}}^2}{m} \cdot \sum_{r \in [m]} \overline{\rho}_{r,i}^{(\tau+1)}$$

$$\leq 24\lambda_{\text{st}} n_{\boldsymbol{\mu}} \text{SNR}_{\boldsymbol{\mu}}^2 \log T^*$$

$$\leq \frac{\kappa_{\text{st}}}{64},$$

where the first equality follows from (S2) at iteration $0, \ldots, \tau$, the second inequality follows from (S1) and the last inequality follows from Condition 3.2. Similarly, we have

$$\frac{1}{m} \sum_{r \in \mathcal{A}_s} \overline{N}_{s,r}^{(\tau+1)} \leq \frac{6\lambda_{\text{st}} n_{\boldsymbol{\nu}} \text{SNR}_{\boldsymbol{\nu}}^2}{m} \cdot \sum_{r \in [m]} \overline{\rho}_{r,i}^{(\tau+1)} \leq 24\lambda_{\text{st}} n_{\boldsymbol{\nu}} \text{SNR}_{\boldsymbol{\nu}}^2 \log T^* \leq \frac{\kappa_{\text{st}}}{64}$$

and

$$-\frac{1}{m} \sum_{r \in \mathcal{B}_s} \overline{N}_{s,r}^{(\tau+1)} \leq \frac{6\lambda_{\text{st}} n_{\boldsymbol{\nu}} \text{SNR}_{\boldsymbol{\nu}}^2}{m} \cdot \sum_{r \in [m]} \overline{\rho}_{r,i}^{(\tau+1)} \leq 24\lambda_{\text{st}} n_{\boldsymbol{\nu}} \text{SNR}_{\boldsymbol{\nu}}^2 \log T^* \leq \frac{\kappa_{\text{st}}}{64}.$$

Therefore, for any $s \in \{\pm 1\}$, due to (6) and the above three inequalities, we have

$$\frac{1}{m} \sum_{r \in [m]} \sigma \left( \left\langle \boldsymbol{w}_{s,r}^{(\tau+1)}, \boldsymbol{\mu}_s \right\rangle \right), \frac{1}{m} \sum_{r \in [m]} \sigma \left( \left\langle \boldsymbol{w}_{s,r}^{(\tau+1)}, \boldsymbol{\nu}_s \right\rangle \right), \frac{1}{m} \sum_{r \in [m]} \sigma \left( \left\langle \boldsymbol{w}_{s,r}^{(\tau+1)}, -\boldsymbol{\nu}_s \right\rangle \right) \leq \frac{\kappa_{\text{st}}}{32}.$$

(14)

Together with applying Lemma C.2 and , we have

$$\left| \hat{y}_i f_{\text{st}} \left( \boldsymbol{W}^{(\tau+1)}, \tilde{\boldsymbol{X}}_i \right) - \frac{1}{m} \sum_{r \in [m]} \overline{\rho}_{r,i}^{(\tau+1)} \right|$$

$$= \left| F_{\hat{y}_i}\left(\boldsymbol{W}_{\hat{y}_i}^{(\tau)}, \tilde{\boldsymbol{X}}_i\right) - \frac{1}{m}\sum_{r\in[m]} \overline{\rho}_{r,i}^{(\tau)} \right| + F_{-\hat{y}_i}\left(\boldsymbol{W}_{-\hat{y}_i}^{(\tau)}, \tilde{\boldsymbol{X}}_i\right)$$

$$\leq \frac{1}{m}\sum_{r\in[m]} \left| \sigma\left(\left\langle \boldsymbol{w}_{\hat{y}_i,r}^{(\tau+1)}, \tilde{\boldsymbol{\xi}}_i \right\rangle - \overline{\rho}_{r,i}^{(\tau+1)}\right) \right| + \frac{1}{m}\sum_{l\in[2]}\sum_{r\in[m]} \sigma\left(\left\langle \boldsymbol{w}_{\hat{y}_i,r}, \tilde{\boldsymbol{v}}_i^{(l)} \right\rangle - \overline{\rho}_{r,i}^{(\tau+1)}\right) + \frac{\kappa_{\text{st}}}{16}$$

$$\leq \frac{\kappa_{\text{st}}}{4}.$$

(S5): We fix $i, j \in [n_{\text{st}}]$ and we assume $\frac{1}{m}\sum_{r\in[m]} \left[ \overline{\rho}_{r,i}^{(\tau)} - \overline{\rho}_{r,j}^{(\tau)} \right] > 0$, without loss of generality.

From the triangular inequality, (8), and (10), we have

$$\left| \frac{1}{m}\sum_{r\in[m]} \left[ \overline{\rho}_{r,i}^{(\tau+1)} - \overline{\rho}_{r,j}^{(\tau+1)} \right] - \frac{1}{m}\sum_{r\in[m]} \left[ \overline{\rho}_{r,i}^{(\tau)} - \overline{\rho}_{r,j}^{(\tau)} \right] \right|$$

$$\leq \frac{1}{m}\sum_{r\in[m]} \left[ \overline{\rho}_{r,i}^{(\tau+1)} - \overline{\rho}_{r,i}^{(\tau)} \right] + \frac{1}{m}\sum_{r\in[m]} \left[ \overline{\rho}_{r,j}^{(\tau+1)} - \overline{\rho}_{r,j}^{(\tau)} \right]$$

$$\leq \frac{\eta}{m n_{\text{st}}} \tilde{g}_i^{(\tau)} \|\tilde{\boldsymbol{\xi}}_i\|^2 + \frac{\eta}{m n_{\text{st}}} \tilde{g}_j^{(\tau)} \|\tilde{\boldsymbol{\xi}}_j\|^2$$

$$\leq \frac{3\eta\sigma_p^2 d}{m n_{\text{st}}}$$

$$\leq \frac{\kappa_{\text{st}}}{2}.$$

Hence, we have $\frac{1}{m}\sum_{r\in[m]} \left[ \overline{\rho}_{r,i}^{(\tau+1)} - \overline{\rho}_{r,j}^{(\tau+1)} \right] > -\frac{\kappa_{\text{st}}}{2}$.

Also, if $\frac{1}{m}\sum_{r\in[m]} \left[ \overline{\rho}_{r,i}^{(\tau)} - \overline{\rho}_{r,j}^{(\tau)} \right] < \frac{\kappa_{\text{st}}}{2}$, then we have

$$\frac{1}{m}\sum_{r\in[m]} \left[ \overline{\rho}_{r,i}^{(\tau+1)} - \overline{\rho}_{r,j}^{(\tau+1)} \right] \leq \frac{1}{m}\sum_{r\in[m]} \left[ \overline{\rho}_{r,i}^{(\tau)} - \overline{\rho}_{r,j}^{(\tau)} \right] + \frac{\kappa_{\text{st}}}{2} \leq \kappa_{\text{st}}.$$

Otherwise, we have $\frac{\kappa_{\text{st}}}{2} \leq \frac{1}{m}\sum_{r\in[m]} \left[ \overline{\rho}_{r,i}^{(\tau)} - \overline{\rho}_{r,j}^{(\tau)} \right] \leq \kappa_{\text{st}}$. Together with applying Lemma C.2 and (14), we have

$$\hat{y}_i f_{\text{st}}\left(\boldsymbol{W}^{(\tau)}, \tilde{\boldsymbol{X}}_i\right) - \hat{y}_j f_{\text{st}}\left(\boldsymbol{W}^{(\tau)}, \tilde{\boldsymbol{X}}_j\right)$$

$$= F_{\hat{y}_i}\left(\boldsymbol{W}_{\hat{y}_i}^{(\tau)}, \tilde{\boldsymbol{X}}_i\right) - F_{-\hat{y}_i}\left(\boldsymbol{W}_{-\hat{y}_i}^{(\tau)}, \tilde{\boldsymbol{X}}_i\right) - F_{\hat{y}_j}\left(\boldsymbol{W}_{\hat{y}_j}^{(\tau)}, \tilde{\boldsymbol{X}}_j\right) + F_{-\hat{y}_j}\left(\boldsymbol{W}_{-\hat{y}_j}^{(\tau)}, \tilde{\boldsymbol{X}}_j\right)$$

$$\geq F_{\hat{y}_i}\left(\boldsymbol{W}_{\hat{y}_i}^{(\tau)}, \tilde{\boldsymbol{X}}_i\right) - F_{\hat{y}_j}\left(\boldsymbol{W}_{\hat{y}_j}^{(\tau)}, \tilde{\boldsymbol{X}}_j\right) - \frac{\kappa_{\text{st}}}{16}$$

$$\geq \frac{1}{m}\sum_{r\in[m]} \left[ \sigma\left(\left\langle \boldsymbol{w}_{\hat{y}_i,r}^{(\tau)}, \tilde{\boldsymbol{\xi}}_i \right\rangle\right) - \sigma\left(\left\langle \boldsymbol{w}_{\hat{y}_j,r}^{(\tau)}, \tilde{\boldsymbol{\xi}}_j \right\rangle\right) \right] - \frac{1}{m}\sum_{l\in[2]}\sum_{r\in[m]} \sigma\left(\left\langle \boldsymbol{w}_{\hat{y}_j,r}^{(\tau)}, \tilde{\boldsymbol{v}}_j^{(l)} \right\rangle\right) - \frac{\kappa_{\text{st}}}{16}$$

$$\geq \frac{1}{m}\sum_{r\in[m]} \left[ \overline{\rho}_{r,i}^{(\tau)} - \overline{\rho}_{r,j}^{(\tau)} \right] - \frac{\kappa_{\text{st}}}{4}$$

$$\geq \frac{\kappa_{\text{st}}}{4}.$$

Therefore, we have

$$\frac{\tilde{g}_i^{(\tau)}}{\tilde{g}_j^{(\tau)}} = \frac{1 + \exp\left(\hat{y}_j f_{\text{st}}\left(\boldsymbol{W}^{(\tau)}, \tilde{\boldsymbol{X}}_j\right)\right)}{1 + \exp\left(\hat{y}_i f_{\text{st}}\left(\boldsymbol{W}^{(\tau)}, \tilde{\boldsymbol{X}}_i\right)\right)}$$

$$= \frac{\exp\left(-\hat{y}_j f_{\text{st}}\left(\boldsymbol{W}^{(\tau)}, \tilde{\boldsymbol{X}}_j\right)\right) + 1}{\exp\left(-\hat{y}_j f_{\text{st}}\left(\boldsymbol{W}^{(\tau)}, \tilde{\boldsymbol{X}}_j\right)\right) + \exp\left(\hat{y}_i f_{\text{st}}\left(\boldsymbol{W}^{(\tau)}, \tilde{\boldsymbol{X}}_i\right) - \hat{y}_j f_{\text{st}}\left(\boldsymbol{W}^{(\tau)}, \tilde{\boldsymbol{X}}_j\right)\right)}$$

$$\leq \frac{\exp\left(-\hat{y}_j f_{\text{st}}\left(\boldsymbol{W}^{(\tau)}, \tilde{\boldsymbol{X}}_j\right)\right) + 1}{\exp\left(-\hat{y}_j f_{\text{st}}\left(\boldsymbol{W}^{(\tau)}, \tilde{\boldsymbol{X}}_j\right)\right) + \exp\left(\kappa_{\text{st}}/4\right)}$$

$$\leq \frac{\exp(\kappa_{\text{st}}/16) + 1}{\exp(\kappa_{\text{st}}/16) + \exp(\kappa_{\text{st}}/4))}$$

$$\leq \exp\left(-\kappa_{\text{st}}/8\right),$$

where the second inequality follows from

$$-\hat{y}_j f_{\text{st}}\left(\boldsymbol{W}^{(\tau)}, \tilde{\boldsymbol{X}}_j\right) \leq F_{-\hat{y}_j}\left(\boldsymbol{W}^{(\tau)}_{-\hat{y}_j}, \tilde{\boldsymbol{X}}_j\right) \leq \frac{\kappa_{\text{st}}}{16}$$

and the last inequality follows from applying $\frac{z(z^3+1)}{z+1} = z(z^2 - z + 1) \geq z^2$ with $z = \exp(\kappa_{\text{st}}/16)$.

Therefore, we have

$$\sum_{r \in [m]} \left[\overline{\rho}_{r,i}^{(\tau+1)} - \overline{\rho}_{r,j}^{(\tau+1)}\right] - \sum_{r \in [m]} \left[\overline{\rho}_{r,i}^{(\tau)} - \overline{\rho}_{r,j}^{(\tau)}\right]$$

$$\leq \frac{\eta}{mn_{\text{st}}} \left(\tilde{g}_i^{(\tau)} m \|\tilde{\boldsymbol{\xi}}_i\|^2 - \tilde{g}_j^{(\tau)} |\mathcal{X}_j| \|\tilde{\boldsymbol{\xi}}_j\|^2\right)$$

$$= \frac{\eta}{mn_{\text{st}}} \tilde{g}_j^{(\tau)} |\mathcal{X}_j| \|\tilde{\boldsymbol{\xi}}_j\|^2 \left(\frac{\tilde{g}_i^{(\tau)} m \|\tilde{\boldsymbol{\xi}}_i\|^2}{\tilde{g}_j^{(\tau)} |\mathcal{X}_j| \|\tilde{\boldsymbol{\xi}}_j\|^2} - 1\right)$$

$$\leq \frac{\eta}{mn_{\text{st}}} \tilde{g}_j^{(\tau)} |\mathcal{X}_j| \|\tilde{\boldsymbol{\xi}}_j\|^2 \left(\exp(-\kappa_{\text{st}}/8) \cdot 4 \cdot (1 + \beta_{\text{st}}/n) - 1\right)$$

$$\leq \frac{\eta}{mn_{\text{st}}} \tilde{g}_j^{(\tau)} |\mathcal{X}_j| \|\tilde{\boldsymbol{\xi}}_j\|^2 \left(12 \exp(-\kappa_{\text{st}}/8) - 1\right)$$

$$= 0,$$

where the third inequality is due to (6) and $1 + z \leq e^z$ for any $z \in \mathbb{R}$. Hence, we have

$$\frac{1}{m} \sum_{r \in [m]} \left[\overline{\rho}_{r,i}^{(\tau+1)} - \overline{\rho}_{r,j}^{(\tau+1)}\right] \leq \frac{1}{m} \sum_{r \in [m]} \left[\overline{\rho}_{r,i}^{(\tau)} - \overline{\rho}_{r,j}^{(\tau)}\right] \leq \kappa_{\text{st}}.$$

(S6): We fix arbitrary $i, j \in [n_{\text{st}}]$ and we assume $\hat{y}_i f_{\text{st}}\left(\boldsymbol{W}^{(\tau+1)}, \tilde{\boldsymbol{X}}_i\right) \geq \hat{y}_j f_{\text{st}}\left(\boldsymbol{W}^{(\tau+1)}, \tilde{\boldsymbol{X}}_j\right)$, without loss of generality. By combining (S4) and (S5) at iteration $\tau + 1$ which we already have shown, we have

$$\hat{y}_i f_{\text{st}}\left(\boldsymbol{W}^{(\tau+1)}, \tilde{\boldsymbol{X}}_i\right) - \hat{y}_j f_{\text{st}}\left(\boldsymbol{W}^{(\tau+1)}, \tilde{\boldsymbol{X}}_j\right)$$

$$\leq \left|\frac{1}{m} \sum_{r \in [m]} \left[\overline{\rho}_{r,i}^{(\tau+1)} - \overline{\rho}_{r,j}^{(\tau+1)}\right]\right|$$

$$+ \left|\hat{y}_i f_{\text{st}}\left(\boldsymbol{W}^{(\tau+1)}, \tilde{\boldsymbol{X}}_i\right) - \frac{1}{m} \sum_{r \in [m]} \overline{\rho}_{r,i}^{(\tau+1)}\right| + \left|\hat{y}_j f_{\text{st}}\left(\boldsymbol{W}^{(\tau+1)}, \tilde{\boldsymbol{X}}_j\right) - \frac{1}{m} \sum_{r \in [m]} \overline{\rho}_{r,j}^{(\tau+1)}\right|$$

$$\leq 2\kappa_{\text{st}}.$$

Then, we have

$$\frac{\tilde{g}_j^{(\tau+1)}}{\tilde{g}_i^{(\tau+1)}} = \frac{1 + \exp\left(\hat{y}_i f_{\text{st}}\left(\boldsymbol{W}^{(\tau+1)}, \tilde{\boldsymbol{X}}_i\right)\right)}{1 + \exp\left(\hat{y}_j f_{\text{st}}\left(\boldsymbol{W}^{(\tau+1)}, \tilde{\boldsymbol{X}}_j\right)\right)}$$

$$\leq \exp\left[\hat{y}_i f_{\text{st}}\left(\boldsymbol{W}^{(\tau+1)}, \tilde{\boldsymbol{X}}_i\right) - \hat{y}_j f_{\text{st}}\left(\boldsymbol{W}^{(\tau+1)}, \tilde{\boldsymbol{X}}_j\right)\right]$$

$$\leq \exp(2\kappa_{\text{st}})$$

$$= \lambda_{\text{st}}.$$

(S7): We fix arbitrary $i \in [n_{\text{st}}]$. From (S7) at iteration $\tau$, we have $\left\langle \boldsymbol{w}_{\hat{y}_i,r}^{(\tau)}, \tilde{\boldsymbol{\xi}}_i \right\rangle > 0$ for any $r \in \mathcal{X}_i$. Therefore, we have

$$\overline{\rho}_{r,i}^{(\tau+1)} = \overline{\rho}_{r,i}^{(\tau)} + \frac{\eta}{mn_{\text{st}}} \tilde{g}_i^{(\tau)} \|\tilde{\boldsymbol{\xi}}_i\|^2$$

and

$$\left\langle \boldsymbol{w}_{\hat{y}_i,r}^{(\tau+1)}, \tilde{\boldsymbol{\xi}}_i \right\rangle - \left\langle \boldsymbol{w}_{\hat{y}_i,r}^{(\tau)}, \tilde{\boldsymbol{\xi}}_i \right\rangle$$

$$= \left( \overline{\rho}_{r,i}^{(\tau+1)} - \overline{\rho}_{r,i}^{(\tau)} \right) + \sum_{j \in [n_{\text{st}}] \setminus \{i\}} \left( \rho_{\hat{y}_i,r,j}^{(\tau+1)} - \rho_{\hat{y}_i,r,j}^{(\tau)} \right) \frac{\langle \tilde{\boldsymbol{\xi}}_i, \tilde{\boldsymbol{\xi}}_j \rangle}{\|\tilde{\boldsymbol{\xi}}_j\|^2}$$

$$\geq \frac{\eta}{mn_{\text{st}}} \tilde{g}_i^{(\tau)} \|\tilde{\boldsymbol{\xi}}_i\|^2 - \frac{\eta}{mn_{\text{st}}} \sum_{j \in [n_{\text{st}}] \setminus \{i\}} \tilde{g}_j^{(\tau)} \left| \langle \tilde{\boldsymbol{\xi}}_i, \tilde{\boldsymbol{\xi}}_j \rangle \right|$$

$$= \frac{\eta}{mn_{\text{st}}} \tilde{g}_i^{(\tau)} \|\tilde{\boldsymbol{\xi}}_i\|^2 \left( 1 - \sum_{j \in [n_{\text{st}}] \setminus \{i\}} \frac{\tilde{g}_j^{(\tau)}}{\tilde{g}_i^{(\tau)}} \cdot \frac{\left| \langle \tilde{\boldsymbol{\xi}}_i, \tilde{\boldsymbol{\xi}}_j \rangle \right|}{\|\tilde{\boldsymbol{\xi}}_i\|^2} \right)$$

$$\geq \frac{\eta}{mn_{\text{st}}} \tilde{g}_i^{(\tau)} \|\tilde{\boldsymbol{\xi}}_i\|^2 (1 - \lambda_{\text{st}} \beta_{\text{st}})$$

$$\geq 0,$$

where we use (S6) at iteration $\tau$, (8) for the second inequality, and (6) for the last inequality. Hence, we have $\left\langle \boldsymbol{w}_{\hat{y}_i,r}^{(\tau+1)}, \tilde{\boldsymbol{\xi}}_i \right\rangle > 0$. Now we prove the second part. For any $r \in \mathcal{X}_i$ and $r' \in [m]$, we have

$$\overline{\rho}_{r',i}^{(\tau+1)} \leq \overline{\rho}_{r',i}^{(\tau)} + \frac{\eta}{mn_{\text{st}}} \tilde{g}_i^{(\tau)} \|\tilde{\boldsymbol{\xi}}_i\|^2 \leq \overline{\rho}_{r,i}^{(\tau)} + \frac{\eta}{mn_{\text{st}}} \tilde{g}_i^{(\tau)} \|\tilde{\boldsymbol{\xi}}_i\|^2 = \overline{\rho}_{r,i}^{(\tau+1)},$$

where the second inequality is due to (S7) with iteration $\tau$.

(S8): From (S7) at iteration $\tau$, we have

$$\frac{1}{m} \sum_{r \in [m]} \overline{\rho}_{r,i}^{(\tau+1)} \geq \frac{1}{m} \sum_{r \in [m]} \overline{\rho}_{r,i}^{(\tau)} + \frac{\eta}{mn_{\text{st}}} \tilde{g}_i^{(\tau)} \cdot \frac{|\mathcal{X}_i|}{m} \cdot \|\tilde{\boldsymbol{\xi}}_i\|^2$$

$$= \frac{1}{m} \sum_{r \in [m]} \overline{\rho}_{r,i}^{(\tau)} + \frac{\eta}{mn_{\text{st}}} \cdot \frac{1}{1 + \exp\left( \hat{y}_i f_{\text{st}} \left( \boldsymbol{W}^{(\tau)}, \tilde{\boldsymbol{X}}_i \right) \right)} \cdot \frac{|\mathcal{X}_i|}{m} \cdot \|\tilde{\boldsymbol{\xi}}_i\|^2.$$

From (S4) at iteration $\tau$, (8), and (9), we have

$$\frac{1}{m} \sum_{r \in [m]} \overline{\rho}_{r,i}^{(\tau+1)} \geq \frac{1}{m} \sum_{r \in [m]} \overline{\rho}_{r,i}^{(\tau)} + \frac{\eta \sigma_p^2 d}{8mn_{\text{st}}} \cdot \frac{1}{1 + \exp(\kappa_{\text{st}}/4) \exp\left( \frac{1}{m} \sum_{r \in [m]} \overline{\rho}_{r,i}^{(\tau)} \right)}.$$

By applying Lemma A.5, the fact that $z + \frac{c}{1+be^z}$ is an increasing function for any $c \in [0, 1], b > 0$, and the comparison theorem, we have our conclusion. $\qquad \square$

## C.2 Convergence of Training Loss

In this subsection, we prove that the training loss converges below $\varepsilon$ within $\mathcal{O}(\eta^{-1} \varepsilon^{-1} n_{\text{st}} m d^{-1} \sigma_p^{-2})$. All the arguments in this subsection are under Condition 3.1 and the event $E_{\text{st}}$.

For any $t \in [0, T^*]$, from the definition of $x_t$, we have

$$x_t \leq \log \left( \frac{\eta \sigma_p^2 d}{8mn_{\text{st}} \exp(\kappa_{\text{st}}/4)} t + 1 \right).$$

Combining the inequality above with the definition of $x_t$, we have

$$\exp(x_t) \geq \frac{\eta \sigma_p^2 d}{8mn_{\text{st}} \exp(\kappa_{\text{st}}/4)} t + 1 - \exp(-\kappa_{\text{st}}/4) \log \left( \frac{\eta \sigma_p^2 d}{8mn_{\text{st}} \exp(\kappa_{\text{st}}/4)} t + 1 \right)$$

$$\geq \frac{\eta \sigma_p^2 d}{8mn_{\mathrm{st}} \exp(\kappa_{\mathrm{st}}/4)} t + 1 - \log\left(\frac{\eta \sigma_p^2 d}{8mn_{\mathrm{st}} \exp(\kappa_{\mathrm{st}}/4)} t + 1\right)$$

$$\geq \frac{\eta \sigma_p^2 d}{16mn_{\mathrm{st}} \exp(\kappa_{\mathrm{st}}/4)} t + \frac{1}{2}$$

$$\geq \frac{\eta \sigma_p^2 d}{16mn_{\mathrm{st}} \exp(\kappa_{\mathrm{st}}/4)} t, \tag{15}$$

where we use the inequality $\log z < \frac{z}{2}$ for any $z > 1$.

For any $t \in [0, T^*]$ and $i \in [n_{\mathrm{st}}]$, by applying (S4) and (S8), we have

$$\hat{y}_i f\left(\boldsymbol{W}^{(t)}, \tilde{\boldsymbol{X}}_i\right) \geq -\frac{\kappa_{\mathrm{st}}}{4} + \frac{1}{m} \sum_{r \in [m]} \overline{\rho}_{r,i}^{(t)}$$

$$\geq -\frac{\kappa_{\mathrm{st}}}{4} + x_t$$

$$\geq -\frac{\kappa_{\mathrm{st}}}{4} + \log\left(\frac{\eta \sigma_p^2 d}{16mn_{\mathrm{st}} \exp(\kappa_{\mathrm{st}}/4)} t\right)$$

$$= \log\left(\frac{\eta \sigma_p^2 d}{16mn_{\mathrm{st}} \exp(\kappa_{\mathrm{st}}/2)} t\right)$$

$$\geq \log\left(\frac{\eta \sigma_p^2 d}{16\lambda_{\mathrm{st}} mn_{\mathrm{st}}} t\right),$$

where the third inequality follows from (15) and the fourth inequality follows from (6). Therefore, we have

$$L_{\mathrm{st}}\left(\boldsymbol{W}^{(t)}\right) \leq \log\left(1 + \frac{16\lambda_{\mathrm{st}} mn_{\mathrm{st}}}{\eta \sigma_p^2 d} \cdot t^{-1}\right) \leq \frac{16\lambda_{\mathrm{st}} mn_{\mathrm{st}}}{\eta \sigma_p^2 d} \cdot t^{-1},$$

where the inequality follows from $\log(1 + z) \leq z$ for $z > 0$. If $t \geq 16\lambda_{\mathrm{st}} \eta^{-1} \varepsilon^{-1} mn_{\mathrm{st}} d^{-1} \sigma_p^{-2}$, then we have $L_{\mathrm{st}}\left(\boldsymbol{W}^{(t)}\right) \leq \varepsilon$. Hence, by defining $T_{\mathrm{st}} := \lceil 16\lambda_{\mathrm{st}} \eta^{-1} \varepsilon^{-1} mn_{\mathrm{st}} d^{-1} \sigma_p^{-2} \rceil$, we have the first conclusion.

### C.3  Test Error

In this subsection, we prove the second part of our conclusion. All the arguments in this subsection are under Condition 3.1 and the event $E_{\mathrm{st}}$.

Define $\boldsymbol{v}^{(1)}$, $\boldsymbol{v}^{(2)}$, and $\boldsymbol{\xi}$ as the signal vectors and the noise vector in the test data $(\boldsymbol{X}, y)$, respectively. We fix an arbitrary iteration $t \in [T_{\mathrm{st}}, T^*]$. From the choice of iteration $t$ and (15), for any $i \in [n_{\mathrm{st}}]$, we have

$$\log\left(\varepsilon^{-1}\right) \leq \log\left(\frac{\eta \sigma_p^2 d}{16\lambda_{\mathrm{st}} mn_{\mathrm{st}}} t\right) \leq \log\left(\frac{\eta \sigma_p^2 d}{16mn_{\mathrm{st}} \exp(\kappa_{\mathrm{st}}/2)} t\right) \leq x_t \leq \frac{1}{m} \sum_{r \in [m]} \overline{\rho}_{r,i}^{(t)}. \tag{16}$$

#### C.3.1  Test Error Upper Bound

We define a function $h : S \to \mathbb{R}$ as $h(\boldsymbol{z}) := \frac{1}{m} \sum_{r \in [m]} \sigma\left(\left\langle \boldsymbol{w}_{-y,r}^{(t)}, \boldsymbol{z} \right\rangle\right)$ for any $\boldsymbol{z} \in S$. It plays a crucial role when we prove the upper bounds on test error. We have

$$\mathbb{E}[h(\boldsymbol{\xi})] = \frac{1}{m} \mathbb{E}_{\mathbf{z}_1,\ldots,\mathbf{z}_m}\left[\sum_{r \in [m]} \sigma(\mathbf{z}_r)\right] = \frac{1}{2m} \mathbb{E}_{\mathbf{z}_1,\ldots,\mathbf{z}_m}\left[\sum_{r \in [m]} |\mathbf{z}_r|\right] = \frac{\sigma_p}{\sqrt{2\pi}m} \sum_{r \in [m]} \left\|\Pi_S \boldsymbol{w}_{-y,r}^{(t)}\right\|,$$

where $\mathbf{z}_r \sim \mathcal{N}\left(0, \sigma_p^2 \left\|\Pi_S \boldsymbol{w}_{-y,r}^{(t)}\right\|^2\right)$ for each $r \in [m]$. Also, for any $\boldsymbol{z}_1, \boldsymbol{z}_2 \in S$, we have

$$|h(\boldsymbol{z}_1) - h(\boldsymbol{z}_2)| \leq \frac{1}{m} \sum_{r \in [m]} \left|\sigma\left(\left\langle \boldsymbol{w}_{-y,r}^{(t)}, \boldsymbol{z}_1 \right\rangle\right) - \sigma\left(\left\langle \boldsymbol{w}_{-y,r}^{(t)}, \boldsymbol{z}_2 \right\rangle\right)\right|$$

$$\leq \frac{1}{m} \sum_{r\in[m]} \left| \left\langle \boldsymbol{w}_{-y,r}^{(t)}, \boldsymbol{z}_1 \right\rangle - \left\langle \boldsymbol{w}_{-y,r}^{(t)}, \boldsymbol{z}_2 \right\rangle \right|$$

$$= \frac{1}{m} \sum_{r\in[m]} \left| \left\langle \Pi_S \boldsymbol{w}_{-y,r}^{(t)}, \boldsymbol{z}_1 \right\rangle - \left\langle \Pi_S \boldsymbol{w}_{-y,r}^{(t)}, \boldsymbol{z}_2 \right\rangle \right|$$

$$\leq \frac{1}{m} \sum_{r\in[m]} \left\| \Pi_S \boldsymbol{w}_{-y,r}^{(t)} \right\| \|\boldsymbol{z}_1 - \boldsymbol{z}_2\|.$$

Hence, $h$ is $\frac{1}{m}\sum_{r\in[m]}\left\|\Pi_S\boldsymbol{w}_{-y,r}^{(t)}\right\|$-Lipschitz.

The following lemma characterizes $\sum_{r\in[m]}\left\|\Pi_S\boldsymbol{w}_{-y,r}^{(t)}\right\|$'s which is related to key properties of $h$.

**Lemma C.4.** *For any $s\in\{\pm 1\}$, it holds that*

$$\sum_{r\in[m]} \left\| \Pi_S \boldsymbol{w}_{s,r}^{(t)} \right\| \leq 20\sigma_p^{-1} d^{-\frac{1}{2}} \left( \sum_{i\in[n_{\mathrm{st}}]} \left( \sum_{r\in[m]} \overline{\rho}_{r,i}^{(t)} \right)^2 \right)^{\frac{1}{2}}.$$

*Proof of Lemma C.4.* From triangular inequality and the event $E_{\mathrm{st}}$, for each $r\in[m]$, we have

$$\left\| \Pi_S \boldsymbol{w}_{s,r}^{(t)} \right\| \leq \left\| \Pi_S \boldsymbol{w}_{s,r}^{(0)} \right\| + \left\| \sum_{i\in[n_{\mathrm{st}}]} \rho_{s,r,i}^{(t)} \tilde{\boldsymbol{\xi}}_i \|\tilde{\boldsymbol{\xi}}_i\|^{-2} \right\| \leq \sqrt{2}\sigma_0 d^{\frac{1}{2}} + \left\| \sum_{i\in[n_{\mathrm{st}}]} \rho_{s,r,i}^{(t)} \tilde{\boldsymbol{\xi}}_i \|\tilde{\boldsymbol{\xi}}_i\|^{-2} \right\|.$$

In addition, we have

$$\left\| \sum_{i\in[n_{\mathrm{st}}]} \rho_{s,r,i}^{(t)} \tilde{\boldsymbol{\xi}}_i \|\tilde{\boldsymbol{\xi}}_i\|^{-2} \right\|^2$$

$$= \sum_{i\in[n_{\mathrm{st}}]} \left( \rho_{s,r,i}^{(t)} \right)^2 \|\tilde{\boldsymbol{\xi}}_i\|^{-2} + \sum_{\substack{i,j\in[n_{\mathrm{st}}]\\ i\neq j}} \rho_{s,r,i}^{(t)} \rho_{s,r,j}^{(t)} \langle \tilde{\boldsymbol{\xi}}_i, \tilde{\boldsymbol{\xi}}_j \rangle \|\tilde{\boldsymbol{\xi}}_i\|^{-2} \|\tilde{\boldsymbol{\xi}}_j\|^{-2}$$

$$\leq 2\sigma_p^{-2} d^{-1} \sum_{i\in[n_{\mathrm{st}}]} \left( \rho_{s,r,i}^{(t)} \right)^2 + 2\beta_{\mathrm{st}} n_{\mathrm{st}}^{-1} \sigma_p^{-2} d^{-1} \sum_{\substack{i,j\in[n_{\mathrm{st}}]\\ i\neq j}} \left| \rho_{s,r,i}^{(t)} \right| \left| \rho_{s,r,j}^{(t)} \right|$$

$$\leq 2\sigma_p^{-2} d^{-1} \sum_{i\in[n_{\mathrm{st}}]} \left( \rho_{s,r,i}^{(t)} \right)^2 + \beta_{\mathrm{st}} n_{\mathrm{st}}^{-1} \sigma_p^{-2} d^{-1} \sum_{\substack{i,j\in[n_{\mathrm{st}}]\\ i\neq j}} \frac{\left( \rho_{s,r,i}^{(t)} \right)^2 + \left( \rho_{s,r,j}^{(t)} \right)^2}{2},$$

$$\leq 4\sigma_p^{-2} d^{-1} \sum_{i\in[n_{\mathrm{st}}]} \left( \rho_{s,r,i}^{(t)} \right)^2$$

where the first inequality follows from (8) and the second inequality follows from AM-GM inequality, and the last inequality follows from (6). From the Cauchy-Schwarz inequality, we have

$$\sum_{r\in[m]} \left\| \sum_{i\in[n_{\mathrm{st}}]} \rho_{s,r,i}^{(t)} \tilde{\boldsymbol{\xi}}_i \|\tilde{\boldsymbol{\xi}}_i\|^{-2} \right\| \leq 2\sigma_p^{-1} d^{-\frac{1}{2}} \sum_{r\in[m]} \left( \sum_{i\in[n_{\mathrm{st}}]} \left( \rho_{s,r,i}^{(t)} \right)^2 \right)^{\frac{1}{2}}$$

$$\leq 2m^{\frac{1}{2}} \sigma_p^{-1} d^{-\frac{1}{2}} \left( \sum_{r\in[m]} \sum_{i\in[n_{\mathrm{st}}]} \left( \rho_{s,r,i}^{(t)} \right)^2 \right)^{\frac{1}{2}}.$$

In addition, from (S1) with iteration $t$, we have

$$\sum_{i\in[n_{\mathrm{st}}]} \sum_{r\in[m]} \left( \rho_{s,r,i}^{(t)} \right)^2 = \sum_{\substack{i\in[n_{\mathrm{st}}]\\ \hat{y}_i=s}} \sum_{r\in[m]} \left( \overline{\rho}_{r,i}^{(t)} \right)^2 + \sum_{\substack{i\in[n_{\mathrm{st}}]\\ \hat{y}_i=-s}} \sum_{r\in[m]} \left( \underline{\rho}_{r,i}^{(t)} \right)^2$$

$$\leq \sum_{\substack{i \in [n_{\mathrm{st}}] \\ \hat{y}_i = s}} \sum_{r \in [m]} \left(\overline{\rho}_{r,i}^{(t)}\right)^2 + (\alpha_{\mathrm{st}} + 5\beta_{\mathrm{st}} \log T^*)^2 m n_{\mathrm{st}}.$$

For any $i \in [n_{\mathrm{st}}]$ such that $\hat{y}_i = s$, we have

$$\sum_{r \in [m]} \left(\overline{\rho}_{r,i}^{(t)}\right)^2 \leq m \left(\max_{r \in [m]} \overline{\rho}_{r,i}^{(t)}\right)^2 \leq 16 m^{-1} \left(\sum_{r \in [m]} \overline{\rho}_{r,i}^{(t)}\right)^2,$$

where the last inequality follows from (S7) and (9). Therefore, we have

$$\sum_{i \in [n_{\mathrm{st}}]} \sum_{r \in [m]} \left(\rho_{s,r,i}^{(t)}\right)^2 \leq 16 m^{-1} \sum_{i \in [n_{\mathrm{st}}]} \left(\sum_{r \in [m]} \overline{\rho}_{r,i}^{(t)}\right)^2 + (\alpha_{\mathrm{st}} + 5\beta_{\mathrm{st}} \log T^*)^2 m n_{\mathrm{st}}$$

$$\leq 25 m^{-1} \sum_{i \in [n_{\mathrm{st}}]} \left(\sum_{r \in [m]} \overline{\rho}_{r,i}^{(t)}\right)^2,$$

where the last inequality follows from (16) and (6). We conclude

$$\sum_{r \in [m]} \left\| \Pi_S \boldsymbol{w}_{s,r}^{(t)} \right\|$$

$$\leq \sqrt{2} m \sigma_0 d^{\frac{1}{2}} + 10 \sigma_p^{-1} d^{-\frac{1}{2}} \left( \sum_{i \in [n_{\mathrm{st}}]} \left( \sum_{r \in [m]} \overline{\rho}_{r,i}^{(t)} \right)^2 \right)^{\frac{1}{2}}$$

$$\leq 20 \sigma_p^{-1} d^{-\frac{1}{2}} \left( \sum_{i \in [n_{\mathrm{st}}]} \left( \sum_{r \in [m]} \overline{\rho}_{r,i}^{(t)} \right)^2 \right)^{\frac{1}{2}},$$

where the second inequality follows from (16), (6), and (C3). $\qquad\square$

By Theorem 5.2.2 in Vershynin (2018), for any $z > 0$, it holds that

$$\mathbb{P}[h(\boldsymbol{\xi}) - \mathbb{E}[h(\boldsymbol{\xi})] \geq z] \leq \exp\left(-\frac{cz^2}{\sigma_p^2 \|h\|_{\mathrm{Lip}}^2}\right)$$

where $c$ is a universal constant and $\|\cdot\|_{\mathrm{Lip}}$ denotes the best Lipschitz constant. Combining with Lemma C.4, we have

$$\mathbb{P}[h(\boldsymbol{\xi}) - \mathbb{E}[h(\boldsymbol{\xi})] \geq z] \leq \exp\left(-\frac{cm^2 d}{400 \sum_{i \in [n_{\mathrm{st}}]} \left(\sum_{r \in [m]} \overline{\rho}_{r,i}^{(t)}\right)^2} z^2\right). \tag{17}$$

Now, we characterize the test error. First, we consider the case $(\boldsymbol{X}, y) \in \mathcal{S}_{\mathrm{e}} \cup \mathcal{S}_{\mathrm{b}}$. We have

$$y f_{\mathrm{st}}\left(\boldsymbol{W}^{(t)}, \boldsymbol{X}\right)$$
$$= F_y\left(\boldsymbol{W}_y^{(t)}, \boldsymbol{X}\right) - F_{-y}\left(\boldsymbol{W}_{-y}^{(t)}, \boldsymbol{X}\right)$$
$$= \frac{1}{m} \sum_{l \in [2]} \sum_{r \in [m]} \sigma\left(\left\langle \boldsymbol{w}_{y,r}^{(t)}, \boldsymbol{v}^{(l)} \right\rangle\right) + \frac{1}{m} \sum_{r \in [m]} \sigma\left(\left\langle \boldsymbol{w}_{y,r}^{(t)}, \boldsymbol{\xi} \right\rangle\right)$$
$$- \frac{1}{m} \sum_{l \in [2]} \sum_{r \in [m]} \sigma\left(\left\langle \boldsymbol{w}_{-y,r}^{(t)}, \boldsymbol{v}^{(l)} \right\rangle\right) - \frac{1}{m} \sum_{r \in [m]} \sigma\left(\left\langle \boldsymbol{w}_{-y,r}^{(t)}, \boldsymbol{\xi} \right\rangle\right)$$

$$\geq -\frac{1}{m} \sum_{r \in [m]} \sigma\left(\left\langle \boldsymbol{w}_{-y,r}^{(t)}, \boldsymbol{\xi} \right\rangle\right) + \sum_{r \in [m]} \sigma\left(\left\langle \boldsymbol{w}_{y,r}^{(t)}, \boldsymbol{\mu}_y \right\rangle\right) - \frac{1}{m} \sum_{l \in [2]} \sum_{r \in [m]} \sigma\left(\left\langle \boldsymbol{w}_{-y,r}^{(t)}, \boldsymbol{v}^{(l)} \right\rangle\right)$$

$$\geq -\frac{1}{m} \sum_{r \in [m]} \sigma\left(\left\langle \boldsymbol{w}_{-y,r}^{(t)}, \boldsymbol{\xi} \right\rangle\right) + \frac{1}{m} \sum_{r \in [m]} \overline{M}_{y,r}^{(t)} - 2(2\alpha_{\mathrm{st}} + \beta_{\mathrm{st}})$$

$$= -h(\boldsymbol{\xi}) + \frac{1}{m} \sum_{r \in [m]} \overline{M}_{y,r}^{(t)} - 2(2\alpha_{\mathrm{st}} + \beta_{\mathrm{st}}),$$

where the second inequality follows from (7) and (S2). From (S3), (S8), and (16), we have

$$\frac{1}{m} \sum_{r \in [m]} \overline{M}_{y,r}^{(t)} \geq \frac{1}{12\lambda_{\mathrm{st}}} n_{\boldsymbol{\mu}} \mathrm{SNR}_{\boldsymbol{\mu}}^2 \cdot x_t$$

$$\geq \frac{1}{12\lambda_{\mathrm{st}}} n_{\boldsymbol{\mu}} \mathrm{SNR}_{\boldsymbol{\mu}}^2 \log\left(\varepsilon^{-1}\right)$$

$$\geq 4(2\alpha_{\mathrm{st}} + \beta_{\mathrm{st}}),$$

where the last inequality follows from (6). Therefore, we have

$$y f_{\mathrm{st}}\left(\boldsymbol{W}^{(t)}, \boldsymbol{X}\right) \geq -h(\boldsymbol{\xi}) + \frac{1}{2m} \sum_{r \in [m]} \overline{M}_{y,r}^{(t)}$$

and thus

$$\mathbb{P}\left[y f_{\mathrm{st}}\left(\boldsymbol{W}^{(t)}, \boldsymbol{X}\right) < 0 \,\middle|\, (\boldsymbol{X}, y) \in \mathcal{S}_{\mathrm{e}} \cup \mathcal{S}_{\mathrm{b}}\right] \leq \mathbb{P}\left[h(\boldsymbol{\xi}) > \frac{1}{2m} \sum_{r \in [m]} \overline{M}_{y,r}^{(t)}\right].$$

From Lemma C.4, we have

$$\frac{1}{2m} \sum_{r \in [m]} \overline{M}_{y,r}^{(t)} - \mathbb{E}[h(\boldsymbol{\xi})]$$

$$= \frac{1}{2m} \sum_{r \in [m]} \overline{M}_{y,r}^{(t)} - \frac{\sigma_p}{\sqrt{2\pi}m} \sum_{r \in [m]} \left\| \Pi_S \boldsymbol{w}_{-y,r}^{(t)} \right\|$$

$$\geq \frac{n_{\boldsymbol{\mu}} \mathrm{SNR}_{\boldsymbol{\mu}}^2}{24\lambda_{\mathrm{st}} m n_{\mathrm{st}}^{\frac{1}{2}}} \left(\sum_{i \in [n_{\mathrm{st}}]} \left(\sum_{r \in [m]} \overline{\rho}_{r,i}^{(t)}\right)^2\right)^{\frac{1}{2}} - \frac{20}{\sqrt{2\pi} m d^{\frac{1}{2}}} \left(\sum_{i \in [n_{\mathrm{st}}]} \left(\sum_{r \in [m]} \overline{\rho}_{r,i}^{(t)}\right)^2\right)^{\frac{1}{2}}$$

$$\geq \frac{n_{\boldsymbol{\mu}} \mathrm{SNR}_{\boldsymbol{\mu}}^2}{48\lambda_{\mathrm{st}} m n_{\mathrm{st}}^{\frac{1}{2}}} \left(\sum_{i \in [n_{\mathrm{st}}]} \left(\sum_{r \in [m]} \overline{\rho}_{r,i}^{(t)}\right)^2\right)^{\frac{1}{2}}$$

where the last inequality follows from the condition $n_{\mathrm{st}} p_{\mathrm{b}}^2 \|\boldsymbol{\nu}\|^4 \geq C_2 \sigma_p^4 d$ and (C5).

From (17), we have

$$\mathbb{P}\left[h(\boldsymbol{\xi}) > \frac{1}{2m} \sum_{r \in [m]} \overline{M}_{s,r}^{(t)}\right] = \mathbb{P}\left[h(\boldsymbol{\xi}) - \mathbb{E}[h(\boldsymbol{\xi})] > \frac{1}{2m} \sum_{r \in [m]} \overline{M}_{y,r}^{(t)} - \mathbb{E}[h(\boldsymbol{\xi})]\right]$$

$$\leq \mathbb{P}\left[h(\boldsymbol{\xi}) - \mathbb{E}[h(\boldsymbol{\xi})] > \frac{n_{\boldsymbol{\mu}} \mathrm{SNR}_{\boldsymbol{\mu}}^2}{48\lambda_{\mathrm{st}} m n_{\mathrm{st}}^{\frac{1}{2}}} \left(\sum_{i \in [n_{\mathrm{st}}]} \left(\sum_{r \in [m]} \overline{\rho}_{r,i}^{(t)}\right)^2\right)^{\frac{1}{2}}\right]$$

$$\leq \exp\left(-\frac{c n_{\boldsymbol{\mu}}^2 \|\boldsymbol{\mu}\|^4}{400 \cdot 48^2 \lambda_{\mathrm{st}}^2 \cdot n_{\mathrm{st}} \sigma_p^4 d}\right)$$

$$\leq \exp\left(-\frac{n_{\mathrm{st}}(2p_{\mathrm{e}} + p_{\mathrm{b}})^2 \|\boldsymbol{\mu}\|^4}{C_3 \sigma_p^4 d}\right),$$

with some constant $C_3 > 0$.

Using a similar argument, we can prove the upper bound on test error for the case $(\boldsymbol{X}, y) \in \mathcal{S}_{\mathrm{h}}$. In this case, we have

$$
\begin{aligned}
&y f_{\mathrm{st}}\left(\boldsymbol{W}^{(t)}, \boldsymbol{X}\right) \\
&= F_y\left(\boldsymbol{W}_y^{(t)}, \boldsymbol{X}\right) - F_{-y}\left(\boldsymbol{W}_{-y}^{(t)}, \boldsymbol{X}\right) \\
&= \frac{1}{m} \sum_{l \in [2]} \sum_{r \in [m]} \sigma\left(\left\langle \boldsymbol{w}_{y,r}^{(t)}, \boldsymbol{v}^{(l)}\right\rangle\right) + \frac{1}{m} \sum_{r \in [m]} \sigma\left(\left\langle \boldsymbol{w}_{y,r}^{(t)}, \boldsymbol{\xi}\right\rangle\right) \\
&\quad - \frac{1}{m} \sum_{l \in [2]} \sum_{r \in [m]} \sigma\left(\left\langle \boldsymbol{w}_{-y,r}^{(t)}, \boldsymbol{v}^{(l)}\right\rangle\right) - \frac{1}{m} \sum_{r \in [m]} \sigma\left(\left\langle \boldsymbol{w}_{-y,r}^{(t)}, \boldsymbol{\xi}\right\rangle\right) \\
&\geq -\frac{1}{m} \sum_{r \in [m]} \sigma\left(\left\langle \boldsymbol{w}_{-y,r}^{(t)}, \boldsymbol{\xi}\right\rangle\right) \\
&\quad + \frac{1}{m} \sum_{l \in [2]} \sum_{r \in [m]} \sigma\left(\left\langle \boldsymbol{w}_{y,r}^{(t)}, \boldsymbol{v}^{(l)}\right\rangle\right) - \frac{1}{m} \sum_{l \in [2]} \sum_{r \in [m]} \sigma\left(\left\langle \boldsymbol{w}_{-y,r}^{(t)}, \boldsymbol{v}^{(l)}\right\rangle\right) \\
&\geq -\frac{1}{m} \sum_{r \in [m]} \sigma\left(\left\langle \boldsymbol{w}_{-y,r}^{(t)}, \boldsymbol{\xi}\right\rangle\right) + \frac{2}{m} \min\left\{\sum_{r \in \mathcal{A}_y} \overline{N}_{y,r}^{(t)}, -\sum_{r \in \mathcal{B}_y} \overline{N}_{y,r}^{(t)}\right\} - 2(2\alpha_{\mathrm{st}} + \beta_{\mathrm{st}}) \\
&= -h(\boldsymbol{\xi}) + \frac{2}{m} \min\left\{\sum_{r \in \mathcal{A}_y} \overline{N}_{y,r}^{(t)}, -\sum_{r \in \mathcal{B}_y} \overline{N}_{y,r}^{(t)}\right\} - 2(2\alpha_{\mathrm{st}} + \beta_{\mathrm{st}})
\end{aligned}
$$

where the first inequality follows from (7) and (S2). From (S3), (S8), and (16), we have

$$
\begin{aligned}
\frac{1}{m} \sum_{r \in \mathcal{A}_y} \overline{N}_{y,r}^{(t)} - \frac{1}{m} \sum_{r \in \mathcal{B}_y} \overline{N}_{y,r}^{(t)} &\geq \frac{1}{12\lambda_{\mathrm{st}}} n_{\boldsymbol{\nu}} \mathrm{SNR}_{\boldsymbol{\nu}}^2 \cdot x_t \\
&\geq \frac{1}{12\lambda_{\mathrm{st}}} n_{\boldsymbol{\nu}} \mathrm{SNR}_{\boldsymbol{\nu}}^2 \cdot \log\left(\frac{\eta \sigma_p^2 d}{16 m n_{\mathrm{st}} \exp(\kappa_{\mathrm{st}}/4)} t\right) \\
&\geq \frac{1}{12\lambda_{\mathrm{st}}} n_{\boldsymbol{\nu}} \mathrm{SNR}_{\boldsymbol{\nu}}^2 \log\left(\varepsilon^{-1}\right) \\
&\geq 4(2\alpha_{\mathrm{st}} + \beta_{\mathrm{st}}),
\end{aligned}
\tag{18}
$$

where the last inequality follows from (6). Therefore, we have

$$
y f_{\mathrm{st}}\left(\boldsymbol{W}^{(t)}, \boldsymbol{X}\right) \geq -h(\boldsymbol{\xi}) + \frac{1}{m} \min\left\{\sum_{r \in \mathcal{A}_y} \overline{N}_{y,r}^{(t)}, -\sum_{r \in \mathcal{B}_y} \overline{N}_{y,r}^{(t)}\right\}
$$

and thus

$$
\mathbb{P}\left[y f_{\mathrm{st}}\left(\boldsymbol{W}^{(t)}, \boldsymbol{X}\right) < 0 \,\Big|\, (\boldsymbol{X}, y) \in \mathcal{S}_{\mathrm{h}}\right] \leq \mathbb{P}\left[h(\boldsymbol{\xi}) > \frac{1}{m} \min\left\{\sum_{r \in \mathcal{A}_y} \overline{N}_{y,r}^{(t)}, -\sum_{r \in \mathcal{B}_y} \overline{N}_{y,r}^{(t)}\right\}\right].
$$

From Lemma C.4 and Condition 3.1, we have

$$
\begin{aligned}
&\frac{1}{m} \min\left\{\sum_{r \in \mathcal{A}_y} \overline{N}_{y,r}^{(t)}, -\sum_{r \in \mathcal{B}_y} \overline{N}_{y,r}^{(t)}\right\} - \mathbb{E}[h(\boldsymbol{\xi})] \\
&= \frac{1}{m} \min\left\{\sum_{r \in \mathcal{A}_y} \overline{N}_{y,r}^{(t)}, -\sum_{r \in \mathcal{B}_y} \overline{N}_{y,r}^{(t)}\right\} - \frac{\sigma_p}{\sqrt{2\pi} m} \sum_{r \in [m]} \left\|\Pi_S \boldsymbol{w}_{-y,r}^{(t)}\right\|
\end{aligned}
$$

$$\geq \frac{1}{12\lambda_{\mathrm{st}}mn_{\mathrm{st}}}n_{\boldsymbol{\nu}}\mathrm{SNR}_{\boldsymbol{\nu}}^2 \cdot \sum_{i\in[n_{\mathrm{st}}]}\sum_{r\in[m]}\overline{\rho}_{r,i}^{(t)} - \frac{3}{\sqrt{2\pi}mn_{\mathrm{st}}^{\frac{1}{2}}d^{\frac{1}{2}}}\sum_{i\in[n_{\mathrm{st}}]}\sum_{r\in[m]}\overline{\rho}_{r,i}^{(t)}$$

$$\geq \frac{1}{24\lambda_{\mathrm{st}}mn_{\mathrm{st}}}n_{\boldsymbol{\nu}}\mathrm{SNR}_{\boldsymbol{\nu}}^2 \sum_{i\in[n_{\mathrm{st}}]}\sum_{r\in[m]}\overline{\rho}_{r,i}^{(t)},$$

where the last inequality follows from the condition given in the statement. From (17), we have

$$\mathbb{P}\left[h(\boldsymbol{\xi}) > \frac{1}{m}\min\left\{\sum_{r\in\mathcal{A}_y}\overline{N}_{y,r}^{(t)}, -\sum_{r\in\mathcal{B}_y}\overline{N}_{y,r}^{(t)}\right\}\right]$$

$$= \mathbb{P}\left[h(\boldsymbol{\xi}) - \mathbb{E}[h(\boldsymbol{\xi})] > \frac{1}{m}\min\left\{\sum_{r\in\mathcal{A}_y}\overline{N}_{y,r}^{(t)}, -\sum_{r\in\mathcal{B}_y}\overline{N}_{y,r}^{(t)}\right\} - \mathbb{E}[h(\boldsymbol{\xi})]\right]$$

$$\leq \mathbb{P}\left[h(\boldsymbol{\xi}) - \mathbb{E}[h(\boldsymbol{\xi})] > \frac{1}{24\lambda_{\mathrm{st}}mn_{\mathrm{st}}}n_{\boldsymbol{\nu}}\mathrm{SNR}_{\boldsymbol{\nu}}^2 \sum_{i\in[n_{\mathrm{st}}]}\sum_{r\in[m]}\overline{\rho}_{r,i}^{(t)}\right]$$

$$\leq \exp\left(-\frac{cn_{\boldsymbol{\nu}}^2\|\boldsymbol{\nu}\|^4}{9\cdot 24^2\lambda_{\mathrm{st}}^2 \cdot n_{\mathrm{st}}\sigma_p^4 d}\right)$$

$$\leq \exp\left(-\frac{n_{\mathrm{st}}p_{\mathrm{b}}^2\|\boldsymbol{\nu}\|^4}{C_3\sigma_p^4 d}\right),$$

with some constant $C_3 > 0$.

### C.3.2 Test Error Lower Bound

We consider the case $(\boldsymbol{X}, y) \in \mathcal{S}_{\mathrm{h}}$. Define $g : S \to \mathbb{R}$ as $g(\boldsymbol{z}) := \frac{1}{m}\sum_{r\in[m]}\sigma\left(\left\langle\boldsymbol{w}_{1,r}^{(t)}, \boldsymbol{z}\right\rangle\right) - \frac{1}{m}\sum_{r\in[m]}\sigma\left(\left\langle\boldsymbol{w}_{-1,r}^{(t)}, \boldsymbol{z}\right\rangle\right)$ for any $\boldsymbol{z} \in S$. Then, we have

$$yf_{\mathrm{st}}\left(\boldsymbol{W}^{(t)}, \boldsymbol{X}\right)$$

$$= F_y\left(\boldsymbol{W}_y^{(t)}, \boldsymbol{X}\right) - F_{-y}\left(\boldsymbol{W}_{-y}^{(t)}, \boldsymbol{X}\right)$$

$$= \frac{1}{m}\sum_{l\in[2]}\sum_{r\in[m]}\sigma\left(\left\langle\boldsymbol{w}_{y,r}^{(t)}, \boldsymbol{v}^{(l)}\right\rangle\right) + \frac{1}{m}\sum_{r\in[m]}\sigma\left(\left\langle\boldsymbol{w}_{y,r}^{(t)}, \boldsymbol{\xi}\right\rangle\right)$$

$$\quad - \frac{1}{m}\sum_{l\in[2]}\sum_{r\in[m]}\sigma\left(\left\langle\boldsymbol{w}_{-y,r}^{(t)}, \boldsymbol{v}^{(l)}\right\rangle\right) - \frac{1}{m}\sum_{r\in[m]}\sigma\left(\left\langle\boldsymbol{w}_{-y,r}^{(t)}, \boldsymbol{\xi}\right\rangle\right)$$

$$\leq \frac{1}{m}\sum_{r\in[m]}\sigma\left(\left\langle\boldsymbol{w}_{y,r}^{(t)}, \boldsymbol{\xi}\right\rangle\right) - \frac{1}{m}\sum_{r\in[m]}\sigma\left(\left\langle\boldsymbol{w}_{-y,r}^{(t)}, \boldsymbol{\xi}\right\rangle\right) + \frac{1}{m}\sum_{l\in[2]}\sum_{r\in[m]}\sigma\left(\left\langle\boldsymbol{w}_{y,r}^{(t)}, \boldsymbol{v}^{(l)}\right\rangle\right)$$

$$\leq yg(\boldsymbol{\xi}) + \frac{2}{m}\max\left\{\sum_{r\in\mathcal{A}_y}\overline{N}_{y,r}^{(t)}, -\sum_{r\in\mathcal{B}_y}\overline{N}_{y,r}^{(t)}\right\} + 2\alpha_{\mathrm{st}}$$

$$\leq yg(\boldsymbol{\xi}) + \frac{3}{m}\max\left\{\sum_{r\in\mathcal{A}_y}\overline{N}_{y,r}^{(t)}, -\sum_{r\in\mathcal{B}_y}\overline{N}_{y,r}^{(t)}\right\}$$

$$\leq yg(\boldsymbol{\xi}) + \frac{3}{m}\max_{s\in\{\pm 1\}}\left\{\sum_{r\in\mathcal{A}_s}\overline{N}_{s,r}^{(t)}, -\sum_{r\in\mathcal{B}_s}\overline{N}_{s,r}^{(t)}\right\},$$

where the second inequality follows from (18). Therefore, we have

$$\mathbb{P}\left[yf_{\mathrm{st}}\left(\boldsymbol{W}^{(t)}, \boldsymbol{X}\right) \,\Big|\, (\boldsymbol{X}, y) \in \mathcal{S}_{\mathrm{h}}\right] \geq \frac{1}{2}\mathbb{P}\left[|g(\boldsymbol{\xi})| \geq \frac{3}{m}\max_{s\in\{\pm 1\}}\left\{\sum_{r\in\mathcal{A}_s}\overline{N}_{s,r}^{(t)}, -\sum_{r\in\mathcal{B}_s}\overline{N}_{s,r}^{(t)}\right\}\right].$$

We define the set

$$\mathbf{\Omega} := \left\{ \boldsymbol{z} \in S : |g(\boldsymbol{z})| \geq \frac{3}{m} \max_{s \in \{\pm 1\}} \left\{ \sum_{r \in \mathcal{A}_s} \overline{N}_{s,r}^{(t)}, - \sum_{r \in \mathcal{B}_s} \overline{N}_{s,r}^{(t)} \right\} \right\}.$$

We immediately obtain $\mathbb{P}\left[ y f_{\mathrm{st}}\left( \boldsymbol{W}^{(t)}, \boldsymbol{X} \right) \big| (\boldsymbol{X}, y) \in \mathcal{S}_{\mathrm{h}} \right] \geq \frac{1}{2}\mathbb{P}[\boldsymbol{\xi} \in \mathbf{\Omega}]$ and thus we will characterize $\mathbb{P}[\boldsymbol{\xi} \in \mathbf{\Omega}]$. Denote $\boldsymbol{\zeta} = C_6 p_{\mathrm{b}} \mathrm{SNR}_{\boldsymbol{\nu}}^2 \cdot \sum_{\substack{i \in [n_{\mathrm{st}}] \\ \hat{y}_i = 1}} \tilde{\boldsymbol{\xi}}_i$, where $C_6 > 0$ is some small constant. Then, we

have

$$\|\boldsymbol{\zeta}\| \leq C_6 p_{\mathrm{b}} \mathrm{SNR}_{\boldsymbol{\nu}}^2 \left( \sum_{i \in [n_{\mathrm{st}}]} \|\tilde{\boldsymbol{\xi}}_i\|^2 + \sum_{i \in [n_{\mathrm{st}}]} \sum_{j \in [n_{\mathrm{st}}] \setminus \{i\}} \left| \langle \tilde{\boldsymbol{\xi}}_i, \tilde{\boldsymbol{\xi}}_j \rangle \right| \right)^{\frac{1}{2}}$$

$$\leq C_6 p_{\mathrm{b}} \mathrm{SNR}_{\boldsymbol{\nu}}^2 \sqrt{\frac{3(1 + \beta_{\mathrm{st}}) n_{\mathrm{st}} \sigma_p^2 d}{2}}$$

$$= \sqrt{\frac{2 C_6^2 n_{\mathrm{st}} p_{\mathrm{b}}^2 \|\boldsymbol{\nu}\|^4}{\sigma_p^2 d}}$$

$$\leq 0.02 \sigma_p, \tag{19}$$

where the first inequality follows from (8) and the last follows from the statement condition $n_{\mathrm{st}} p_{\mathrm{b}}^2 \|\boldsymbol{\nu}\|^4 \leq C_4 \sigma_p^4 d$ and the small choice of $C_6$. Also, for any $r \in [m]$, we have

$$\sigma\left( \left\langle \boldsymbol{w}_{1,r}^{(t)}, \boldsymbol{\xi} + \boldsymbol{\zeta} \right\rangle \right) - \sigma\left( \left\langle \boldsymbol{w}_{1,r}^{(t)}, \boldsymbol{\xi} \right\rangle \right) + \sigma\left( \left\langle \boldsymbol{w}_{1,r}^{(t)}, -\boldsymbol{\xi} + \boldsymbol{\zeta} \right\rangle \right) - \sigma\left( \left\langle \boldsymbol{w}_{1,r}^{(t)}, -\boldsymbol{\xi} \right\rangle \right)$$

$$\geq \mathbb{1}\left[ \left\langle \boldsymbol{w}_{1,r}^{(t)}, \boldsymbol{\xi} \right\rangle > 0 \right] \left\langle \boldsymbol{w}_{1,r}^{(t)}, \boldsymbol{\zeta} \right\rangle + \mathbb{1}\left[ \left\langle \boldsymbol{w}_{1,r}^{(t)}, -\boldsymbol{\xi} \right\rangle > 0 \right] \left\langle \boldsymbol{w}_{1,r}^{(t)}, \boldsymbol{\zeta} \right\rangle$$

$$= \left\langle \boldsymbol{w}_{1,r}^{(t)}, \boldsymbol{\zeta} \right\rangle$$

$$= C_6 p_{\mathrm{b}} \mathrm{SNR}_{\boldsymbol{\nu}}^2 \left[ \sum_{\substack{i \in [n_{\mathrm{st}}] \\ \hat{y}_i = 1}} \overline{\rho}_{r,i}^{(t)} - \sum_{\substack{i \in [n_{\mathrm{st}}] \\ \hat{y}_i = 1}} \sum_{j \in [n_{\mathrm{st}}] \setminus \{i\}} \rho_{1,r,j}^{(t)} \frac{\langle \tilde{\boldsymbol{\xi}}_i, \tilde{\boldsymbol{\xi}}_j \rangle}{\|\tilde{\boldsymbol{\xi}}_j\|^2} + \sum_{\substack{i \in [n_{\mathrm{st}}] \\ \hat{y}_i = 1}} \left\langle \boldsymbol{w}_{1,r}^{(0)}, \tilde{\boldsymbol{\xi}}_i \right\rangle \right]$$

$$\geq C_6 p_{\mathrm{b}} \mathrm{SNR}_{\boldsymbol{\nu}}^2 \left[ \sum_{\substack{i \in [n_{\mathrm{st}}] \\ \hat{y}_i = 1}} \overline{\rho}_{r,i}^{(t)} - 4\beta_{\mathrm{st}} \log T^* - n_{\mathrm{st}} \alpha_{\mathrm{st}} \right]$$

where the first inequality follows from the convexity of ReLU, and the second inequality follows from (S1), (7), and (8). In addition, for any $r \in [m]$, we have

$$\sigma\left( \left\langle \boldsymbol{w}_{-1,r}^{(t)}, \boldsymbol{\xi} + \boldsymbol{\zeta} \right\rangle \right) - \sigma\left( \left\langle \boldsymbol{w}_{-1,r}^{(t)}, \boldsymbol{\xi} \right\rangle \right) + \sigma\left( \left\langle \boldsymbol{w}_{-1,r}^{(t)}, -\boldsymbol{\xi} + \boldsymbol{\zeta} \right\rangle \right) - \sigma\left( \left\langle \boldsymbol{w}_{-1,r}^{(t)}, -\boldsymbol{\xi} \right\rangle \right)$$

$$\leq 2 \left| \left\langle \boldsymbol{w}_{-1,r}^{(t)}, \boldsymbol{\zeta} \right\rangle \right|$$

$$\leq 2\lambda \left[ \sum_{\substack{i \in [n_{\mathrm{st}}] \\ \hat{y}_i = 1}} \left| \rho_{-r,i}^{(t)} \right| + \sum_{\substack{i \in [n_{\mathrm{st}}] \\ \hat{y}_i = 1}} \sum_{j \in [n_{\mathrm{st}}] \setminus \{i\}} \left| \rho_{-1,r,j}^{(t)} \right| \frac{\left| \langle \tilde{\boldsymbol{\xi}}_i, \tilde{\boldsymbol{\xi}}_j \rangle \right|}{\|\tilde{\boldsymbol{\xi}}_j\|^2} + \sum_{\substack{i \in [n_{\mathrm{st}}] \\ \hat{y}_i = 1}} \left| \left\langle \boldsymbol{w}_{-1,r}^{(0)}, \tilde{\boldsymbol{\xi}}_i \right\rangle \right| \right]$$

$$\leq 2 C_6 p_{\mathrm{b}} \mathrm{SNR}_{\boldsymbol{\nu}}^2 \left( n_{\mathrm{st}}(\alpha_{\mathrm{st}} + 5\beta_{\mathrm{st}} \log T^*) + 4\beta_{\mathrm{st}} \log T^* + n_{\mathrm{st}} \alpha_{\mathrm{st}} \right)$$

$$= 2 C_6 p_{\mathrm{b}} \mathrm{SNR}_{\boldsymbol{\nu}}^2 n_{\mathrm{st}}(2\alpha_{\mathrm{st}} + 9\beta_{\mathrm{st}} \log T^*),$$

where the first inequality holds since ReLU is 1-Lipschitz and the second inequality follows from (S1), (7), and (8). Therefore, we have

$$g(\boldsymbol{\xi} + \boldsymbol{\zeta}) - g(\boldsymbol{\xi}) + g(-\boldsymbol{\xi} + \boldsymbol{\zeta}) - g(-\boldsymbol{\xi})$$

$$\geq \frac{C_6 p_{\mathrm{b}} \mathrm{SNR}_{\boldsymbol{\nu}}^2}{m} \left[ \sum_{\substack{i \in [n_{\mathrm{st}}] \\ \hat{y}_i = 1}} \overline{\rho}_{r,i}^{(t)} - n_{\mathrm{st}} (7\alpha_{\mathrm{st}} + 12\beta_{\mathrm{st}} \log T^*) \right]$$

$$\geq \frac{C_6 p_{\mathrm{b}} \mathrm{SNR}_{\boldsymbol{\nu}}^2}{2m} \sum_{\substack{i \in [n_{\mathrm{st}}] \\ \hat{y}_i = 1}} \overline{\rho}_{r,i}^{(t)}$$

$$\geq \frac{C_6 p_{\mathrm{b}} \mathrm{SNR}_{\boldsymbol{\nu}}^2}{2m} \cdot \frac{\left| \mathcal{C}_{\boldsymbol{\mu}_1}^{(1)} \right| + \left| \mathcal{C}_{\boldsymbol{\nu}_1}^{(1)} \right| + \left| \mathcal{C}_{-\boldsymbol{\nu}_1}^{(1)} \right|}{3\lambda_{\mathrm{st}} n_{\boldsymbol{\nu}} \mathrm{SNR}_{\boldsymbol{\nu}}^2} \cdot \max_{s \in \{\pm 1\}} \left\{ \sum_{r \in \mathcal{A}_s} \overline{N}_{s,r}^{(t)}, - \sum_{r \in \mathcal{B}_s} \overline{N}_{s,r}^{(t)} \right\}$$

$$\geq \frac{12}{m} \max_{s \in \{\pm 1\}} \left\{ \sum_{r \in \mathcal{A}_s} \overline{N}_{s,r}^{(t)}, - \sum_{r \in \mathcal{B}_s} \overline{N}_{s,r}^{(t)} \right\},$$

where the second inequality follows from (16) and (6), the third inequality follows from (S3) and the last inequality follows from the choice of $C_6 > 0$ and

$$\left| \mathcal{C}_{\boldsymbol{\mu}_1}^{(1)} \right| + \left| \mathcal{C}_{\boldsymbol{\nu}_1}^{(1)} \right| + \left| \mathcal{C}_{-\boldsymbol{\nu}_1}^{(1)} \right| \geq \left( 1 - C_{\mathrm{st}}^{-1} \right) \cdot n_{\boldsymbol{\mu}} + 2(1 - C_{\mathrm{st}}^{-1}) n_{\boldsymbol{\nu}} = \frac{\left( 1 - C_{\mathrm{st}}^{-1} \right) (p_{\mathrm{e}} + p_{\mathrm{b}}) n_{\mathrm{st}}}{2} \geq \frac{n_{\mathrm{st}}}{8}.$$

By the pigeonhole principle, it implies that at least one of $\boldsymbol{\xi}, -\boldsymbol{\xi}, \boldsymbol{\xi} + \boldsymbol{\zeta}, -\boldsymbol{\xi} + \boldsymbol{\zeta}$ belongs to $\boldsymbol{\Omega}$. Hence,

$$\mathbb{P}[\boldsymbol{\xi} \in \boldsymbol{\Omega}] + \mathbb{P}[-\boldsymbol{\xi} \in \boldsymbol{\Omega}] + \mathbb{P}[\boldsymbol{\xi} + \boldsymbol{\zeta} \in \boldsymbol{\Omega}] + \mathbb{P}[-\boldsymbol{\xi} + \boldsymbol{\zeta} \in \boldsymbol{\Omega}] \geq 1.$$

Also, from symmetry, we have $\mathbb{P}[\boldsymbol{\xi} \in \boldsymbol{\Omega}] = \mathbb{P}[-\boldsymbol{\xi} \in \boldsymbol{\Omega}]$ and $\mathbb{P}[-\boldsymbol{\xi} + \boldsymbol{\zeta} \in \boldsymbol{\Omega}] = \mathbb{P}[\boldsymbol{\xi} - \boldsymbol{\zeta} \in \boldsymbol{\Omega}]$. The following lemma allows us to relate the probability $\mathbb{P}[\boldsymbol{\xi} \in \boldsymbol{\Omega}]$ to the probabilities $\mathbb{P}[\boldsymbol{\xi} \pm \boldsymbol{\zeta} \in \boldsymbol{\Omega}]$.

**Lemma C.5** (Direct from Proposition 2.1 in Devroye et al. (2018)). *For any $\boldsymbol{v} \in S$ the total variation distance* $\mathrm{TV}(\cdot, \cdot)$ *between* $\mathcal{N}(0, \sigma_p^2 \boldsymbol{\Lambda})$ *and* $\mathcal{N}(\boldsymbol{v}, \sigma_p^2 \boldsymbol{\Lambda})$ *is smaller than* $\frac{\|\boldsymbol{v}\|}{2\sigma_p}$.

By Lemma C.5 and (19), we have

$$\left| \mathbb{P}[\boldsymbol{\xi} \in \Omega] - \mathbb{P}[\boldsymbol{\xi} \in \Omega \pm \boldsymbol{\zeta}] \right| \leq \mathrm{TV} \left( \mathcal{N}(\boldsymbol{0}, \sigma_p^2 \boldsymbol{\Lambda}), \mathcal{N}(\pm \boldsymbol{\zeta}, \sigma_p^2 \boldsymbol{\Lambda}) \right) \leq \frac{\|\boldsymbol{\zeta}\|}{2\sigma_p} \leq 0.01.$$

Therefore, we have

$$1 \leq \mathbb{P}[\boldsymbol{\xi} \in \boldsymbol{\Omega}] + \mathbb{P}[-\boldsymbol{\xi} \in \boldsymbol{\Omega}] + \mathbb{P}[\boldsymbol{\xi} + \boldsymbol{\zeta} \in \boldsymbol{\Omega}] + \mathbb{P}[-\boldsymbol{\xi} + \boldsymbol{\zeta} \in \boldsymbol{\Omega}] \leq 4\mathbb{P}[\boldsymbol{\xi} \in \boldsymbol{\Omega}] + 0.02$$

and thus $\mathbb{P}[\boldsymbol{\xi} \in \boldsymbol{\Omega}] \geq 0.24$. We conclude that

$$\mathbb{P} \left[ y f_{\mathrm{st}} \left( \boldsymbol{W}^{(t)}, \boldsymbol{X} \right) \middle| (\boldsymbol{X}, y) \in \mathcal{S}_{\mathrm{h}} \right] \geq 0.12.$$

# D   Proof of Theorem 3.6

It suffices to prove the following restatements of Theorem 3.6.

**Theorem D.1** (Weak-to-Strong Training, Data-Abundant Regime). *Let $\boldsymbol{W}^{(t)}$ be the iterates of the weak-to-strong training, with the weak model $f_{\mathrm{wk}}(\boldsymbol{w}^*, \cdot)$ satisfying the conclusion of Theorem 3.3. For any $\delta \in (0, 1)$ satisfying Condition 3.5, with probability at least $1 - \delta$, there exists early stopping time $T_{\mathrm{es}} = \mathcal{O}(\eta^{-1} m (2p_{\mathrm{e}} + p_{\mathrm{b}})^{-1} \|\boldsymbol{\mu}\|^{-2})$ such that the following statements hold:*

1. *The early stopped strong model $f_{\mathrm{st}}\left(\boldsymbol{W}^{(T_{\mathrm{es}})}, \cdot\right)$ perfectly fits training data having correct label (i.e. $\hat{y}_i = \tilde{y}_i$) but fails to training data with flipped label (i.e. $\hat{y}_i \neq \tilde{y}_i$). In other words, the model predicts the true label $\tilde{y}_i$ for any training data point $\tilde{\boldsymbol{X}}_i$.*

2. *Let $(\boldsymbol{X}, y) \sim \mathcal{D}$ be an unseen test example, independent of the training set $\{(\tilde{\boldsymbol{X}}_i, \hat{y}_i)\}_{i=1}^{n_{\mathrm{st}}}$. We have*

$$\mathbb{P}\left[y f_{\mathrm{st}}\left(\boldsymbol{W}^{(T_{\mathrm{es}})}, \boldsymbol{X}\right) < 0 \,\middle|\, (\boldsymbol{X}, y) \in \mathcal{S}_{\mathrm{e}} \cup \mathcal{S}_{\mathrm{b}}\right] \leq \exp\left(-\frac{n_{\mathrm{st}}(2p_{\mathrm{e}} + p_{\mathrm{b}})^2 \|\boldsymbol{\mu}\|^4}{C_5 \sigma_p^4 d}\right),$$

*and*

$$\mathbb{P}\left[y f_{\mathrm{st}}\left(\boldsymbol{W}^{(T_{\mathrm{es}})}, \boldsymbol{X}\right) < 0 \,\middle|\, (\boldsymbol{X}, y) \in \mathcal{S}_{\mathrm{h}}\right] \leq \exp\left(-\frac{n_{\mathrm{st}} p_{\mathrm{b}}^2 \|\boldsymbol{\nu}\|^4}{C_5 \sigma_p^4 d}\right),$$

*Here, $C_5 > 0$ is a constant.*

For the proof, we first analyze the early training dynamics and characterize the early stopping iteration (Appendix D.1). We then show that the early-stopped model perfectly fits the training data with true labels (Appendix D.2), and finally, we establish a bound on the test error (Appendix D.3).

## D.1   Analyzing Early Phase

First, we establish upper bounds on the noise coefficients.

**Lemma D.2.** *Under Condition 3.5 and the event $E_{\mathrm{st}}$, for any $t \in [0, T^*]$, $s \in \{\pm 1\}$, $r \in [m]$ and $i \in [n_{\mathrm{st}}]$, it holds that*

$$\left|\rho_{s,r,i}^{(t)}\right| \leq \frac{3\eta\sigma_p^2 d}{2mn_{\mathrm{st}}} t, \quad \left|\left\langle \boldsymbol{w}_{s,r}^{(t)}, \tilde{\boldsymbol{\xi}}_i \right\rangle\right| \leq \alpha_{\mathrm{st}} + \frac{3\eta\sigma_p^2 d}{mn_{\mathrm{st}}} t.$$

*Proof of Lemma D.2.* We fix arbitrary $s \in \{\pm 1\}$, $r \in [m]$ and $i \in [n_{\mathrm{st}}]$. For any iteration $0 < t \leq T^*$, we have

$$\left|\rho_{s,r,i}^{(t)}\right| \leq \left|\rho_{s,r,i}^{(t-1)}\right| + \frac{\eta}{mn_{\mathrm{st}}} \tilde{g}_i^{(t-1)} \|\tilde{\boldsymbol{\xi}}_i\|^2 \leq \left|\rho_{s,r,i}^{(t-1)}\right| + \frac{3\eta\sigma_p^2 d}{2mn_{\mathrm{st}}} \leq \cdots \leq \left|\rho_{s,r,i}^{(0)}\right| + \frac{3\eta\sigma_p^2 d}{2mn_{\mathrm{st}}} t = \frac{3\eta\sigma_p^2 d}{2mn_{\mathrm{st}}} t,$$

where the first inequality is due to the triangular inequality and the others are due to (8). Therefore, we have

$$\left|\left\langle \boldsymbol{w}_{s,r}^{(t)}, \tilde{\boldsymbol{\xi}}_i \right\rangle\right| \leq \left|\left\langle \boldsymbol{w}_{s,r}^{(0)}, \tilde{\boldsymbol{\xi}}_i \right\rangle\right| + \left|\rho_{s,r,i}^{(t)}\right| + \sum_{j \in [n_{\mathrm{st}}] \setminus \{i\}} \left|\rho_{s,r,j}^{(t)}\right| \frac{\left|\langle \tilde{\boldsymbol{\xi}}_i, \tilde{\boldsymbol{\xi}}_j \rangle\right|}{\|\tilde{\boldsymbol{\xi}}_j\|^2}$$

$$\leq \alpha_{\mathrm{st}} + \frac{3\eta\sigma_p^2 d}{2mn_{\mathrm{st}}} t (1 + \beta_{\mathrm{st}})$$

$$\leq \alpha_{\mathrm{st}} + \frac{3\eta\sigma_p^2 d}{mn_{\mathrm{st}}} t,$$

where the second inequality follows from (7) and (8). □

The following lemma can be inductively applied when we characterize the early phase of learning dynamics.

**Lemma D.3.** *Suppose the iteration $\tau \in \left[0, \frac{mn_{\mathrm{st}}}{\eta\sigma_p^2 d \log T^*}\right]$ satisfy the following:*

1. $\frac{1}{m} \sum_{r \in [m]} \overline{M}_{1,r}^{(\tau)}, \frac{1}{m} \sum_{r \in [m]} \overline{M}_{-1,r}^{(\tau)} < \frac{1}{2}.$

2. For each $s \in \{\pm 1\}$, it holds that $\overline{M}_{s,r}^{(\tau)}, \left\langle \boldsymbol{w}_{s,r}^{(\tau)}, \boldsymbol{\mu}_s \right\rangle > 0$ if $r \in \mathcal{M}_s$ and $\overline{M}_{s,r}^{(\tau)} = 0$ if $r \notin \mathcal{M}_s$.

3. For each $s \in \{\pm 1\}$, it holds that $\overline{N}_{s,r}^{(\tau)}, \left\langle \boldsymbol{w}_{s,r}^{(\tau)}, \boldsymbol{\nu}_s \right\rangle > 0$ if $r \in \mathcal{A}_s$ and $\overline{N}_{s,r}^{(\tau)}, \left\langle \boldsymbol{w}_{s,r}^{(\tau)}, \boldsymbol{\nu}_s \right\rangle < 0$ if $r \in \mathcal{B}_s$.

4. $\frac{1}{60} \sum_{r \in [m]} \overline{M}_{-1,r}^{(\tau)} \leq \sum_{r \in [m]} \overline{M}_{1,r}^{(\tau)} \leq 60 \sum_{r \in [m]} \overline{M}_{-1,r}^{(\tau)}.$

5. For each $s, s' \in \{\pm 1\}$,

$$\frac{p_{\mathrm{b}} \|\boldsymbol{\nu}\|^2}{120(2p_{\mathrm{e}} + p_{\mathrm{b}}) \|\boldsymbol{\mu}\|^2} \sum_{r \in [m]} \overline{M}_{s',r}^{(\tau)} \leq \sum_{r \in \mathcal{A}_s} \overline{N}_{s,r}^{(\tau)}, \quad -\sum_{r \in \mathcal{B}_s} \overline{N}_{s,r}^{(\tau)} \leq \sum_{r \in [m]} \overline{M}_{s',r}^{(\tau)}$$

6. For any $s \in \{\pm 1\}$ and $r \in [m]$, it holds that $\left| \underline{M}_{s,r}^{(\tau)} \right|, \left| \underline{N}_{s,r}^{(\tau)} \right| \leq \alpha_{\mathrm{st}} + \beta_{\mathrm{st}}.$

*Then under Condition 3.5 and the event $E_{\mathrm{st}}$, the following hold:*

1. For any $s \in \{\pm 1\}$, it holds that $\overline{M}_{s,r}^{(\tau+1)} \geq \overline{M}_{s,r}^{(\tau)}$ if $r \in [m]$, $\overline{N}_{s,r}^{(\tau+1)} \geq \overline{N}_{s,r}^{(\tau)}$ if $r \in \mathcal{A}_s$, and $\overline{N}_{s,r}^{(\tau+1)} \leq \overline{N}_{s,r}^{(\tau)}$ if $r \in \mathcal{B}_s$.

2. For each $s \in \{\pm 1\}$, it holds that $\overline{M}_{s,r}^{(\tau+1)}, \left\langle \boldsymbol{w}_{s,r}^{(\tau+1)}, \boldsymbol{\mu}_s \right\rangle > 0$ if $r \in \mathcal{M}_s$ and $\overline{M}_{s,r}^{(\tau+1)} = 0$ if $r \notin \mathcal{M}_s$.

3. For each $s \in \{\pm 1\}$, it holds that $\overline{N}_{s,r}^{(\tau+1)}, \left\langle \boldsymbol{w}_{s,r}^{(\tau+1)}, \boldsymbol{\nu}_s \right\rangle > 0$ if $r \in \mathcal{A}_s$ and $\overline{N}_{s,r}^{(\tau+1)}, \left\langle \boldsymbol{w}_{s,r}^{(\tau+1)}, \boldsymbol{\nu}_s \right\rangle < 0$ if $r \in \mathcal{B}_s$.

4. For each $s \in \{\pm 1\}$,

$$\frac{1}{m} \sum_{r \in [m]} \overline{M}_{s,r}^{(\tau+1)} \geq \frac{1}{m} \sum_{r \in [m]} \overline{M}_{s,r}^{(\tau)} + \frac{\eta(2p_{\mathrm{e}} + p_{\mathrm{b}}) \|\boldsymbol{\mu}\|^2}{80m}.$$

5. For each $s \in \{\pm 1\}$,

$$\frac{1}{m} \sum_{r \in \mathcal{A}_s} \overline{N}_{s,r}^{(\tau+1)} - \frac{1}{m} \sum_{r \in \mathcal{A}_s} \overline{N}_{s,r}^{(\tau)} \geq \frac{\eta p_{\mathrm{b}} \|\boldsymbol{\nu}\|^2}{160m}, \quad -\frac{1}{m} \sum_{r \in \mathcal{B}_s} \overline{N}_{s,r}^{(\tau+1)} + \frac{1}{m} \sum_{r \in \mathcal{B}_s} \overline{N}_{s,r}^{(\tau)} \geq \frac{\eta p_{\mathrm{b}} \|\boldsymbol{\nu}\|^2}{160m}.$$

6. $\frac{1}{60} \sum_{r \in [m]} \overline{M}_{-1,r}^{(\tau+1)} \leq \sum_{r \in [m]} \overline{M}_{1,r}^{(\tau+1)} \leq 60 \sum_{r \in [m]} \overline{M}_{-1,r}^{(\tau+1)}.$

7. For each $s, s' \in \{\pm 1\}$,

$$\frac{p_{\mathrm{b}} \|\boldsymbol{\nu}\|^2}{120(2p_{\mathrm{e}} + p_{\mathrm{b}}) \|\boldsymbol{\mu}\|^2} \sum_{r \in [m]} \overline{M}_{s,r}^{(\tau+1)} \leq \sum_{r \in \mathcal{A}_s} \overline{N}_{s',r}^{(\tau+1)}, \quad -\sum_{r \in \mathcal{B}_s} \overline{N}_{s',r}^{(\tau+1)} \leq \sum_{r \in [m]} \overline{M}_{s,r}^{(\tau+1)}.$$

8. For any $s \in \{\pm 1\}$ and $r \in [m]$, $\left| \underline{M}_{s,r}^{(\tau+1)} \right|, \left| \underline{N}_{s,r}^{(\tau+1)} \right| \leq \alpha_{\mathrm{st}} + \beta_{\mathrm{st}}.$

*Proof of Lemma D.3.* For any $i \in [n_{\mathrm{st}}]$, we have

$$\hat{y}_i f_{\mathrm{st}} \left( \boldsymbol{W}^{(\tau)}, \tilde{\boldsymbol{X}}_i \right) = F_{\hat{y}_i} \left( \boldsymbol{W}_{\hat{y}_i}^{(\tau)}, \tilde{\boldsymbol{X}}_i \right) - F_{-\hat{y}_i} \left( \boldsymbol{W}_{\hat{y}_i}^{(\tau)}, \tilde{\boldsymbol{X}}_i \right)$$

$$\leq F_{\hat{y}_i} \left( \boldsymbol{W}_{\hat{y}_i}^{(\tau)}, \tilde{\boldsymbol{X}}_i \right)$$

$$= \frac{1}{m} \sum_{l \in [2]} \sum_{r \in [m]} \sigma \left( \left\langle \boldsymbol{w}_{\hat{y}_i, r}^{(\tau)}, \tilde{\boldsymbol{v}}_i^{(l)} \right\rangle \right) + \frac{1}{m} \sum_{r \in [m]} \sigma \left( \left\langle \boldsymbol{w}_{\hat{y}_i, r}^{(\tau)}, \tilde{\boldsymbol{\xi}}_i \right\rangle \right).$$

For each $i \in [n_{\mathrm{st}}]$ and $l \in [2]$, we have

$$\frac{1}{m} \sum_{r \in [m]} \sigma \left( \left\langle \boldsymbol{w}_{\hat{y}_i, r}^{(\tau)}, \tilde{\boldsymbol{v}}_i^{(l)} \right\rangle \right)$$

$$\leq \frac{1}{m} \sum_{r \in [m]} \sigma \left( \left\langle \boldsymbol{w}_{\hat{y}_i, r}^{(0)}, \tilde{\boldsymbol{v}}_i^{(l)} \right\rangle + \max_{s \in \{\pm 1\}} \left\{ \overline{M}_{s,r}^{(\tau)}, \pm \overline{N}_{s,r}^{(\tau)} \right\} \right)$$

$$\leq \frac{1}{m} \sum_{r \in [m]} \left[ \sigma \left( \left\langle \boldsymbol{w}_{\hat{y}_i, r}^{(0)}, \tilde{\boldsymbol{v}}_i^{(l)} \right\rangle \right) + \sigma \left( \max_{s \in \{\pm 1\}} \left\{ \overline{M}_{s,r}^{(\tau)}, \pm \overline{N}_{s,r}^{(\tau)} \right\} \right) \right]$$

$$\leq \frac{1}{m} \sum_{r \in [m]} \left[ \sigma \left( \left\langle \boldsymbol{w}_{\hat{y}_i, r}^{(0)}, \tilde{\boldsymbol{v}}_i^{(l)} \right\rangle \right) + \max_{s \in \{\pm 1\}} \left\{ \sigma \left( \overline{M}_{s,r}^{(\tau)} \right), \sigma \left( \overline{N}_{s,r}^{(\tau)} \right) \sigma \left( -\overline{N}_{s,r}^{(\tau)} \right) \right\} \right]$$

$$\leq \frac{1}{m} \sum_{r \in [m]} \sigma \left( \left\langle \boldsymbol{w}_{\hat{y}_i, r}^{(0)}, \tilde{\boldsymbol{v}}_i^{(l)} \right\rangle \right) + \max_{s \in \{\pm 1\}} \left\{ \frac{1}{m} \sum_{r \in [m]} \overline{M}_{s,r}^{(\tau)}, \frac{1}{m} \sum_{r \in \mathcal{A}_s} \overline{N}_{s,r}^{(\tau)}, -\frac{1}{m} \sum_{r \in \mathcal{B}_s} \overline{N}_{s,r}^{(\tau)} \right\}$$

$$\leq \alpha_{\mathrm{st}} + \frac{1}{2}.$$

Combining with Lemma D.2, for any $i \in [n_{\mathrm{st}}]$, we have

$$\hat{y}_i f_{\mathrm{st}} \left( \boldsymbol{W}^{(\tau)}, \tilde{\boldsymbol{X}}_i \right) \leq 2 \cdot \left( \alpha_{\mathrm{st}} + \frac{1}{2} \right) + \alpha_{\mathrm{st}} + \frac{3}{\log T^*} \leq 2,$$

where the last inequality follows from (6) and thus we have

$$1 \geq \tilde{g}_i^{(\tau)} = \frac{1}{1 + \exp \left( \hat{y}_i f_{\mathrm{st}} \left( \boldsymbol{W}^{(\tau)}, \tilde{\boldsymbol{X}}_i \right) \right)} \geq \frac{1}{1 + \exp(2)} \geq \frac{1}{9}, \tag{20}$$

for any $i \in [n_{\mathrm{st}}]$.

From Lemma A.3 and the event $E_{\mathrm{st}}$, for any $s \in \{\pm 1\}$ and $r \in [m]$, we obtain

$$\overline{M}_{s,r}^{(\tau+1)} - \overline{M}_{s,r}^{(\tau)} = \frac{\eta}{m n_{\mathrm{st}}} \sum_{l \in [2]} \left( \sum_{i \in \mathcal{C}_{\boldsymbol{\mu}_s}^{(l)}} \tilde{g}_i^{(\tau)} - \sum_{i \in \mathcal{F}_{\boldsymbol{\mu}_s}^{(l)}} \tilde{g}_i^{(\tau)} \right) \|\boldsymbol{\mu}\|^2 \cdot \mathbb{1} \left[ \left\langle \boldsymbol{w}_{s,r}^{(\tau)}, \boldsymbol{\mu}_s \right\rangle > 0 \right]$$

$$\geq \frac{\eta}{m n_{\mathrm{st}}} \sum_{l \in [2]} \left( \frac{1}{9} \left| \mathcal{C}_{\boldsymbol{\mu}_s}^{(l)} \right| - \left| \mathcal{F}_{\boldsymbol{\mu}_s}^{(l)} \right| \right) \|\boldsymbol{\mu}\|^2 \cdot \mathbb{1} \left[ \left\langle \boldsymbol{w}_{s,r}^{(\tau)}, \boldsymbol{\mu}_s \right\rangle > 0 \right]$$

$$\geq \frac{2\eta}{m n_{\mathrm{st}}} \left( \frac{1 - C_{\mathrm{st}}^{-1}}{9} \cdot n_{\boldsymbol{\mu}} - C_{\mathrm{st}}^{-1} \cdot n_{\boldsymbol{\mu}} \right) \|\boldsymbol{\mu}\|^2 \cdot \mathbb{1} \left[ \left\langle \boldsymbol{w}_{s,r}^{(\tau)}, \boldsymbol{\mu}_s \right\rangle > 0 \right]$$

$$\geq \frac{\eta}{5 m n_{\mathrm{st}}} n_{\boldsymbol{\mu}} \|\boldsymbol{\mu}\|^2 \cdot \mathbb{1} \left[ \left\langle \boldsymbol{w}_{s,r}^{(\tau)}, \boldsymbol{\mu}_s \right\rangle > 0 \right]$$

$$= \frac{\eta(2 p_{\mathrm{e}} + p_{\mathrm{b}})}{20 m} \|\boldsymbol{\mu}\|^2 \cdot \mathbb{1} \left[ \left\langle \boldsymbol{w}_{s,r}^{(\tau)}, \boldsymbol{\mu}_s \right\rangle > 0 \right]$$

$$\geq 0.$$

Hence, if $r \in \mathcal{M}_s$, we have

$$\left\langle \boldsymbol{w}_{s,r}^{(\tau+1)}, \boldsymbol{\mu}_s \right\rangle = \left\langle \boldsymbol{w}_{s,r}^{(0)}, \boldsymbol{\mu}_s \right\rangle + \overline{M}_{s,r}^{(\tau+1)} \geq \left\langle \boldsymbol{w}_{s,r}^{(0)}, \boldsymbol{\mu}_s \right\rangle + \overline{M}_{s,r}^{(\tau)} = \left\langle \boldsymbol{w}_{s,r}^{(\tau)}, \boldsymbol{\mu}_s \right\rangle > 0$$

and if $r \notin \mathcal{M}_s$, we have $\overline{M}_{s,r}^{(\tau+1)} = \overline{M}_{s,r}^{(\tau)} = 0$.

In addition, we have

$$\frac{1}{m} \sum_{r \in [m]} \overline{M}_{s,r}^{(\tau+1)} \geq \frac{1}{m} \sum_{r \in [m]} \overline{M}_{s,r}^{(\tau)} + \frac{\eta(2 p_{\mathrm{e}} + p_{\mathrm{b}})}{20 m} \|\boldsymbol{\mu}\|^2 \cdot \frac{1}{m} \sum_{r \in [m]} \mathbb{1} \left[ \left\langle \boldsymbol{w}_{s,r}^{(\tau)}, \boldsymbol{\mu}_s \right\rangle > 0 \right]$$

$$\geq \frac{1}{m} \sum_{r \in [m]} \overline{M}_{s,r}^{(\tau)} + \frac{\eta(2p_{\mathrm{e}} + p_{\mathrm{b}})}{20m} \|\boldsymbol{\mu}\|^2 \cdot \frac{|\mathcal{M}_s|}{m}$$

$$\geq \frac{1}{m} \sum_{r \in [m]} \overline{M}_{s,r}^{(\tau)} + \frac{\eta(2p_{\mathrm{e}} + p_{\mathrm{b}})}{80m} \|\boldsymbol{\mu}\|^2 ,$$

where the last inequality follows from (9).

Similarly, for any $s \in \{\pm 1\}$ and $r \in \mathcal{A}_s$ we obtain

$$\overline{N}_{s,r}^{(\tau+1)} - \overline{N}_{s,r}^{(\tau)} = \frac{\eta}{mn_{\mathrm{st}}} \sum_{l \in [2]} \left( \sum_{i \in \mathcal{C}_{\boldsymbol{\nu}_s}^{(l)}} \tilde{g}_i^{(\tau)} - \sum_{i \in \mathcal{F}_{\boldsymbol{\nu}_s}^{(l)}} \tilde{g}_i^{(\tau)} \right) \|\boldsymbol{\nu}\|^2$$

$$\geq \frac{\eta}{mn_{\mathrm{st}}} \sum_{l \in [2]} \left( \frac{1}{9} \left| \mathcal{C}_{\boldsymbol{\nu}_s}^{(l)} \right| - \left| \mathcal{F}_{\boldsymbol{\nu}_s}^{(l)} \right| \right) \|\boldsymbol{\nu}\|^2$$

$$\geq \frac{2\eta}{mn_{\mathrm{st}}} \left( \frac{1 - C_{\mathrm{st}}^{-1}}{9} \cdot n_{\boldsymbol{\nu}} - C_{\mathrm{st}}^{-1} \cdot n_{\boldsymbol{\nu}} \right) \|\boldsymbol{\nu}\|^2$$

$$\geq \frac{\eta}{5mn_{\mathrm{st}}} n_{\boldsymbol{\nu}} \|\boldsymbol{\nu}\|^2$$

$$= \frac{\eta p_{\mathrm{b}}}{40m} \|\boldsymbol{\nu}\|^2$$

$$\geq 0.$$

Hence, if $r \in \mathcal{A}_s$, we have

$$\left\langle \boldsymbol{w}_{s,r}^{(\tau+1)}, \boldsymbol{\nu}_s \right\rangle = \left\langle \boldsymbol{w}_{s,r}^{(0)}, \boldsymbol{\nu}_s \right\rangle + \overline{N}_{s,r}^{(\tau+1)} \geq \left\langle \boldsymbol{w}_{s,r}^{(0)}, \boldsymbol{\nu}_s \right\rangle + \overline{N}_{s,r}^{(\tau)} = \left\langle \boldsymbol{w}_{s,r}^{(\tau)}, \boldsymbol{\nu}_s \right\rangle > 0.$$

In addition, we have

$$\frac{1}{m} \sum_{r \in \mathcal{A}_s} \overline{N}_{s,r}^{(\tau+1)} \geq \frac{1}{m} \sum_{r \in \mathcal{A}_s} \overline{N}_{s,r}^{(\tau)} + \frac{\eta p_{\mathrm{b}}}{40} \|\boldsymbol{\nu}\|^2 \cdot \frac{|\mathcal{A}_s|}{m}$$

$$\geq \frac{1}{m} \sum_{r \in \mathcal{A}_s} \overline{N}_{s,r}^{(\tau)} + \frac{\eta p_{\mathrm{b}}}{160m} \|\boldsymbol{\nu}\|^2 .$$

We can obtain similar conclusions for $\mathcal{B}_s$. Thus, we obtain the first five statements.

For any $s \in \{\pm 1\}$, we have

$$\frac{1}{m} \sum_{r \in [m]} \overline{M}_{s,r}^{(\tau+1)} \leq \frac{1}{m} \sum_{r \in [m]} \overline{M}_{s,r}^{(\tau)} + \frac{\eta}{mn_{\mathrm{st}}} \left( \left| \mathcal{C}_{\boldsymbol{\mu}_s}^{(1)} \right| + \left| \mathcal{C}_{\boldsymbol{\mu}_s}^{(2)} \right| \right) \|\boldsymbol{\mu}\|^2$$

$$\leq \frac{1}{m} \sum_{r \in [m]} \overline{M}_{s,r}^{(\tau)} + \frac{2 \left( 1 + C_{\mathrm{st}}^{-1} \right) \eta n_{\boldsymbol{\mu}}}{mn_{\mathrm{st}}} \|\boldsymbol{\mu}\|^2$$

$$\leq \frac{1}{m} \sum_{r \in [m]} \overline{M}_{s,r}^{(\tau)} + \frac{3\eta(2p_{\mathrm{e}} + p_{\mathrm{b}})}{4m} \|\boldsymbol{\mu}\|^2 .$$

In addition, we have

$$\frac{1}{m} \sum_{r \in \mathcal{A}_s} \overline{N}_{s,r}^{(\tau+1)} \leq \frac{1}{m} \sum_{r \in \mathcal{A}_s} \overline{N}_{s,r}^{(\tau)} + \frac{\eta}{mn_{\mathrm{st}}} \left( \left| \mathcal{C}_{\boldsymbol{\nu}_s}^{(1)} \right| + \left| \mathcal{C}_{\boldsymbol{\nu}_s}^{(2)} \right| \right) \|\boldsymbol{\nu}\|^2$$

$$\leq \frac{1}{m} \sum_{r \in \mathcal{A}_s} \overline{N}_{s,r}^{(\tau)} + \frac{2 \left( 1 + C_{\mathrm{st}}^{-1} \right) \eta n_{\boldsymbol{\nu}}}{mn_{\mathrm{st}}} \|\boldsymbol{\nu}\|^2$$

$$\leq \frac{1}{m} \sum_{r \in \mathcal{A}_s} \overline{N}_{s,r}^{(\tau)} + \frac{3\eta p_{\mathrm{b}}}{8m} \|\boldsymbol{\nu}\|^2 .$$

Similarly, we have

$$-\frac{1}{m}\sum_{r\in\mathcal{B}_s}\overline{N}_{s,r}^{(\tau+1)} \le -\frac{1}{m}\sum_{r\in\mathcal{B}_s}\overline{N}_{s,r}^{(\tau)} + \frac{\eta}{mn_{\text{st}}}\left(\left|\mathcal{C}_{-\boldsymbol{\nu}_s}^{(1)}\right| + \left|\mathcal{C}_{-\boldsymbol{\nu}_s}^{(2)}\right|\right)\|\boldsymbol{\nu}\|^2$$

$$\le -\frac{1}{m}\sum_{r\in\mathcal{B}_s}\overline{N}_{s,r}^{(\tau)} + \frac{2\left(1 + C_{\text{st}}^{-1}\right)\eta n_{\boldsymbol{\nu}}}{mn_{\text{st}}}\|\boldsymbol{\nu}\|^2$$

$$\le -\frac{1}{m}\sum_{r\in\mathcal{B}_s}\overline{N}_{s,r}^{(\tau)} + \frac{3p_{\text{b}}\eta}{8m}\|\boldsymbol{\nu}\|^2.$$

Using these, we have

$$\sum_{r\in[m]}\overline{M}_{1,r}^{(\tau+1)} = \sum_{r\in[m]}\overline{M}_{1,r}^{(\tau)} + \left[\sum_{r\in[m]}\overline{M}_{1,r}^{(\tau+1)} - \sum_{r\in[m]}\overline{M}_{1,r}^{(\tau)}\right]$$

$$\ge \sum_{r\in[m]}\overline{M}_{1,r}^{(\tau)} + \frac{\eta(2p_{\text{e}} + p_{\text{b}})}{80}\|\boldsymbol{\mu}\|^2$$

$$\ge \sum_{r\in[m]}\overline{M}_{1,r}^{(\tau)} + \frac{1}{60}\left[\sum_{r\in[m]}\overline{M}_{-1,r}^{(\tau+1)} - \sum_{r\in[m]}\overline{M}_{-1,r}^{(\tau)}\right]$$

$$\ge \frac{1}{60}\sum_{r\in[m]}\overline{M}_{-1,r}^{(\tau)} + \frac{1}{60}\left[\sum_{r\in[m]}\overline{M}_{-1,r}^{(\tau+1)} - \sum_{r\in[m]}\overline{M}_{-1,r}^{(\tau)}\right]$$

$$= \frac{1}{60}\sum_{r\in[m]}\overline{M}_{-1,r}^{(\tau+1)}.$$

By using symmetric arguments, we can obtain $\sum_{r\in[m]}\overline{M}_{1,r}^{(\tau+1)} \le 60\sum_{r\in[m]}\overline{M}_{-1,r}^{(\tau+1)}$.

Similarly, for any $s, s' \in \{\pm 1\}$, we have

$$\sum_{r\in\mathcal{A}_s}\overline{N}_{s,r}^{(\tau+1)} = \sum_{r\in\mathcal{A}_s}\overline{N}_{s,r}^{(\tau)} + \left[\sum_{r\in\mathcal{A}_s}\overline{N}_{s,r}^{(\tau+1)} - \sum_{r\in\mathcal{A}_s}\overline{N}_{s,r}^{(\tau)}\right]$$

$$\le \sum_{r\in\mathcal{A}_s}\overline{N}_{s,r}^{(\tau)} + \frac{3\eta p_{\text{b}}}{8}\|\boldsymbol{\nu}\|^2$$

$$\le \sum_{r\in\mathcal{A}_s}\overline{N}_{s,r}^{(\tau)} + \frac{\eta(2p_{\text{e}} + p_{\text{b}})}{80}\|\boldsymbol{\mu}\|^2$$

$$\le \sum_{r\in[m]}\overline{M}_{s',r}^{(\tau)} + \left[\sum_{r\in[m]}\overline{M}_{s',r}^{(\tau+1)} - \sum_{r\in[m]}\overline{M}_{s',r}^{(\tau)}\right]$$

$$= \sum_{r\in[m]}\overline{M}_{s',r}^{(\tau+1)}.$$

In addition, we have

$$\sum_{r\in\mathcal{A}_s}\overline{N}_{s,r}^{(\tau+1)}$$

$$= \sum_{r\in\mathcal{A}_s}\overline{N}_{s,r}^{(\tau)} + \left[\sum_{r\in\mathcal{A}_s}\overline{N}_{s,r}^{(\tau+1)} - \sum_{r\in\mathcal{A}_s}\overline{N}_{s,r}^{(\tau)}\right]$$

$$\ge \sum_{r\in\mathcal{A}_s}\overline{N}_{s,r}^{(\tau)} + \frac{p_{\text{b}}\eta}{160}\|\boldsymbol{\nu}\|^2$$

$$= \sum_{r \in \mathcal{A}_s} \overline{N}_{s,r}^{(\tau)} + \frac{p_{\mathrm{b}} \|\boldsymbol{\nu}\|^2}{120(2p_{\mathrm{e}} + p_{\mathrm{b}}) \|\boldsymbol{\mu}\|^2} \cdot \frac{3\eta(2p_{\mathrm{e}} + p_{\mathrm{b}}) \|\boldsymbol{\mu}\|^2}{4m}$$

$$\geq \frac{p_{\mathrm{b}} \|\boldsymbol{\nu}\|^2}{120(2p_{\mathrm{e}} + p_{\mathrm{b}}) \|\boldsymbol{\mu}\|^2} \sum_{r \in [m]} \overline{M}_{s',r}^{(\tau)} + \frac{p_{\mathrm{b}} \|\boldsymbol{\nu}\|^2}{120(2p_{\mathrm{e}} + p_{\mathrm{b}}) \|\boldsymbol{\mu}\|^2} \cdot \left[ \sum_{r \in [m]} \overline{M}_{s',r}^{(\tau+1)} - \sum_{r \in [m]} \overline{M}_{s',r}^{(\tau)} \right]$$

$$= \frac{p_{\mathrm{b}} \|\boldsymbol{\nu}\|^2}{120(2p_{\mathrm{e}} + p_{\mathrm{b}}) \|\boldsymbol{\mu}\|^2} \sum_{r \in [m]} \overline{M}_{s',r}^{(\tau+1)}.$$

Now, we prove the last statement. For any $r \in [m]$, if $\underline{M}_{s,r}^{(\tau)} \leq -\alpha_{\mathrm{st}}$, then we have $\left\langle \boldsymbol{w}_{s,r}^{(\tau)}, \boldsymbol{\mu}_{-s} \right\rangle < 0$.
Hence, $\left| \underline{M}_{s,r}^{(\tau+1)} \right| = \left| \underline{M}_{s,r}^{(\tau)} \right| \leq \alpha_{\mathrm{st}} + \beta_{\mathrm{st}}$ by Lemma A.3. Otherwise, $\underline{M}_{s,r}^{(\tau)} > -\alpha_{\mathrm{st}}$ implies that

$$\frac{mn_{\mathrm{st}}}{\eta\|\boldsymbol{\mu}\|^2} \left( \underline{M}_{s,r}^{(\tau+1)} - \underline{M}_{s,r}^{(\tau)} \right)$$

$$= -\sum_{l \in [2]} \left( \sum_{j \in \mathcal{C}_{\boldsymbol{\mu}_{-s}}^{(l)}} \tilde{g}_j^{(\tau)} - \sum_{j \in \mathcal{F}_{\boldsymbol{\mu}_{-s}}^{(l)}} \tilde{g}_j^{(\tau)} \right) \cdot \mathbb{1}\left[ \left\langle \boldsymbol{w}_{s,r}^{(\tau)}, \boldsymbol{\mu}_{-s} \right\rangle > 0 \right]$$

$$\leq -\sum_{l \in [2]} \left( \frac{1}{9} \left| \mathcal{C}_{\boldsymbol{\mu}_{-s}}^{(l)} \right| - \left| \mathcal{F}_{\boldsymbol{\mu}_{-s}}^{(l)} \right| \right) \cdot \mathbb{1}\left[ \left\langle \boldsymbol{w}_{s,r}^{(\tau)}, \boldsymbol{\mu}_{-s} \right\rangle > 0 \right]$$

$$\leq 0,$$

where the first inequality follows from (20) and the last inequality follows from the event $E_{\mathrm{st}}$. Thus, $\underline{M}_{s,r}^{(\tau+1)} \leq \underline{M}_{s,r}^{(\tau)} \leq \alpha_{\mathrm{st}} + \beta_{\mathrm{st}}$. In addition, we have

$$\frac{mn_{\mathrm{st}}}{\eta\|\boldsymbol{\mu}\|^2} \left( \underline{M}_{s,r}^{(\tau+1)} - \underline{M}_{s,r}^{(\tau)} \right)$$

$$= -\sum_{l \in [2]} \left( \sum_{j \in \mathcal{C}_{\boldsymbol{\mu}_{-s}}^{(l)}} \tilde{g}_j^{(\tau)} - \sum_{j \in \mathcal{F}_{\boldsymbol{\mu}_{-s}}^{(l)}} \tilde{g}_j^{(\tau)} \right) \cdot \mathbb{1}\left[ \left\langle \boldsymbol{w}_{s,r}^{(\tau)}, \boldsymbol{\mu}_{-s} \right\rangle > 0 \right]$$

$$\geq -\sum_{l \in [2]} \left| \mathcal{C}_{\boldsymbol{\mu}_s}^{(l)} \right| \cdot \mathbb{1}\left[ \left\langle \boldsymbol{w}_{s,r}^{(\tau)}, \boldsymbol{\mu}_{-s} \right\rangle > 0 \right]$$

$$\geq -2n_{\mathrm{st}}.$$

Therefore, we have

$$\underline{M}_{s,r}^{(\tau+1)} \geq \underline{M}_{s,r}^{(\tau)} - \frac{2\eta\|\boldsymbol{\mu}\|^2}{m} \geq -\alpha_{\mathrm{st}} - \frac{2\eta\|\boldsymbol{\mu}\|^2}{m} \geq -\alpha_{\mathrm{st}} - \beta_{\mathrm{st}},$$

where the last inequality follows from (10).

From Lemma A.3, for any $r \in [m]$,

$$\left| \underline{N}_{s,r}^{(\tau+1)} - \underline{N}_{s,r}^{(\tau)} \right| \leq \frac{2\eta \|\boldsymbol{\nu}\|^2}{m} \leq \alpha_{\mathrm{st}}.$$

Therefore, it suffices to show that $\underline{N}_{s,r}^{(\tau+1)} \leq \underline{N}_{s,r}^{(\tau)}$ when $\underline{N}_{s,r}^{(\tau)} > \alpha_{\mathrm{st}}$ and $\underline{N}_{s,r}^{(\tau+1)} \geq \underline{N}_{s,r}^{(\tau)}$ when $\underline{N}_{s,r}^{(\tau)} < -\alpha_{\mathrm{st}}$. If $\underline{N}_{s,r}^{(\tau)} > \alpha_{\mathrm{st}}$, then we have

$$\left\langle \boldsymbol{w}_{s,r}^{(\tau)}, \boldsymbol{\nu}_{-s} \right\rangle = \left\langle \boldsymbol{w}_{s,r}^{(0)}, \boldsymbol{\nu}_{-s} \right\rangle + \underline{N}_{s,r}^{(\tau)} > 0.$$

Hence, we have

$$\frac{mn_{\mathrm{st}}}{\eta\|\boldsymbol{\nu}\|^2} \left( \underline{N}_{s,r}^{(\tau+1)} - \underline{N}_{s,r}^{(\tau)} \right)$$

$$= -\sum_{l \in [2]} \left( \sum_{j \in \mathcal{C}_{\boldsymbol{\nu}_{-s}}^{(l)}} \tilde{g}_j^{(\tau)} - \sum_{j \in \mathcal{F}_{\boldsymbol{\nu}_{-s}}^{(l)}} \tilde{g}_j^{(\tau)} \right)$$

$$\leq -\sum_{l \in [2]} \left( \frac{1}{9} \left| \mathcal{C}_{\boldsymbol{\nu}_{-s}}^{(l)} \right| - \left| \mathcal{F}_{\boldsymbol{\nu}_{-s}}^{(l)} \right| \right)$$

$$\leq -2 \left( \frac{(1 - C_{\mathrm{st}}^{-1})}{9} \cdot n_{\boldsymbol{\nu}} - C_{\mathrm{st}}^{-1} \cdot n_{\boldsymbol{\nu}} \right)$$

$$\leq 0,$$

where the first inequality follows from (20) and the last inequality follows from the event $E_{\mathrm{st}}$. Using a similar argument, we can also show that $\underline{N}_{s,r}^{(\tau+1)} \geq \underline{N}_{s,r}^{(\tau)}$ when $\underline{N}_{s,r}^{(\tau)} < -\alpha_{\mathrm{st}}$ and we have desired conclusion. $\qquad\square$

Next, we characterize the early-phase learning dynamics of easy signals.

**Lemma D.4.** *Under Condition 3.5 and the event $E_{\mathrm{st}}$, there exists the smallest iteration $T_{\mathrm{es}} \in \left[0, \frac{200m}{\eta(2p_{\mathrm{e}}+p_{\mathrm{b}})\|\boldsymbol{\mu}\|^2}\right]$ such that*

$$\max \left\{ \frac{1}{m} \sum_{r \in [m]} \overline{M}_{1,r}^{(T_{\mathrm{es}})}, \frac{1}{m} \sum_{r \in [m]} \overline{M}_{-1,r}^{(T_{\mathrm{es}})} \right\} \geq \frac{1}{2}.$$

*Proof of Lemma D.4.* Suppose there is no such iteration. We fix an arbitrary $s \in \{\pm 1\}$. Note that from Condition 3.5, $\frac{100m}{\eta(2p_{\mathrm{e}}+p_{\mathrm{b}})\|\boldsymbol{\mu}\|^2} \leq \frac{mn_{\mathrm{st}}}{\eta\sigma_p^2 d \log T^*}$. Thus, we can apply Lemma D.3 and for any $t \in \left[0, \frac{100m}{\eta(2p_{\mathrm{e}}+p_{\mathrm{b}})\|\boldsymbol{\mu}\|^2}\right]$, we have

$$\frac{1}{m} \sum_{r \in [m]} \overline{M}_{s,r}^{(t)} \geq \frac{1}{m} \sum_{r \in [m]} \overline{M}_{s,r}^{(t-1)} + \frac{\eta(2p_{\mathrm{e}}+p_{\mathrm{b}})}{80m} \|\boldsymbol{\mu}\|^2$$

$$\vdots$$

$$\geq \frac{1}{m} \sum_{r \in [m]} \overline{M}_{s,r}^{(0)} + \frac{\eta(2p_{\mathrm{e}}+p_{\mathrm{b}})}{80m} \|\boldsymbol{\mu}\|^2 t$$

$$= \frac{\eta(2p_{\mathrm{e}}+p_{\mathrm{b}})}{80} \|\boldsymbol{\mu}\|^2 t.$$

By choosing $t = \frac{40m}{\eta(2p_{\mathrm{e}}+p_{\mathrm{b}})\|\boldsymbol{\mu}\|^2} \in \left[0, \frac{100m}{\eta(2p_{\mathrm{e}}+p_{\mathrm{b}})\|\boldsymbol{\mu}\|^2}\right]$, we obtain contradiction. Therefore, there exists an iteration $t \in \left[0, \frac{100m}{\eta(2p_{\mathrm{e}}+p_{\mathrm{b}})\|\boldsymbol{\mu}\|^2}\right]$ such that

$$\max \left\{ \frac{1}{m} \sum_{r \in [m]} \overline{M}_{1,r}^{(t)}, \frac{1}{m} \sum_{r \in [m]} \overline{M}_{-1,r}^{(t)} \right\} \geq \frac{1}{2}.$$

We then define $T_{\mathrm{es}}$ as the smallest such iteration. $\qquad\square$

We will show that iteration $T_{\mathrm{es}}$ obtained from Lemma D.4 is our desired stopping time. By sequentially applying Lemma D.3, for any $s \in \{\pm 1\}$, we have $\overline{M}_{s,r}^{(T_{\mathrm{es}})} \geq 0$ for all $r \in [m]$, $\overline{N}_{s,r}^{(T_{\mathrm{es}})} \geq 0$ if $r \in \mathcal{A}_s$, and $\overline{N}_{s,r}^{(T_{\mathrm{es}})} \leq 0$ if $r \in \mathcal{B}_s$. Furthermore, we have

$$\frac{1}{m} \sum_{r \in [m]} \overline{M}_{s,r}^{(T_{\mathrm{es}})} \geq \frac{1}{120}, \quad \frac{1}{m} \sum_{r \in \mathcal{A}_s} \overline{N}_{s,r}^{(T_{\mathrm{es}})}, -\frac{1}{m} \sum_{r \in \mathcal{B}_s} \overline{N}_{s,r}^{(T_{\mathrm{es}})} \geq \frac{p_{\mathrm{b}} \|\boldsymbol{\nu}\|^2}{240(2p_{\mathrm{e}}+p_{\mathrm{b}}) \|\boldsymbol{\mu}\|^2} \qquad (21)$$

and for any $r \in [m]$, we have

$$\left| \underline{M}_{s,r}^{(T_{\text{es}})} \right|, \left| \underline{N}_{s,r}^{(T_{\text{es}})} \right| \leq \alpha_{\text{st}} + \beta_{\text{st}}. \tag{22}$$

Combining the upper bound on $T_{\text{es}}$ and Lemma D.2 leads to the following bound: for any $s \in \{\pm 1\}, r \in [m]$, and $i \in [n_{\text{st}}]$,

$$\left| \rho_{s,r,i}^{(T_{\text{es}})} \right|, \left| \left\langle \boldsymbol{w}_{s,r}^{(T_{\text{es}})}, \tilde{\boldsymbol{\xi}}_i \right\rangle \right| \leq \alpha_{\text{st}} + \frac{3\eta\sigma_p^2 d}{mn_{\text{st}}} \cdot \frac{100m}{\eta(2p_{\text{e}} + p_{\text{b}}) \|\boldsymbol{\mu}\|^2} \leq \frac{400\sigma_p^2 d}{(2p_{\text{e}} + p_{\text{b}})n_{\text{st}} \|\boldsymbol{\mu}\|^2}, \tag{23}$$

where the last inequality follows from (6).

## D.2 Train Error

In this subsection, we prove the first conclusion conditioned on the event $E_{\text{st}}$. For any $i \in [n_{\text{st}}]$, we have

$$\tilde{y}_i f_{\text{st}} \left( \boldsymbol{W}^{(T_{\text{es}})}, \tilde{\boldsymbol{X}}_i \right)$$

$$= \frac{1}{m} \sum_{l \in [2]} \sum_{r \in [m]} \phi \left( \left\langle \boldsymbol{w}_{\tilde{y}_i,r}^{(T_{\text{es}})}, \tilde{\boldsymbol{v}}_i^{(l)} \right\rangle \right) - \frac{1}{m} \sum_{l \in [2]} \sum_{r \in [m]} \phi \left( \left\langle \boldsymbol{w}_{-\tilde{y}_i,r}^{(T_{\text{es}})}, \tilde{\boldsymbol{v}}_i^{(l)} \right\rangle \right)$$

$$+ \frac{1}{m} \sum_{r \in [m]} \phi \left( \left\langle \boldsymbol{w}_{\tilde{y}_i,r}^{(T_{\text{es}})}, \tilde{\boldsymbol{\xi}}_i \right\rangle \right) - \frac{1}{m} \sum_{r \in [m]} \phi \left( \left\langle \boldsymbol{w}_{-\tilde{y}_i,r}^{(T_{\text{es}})}, \tilde{\boldsymbol{\xi}}_i \right\rangle \right)$$

$$\geq \frac{1}{m} \sum_{l \in [2]} \sum_{r \in [m]} \phi \left( \left\langle \boldsymbol{w}_{\tilde{y}_i,r}^{(T_{\text{es}})}, \tilde{\boldsymbol{v}}_i^{(l)} \right\rangle \right) - 2 \cdot (\alpha_{\text{st}} + \alpha_{\text{st}} + \beta_{\text{st}}) - \frac{400\sigma_p^2 d}{(2p_{\text{e}} + p_{\text{b}})n_{\text{st}} \|\boldsymbol{\mu}\|^2}$$

$$\geq \frac{2}{m} \min \left\{ \sum_{r \in [m]} \overline{M}_{\tilde{y}_i,r}^{(T_{\text{es}})}, \sum_{r \in [m]} \overline{N}_{\tilde{y}_i,r}^{(T_{\text{es}})}, - \sum_{r \in \mathcal{B}_{\tilde{y}_i}} \overline{N}_{\tilde{y}_i,r}^{(T_{\text{es}})} \right\} - 2(2\alpha_{\text{st}} + \beta_{\text{st}}) - \frac{400\sigma_p^2 d}{(2p_{\text{e}} + p_{\text{b}})n_{\text{st}} \|\boldsymbol{\mu}\|^2}$$

$$\geq \frac{p_{\text{b}} \|\boldsymbol{\nu}\|^2}{120(2p_{\text{e}} + p_{\text{b}}) \|\boldsymbol{\mu}\|^2} - 2(\alpha_{\text{st}} + \beta_{\text{st}}) - \frac{400\sigma_p^2 d}{(2p_{\text{e}} + p_{\text{b}})n_{\text{st}} \|\boldsymbol{\mu}\|^2}$$

$$> 0,$$

where the first inequality follows from (22) and (23), the third inequality follows from (21), and the last inequality follows from (6) and Condition 3.5. □

## D.3 Test Error

In this subsection, we characterize the test error of the strong model. All arguments in this subsection are under the event $E_{\text{st}}$. Define $\boldsymbol{v}^{(1)}, \boldsymbol{v}^{(2)}$, and $\boldsymbol{\xi}$ as the signal vectors and the noise vector in the test data $(\boldsymbol{X}, y)$, respectively.

We define a function $h : S \to \mathbb{R}$ as $h(\boldsymbol{z}) := \frac{1}{m} \sum_{r \in [m]} \sigma \left( \left\langle \boldsymbol{w}_{-y,r}^{(T_{\text{es}})}, \boldsymbol{z} \right\rangle \right)$ for any $\boldsymbol{z} \in S$. It plays a crucial role when we prove the upper bounds on test error. We have

$$\mathbb{E}[h(\boldsymbol{\xi})] = \frac{1}{m} \mathbb{E}_{\boldsymbol{z}_1,\ldots,\boldsymbol{z}_m} \left[ \sum_{r \in [m]} \sigma(\boldsymbol{z}_r) \right] = \frac{1}{2m} \mathbb{E}_{\boldsymbol{z}_1,\ldots,\boldsymbol{z}_m} \left[ \sum_{r \in [m]} |\boldsymbol{z}_r| \right] = \frac{\sigma_p}{\sqrt{2\pi}m} \sum_{r \in [m]} \left\| \Pi_S \boldsymbol{w}_{-y,r}^{(T_{\text{es}})} \right\|,$$

where $\boldsymbol{z}_r \sim \mathcal{N} \left( 0, \sigma_p^2 \left\| \Pi_S \boldsymbol{w}_{-y,r}^{(T_{\text{es}})} \right\|^2 \right)$ for each $r \in [m]$. Also, for any $\boldsymbol{z}_1, \boldsymbol{z}_2 \in S$, we have

$$|h(\boldsymbol{z}_1) - h(\boldsymbol{z}_2)| \leq \frac{1}{m} \sum_{r \in [m]} \left| \sigma \left( \left\langle \boldsymbol{w}_{-y,r}^{(T_{\text{es}})}, \boldsymbol{z}_1 \right\rangle \right) - \sigma \left( \left\langle \boldsymbol{w}_{-y,r}^{(T_{\text{es}})}, \boldsymbol{z}_2 \right\rangle \right) \right|$$

$$\leq \frac{1}{m} \sum_{r \in [m]} \left| \left\langle \boldsymbol{w}_{-y,r}^{(T_{\text{es}})}, \boldsymbol{z}_1 \right\rangle - \left\langle \boldsymbol{w}_{-y,r}^{(T_{\text{es}})}, \boldsymbol{z}_2 \right\rangle \right|$$

$$= \frac{1}{m} \sum_{r \in [m]} \left| \left\langle \Pi_S \boldsymbol{w}_{-y,r}^{(T_{\mathrm{es}})}, \boldsymbol{z}_1 \right\rangle - \left\langle \Pi_S \boldsymbol{w}_{-y,r}^{(T_{\mathrm{es}})}, \boldsymbol{z}_2 \right\rangle \right|$$

$$\leq \frac{1}{m} \sum_{r \in [m]} \left\| \Pi_S \boldsymbol{w}_{-y,r}^{(T_{\mathrm{es}})} \right\| \|\boldsymbol{z}_1 - \boldsymbol{z}_2\|.$$

Hence, $h$ is $\frac{1}{m} \sum_{r \in [m]} \left\| \Pi_S \boldsymbol{w}_{-y,r}^{(T_{\mathrm{es}})} \right\|$-Lipschitz.

The following lemma characterizes $\left\| \Pi_S \boldsymbol{w}_{-y,r}^{(T_{\mathrm{es}})} \right\|$'s which is related to key properties of $h$.

**Lemma D.5.** *For any $s \in \{\pm 1\}$, it holds that*

$$\sum_{r \in [m]} \left\| \Pi_S \boldsymbol{w}_{s,r}^{(T_{\mathrm{es}})} \right\| \leq \frac{900 m \sigma_p d^{\frac{1}{2}}}{(2p_{\mathrm{e}} + p_{\mathrm{b}}) n^{\frac{1}{2}} \|\boldsymbol{\mu}\|^2}.$$

*Proof of Lemma D.5.* From Lemma A.3 and triangular inequality, we have

$$\left\| \Pi_S \boldsymbol{w}_{s,r}^{(T_{\mathrm{es}})} \right\| \leq \left\| \Pi_S \boldsymbol{w}_{s,r}^{(0)} \right\| + \left\| \sum_{i \in [n_{\mathrm{st}}]} \rho_{s,r,i}^{(T_{\mathrm{es}})} \tilde{\boldsymbol{\xi}}_i \|\tilde{\boldsymbol{\xi}}_i\|^{-2} \right\| \leq \sqrt{2} \sigma_0 d^{\frac{1}{2}} + \left\| \sum_{i \in [n_{\mathrm{st}}]} \rho_{s,r,i}^{(T_{\mathrm{es}})} \tilde{\boldsymbol{\xi}}_i \|\tilde{\boldsymbol{\xi}}_i\|^{-2} \right\|.$$

We have

$$\left\| \sum_{i \in [n_{\mathrm{st}}]} \rho_{s,r,i}^{(T_{\mathrm{es}})} \tilde{\boldsymbol{\xi}}_i \|\tilde{\boldsymbol{\xi}}_i\|^{-2} \right\|^2$$

$$= \sum_{i \in [n_{\mathrm{st}}]} \left( \rho_{s,r,i}^{(T_{\mathrm{es}})} \right)^2 \|\tilde{\boldsymbol{\xi}}_i\|^{-2} + \sum_{\substack{i,j \in [n_{\mathrm{st}}] \\ i \neq j}} \rho_{s,r,i}^{(T_{\mathrm{es}})} \rho_{s,r,j}^{(T_{\mathrm{es}})} \langle \tilde{\boldsymbol{\xi}}_i, \tilde{\boldsymbol{\xi}}_j \rangle \|\tilde{\boldsymbol{\xi}}_i\|^{-2} \|\tilde{\boldsymbol{\xi}}_j\|^{-2}$$

$$\leq \sum_{i \in [n_{\mathrm{st}}]} \left( \rho_{s,r,i}^{(T_{\mathrm{es}})} \right)^2 \|\tilde{\boldsymbol{\xi}}_i\|^{-2} + \sum_{\substack{i,j \in [n_{\mathrm{st}}] \\ i \neq j}} \left| \rho_{s,r,i}^{(T_{\mathrm{es}})} \rho_{s,r,j}^{(T_{\mathrm{es}})} \right| \left| \langle \tilde{\boldsymbol{\xi}}_i, \tilde{\boldsymbol{\xi}}_j \rangle \right| \|\tilde{\boldsymbol{\xi}}_i\|^{-2} \|\tilde{\boldsymbol{\xi}}_j\|^{-2}$$

$$\leq \sum_{i \in [n_{\mathrm{st}}]} \left( \rho_{s,r,i}^{(T_{\mathrm{es}})} \right)^2 \|\tilde{\boldsymbol{\xi}}_i\|^{-2} + \frac{1}{2} \sum_{\substack{i,j \in [n_{\mathrm{st}}] \\ i \neq j}} \left( \left( \rho_{s,r,i}^{(T_{\mathrm{es}})} \right)^2 + \left( \rho_{s,r,j}^{(T_{\mathrm{es}})} \right)^2 \right) \left| \langle \tilde{\boldsymbol{\xi}}_i, \tilde{\boldsymbol{\xi}}_j \rangle \right| \|\tilde{\boldsymbol{\xi}}_i\|^{-2} \|\tilde{\boldsymbol{\xi}}_j\|^{-2}$$

$$\leq (1 + \beta_{\mathrm{st}}) \sum_{i \in [n_{\mathrm{st}}]} \left( \rho_{s,r,i}^{(T_{\mathrm{es}})} \right)^2 \|\tilde{\boldsymbol{\xi}}_i\|^{-2}$$

$$\leq \left( \frac{800 \sigma_p d^{\frac{1}{2}}}{(2p_{\mathrm{e}} + p_{\mathrm{b}}) n_{\mathrm{st}}^{\frac{1}{2}} \|\boldsymbol{\mu}\|^2} \right)^2$$

where the third inequality follows from (8) and the fourth inequality follows from (23) and (8). Therefore, we have

$$\sum_{r \in [m]} \left\| \Pi_S \boldsymbol{w}_{s,r}^{(T_{\mathrm{es}})} \right\| \leq \sqrt{2} m \sigma_0 d^{\frac{1}{2}} + \frac{800 m \sigma_p d^{\frac{1}{2}}}{(2p_{\mathrm{e}} + p_{\mathrm{b}}) n_{\mathrm{st}}^{\frac{1}{2}} \|\boldsymbol{\mu}\|^2} \leq \frac{900 m \sigma_p d^{\frac{1}{2}}}{(2p_{\mathrm{e}} + p_{\mathrm{b}}) n_{\mathrm{st}}^{\frac{1}{2}} \|\boldsymbol{\mu}\|^2},$$

where the last inequality follows from (C3). $\qquad \square$

By Theorem 5.2.2 in Vershynin (2018), for any $z > 0$, it holds that

$$\mathbb{P}[h(\boldsymbol{\xi}) - \mathbb{E}[h(\boldsymbol{\xi})] \geq z] \leq \exp \left( -\frac{cz^2}{\sigma_p^2 \|h\|_{\mathrm{Lip}}^2} \right)$$

where $c$ is a universal constant and $\|\cdot\|_{\mathrm{Lip}}$ denotes the best Lipschitz constant. Combining with Lemma C.4, we have

$$\mathbb{P}[h(\boldsymbol{\xi}) - \mathbb{E}[h(\boldsymbol{\xi})] \geq z] \leq \exp\left(-\frac{c(2p_{\mathrm{e}} + p_{\mathrm{b}})^2 \|\boldsymbol{\mu}\|^4}{900^2 \sigma_p^4 d} z^2\right). \tag{24}$$

Now, we characterize the test error. First, we consider the case $(\boldsymbol{X}, y) \in \mathcal{S}_{\mathrm{e}} \cup \mathcal{S}_{\mathrm{b}}$. We have

$$
\begin{aligned}
&y f_{\mathrm{st}}\left(\boldsymbol{W}^{(T_{\mathrm{es}})}, \boldsymbol{X}\right) \\
&= F_y\left(\boldsymbol{W}_y^{(T_{\mathrm{es}})}, \boldsymbol{X}\right) - F_{-y}\left(\boldsymbol{W}_{-y}^{(T_{\mathrm{es}})}, \boldsymbol{X}\right) \\
&= \frac{1}{m} \sum_{l \in [2]} \sum_{r \in [m]} \sigma\left(\left\langle \boldsymbol{w}_{y,r}^{(T_{\mathrm{es}})}, \boldsymbol{v}^{(l)}\right\rangle\right) + \frac{1}{m} \sum_{r \in [m]} \sigma\left(\left\langle \boldsymbol{w}_{y,r}^{(T_{\mathrm{es}})}, \boldsymbol{\xi}\right\rangle\right) \\
&\quad - \frac{1}{m} \sum_{l \in [2]} \sum_{r \in [m]} \sigma\left(\left\langle \boldsymbol{w}_{-y,r}^{(T_{\mathrm{es}})}, \boldsymbol{v}^{(l)}\right\rangle\right) - \frac{1}{m} \sum_{r \in [m]} \sigma\left(\left\langle \boldsymbol{w}_{-y,r}^{(T_{\mathrm{es}})}, \boldsymbol{\xi}\right\rangle\right) \\
&\geq \frac{1}{m} \sum_{r \in [m]} \sigma\left(\left\langle \boldsymbol{w}_{y,r}^{(T_{\mathrm{es}})}, \boldsymbol{\xi}\right\rangle\right) - \frac{1}{m} \sum_{r \in [m]} \sigma\left(\left\langle \boldsymbol{w}_{-y,r}^{(T_{\mathrm{es}})}, \boldsymbol{\xi}\right\rangle\right) + \frac{1}{m} \sum_{r \in [m]} \overline{M}_{y,r}^{(T_{\mathrm{es}})} - 4\alpha_{\mathrm{st}} \\
&\geq -\frac{1}{m} \sum_{r \in [m]} \sigma\left(\left\langle \boldsymbol{w}_{-y,r}^{(T_{\mathrm{es}})}, \boldsymbol{\xi}\right\rangle\right) + \frac{1}{120} - 4\alpha_{\mathrm{st}} \\
&\geq -h(\boldsymbol{\xi}) + \frac{1}{200},
\end{aligned}
$$

where the first inequality follows from (7) and (22). From (24) and Lemma D.5, we have

$$
\begin{aligned}
&\mathbb{P}\left[y f_{\mathrm{st}}\left(\boldsymbol{W}^{(T_{\mathrm{es}})}, \boldsymbol{X}\right) < 0 \,\middle|\, (\boldsymbol{X}, y) \in \mathcal{S}_{\mathrm{e}} \cup \mathcal{S}_{\mathrm{b}}\right] \\
&\leq \mathbb{P}\left[h(\boldsymbol{\xi}) > \frac{1}{200}\right] = \mathbb{P}\left[h(\boldsymbol{\xi}) - \mathbb{E}[h(\boldsymbol{\xi})] > \frac{1}{200} - \mathbb{E}[h(\boldsymbol{\xi})]\right] \\
&\leq \mathbb{P}\left[h(\boldsymbol{\xi}) - \mathbb{E}[h(\boldsymbol{\xi})] > \frac{1}{200} - \frac{900\sigma_p d^{\frac{1}{2}}}{(2p_{\mathrm{e}} + p_{\mathrm{b}}) n_{\mathrm{st}}^{\frac{1}{2}} \|\boldsymbol{\mu}\|^2}\right] \\
&\leq \mathbb{P}\left[h(\boldsymbol{\xi}) - \mathbb{E}[h(\boldsymbol{\xi})] > \frac{1}{250}\right] \\
&\leq \exp\left(-\frac{n_{\mathrm{st}}(2p_{\mathrm{e}} + p_{\mathrm{b}})^2 \|\boldsymbol{\mu}\|^4}{C_5 \sigma_p^4 d}\right),
\end{aligned}
$$

with some constant $C_5 > 0$.

Using a similar argument, we can prove the upper bound on the test error for the case $(\boldsymbol{X}, y) \in \mathcal{S}_{\mathrm{h}}$. In this case, we have

$$
\begin{aligned}
&y f_{\mathrm{st}}\left(\boldsymbol{W}^{(T_{\mathrm{es}})}, \boldsymbol{X}\right) \\
&= F_y\left(\boldsymbol{W}_y^{(T_{\mathrm{es}})}, \boldsymbol{X}\right) - F_{-y}\left(\boldsymbol{W}_{-y}^{(T_{\mathrm{es}})}, \boldsymbol{X}\right) \\
&= \frac{1}{m} \sum_{l \in [2]} \sum_{r \in [m]} \sigma\left(\left\langle \boldsymbol{w}_{y,r}^{(T_{\mathrm{es}})}, \boldsymbol{v}^{(l)}\right\rangle\right) + \frac{1}{m} \sum_{r \in [m]} \sigma\left(\left\langle \boldsymbol{w}_{y,r}^{(T_{\mathrm{es}})}, \boldsymbol{\xi}\right\rangle\right) \\
&\quad - \frac{1}{m} \sum_{l \in [2]} \sum_{r \in [m]} \sigma\left(\left\langle \boldsymbol{w}_{-y,r}^{(T_{\mathrm{es}})}, \boldsymbol{v}^{(l)}\right\rangle\right) - \frac{1}{m} \sum_{r \in [m]} \sigma\left(\left\langle \boldsymbol{w}_{-y,r}^{(T_{\mathrm{es}})}, \boldsymbol{\xi}\right\rangle\right) \\
&\geq \frac{1}{m} \sum_{r \in [m]} \sigma\left(\left\langle \boldsymbol{w}_{y,r}^{(T_{\mathrm{es}})}, \boldsymbol{\xi}\right\rangle\right) - \frac{1}{m} \sum_{r \in [m]} \sigma\left(\left\langle \boldsymbol{w}_{-y,r}^{(T_{\mathrm{es}})}, \boldsymbol{\xi}\right\rangle\right) \\
&\quad + \frac{2}{m} \min\left\{\sum_{r \in \mathcal{A}_y} \overline{N}_{y,r}^{(T_{\mathrm{es}})}, -\sum_{r \in \mathcal{B}_y} \overline{N}_{y,r}^{(T_{\mathrm{es}})}\right\} - 2(\alpha_{\mathrm{st}} + \beta_{\mathrm{st}})
\end{aligned}
$$

$$\geq -\frac{1}{m}\sum_{r\in[m]}\sigma\left(\left\langle \boldsymbol{w}_{-y,r}^{(T_{\mathrm{es}})},\boldsymbol{\xi}\right\rangle\right) + \frac{2}{m}\min\left\{\sum_{r\in\mathcal{A}_y}\overline{N}_{y,r}^{(T_{\mathrm{es}})}, -\sum_{r\in\mathcal{B}_y}\overline{N}_{y,r}^{(T_{\mathrm{es}})}\right\} - 2(\alpha_{\mathrm{st}}+\beta_{\mathrm{st}})$$

$$\geq -h(\boldsymbol{\xi}) + \frac{p_{\mathrm{b}}\|\boldsymbol{\nu}\|^2}{120(2p_{\mathrm{e}}+p_{\mathrm{b}})\|\boldsymbol{\mu}\|^2} - 2(\alpha_{\mathrm{st}}+\beta_{\mathrm{st}})$$

$$\geq -h(\boldsymbol{\xi}) + \frac{p_{\mathrm{b}}\|\boldsymbol{\nu}\|^2}{200(2p_{\mathrm{e}}+p_{\mathrm{b}})\|\boldsymbol{\mu}\|^2}$$

where the first inequality follows from (7) and (22), the third inequality follows from (21), and the last inequality follows from (6) and Condition 3.5.

From (24) and Lemma D.5, we have

$$\mathbb{P}\left[yf_{\mathrm{st}}\left(\boldsymbol{W}^{(T_{\mathrm{es}})},\boldsymbol{X}\right)<0\,\Big|\,(\boldsymbol{X},y)\in\mathcal{S}_{\mathrm{h}}\right]$$

$$\leq \mathbb{P}\left[h(\boldsymbol{\xi}) > \frac{p_{\mathrm{b}}\|\boldsymbol{\nu}\|^2}{200(2p_{\mathrm{e}}+p_{\mathrm{b}})\|\boldsymbol{\mu}\|^2}\right]$$

$$= \mathbb{P}\left[h(\boldsymbol{\xi}) - \mathbb{E}[h(\boldsymbol{\xi})] > \frac{p_{\mathrm{b}}\|\boldsymbol{\nu}\|^2}{200(2p_{\mathrm{e}}+p_{\mathrm{b}})\|\boldsymbol{\mu}\|^2} - \mathbb{E}[h(\boldsymbol{\xi})]\right]$$

$$\leq \mathbb{P}\left[h(\boldsymbol{\xi}) - \mathbb{E}[h(\boldsymbol{\xi})] > \frac{p_{\mathrm{b}}\|\boldsymbol{\nu}\|^2}{200(2p_{\mathrm{e}}+p_{\mathrm{b}})\|\boldsymbol{\mu}\|^2} - \frac{900\sigma_p d^{\frac{1}{2}}}{(2p_{\mathrm{e}}+p_{\mathrm{b}})n_{\mathrm{st}}^{\frac{1}{2}}\|\boldsymbol{\mu}\|^2}\right]$$

$$\leq \mathbb{P}\left[h(\boldsymbol{\xi}) - \mathbb{E}[h(\boldsymbol{\xi})] > \frac{p_{\mathrm{b}}\|\boldsymbol{\nu}\|^2}{250(2p_{\mathrm{e}}+p_{\mathrm{b}})\|\boldsymbol{\mu}\|^2}\right]$$

$$\leq \exp\left(-\frac{n_{\mathrm{st}}p_{\mathrm{b}}^2\|\boldsymbol{\nu}\|^4}{C_5\sigma_p^4 d}\right),$$

with some constant $C_5 > 0$.

