# OpenReview forum: "From Linear to Nonlinear: Provable Weak-to-Strong Generalization through Feature Learning"
_NeurIPS.cc/2025/Conference — NeurIPS 2025 poster_

### Official Review · Reviewer_yKoW · 2025-06-20

**Clarity:** 2
**Significance:** 2
**Originality:** 3
**Rating:** 4
**Confidence:** 3

**Summary:**

The paper introduces a theoretical framework for studying weak-to-strong generalization via feature learning. It shows how a two-layer ReLU CNN learns from a linear CNN model, and characterize two regimes depending on the number of samples for training the stronger model. Empirical simulation match the derived theories.

**Questions:**

1. What is the role of weak model? It seems the weak model will learn to correctly classify both the easy-only and both-signal samples (with $o(1)$ error) while random guess the hard-only samples. Then the training of stronger model would be on a dataset with label noise almost only on hard-only samples. Is my understanding correct? If so, could the author comment on what is the difference to paper [2], which also considers label noise in their data model.

2. How practical the data distribution is and how likely the theoretical results hold in practice? Could the authors provide some experiments done in real-world scenarios?

3. Could the authors provide more discussions in relation to the existing theoretical analysis of weak-to-strong generalization? Currently the paper only mentions the difference in terms modelling assumption, without an in-depth discussion how the derived results in the paper show similar/different insights compared to existing works.

4. It seems the settings and analysis techniques are based on [1,2]. It would be good if the authors could comment on the differences in terms of analysis and training dynamics, which could further help understand the insights as well as the technical contributions of the paper (if any).

[1] Benign Overfitting in Two-layer Convolutional Neural Networks.

[2] Benign Overfitting in Two-layer ReLU Convolutional Neural Networks.

**Ethical Concerns:**

["NO or VERY MINOR ethics concerns only"]

**Final Justification:**

I remain positive for the paper as most of my concerns are addressed during rebuttal.

**Limitations:**

The discussions of limitation is insufficient. Authors are encouraged to add a separate paragraph outlining the limitations.

**Quality:**

3

**Strengths And Weaknesses:**

In terms of strengths, the paper studies weak-to-strong generalization with nonlinear models, in contrast to existing analysis with linear model or random feature model. Also, the theoretical analysis appears to be complete in terms of characterizing a transition from generalization to overfitting. For the weakness, refer to the question section.

---

> ### Author Rebuttal · Authors · 2025-07-31
>
> We express our gratitude for your valuable comments. In the following, we address the points raised by the reviewer.
>
> ## **Q1, Q4. Comparison to [1],[2]**
> Your understanding on the role of weak model is correct. We would like to provide differences between our work and previous works [1, 2].
>
> * We focus on a training scenario that is different from the one considered in [1, 2]: training a single type of network. Specifically, we study a two-stage training process: first training a weak model, then training a strong model supervised by the pretrained weak model. This difference in training scenarios leads to some technical differences.
>
> * While [1,2] considers a single type of signal for each class, our setting involves two different types of signals per class. Moreover, the label flipping probability in [2] is independent of the input, whereas in our setting, it depends on the input because it is introduced by a pretrained weak model. These differences require more careful analysis in our proofs.
>
> * While working on this paper, we also identified technical issues in [2]. Specifically, [2] often uses the sum of activations as the model output, whereas the correct formulation should use the average of activations. To correct this, in our Lemma C.3 (S7), we show that the filter whose initialization has a positive inner product with the noise tends to have a noise coefficient close to the maximum among all filters. This is an analysis not previously presented in [2].
>
> * Condition 4.1 in [2] considers a regime where noise learning dominates signal learning, which corresponds to the data-scarce regime in our work. However, we additionally consider the data-abundant regime and reveal that a different behavior emerges in this setting.
>
> ## **Q2. Practical Insights and Experiments on Real-World Data.**
> Our data distribution is inspired by real-world characteristics, consisting of label-relevant information (corresponding to the signal in our work) and label-irrelevant information (noise in our work). Moreover, the label-relevant information varies in learning difficulty due to differences in its strength or complexity. The distribution is designed to reflect these properties while maintaining mathematical tractability. We also emphasize that analysis in such simplified settings can still provide valuable insights into the underlying mechanisms.
>
> Our key intuition behind weak-to-strong generalization is as follows: there exists a subset of data containing hard-to-learn information that is nevertheless labeled correctly due to the presence of easy-to-learn information (referred to as easy signals in our setting). This subset of data, termed both-signal data in our work, provides gradient updates that enable the strong model to learn the hard-to-learn information.
>
> We additionally conducted experiments on a simplified real-world dataset to support our findings. Specifically, we slightly modified the MNIST dataset to highlight the roles of key components such as signal and noise. The results demonstrate that our theoretical insights also hold beyond our settings. Detailed experimental settings and results are provided in our responses to Reviewer ABZC’s W1 and Q2.
>
> ## **Q3. Comparison to Previous Works on Theory of Weak-to-Strong Generalization**
> We would like to include more discussion on some prior works related to the theory of weak-to-strong generalization, in addition to what is already presented in our draft.
>
> The most closely related prior work is [3], which shares a similar intuition with ours—namely, that data containing both easy-to-learn and hard-to-learn information is essential for weak-to-strong generalization. However, their theoretical analysis relies heavily on the abstract framework introduced in [4], and it remains unclear whether gradient-based training can indeed lead to weak-to-strong generalization in their setting.
>
> Similar to our results, [5] demonstrates that weak-to-strong generalization can emerge through benign overfitting. They consider linear models and analyze the minimum $\ell_2$-norm interpolating solution, which is attainable via gradient descent. We view this as a step forward from analyses based purely on abstract frameworks. However, this work still relies on the assumption that the weak and strong models have access to different sets of features different access to feature space, an assumption we consider to be somewhat unnatural. Moreover, this work does not consider scenarios with a large amount of data, which we explicitly analyze as the data-abundant regime in our work.
>
> [6] considers a more natural setting, where training is performed using gradient-based methods and the weak/strong models are chosen as random feature models with different numbers of hidden nodes. However, they analyze training on the population loss, where early stopping becomes crucial, as overtraining can lead the strong model to produce outputs similar to those of the weak model on the population. In contrast, our analysis is based on finite-sample training and explicitly reveals the role of early stopping in the finite but data-abundant regime.
>
> We believe that our contribution is essential in this line of work, as it makes the theoretical analysis more aligned with realistic training settings.
>
> ## **Limitations**
> Thank you for your suggestions. We plan to discuss our limitations more explicitly in the next revision.
>
> ---
>
> Thank you very much for your time. If there is anything that requires further clarification, please do let us know.
>
> Best regards,
>
> Authors
>
> ---
>
> **References**
>
> [1] Cao et al. Benign Overfitting in Two-layer Convolutional Neural Networks. NeurIPS 2022
>
> [2] Kou et al. Benign Overfitting in Two-layer ReLU Convolutional Neural Networks. ICML 2023
>
> [3] Shin et al. Weak-to-Strong Generalization through the Data-Centric Lens. ICLR 2025
>
> [4] Lang et al. Theoretical Analysis of Weak-to-Strong Generalization. NeurIPS 2024
>
> [5] Wu & Sahai. Provable Weak-to-Strong Generalization via Benign Overfitting. ICLR 2025
>
> [6] Medevedev et al. Weak-to-Strong Generalization Even in Random Feature Networks, Provably. ICML 2025

---

> > ### Comment · Reviewer_yKoW · 2025-08-03
> >
> > I thank the authors for the reply. Most of my concerns are addressed and I maintain my positive rating.

---

> > > ### Author Response · Authors · 2025-08-04
> > >
> > > Thank you for your response. We are glad to hear that our response was helpful and resolved your concerns. If you have any further questions, please feel free to ask.
> > >
> > > Best regards,
> > >
> > > Authors

---

### Official Review · Reviewer_YK97 · 2025-06-28

**Clarity:** 3
**Significance:** 2
**Originality:** 3
**Rating:** 4
**Confidence:** 4

**Summary:**

This paper provides a theoretical analysis for weak-to-strong generalization in binary classification based on special structured data consisting of signal and noise patches (a variation of the multi-view dataset introduced in Allen-Zhu and Li (2020)). The authors take a linear CNN as the weak model and a two-layer ReLU CNN as the strong model. The main theoretical findings are the following:
* A separation result on the generalization of the weak and strong models on the special binary classification task, showing that there exists some parametrization of the strong model that achieves zero test error, whereas the weak model cannot achieve zero test error.
* A generalization bound for the supervised training of the weak model with gradient descent, providing a sample complexity of $n_{wk} \gtrsim \frac{\sigma_p^2 d}{(2 p_e + p_b)^2 \|\mu\|^4}$ for the weak model to achieve a close-to-optimal test error on the binary classification task, where $\sigma_p^2$ is the variance of the noise patches, $p_e$ and $p_b$ are the probabilities of the easy-only and mixture of easy and hard patches, $d$ is the feature dimension, and $\mu$ is the easy feature norm.
* A phase transition analysis for weak-to-strong generalization with respect to the weak-to-strong training sample size $n_{st}$ in the "data-scarce regime", showing that when the strong model is trained with weak supervision, it can either generalize in the case of benign overfitting or fail to generalize in the case of harmful overfitting.
* An analysis of the "data-abundant regime", showing that weak-to-strong generalization can arise via early stopping but vanishes when the strong model is overtrained with weak supervision.

**Questions:**

* It is unclear from the setup in Section 2.1 how the "easy" and "hard" signals are defined and distinguished for various statements.
    - From the proof of Proposition 2.2, the mixture of signs where the hard signals are sampled uniformly seems to be the only key difference between the easy and hard signals that leads to the separation of the algorithm-agnostic capacity gap between the weak and strong models (which I think should be highlighted explicitly).
    - However, additional assumptions on the sampling probabilities $p_e, p_h, p_b$ and signal strength $\|\mu\|,\|\nu\|$ are needed (e.g., Condition (C5)) for weak-to-strong generalization when analyzing the training dynamics of GD. The current organization of these assumptions is somehow confusing. It may be helpful to partition the assumptions in Condition 3.1 into several parts. For example, C5 and C6 may be separated as a stand-alone additional assumption on the easy v.s. hard signals for the training dynamics analysis.
* The phase transition in the "data-scarce" regime with respect to $n_{st}$ is not very clear from the statement of Theorem 3.4. In particular, Theorem 3.4 shows that $n_{st} \ge C_2 \frac{\sigma_p^4 d}{p_b^2 \|\nu\|^4}$ leads to benign overfitting, while $n_{st} \le C_4 \frac{\sigma_p^4 d}{p_b^2 \|\nu\|^4}$ leads to harmful overfitting. Here, the upper and lower bounds are asymptotically the same, and there are no discussions on conditions like $C_2 \approx C_4$, showing that this indeed leads to a phase transition. The simulation in Section 5 shows one synthetic example with three seemingly arbitrary $n_{st}$, without investigating the scaling of weak-to-strong generalization with respect to $n_{st}$ that could demonstrate the phase transition.

**Ethical Concerns:**

["NO or VERY MINOR ethics concerns only"]

**Final Justification:**

I would like to thank the authors for their efforts to address my questions and concerns. I will maintain my positive evaluation.

**Limitations:**

Major limitations are discussed in the weaknesses section. Here are some minor points:
* The explanations for some assumptions (specifically, C1-C4) in Condition 3.1 is vague.
* Section 4 may be partitioned into subsections of "data-scarce" and "data-abundant" regimes to be more organized.
* The meaning of "filters" in line 312 is not clear. The explanation in lines 310-312 is not very clear either.

**Paper Formatting Concerns:**

There are no standing-out formatting concerns.

**Quality:**

3

**Strengths And Weaknesses:**

### Strengths
* While mainly leveraging existing problem setup and technical tools, this paper provides a fresh non-linear feature learning analysis of weak-to-strong generalization, which offers new angles to understand weak-to-strong generalization.
* The setting and main takeaways from the theoretical analysis are clearly organized and presented.
* The key theoretical insights outlined in Section 4 is helpful for understanding the intuitions behind the theoretical results and the implications of the results for weak-to-strong generalization.

### Weaknesses
* Some detailed settings and statements seem vague and could be better clarified. See the "Questions" section below for more details.
* The empirical evidence of this paper seems limited, even given its theoretical nature. Section 5 only provides a single set of experiments on the synthetic binary classification task and does not investigate the scaling of weak-to-strong generalization with respect to $n_{st}$ that seems essential for demonstrating the phase transition in the "data-scarce" regime (see the second point in the "Questions" section below). That is to say, I don't think strong experiments on real data and models are necessary for this work, given its theoretical nature. But given the (possibly inevitable) limitations of some theoretical statements in the paper (e.g., Theorem 3.4, see the second point in the "Questions" section below), it would be helpful to at least provide thorough simulations on the synthetic binary classification task.
* A potential limitation inherited in the problem setup of this work is that the capacity gap between the weak and strong model completely comes from the architecture of the weak and strong models with respect to the special structured data. This is fairly different from weak-to-strong generalization, or the superalignment problem that it aims to address, in practice. The capacity gaps between large foundation models are often due to many other factors beyond the architecture (e.g., pretraining data and methods) and shared across various tasks, more like differences in the qualities of their representations. While I think the learning dynamics analysis of weak-to-strong generalization from linear to non-linear models on special structured data is theoretically interesting and valuable, I subjectively believe that it has a considerable gap from the practical weak-to-strong generalization problem. I think such a gap should be carefully discussed, maybe as a part of the conclusion/limitations at the end.

---

> ### Author Rebuttal · Authors · 2025-07-31
>
> We express our gratitude for your valuable comments. In the following, we address the points raised by the reviewer.
>
> ## **W2, Q2. Experiments on Synthetic Data**
>
> Admittedly, a precise calculation of the constants $C_2$ and $C_4$ is complex; our analysis focuses on an asymptotic regime where parameters such as the number of data points and dimension are sufficiently large. In this regime, the upper and lower bounds for the transition match asymptotically up to constants, creating a sharp threshold. In the small sample settings of our experiments, this transition appears more gradual, rather than a sharp phase transition. However, by varying $n_{st}$ and the dimension d, we can still observe and verify the transition dynamics in our theoretical analysis.
>
> We ran new experiments with the same setup as described in the paper, while varying $n_{st}$ and dimension $d$. The same pretrained weak model was used to generate pseudolabels across all runs within the same seed. We ran experiments for 5 different random seeds. Below is the average test accuracy with standard deviation across different seeds.
>
> As shown in the table, the test accuracy of the weak-to-strong training generally improves as $n_{st}$ increases for both $d = 2000$ and $d = 4000$. This demonstrates the transition in the data-scarce regime. The results also suggest that the transition point scales with the dimension $d$, as a larger $n_{st}$ is required to achieve similar performance when $d$ is doubled, which aligns with our theoretical findings.
>
> | | $d=2000$ (weak test acc: 0.8508) | | $d=4000$ (weak test acc: 0.8494) |
> |:---:|:---:|:---:|:---:|
> | **$n_{st}$** | **weak-to-strong test acc** | **$n_{st}$** | **weak-to-strong test acc** |
> | 75 | 0.8829 ± 0.0448 | 150 | 0.8566 ± 0.0274 |
> | 150 | 0.8830 ± 0.0236 | 300 | 0.8888 ± 0.0267 |
> | 225 | 0.9022 ± 0.0204 | 450 | 0.8963 ± 0.0172 |
> | 300 | 0.9013 ± 0.0310 | 600 | 0.9019 ± 0.0249 |
> | 375 | 0.8831 ± 0.0428 | 750 | 0.8947 ± 0.0099 |
> | 450 | 0.8970 ± 0.0260 | 900 | 0.9054 ± 0.0178 |
> | 525 | 0.9135 ± 0.0124 | 1050 | 0.8845 ± 0.0145 |
> | 600 | 0.9022 ± 0.0281 | 1200 | 0.8846 ± 0.0317 |
> | 675 | 0.9126 ± 0.0200 | 1350 | 0.9128 ± 0.0124 |
> | 750 | 0.9268 ± 0.0173 | 1500 | 0.9036 ± 0.0106 |
> | 825 | 0.9122 ± 0.0119 | 1650 | 0.9112 ± 0.0277 |
> | 900 | 0.9228 ± 0.0044 | 1800 | 0.9001 ± 0.0253 |
> | 975 | 0.9053 ± 0.0115 | 1950 | 0.9079 ± 0.0132 |
> | 1050 | 0.9225 ± 0.0208 | 2100 | 0.9168 ± 0.0166 |
> | 1125 | 0.9254 ± 0.0191 | 2250 | 0.9205 ± 0.0254 |
> | 1200 | 0.9340 ± 0.0141 | 2400 | 0.9094 ± 0.0105 |
> | 1275 | 0.9176 ± 0.0293 | 2550 | 0.9275 ± 0.0194 |
> | 1350 | 0.9082 ± 0.0169 | 2700 | 0.9093 ± 0.0210 |
> | 1425 | 0.9223 ± 0.0148 | 2850 | 0.9066 ± 0.0228 |
> | 1500 | 0.9121 ± 0.0188 | 3000 | 0.9132 ± 0.0255 |
> | 1575 | 0.9205 ± 0.0150 | 3150 | 0.9096 ± 0.0277 |
> | 1650 | 0.9310 ± 0.0176 | 3300 | 0.9263 ± 0.0123 |
> | 1725 | 0.9262 ± 0.0138 | 3450 | 0.9197 ± 0.0157 |
> | 1800 | 0.9380 ± 0.0250 | 3600 | 0.9280 ± 0.0161 |
> | 1875 | 0.9148 ± 0.0250 | 3750 | 0.9343 ± 0.0144 |
> | 1950 | 0.9226 ± 0.0156 | 3900 | 0.9253 ± 0.0096 |
> | 2025 | 0.9230 ± 0.0221 | 4050 | 0.9169 ± 0.0081 |
> | 2100 | 0.9245 ± 0.0298 | 4200 | 0.9224 ± 0.0088 |
> | 2175 | 0.9248 ± 0.0143 | 4350 | 0.9133 ± 0.0136 |
> | 2250 | 0.9304 ± 0.0118 | 4500 | 0.9123 ± 0.0130 |
>
> ## **W3. Architectural Differences Between Weak and Strong Models**
> It is true that architectural differences are not the only factors contributing to the differing capabilities of weak and strong models. However, we believe that one key difference lies in expressive power, as exemplified by the weak-to-strong training experiment from GPT-2 to GPT-4 presented in [1].  While we focus on architectural differences in this work, we believe that the core intuition behind our analysis can be extended to more general scenarios, and we will add this discussion to the Conclusion section as you suggested.
>
> ## **Q1. Conditions on Easy and Hard Signal**
> We agree with the reviewer that the conditions on easy and hard signals appear in multiple places in the draft and may cause confusion. In Section 2.1, line 116, we mentioned that easy and hard signals differ in their learning difficulty.
>
> As you mentioned, the XOR structure of the hard signals creates a capacity gap between our weak and strong models. We agree that this point should be mentioned earlier, and we will explicitly add it around line 116.
>
> In addition, as you mentioned, Conditions (C5) and (C6) are related to the distinction between easy and hard signals and to the weak-to-strong learning scenario, which may differ in nature from Conditions (C1)–(C4). We grouped them together for compactness, but we will consider restructuring this part to improve clarity.
>
> ## **Limitations. Suggestions for Clarification**
> Thank you for your suggestions for improving our paper. We will address the points in the limitations part one by one, following the order in which you raised them.
>
> * We would like to provide further explanation of Condition 3.1. In (C1) and (C2), we assume a sufficiently large dimension and large sample sizes. These assumptions allow us to apply concentration inequalities to quantities such as norms and correlations involving Gaussian noise, as well as the number of each type of data. In (C3) and (C4), we assume a sufficiently small initialization scale and learning rate. These assumptions ensure that the effect of initialization can be neglected compared to the gradient descent updates, and that each update step remains small, which enables stable training dynamics. Lastly, we would like to additionally clarify that the motivation behind (C5) is that the difficulty of learning a signal is determined by its frequency and strength.
> * In our draft, we didn’t separate the discussion of the two regimes into distinct subsections, given that they share overlapping insights. Instead, we plan to add clearer transitional markers for the reader’s convenience in the next revision.
> * In line 312, “filters” refers to $w_{s,r}$ as defined in Definition 3. We would like to provide a more detailed explanation of the discussion in lines 310–312.
> For a filter $w_{s,r}$ that is initially positively aligned with $\nu_s$, the first term in the block equation between lines 304 and 305 is positive, while the second term is zero. As a result, the inner product between this filter and $\nu_s$ increases over time.
> In contrast, for a filter that is initially negatively aligned with $\nu_s$, the first term becomes zero, and the second term is negative. This leads to a decrease in the inner product.
> This contrasting behavior between positively and negatively aligned filters allows the strong model to learn both $\pm \nu_s$.
>
> We will include these in the next revision.
>
> ---
>
> Thanks for your time and efforts. Please let us know if you have further questions.
>
> Best regards,
>
> Authors
>
> ---
>
> **References**
>
> [1] Burns et al. Weak-to-Strong Generalization: Eliciting Strong Capabilities with Weak Supervision. ICML 2024

---

### Official Review · Reviewer_bMLu · 2025-06-28

**Clarity:** 3
**Significance:** 3
**Originality:** 3
**Rating:** 4
**Confidence:** 4

**Summary:**

This paper provides a theoretical and empirical analysis of weak-to-strong generalization, where a stronger model trained under supervision from a weaker model can outperform the teacher. This work fills a gap by analyzing a special setting: weak-to-strong training from a linear CNN to a two-layer ReLU CNN on patch data.

**Questions:**

1. Why does the data distribution assign only the positive direction $\mu_y$ for easy signals, while it allows both $\pm\nu_y$ directions for hard signals? In Definition 1, the easy signals are always set to $\mu_y$, whereas the hard signals are drawn uniformly from all sign combinations $\pm\nu_y$. This introduces a structural asymmetry. I understand that this may be designed to ensure the weak model can reliably learn easy signals while remaining ineffective on hard signals, but I would appreciate clarification on whether this choice is essential to the theoretical guarantees, or whether the same qualitative conclusions would hold under a more symmetric signal setup (e.g., also randomizing the sign of $\mu_y$).
2. How sensitive is the theory to the assumption that the weak model labels are not adversarial, especially in the data-abundant regime? Can small adversarial label flips cause failure of weak-to-strong generalization?

**Ethical Concerns:**

["NO or VERY MINOR ethics concerns only"]

**Final Justification:**

This paper offers a clean and technically solid analysis of weak-to-strong generalization in a synthetic setting. While its scope is narrow and some assumptions are idealized, the corrected proof and added insights meaningfully contribute to the theoretical understanding of the phenomenon. I recommend a weak accept based on its rigor, the technical fix to prior work, and the relevance of the topic to current interest in model distillation and student-teacher dynamics.

**Limitations:**

While the paper presents a rigorous analysis, its setting is highly stylized and somewhat artificial. The data distribution is carefully engineered to contain label-aligned "easy" and "hard" signals with hand-picked orthogonality, which may not reflect the structure or complexity of real-world data. As a result, it is unclear how broadly the insights or guarantees would transfer to practical scenarios, such as vision or NLP tasks. Moreover, much of the technical analysis builds directly on tools and mechanisms developed in prior work on benign overfitting and feature learning (e.g., Cao et al. 2022; Kou et al. 2023), such as signal-noise decompositions and patch-wise convolutional structure. While the application to weak-to-strong generalization is interesting, the paper feels more like a thoughtful reuse of existing techniques in a contrived setting tailored to fit them, rather than a technically novel contribution in its own right. A more ambitious advance would involve analyzing weak-to-strong generalization in less restrictive or more realistic settings.

**Paper Formatting Concerns:**

No.

**Quality:**

3

**Strengths And Weaknesses:**

Strengths:
1. The paper provides a detailed and mathematically sound characterization of weak-to-strong generalization in both data-scarce and data-abundant regimes, including tight bounds and transition conditions between benign and harmful overfitting.

Weaknesses:
1. The data distribution is highly engineered, with orthogonal easy/hard signal vectors and shuffled patch structures. While analytically convenient, it lacks realism and limits the applicability of the results to real-world tasks.
2. Much of the analytical machinery (e.g., signal-noise decomposition, patch-wise CNN analysis, benign overfitting dynamics) closely follows prior work. The contribution lies more in adapting known tools to a new but contrived setup, rather than developing fundamentally new techniques.
3. All experiments are conducted on synthetic data tailored to the theoretical setup. Even simple real-data tests (e.g., CIFAR with weak supervision) would help assess whether the observed phenomena persist beyond the toy model.

---

> ### Author Rebuttal · Authors · 2025-07-31
>
> We would like to express our appreciation for your valuable comments. In the following, we address the points raised by the reviewer.
>
> ## **W1, L1. Our Simple Problem Settings**
> We acknowledge that our problem setting is simplified to enable rigorous theoretical analysis. In particular, assumptions such as the orthogonality of signal vectors allow us to make the learning dynamics mathematically tractable. However, similar simplifications are commonly employed in the recent feature learning theory literature. These line of works utilize similar settings to gain theoretical understandings on various phenomena—such as how model ensembling improves generalization [1], how benign overfitting arises [2,3], how different optimizers affect generalization [4–6], how architectural choices influence generalization [7,8], and how data augmentation contributes to generalization [9–11].
>
> While our assumptions may not hold in practical settings, we believe that they still offer valuable insights into the mechanisms behind weak-to-strong generalization. We would like to offer a more intuitive, high-level explanation for our insights presented in Section 4. A weak model may fail to generalize well because it cannot learn certain types of information that are difficult to capture, what we refer to as hard signals. Now consider training a stronger model—one that is strong enough to learn these hard signals—under the supervision of the weak model. In this setting, the strong model learns from pseudo-labeled data, and its success depends on how often the weak model assigns correct labels.
>
> The key idea is this: there exists a subset of training data that contains both easy and hard signals. Thanks to the easy signals, the weak model is likely to label these examples correctly, even though it cannot learn the hard signals on its own. These correctly labeled examples—what we call both-signal data—provide useful gradient updates that guide the strong model to learn the hard signals. Although the framework we propose is simple, we believe it captures essential intuition that can be extended to more general settings. Also, in our response to W3 (see below), we have empirically shown that our insights can be applied to real-world data.
>
> ## **W3. Experiments on Real World Data**
> We additionally conducted experiments on a simplified real-world dataset to support our findings. Specifically, we slightly modified the MNIST dataset to highlight the roles of key components such as signal and noise. The results demonstrate that our theoretical insights also hold beyond our settings. Detailed experimental settings and results are provided in our responses to Reviewer ABZC’s W1 and Q2.
>
> ## **W2, L2. Novel Contributions**
> Even though we use techniques from [3], we would like to highlight some key differences between [3] and our work.
>
> * While [3] considers a single type of signal for each class, our setting involves two different types of signals per class. Moreover, the label flipping probability in [3] is independent of the input, whereas in our setting, it depends on the input because it is introduced by a pretrained weak model. These differences require more careful analysis in our proofs.
>
> * While working on this paper, we also identified technical issues in [3]. Specifically, [3] often uses the sum of activations as the model output, whereas the correct formulation should use the average of activations. To correct this, in our Lemma C.3 (S7), we show that the filter whose initialization has a positive inner product with the noise tends to have a noise coefficient close to the maximum among all filters. This is an analysis not previously presented in [3].
>
> * Condition 4.1 in [3] considers a regime where noise learning dominates signal learning, which corresponds to the data-scarce regime in our work. However, we additionally consider the data-abundant regime and reveal that a different behavior emerges in this setting.
>
> Moreover, we would like to emphasize that our work not only advances techniques from the feature learning literature, but also addresses several limitations of prior theoretical studies on weak-to-strong generalization by incorporating the following elements: consideration of gradient-based training dynamics, a more natural definition of weak and strong models, analysis based on finite samples rather than population loss, and identification of distinct data regimes. We provide further discussion on these points in our response to Reviewer yKoW’s Q3.
>
> ## **Q1. Choice of Asymmetric Signals**
> Your understanding is correct. Our asymmetric choice of signal directions is intended to highlight the fundamental gap between the weak and strong models. We can also consider a symmetric case where both easy and hard signals involve no sign flips. However, in such a setting, we believe that a linear CNN and a ReLU CNN would require asymptotically the same sample complexity for generalization. As a result, it would be difficult to argue that the two models have different capabilities. In such cases, we believe that changing the architectures of the weak and strong models can lead to similar conclusions. For example, one could adopt a simple CNN as the weak model and a simple ViT as the strong model, whose learning dynamics and benign overfitting behaviors have been studied in [2] and [8]. As noted in [8], a simple ViT has a superior ability to learn signals compared to a simple CNN. Therefore, if we consider a regime where easy signals are learnable by both architectures while hard signals lie in the gap between the two, our theoretical insights discussed above may still apply.
>
> However, we believe that our current simplified setting is more effective in delivering clear and interpretable insights.
>
> ## **Q2. Assumption on Supervising Weak Model**
> We believe there may be a misunderstanding regarding our assumption about supervising the weak model. We do not assume that the weak model's labels are non-adversarial; we only assume that the supervising weak model satisfies certain error bounds. We would like to note that our results also extend to supervisors that assign labels adversarially, as long as they remain within the assumed error budget, since the only requirement is that the weak model satisfies the specified error bounds.
>
> ---
>
> Thanks for your time and consideration. Let us know if you have remaining questions; we are happy to discuss more.
>
> Best regards,
>
> Authors
>
> ---
>
> **References**
>
> [1] Allen-Zhu & Li. Toward Understanding Ensemble, Knowledge Distillation and Self-Distillation in Deep Learning. ICLR 2023
>
> [2] Cao et al. Benign Overfitting in Two-layer Convolutional Neural Networks. NeurIPS 2022
>
> [3] Kou et al. Benign Overfitting in Two-layer ReLU Convolutional Neural Networks. ICML 2023
>
> [4] Zou et al. Understanding the Generalization of Adam in Learning Neural Networks with Proper Regularization. ICLR 2023
>
> [5] Jelassi et al. Towards Understanding How Momentum Improves Generalization in Deep Learning. ICML 2022
>
> [6] Chen et al. Why Does Sharpness-Aware Minimization Generalize Better than SGD. NeurIPS 2023
>
> [7] Huang et al. Quantifying the Optimization and Generalization Advantages of Graph Neural Networks Over Multilayer Perceptrons. AISTATS 2025
>
> [8] Jiang et al. Unveil Benign Overfitting for Transformer in Vision: Training Dynamics, Convergence, and Generalization. NeurIPS 2024
>
> [9] Shen et al. Data Augmentation as Feature Manipulation. ICML 2022
>
> [10] Zou et al. The Benefit of Mixup for Feature Learning. ICML 2023
>
> [11] Oh & Yun. Provable Benefit of Cutout and CutMix for Feature Learning. NeurIPS 2024

---

> > ### Comment · Reviewer_bMLu · 2025-08-05
> >
> > Thank you for the detailed rebuttal. Regarding your comment that reference [3] contains a technical issue due to using the sum of activations rather than the average: upon reviewing [3], the model appears to be defined using the average over filters, not the sum. For instance, in Section 3 of [3], the network output is explicitly defined with a $1/m$ normalization over the number of filters.
> >
> > Since your rebuttal presents the identification of this issue as part of your paper’s technical novelty, I encourage you to elaborate more clearly on this point. If you believe there is a specific step or equation in [3] where the use of the sum leads to a technical flaw or inconsistency, please point to the exact location. As it stands, this appears to be a modeling difference rather than a flaw, and the current claim may be based on a misreading.
> >
> > Clarifying this point would help evaluate the significance and correctness of the contribution.

---

> > > ### Author Response · Authors · 2025-08-06
> > >
> > > Thank you for your response. We would like to clarify the point regarding technical flaws in [3].
> > >
> > > As you mentioned, [3] considers the average of activations (which aligns with our strong model), not the sum. However, in their proof, they frequently alternate between using the average and the sum, which leads to technical inconsistencies. While many of these can be resolved by adding a missing factor of $1/m$ in the analysis, there are also more substantial issues. For example, in Proposition C.2, they aim to show that the noise coefficients $\bar{\rho}\_{j,r,i}^{(t)}$ are upper bounded by $\alpha$ via induction. To do so, they decompose $\bar{\rho}\_{j,r,i}^{(t)}$ into several terms and attempt to bound each one individually (as in Equation (C.23)). However, when bounding the $I_8$ term (middle block equation on page 28), the factor of $1/m$ multiplying $\sigma(\langle w_{j,r}^{(t)}, \xi_i \rangle)$ is missing in the first line. If we naively correct this by inserting the $1/m$ factor, the final bound on $\bar{\rho}\_{j,r,i}^{(t)}$ becomes $m\alpha$ instead of $\alpha$, which would require a stronger assumption on $m$. To address this, we introduced an additional technical result (Lemma C.3 (S7)), which is not covered in [3].
> > >
> > > We hope this clarification is helpful. If you need any further discussion, feel free to reach out.
> > >
> > > Best regards,
> > >
> > > Authors

---

> > > > ### Comment · Reviewer_bMLu · 2025-08-06
> > > >
> > > > Thank you for your detailed clarification. I have carefully reviewed the point you raised regarding Proposition C.2 in [3], and I agree with your observation. I appreciate that your Lemma C.3 (S7) explicitly resolves this issue. Given that this correction is not merely cosmetic but addresses a gap in the inductive proof, I now view this aspect of your work as a meaningful technical contribution. As such, I am willing to revise my score to 4 (Weak Accept).

---

> > > > > ### Author Response · Authors · 2025-08-07
> > > > >
> > > > > Thank you for your response and for reconsidering the score. We are glad to hear that our response was helpful. We would also be happy to hear if you have any additional comments.
> > > > >
> > > > > Best regards,
> > > > >
> > > > > Authors

---

### Official Review · Reviewer_ABZC · 2025-07-02

**Clarity:** 4
**Significance:** 3
**Originality:** 3
**Rating:** 5
**Confidence:** 3

**Summary:**

This paper studies weak-to-strong generalization, a phenomenon where a strong model trained with the supervision from a weaker model outperforms the weaker one. This work focuses on the theoretical analysis to interpret the model behavior in weak-to-strong generalization with a linear CNN as a weak model and a two-layer ReLU CNN as a strong model. Depending on the amount of data, this work provides rigorous explanations for the performance in weak-to-strong generalization. Specifically, when the data is scarce, the generalization can occur via benign overfitting, or fail due to harmful overfitting; when the data is abundant, the generalization occurs at the early stage of model training and diminishes as the training progresses.

**Questions:**

1. Some empirical insights might be helpful, such as identifying key aspects of real data that trigger benign or harmful overfitting in the data-scarce regime. For example, does a smaller dimension for data $d$, a larger proportion of mixed data $p_b$, or a larger number of samples for strong models $n_{st}$, tend to trigger benign overfitting more easily?

2. Is it possible to include experiments with simple real data? What would be the challenges or limitations in validating the theoretical results in real data?

3. As the weak-to-strong generalization refers to the stronger model outperforming the weaker model, is it needed to compare the performance of weaker and stronger models in the theoretical analysis in Theorem 3.4? It doesn't seem to be an obvious comparison in terms of the test errors.

4. How are the conditions 3.1 obtained, and how do they ensure the training properties as discussed in lines 187-192?

**Ethical Concerns:**

["NO or VERY MINOR ethics concerns only"]

**Final Justification:**

My major concern is that this work mainly presents theoretical insights without much empirical validation. The rebuttal of the authors showed the results on synthetic datasets and explained the challenges of empirical experiments, which addressed my concerns, and I would like to keep my positive rating.

**Limitations:**

The authors didn't mention the limitations or potential negative societal impact of their work in the main paper. Discussing the possible scope of applying the modeling or constraint to real data might be helpful.

**Paper Formatting Concerns:**

There are no major formatting issues.

**Quality:**

3

**Strengths And Weaknesses:**

Strengths:

1. This paper is well-written and clearly organized. The writing is concise and effective in conveying the claims. Overall, it's of high quality in terms of clarity.

2. This work provides a theoretical and detailed analysis of the weak-to-strong generalization, specifically from the aspects of gradient descent with linear/nonlinear models, which has the potential to inspire such modeling or further research in the community.

Weaknesses:

1. This paper mainly focuses on the theoretical analysis, and the experiments are limited to synthetic data modeled similarly in the setup. It would be better to include more empirical insights or experiments with simple real data.

2. It would be better for this paper if more discussions on the empirical insights could be added, such as identifying key aspects of data that we can adjust or change in real data to make weak-to-strong generalization occur.

---

> ### Author Rebuttal · Authors · 2025-07-31
>
> Thank you for your valuable comments. In the following, we address the points raised by the reviewer.
> ## **W1, W2. High-level Intuition from Our Analysis**
> We would like to restate the intuition behind our analysis presented in Section 4 at a higher level and emphasize that these insights can be extended to more general scenarios.
>
> A weak model may sometimes perform poorly because it fails to capture certain difficult-to-learn features, which we refer to as hard signals in our setting. Now, imagine training a strong model—capable of learning these hard signals that the weak model misses—using supervision from the weak model. The strong model can only acquire meaningful information from data points that are correctly pseudo-labeled by the weak model.
>
> Our main intuition for why the strong model can outperform the weak model is that there exists a subset of data containing these hard signals but that is still labeled correctly thanks to the presence of easier-to-learn features (or easy signals) that the weak model has successfully learned. This subset, which we call both-signal data, provides the necessary gradient information to help the strong model learn the hard signals. While our framework is simple, we believe the insights gained here can be applied to understand broader scenarios.
>
> Based on the intuition that using more both-signal data is crucial for weak-to-strong generalization, we believe that data selection techniques, such as uncertainty-based methods, can be designed to preferentially select such both-signal data. This is because both-signal data may exhibit higher uncertainty due to the weak model’s inability to fully capture the hard signals. We leave empirical investigations of this approach for future work.
>
> ## **Q1. Key Factors Triggering the Transition from Harmful to Benign Overfitting**
> You're almost right. In fact, noise level matters more than $d$ in general settings. More precisely, the transition from harmful to benign overfitting in weak-to-strong training is mainly governed by the number of both-signal samples, which depends on both the proportion of mixed data $p_b$ and the number of strong model training samples $n_{st}$. Additionally, the noise level in the dataset—controlled by $\sigma_p$ and dimension $d$ in our setting—also plays a key role. This is because only the both-signal data contributes to learning the hard signals, and a higher noise level requires more such data to avoid harmful overfitting.
>
> ## **W1, Q2. Experiments of Real-World Data**
> We additionally conducted experiments on simple real-world data to support our findings. Since it is hard to clearly delineate signal vs noise in real data, we modified the MNIST dataset to emphasize their roles.
>
> First, we multiply each pixel in the original images of digits 4, 5, 6, 7, 8, and 9 by 0.02, while keeping the images of other digits unchanged. This corresponds to the presence of hard signals, with digits 4–9 serving as the hard signals. To emphasize the role of noise, we replace the border region of each 28×28 image—a 5-pixel-wide frame along the edges—with standard Gaussian noise. This results in images where the central 18×18 region contains the digit, surrounded by Gaussian noise. Finally, we randomly concatenate two such modified images that share the same parity (i.e., both even or both odd), producing 28×56 images. We assign binary labels based on their parity.
>
> The resulting data includes a variety of signal types: some pairs contain two bright digits, others contain one bright and one dark digit, and some consist of two dark digits. These types serve as easy-only data, both-signal data, hard-only data in our setting.
>
> Using this modified real-world dataset, we investigate the weak-to-strong training scenario beyond our theoretical settings. For the weak model, we use an MLP consisting of a single hidden layer with 128 units followed by a ReLU activation. For the strong model, we use a CNN with three convolutional layers of increasing channels (64, 128, 256), each followed by batch normalization, ReLU activation, and max pooling. The extracted features are then flattened and passed through a fully connected layer with 512 units.
>
> We first trained the weak model using 500 samples. Then, we trained the strong model using labels predicted by the trained weak model, with varying numbers of training samples $n_{st} = 500, 1000, 1500, 2000, 2500$. We used the Adam optimizer with default parameters and trained each model for 300 epochs. Each experiment was repeated five times for every $n_{st}$, and we report the mean and standard deviation of the test accuracy for both the weak model and the resulting weak-to-strong model.
>
> | $n_{st}$ | weak test acc  | weak-to-strong test acc |
> |:---:|:---:|:---:|
> | 500 |  0.8606 ± 0.0045 | 0.8462  ± 0.0395 |
> | 1000 |  0.8593 ± 0.0013 | 0.8834 ± 0.0128 |
> | 1500 |   0.8642 ± 0.0150  | 0.8846 ± 0.0285  |
> | 2000 |  0.8669 ± 0.0052 | 0.8744 ± 0.0273 |
> | 2500 |  0.8586 ± 0.0139  | 0.8655 ± 0.0122 |
>
> We observe a trend in which the weak-to-strong gain increases with $n_{st}$ and then decreases. These observations are consistent with our theoretical findings, which describe a transition from harmful overfitting to benign overfitting, and eventually to the data-abundant regime.
>
> ## **Q3. Comparison of Test Error Between Weak Model Training and Weak-to-Strong Training**
> Here, we provide a comparison between the test errors presented in Theorem 3.4 and that of the weak model. In the benign overfitting case, our theoretical bound includes exponentially decreasing terms in the proportions of all three types of data—$p_e, p_b,$ and $p_h$. This implies that the model achieves low test error across all data types. Since we have shown in Proposition 2.1 that the weak model fails on hard-only data, this result demonstrates that the weak-to-strong model improves upon the weak model by a test error gap of $p_h/2 + o(1)$ via benign overfitting.
>
> In the harmful overfitting case, the test error is lower bounded by $0.12p_h$, which implies that the gap between the test errors of the weak and weak-to-strong models is upper bounded by a constant multiple (less than $1/2$) of $p_h$.
>
> We will clarify this comparison more explicitly in the next revision to avoid any potential confusion.
>
> ## **Q4. Regarding the Role of Condition 3.1**
> We provide additional clarification regarding Condition 3.1, which reflects standard assumptions commonly adopted in the feature learning theory literature (e.g., [1], [2]).
> Conditions (C1) and (C2) allow us to apply concentration inequalities to key quantities such as norms and correlations involving Gaussian noise, as well as to accurately track the relative proportions of different types of data. These assumptions are essential for analyzing training dynamics—for example, by enabling control over inner products between filters and Gaussian noise vectors.
> Conditions (C3) and (C4) assume a sufficiently small initialization scale and learning rate. This ensures that the influence of initialization is dominated by gradient descent updates, and that each step remains moderate, leading to stable training behavior.
> We would like to reiterate that Condition (C5) is motivated by the intuition that the learnability of a signal depends on both its frequency and strength. The lower bound on p_b imposed by Condition (C6) is also essential for weak-to-strong generalization, as both-signal data play a central role in our analysis.
>
> ## **Limitations**
> Thank you for your suggestions. We plan to discuss our limitations more explicitly in the next revision.
>
> ---
>
> Thank you very much for your time and consideration. Please let us know if anything remains unclear.
>
> Best regards,
>
> Authors
>
> ---
>
> **References**
>
> [1] Cao et al. Benign Overfitting in Two-layer Convolutional Neural Networks. NeurIPS 2022
>
> [2] Kou et al. Benign Overfitting in Two-layer ReLu Convolutional Neural Networks. ICML 2023

---

> > ### Comment · Reviewer_ABZC · 2025-08-04
> >
> > Thank the authors for their detailed rebuttal, which addressed my concerns. The clarification of the intuitive is clear, and the MNIST experiments validated the theoretical insights, which would be greatly helpful to include in the main paper as a synthetic dataset along with a discussion on the challenges of further empirical validation. I will maintain my positive rating of this work.

---

> > > ### Author Response · Authors · 2025-08-04
> > >
> > > Thank you for your response. We are glad to hear that our rebuttal addressed your concerns and was helpful. If you have any further questions, please don’t hesitate to reach out.
> > >
> > > Best regards,
> > >
> > > Authors

---

### Official Review · Reviewer_twxS · 2025-07-02

**Clarity:** 2
**Significance:** 2
**Originality:** 2
**Rating:** 4
**Confidence:** 4

**Summary:**

This paper attempts to find a simple setting in which weak to strong generalization provably occurs.  Their weak learner is a linear network with 3 copied weights (what they call a linear CNN).  Their strong learner is a 1-hidden layer ReLU CNN in which the second layer weights are set to one and only the first layer weights learn.  The ReLU CNN has three types of hidden units in correspondence with three patches in the data. Their model of data is a very contrived data distribution consisting of 3 different patches (matched to both the architecture of the linear and one-hidden layer CNN) with a set of easy, and hard signals in each patch, or noise.  They ask if the weak learner learns on actual data first, then the strong learner learns on data generated by the weak learner, when will the strong learner outperform the weak learner?

The types of results they can prove relevant to this are:

1) In a data scarce regime (not much data but above a threshold), the weak learner performs as well as expected, and the strong learner can perform better than the weak learner due to benign overfitting, in which it does not overfit to the mistakes of the weak learner.

2) In a data abundant regime, the strong learner provably exhibits weak to strong generalization, if the strong learner's learning is stopped before an early stopping time.  If the strong learner learns for too long it could potentially overfit to the mistakes of the weak learner (though no theoretical guarantees are derived at the convergence point of strong learning.  But early stopping of the strong learner provably prevents this.

3) They have simulations in which they train the strong learner for longer than the early stopping time and find performance degrades then plateaus. No theory is provided for this.

**Questions:**

What are the prospects of generalizing your theory methods to more general structures in data and lack of match to architectures?

**Ethical Concerns:**

["NO or VERY MINOR ethics concerns only"]

**Final Justification:**

While the setting in which the theorems are proved are quite simple, the phenomenon of weak to strong generalization is interesting enough that having a simple setting in which it provably occurs may be valuable, and the intuitions provided in the rebuttal as to what we can learn in general from this simple setting are helpful.

**Limitations:**

Yes. But further discussion on the limitations of their data assumptions would be helpful.

**Paper Formatting Concerns:**

None.

**Quality:**

2

**Strengths And Weaknesses:**

Strengths:

1) The paper provides a simple example in which weak to strong generalization is provable.
2) Some simulations bear out the predictions of the theory, and also reveal other phenomena at late times in learning that are beyond the reach of this theory.

Weakness

1) A major weakness is the highly contrived nature of the data and its match to the architecture.  The data comes in 3 patches, and the architecture comes with hidden units (or linear units) with receptive fields matched exactly to one of the three patches.  What goes into each patch has a strange combinatorial nature out of a finite set of objects (fixed orthogonal vectors).  Only first layer weights learn; second layer weights are fixed.  This strong match between data and architecture is unrealistic.  It seems that the analysis is tied to this match.  It is less clear what one can conclude about how the structure of realistic data impacts weak to strong generalization, and what is going on when there is not such contrived match, or such a toy dataset.  It is unclear how their derivations could be generalized.
2) Less of a concern but an area for improvement in exposition: the authors need 6 conditions C1-C6 in section 3.  Some of them are well explained intuitively whereas others are not.  More discussion of these conditions and their roles would help.

---

> ### Author Rebuttal · Authors · 2025-07-31
>
> We would like to express our gratitude for your valuable comments. In the following, we address the points raised by the reviewer.
>
> ## **W1, Q. Extensions and Practical Insights Based on Our Theory**
> We agree with the reviewer that our theoretical framework may look overly simplified compared to realistic settings. As described in lines 82–90, our work builds on the recent line of research in feature learning theory, where the adopted settings allow for mathematically tractable analysis of learning dynamics. While our setting may appear simplified, similar assumptions have been widely used to gain theoretical understandings on various phenomena—such as how model ensembling improves generalization [1], how benign overfitting arises [2,3], how different optimizers affect generalization [4–6], how architectural choices influence generalization [7,8], and how data augmentation contributes to generalization [9–11].
>
> Here, we provide a more high-level explanation of the intuition behind the analysis presented in Section 4. A weak model may sometimes exhibit low performance because it fails to learn certain hard-to-learn information (referred to as hard signals in our setting). Now, consider training a strong model—strong enough to capture the hard-to-learn information that the weak model fails to learn—under the supervision of the weak model. The strong model can only learn useful information from correctly pseudo-labeled data.
>
> Our key intuition behind how the strong model can outperform the weak model is as follows: there exists a subset of data that contains hard-to-learn information but is still labeled correctly due to the presence of easy-to-learn information (easy signals in our setting). This subset of data (termed both-signal data) provides gradient updates that enable the strong model to learn the hard-to-learn information.
>
> Our theoretical framework may seem simple at first glance, but analyzing its training dynamics is technically involved and the fomal proof spans 50 pages in the supplementary material. Also, the intuition from this setting can help us understand more general scenarios. One such example is early stopping, an often adopted practical choice for weak-to-strong generalization. The discussion in lines 313–322 explains the role of this strategy.
>
> Regarding our three-patch data setting, while it could in principle be extended to incorporate more patches, we deliberately choose the minimal number (three) that is sufficient to capture all essential types of signal and noise vectors defined in our framework, in order to maintain mathematical simplicity. In addition, we would like to note that our analysis can be potentially extended to more general architectures—for example, by adopting a simple CNN as the weak model and a simple ViT as the strong model, whose learning dynamics and benign overfitting have been studied in [3] and [8]. However, we believe that our simplified setting is more effective for delivering clear insights. For more details, you can check our response to Reviewer bMLu‘s Q1.
>
> We additionally conducted experiments on a simplified real-world dataset to support our findings. Specifically, we slightly modified the MNIST dataset to highlight the roles of key components such as signal and noise. The results demonstrate that our theoretical insights also hold beyond our settings. Detailed experimental settings and results are provided in our responses to Reviewer ABZC’s W1 and Q2.
> ## **W2. Further Explanation on Condition 3.1**
> We would like to provide further explanation of Condition 3.1. These types of conditions are widely adopted in feature learning theory literature that we have discussed above. In (C1) and (C2), we assume a sufficiently large dimension and large sample sizes. These assumptions allow us to apply concentration inequalities to quantities such as norms and correlations involving Gaussian noise, as well as the number of each type of data.
>
> In (C3) and (C4), we assume a sufficiently small initialization scale and learning rate. These assumptions ensure that the effect of initialization can be neglected compared to the gradient descent updates, and that each update step remains small, which enables stable training dynamics.
>
> Lastly, we provide explanations for (C5) and (C6) in lines 189–192 of the draft, and we would like to further clarify that the motivation behind (C5) is that the difficulty of learning a signal is determined by its frequency and strength.
>
> We will discuss these points more clearly in the next revision.
>
> ---
>
> Thanks for your time and consideration. We are happy to discuss further if anything remains unclear.
>
> Best regards,
>
> Authors
>
> ---
>
> **References**
>
> [1] Allen-Zhu & Li. Toward Understanding Ensemble, Knowledge Distillation and Self-Distillation in Deep Learning. ICLR 2023
>
> [2] Cao et al. Benign Overfitting in Two-layer Convolutional Neural Networks. NeurIPS 2022
>
> [3] Kou et al. Benign Overfitting in Two-layer ReLU Convolutional Neural Networks. ICML 2023
>
> [4] Zou et al. Understanding the Generalization of Adam in Learning Neural Networks with Proper Regularization. ICLR 2023
>
> [5] Jelassi et al. Towards Understanding How Momentum Improves Generalization in Deep Learning. ICML 2022
>
> [6] Chen et al. Why Does Sharpness-Aware Minimization Generalize Better than SGD. NeurIPS 2023
>
> [7] Huang et al. Quantifying the Optimization and Generalization Advantages of Graph Neural Networks Over Multilayer Perceptrons. AISTATS 2025
>
> [8] Jiang et al. Unveil Benign Overfitting for Transformer in Vision: Training Dynamics, Convergence, and Generalization. NeurIPS 2024
>
> [9] Shen et al. Data Augmentation as Feature Manipulation. ICML 2022
>
> [10] Zou et al. The Benefit of Mixup for Feature Learning. ICML 2023
>
> [11] Oh & Yun. Provable Benefit of Cutout and CutMix for Feature Learning. NeurIPS 2024

---

> > ### Author Response · Authors · 2025-08-07
> >
> > Dear Reviewer twxS
> >
> > Thank you for your time and effort in reviewing our work. We greatly appreciate your valuable and constructive feedback. Given your busy schedule, we would be grateful if you could review our responses to ensure we've adequately addressed your concerns. If you have any further questions or suggestions, please let us know, and we'll be happy to address them.
> >
> > Thank you again for your valuable contribution to our research.
> >
> > Best regards,
> >
> > Authors

---

> > ### Comment · Area_Chair_NJoG · 2025-08-07
> >
> > Dear Reviewer twxS,
> >
> > Thank you for your previous input. As the discussion deadline is approaching, we kindly remind you to review the author's response and continue the discussion if you have any further comments or questions.
> >
> > Your timely feedback will be greatly appreciated to ensure a thorough review process.
> >
> > Thank you very much for your attention!
> >
> > Best regards,
> >
> > AC

---

> > ### Comment · Reviewer_twxS · 2025-08-08
> > **Response to rebuttal**
> >
> > Thank you for your response.  I find it much clearer than the paper itself.  Incorporating parts of it into the revision would help.  I understand simplifications need to be made to prove theorems, and the question always is, what can we learn from this simplified setting that might generalize to other settings.  Discussing this more in the paper would help.  The MNIST results help.  I will raise my score accordingly to borderline accept.

---

> > > ### Author Response · Authors · 2025-08-09
> > >
> > > Thank you for your response. We are glad to hear that our rebuttal addressed your concerns. We also appreciate your suggestions for improvement and plan to incorporate them into our next revision.
> > >
> > > Best regards,
> > >
> > > Authors

---

### Comment · Area_Chair_NJoG · 2025-08-04

Dear Reviewers,

Thank you to those who have already engaged in the discussion. I encourage the remaining reviewers to read the author responses and join the discussion during this period.

Your participation is important for a fair and comprehensive review process, and early engagement allows for more meaningful exchanges.

Thank you all for your time and contributions.

Best regards,

AC

---

### Decision · Program_Chairs · 2025-09-17

**Decision:**

Accept (poster)

**Comment:**

This paper develops a theoretical framework for weak-to-strong generalization using a stylized setting where a linear CNN serves as the weak model and a two-layer ReLU CNN serves as the strong model. The authors prove that in data-scarce regimes, weak-to-strong generalization can occur via benign overfitting, while in abundant data regimes it arises under early stopping but may vanish with overtraining. Simulations support these results and highlight additional behaviors beyond theory.

The main strengths are the technical rigor, the clear theoretical characterization of different learning regimes, and the identification of precise conditions under which weak-to-strong generalization succeeds or fails. The rebuttal added valuable intuition, clarified assumptions, and provided new experiments on a modified MNIST dataset, demonstrating that the theoretical insights extend beyond purely synthetic cases. Importantly, the authors also corrected and extended aspects of prior feature-learning analyses, which reviewers recognized as a meaningful contribution. The weaknesses are primarily the highly engineered data distribution and limited real-world applicability, as well as an exposition that could better emphasize the roles of key assumptions.

During the rebuttal, the reviewers agreed that the paper makes a technically solid and timely contribution. The authors’ clarifications, additional intuition, and new experiments satisfactorily addressed the main concerns, and some reviewers raised their scores accordingly. While the narrow and stylized scope limits its suitability for a spotlight, the work provides rigorous and valuable insights into weak-to-strong generalization, a phenomenon of broad current interest. I therefore recommend acceptance.